# EQUIVARIANT METANETWORKS FOR MIXTURE-OF-EXPERTS WEIGHTS

## ABSTRACT

In neural networks, the parameter space serves as a proxy for the function class realized during training; however, the degree to which this parameterization provides a faithful and injective encoding of the underlying functional landscape remains insufficiently understood. A central challenge in this regard is the phenomenon of *functional equivalence*, wherein distinct parameter configurations give rise to identical input–output mappings, thereby revealing the inherent non-injectivity of the parameter-to-function correspondence. While this issue has been extensively studied in classical architectures-such as fully connected and convolutional neural networks with varying widths and activation functions—recent research has increasingly extended to modern architectures, particularly those utilizing multihead attention mechanisms. Motivated by this line of inquiry, we undertake a formal investigation of functional equivalence in Mixture-of-Experts-a class of architectures widely recognized for their scalability and efficiency. We analyze both dense and sparse gating regimes and demonstrate that functional equivalence in the Mixture-of-Experts architecture is fully characterized by permutation symmetries acting on both the expert modules and the gating mechanism. These findings have direct implications for the design of equivariant metanetworks-neural architectures that operate on pretrained weights to perform downstream tasks-where reasoning about functional identity is essential. Our results highlight the importance of analyzing functional equivalence in uncovering model symmetries and informing the development of more principled and robust metanetwork architectures.

## 1 INTRODUCTION

Despite the practical success of deep learning, many underlying mechanisms remain elusive. A particularly intriguing phenomenon is the ability of highly overparameterized neural networks-those with more parameters than training samples-to generalize well to unseen data, rather than overfit (Cybenko, 1989; Hornik et al., 1989). This observation challenges conventional expectations. While classical results suggest that shallow networks can approximate any function, empirical evidence consistently shows that deeper, complex architectures perform better (Zhang et al., 2017; Allen-Zhu et al., 2019). These apparent contradictions have spurred growing interest in understanding overparameterization and its broader implications for optimization, generalization, and model expressivity (Du et al., 2019; Frankle & Carbin, 2019; Neyshabur et al., 2019; Novak et al., 2018).

An important feature of overparameterized neural networks is their *functional equivalence*-the fact that multiple distinct parameter configurations can realize the same input-output function. This redundancy raises fundamental questions about how neural networks encode, optimize, and generalize learned representations (Allen-Zhu et al., 2019; Belkin et al., 2019; Du et al., 2019; Frankle & Carbin, 2018; Novak et al., 2018). The notion of functional equivalence has found many applications in different areas such as weight generation using diffusions (Soro et al., 2024; Saragih et al., 2025; Wang et al., 2025; Xie et al., 2024; Meynent et al., 2025; Andreis et al., 2024), model ensembling (Wortsman et al., 2022; Ganaie et al., 2022; Lakshminarayanan et al., 2017; Mohammed & Kora, 2023), and exploring mode connectivity (Goodfellow et al., 2014; Keskar et al., 2016; Sagun et al., 2017; Venturi et al., 2019; Neyshabur et al., 2020; Tatro et al., 2020; Yunis et al., 2022; Zhou et al., 2023). Functional equivalence has also recently been applied to the design of equivariant metanetworks (Tran et al., 2024b;a; Vo et al., 2025; Zhou et al., 2024c;b;a; Navon et al., 2023).

These metanetworks operate on internal components such as weights or gradients—rather than raw weights themselves—and have been used in a variety of tasks including learnable optimization (Bengio et al., 2013; Runarsson & Jonsson, 2000; Andrychowicz et al., 2016; Metz et al., 2022), feature extraction from implicit representations (Müller et al., 2023; Stanley, 2007; Mildenhall et al., 2021), model editing (Sinitsin et al., 2020; Cao et al., 2021; Mitchell et al., 2022), policy evaluation (Harb et al., 2020), and Bayesian inference (Sokota et al., 2021).

The problem of determining the functional equivalence of multilayer perceptrons (MLPs) was initially posed by Hecht-Nielsen (Hecht-Nielsen, 1990). It was observed that interchanging weights of two units in a hidden layer of an MLP does not change the network's input-output function, provided corresponding weights in the subsequent layer are adjusted accordingly (Allen-Zhu et al., 2019; Du et al., 2019; Frankle & Carbin, 2018; Belkin et al., 2019; Neyshabur et al., 2018). For the same class of MLPs, Fefferman and Markel (Fefferman & Markel, 1993) proved a strong result, showing that input-output mapping of an MLP with $\texttt{tanh}$ activations determines both architecture and weights, up to permutations and sign flips. Since then, a variety of results under different settings have been established for MLPs (Albertini & Sontag, 1993b;a; Bui Thi Mai & Lampert, 2020; Chen et al., 1993; Kurkova & Kainen, 1994), and similarly for convolutional neural networks (CNNs) (Brea et al., 2019; Novak et al., 2018; Bui Thi Mai & Lampert, 2020; Tran et al., 2024a; Vo et al., 2024).

While functional equivalence has been well studied in traditional architectures such as MLPs and CNNs, its characterization in modern architectures like Transformers (Vaswani et al., 2017; Devlin et al., 2018; Brown et al., 2020) and Mixture-of-Experts (MoE) (Jacobs et al., 1991; Shazeer et al., 2017; Lepikhin et al., 2020; Fedus et al., 2022) remains underexplored. For Transformers, recent work (Tran et al., 2025; Knyazev et al., 2024) has identified the maximal symmetry group of the multihead attention and established necessary and sufficient conditions for functional equivalence. In contrast, the functional characterization of MoE architectures remains an open problem.

**Contributions.** Inspired by this line of inquiry, we propose a comprehensive framework for constructing equivariant metanetworks for MoE architecture, based on the functional behavior. The paper is organized as follows:

1. In Section 2, we introduce the notion of the weight space associated with an MoE model and construct a group action that preserves its functional behavior. This formulation applies to both dense and sparse gating scenarios.

2. In Section 3, we establish two key theoretical results demonstrating that the proposed group action characterizes *all* universal symmetries inherent to the gating mechanism of MoE models. These results are supported by rigorous formal proofs.

3. In Section 4, we apply these theoretical findings to the design of equivariant metanetworks for MoE Transformer architectures. We introduce a metanetwork that is equivariant under the group action induced by the structure of the multi-head attention and MoE modules. We also release the *MoE Transformer Zoos dataset*, containing 179,000 MoE Transformer checkpoints, to support future research on MoE weight spaces. Experimental results demonstrate that our equivariant metanetwork consistently outperforms baseline models across datasets.

Additional materials—including a table of notation, theoretical derivations, detailed proofs, and experimental configurations—are provided in the Appendix.

## 2 WEIGHT SPACE OF MIXTURE-OF-EXPERTS AND ITS GROUP ACTION

This section provides a concise overview of the MoE architecture. We define the associated weight space and introduce a group action on this space that preserves the overall functionality. A comprehensive and formal treatment of these concepts is presented in Appendix A.

### 2.1 BACKGROUND ON MIXTURE-OF-EXPERTS

Throughout the paper, we denote by $\sigma$ the ReLU activation function.

**Mixture-of-Experts.** Let $D$ denote the token dimension and $D_e$ the hidden width. We consider *Expert maps* implemented as single-hidden-layer ReLU networks, $\mathrm{E}\colon \mathbb{R}^D \to \mathbb{R}^D$, defined as:

$$\mathrm{E}(x; W^{(A)}, b^{(A)}, W^{(B)}, b^{(B)}) = \sigma(xW^{(A)} + b^{(A)})W^{(B)} + b^{(B)}, \tag{1}$$

with parameters $(W^{(A)}, b^{(A)}, W^{(B)}, b^{(B)}) \in \mathbb{R}^{D \times D_e} \times \mathbb{R}^{1 \times D_e} \times \mathbb{R}^{D_e \times D} \times \mathbb{R}^{1 \times D}$. Given $n_e$ denoting the number of experts, an *MoE* is defined as a map $\mathrm{MoE} \colon \mathbb{R}^D \to \mathbb{R}^D$:

$$\mathrm{MoE}\Big(x; \big\{W^{(G,i)}, b^{(G,i)}, W^{(A,i)}, b^{(A,i)}, W^{(B,i)}, b^{(B,i)}\big\}_{i=1}^{n_e}\Big)$$

$$= \sum_{i=1}^{n_e} \mathrm{softmax}_i\left(\big\{W^{(G,i)}x + b^{(G,i)}\big\}_{i=1}^{n_e}\right) \cdot \mathrm{E}\Big(x; W^{(A,i)}, b^{(A,i)}, W^{(B,i)}, b^{(B,i)}\Big). \quad (2)$$

Here, $(W^{(A,i)}, b^{(A,i)}, W^{(B,i)}, b^{(B,i)}) \in \mathbb{R}^{D \times D_e} \times \mathbb{R}^{1 \times D_e} \times \mathbb{R}^{D_e \times D} \times \mathbb{R}^{1 \times D}$ are the parameters of the $i^{\text{th}}$ expert, while $(W^{(G,i)}, b^{(G,i)}) \in \mathbb{R}^D \times \mathbb{R}$ are the corresponding *gating* parameters. The vector $\mathrm{softmax}(W^{(G,i)}x + b^{(G,i)}{}_{i=1}^{n_e})$ sets the contribution of each expert to the final MoE output.

**Sparse Mixture-of-Experts.** Given a positive integer $K \le n_e$, the Top-$K$ map is defined by Top-$K(x) = \{i_1, \ldots, i_K\}$ for $x = (x_1, \ldots, x_n) \in \mathbb{R}^n$, where $i_1, \ldots, i_K$ are the indices corresponding to the $K$ largest components of $x$. In the event of ties, we select smaller indices first. Using this, a *Sparse Mixture-of-Experts* (SMoE) is the map $\mathrm{SMoE} \colon \mathbb{R}^D \to \mathbb{R}^D$ defined by:

$$\mathrm{SMoE}\Big(x; \big\{W^{(G,i)}, b^{(G,i)}, W^{(A,i)}, b^{(A,i)}, W^{(B,i)}, b^{(B,i)}\big\}_{i=1}^{n_e}\Big)$$

$$= \sum_{i \in T(x)} \mathrm{softmax}_i\left(\big\{W^{(G,i)}x + b^{(G,i)}\big\}_{i \in T(x)}\right) \cdot \mathrm{E}\Big(x; W^{(A,i)}, b^{(A,i)}, W^{(B,i)}, b^{(B,i)}\Big), \quad (3)$$

where $T(x) = T(x; \{W^{(G,i)}, b^{(G,i)}\}_{i=1}^{n_e}) = \text{Top-}K((W^{(G,i)}x + b^{(G,i)})_{i=1}^{n_e})$.

## 2.2 Weight Space of Mixture-of-Experts

The map MoE is parameterized as $\mathrm{MoE}(x; \theta)$ where

$$\theta = \Big(\big(W^{(G,i)}, b^{(G,i)}\big), \big(W^{(A,i)}, b^{(A,i)}\big), \big(W^{(B,i)}, b^{(B,i)}\big)\Big)_{i=1,\ldots,n_e}$$

$$\in \Theta(n_e) := \Big(\big(\mathbb{R}^D \times \mathbb{R}\big) \times \big(\mathbb{R}^{D \times D_e} \times \mathbb{R}^{1 \times D_e}\big) \times \big(\mathbb{R}^{D_e \times D} \times \mathbb{R}^{1 \times D}\big)\Big)^{n_e}. \quad (4)$$

Here, $\Theta(n_e)$ is called the *weight space* of a Mixture-of-$n_e$-experts. Varying the number of experts leads to an MoE weight space that spans across expert sets of different sizes, denoted by

$$\Theta = \bigsqcup_{n_e=1}^{\infty} \Theta(n_e) = \bigsqcup_{n_e=1}^{\infty} \Big(\big(\mathbb{R}^D \times \mathbb{R}\big) \times \big(\mathbb{R}^{D \times D_e} \times \mathbb{R}^{1 \times D_e}\big) \times \big(\mathbb{R}^{D_e \times D} \times \mathbb{R}^{1 \times D}\big)\Big)^{n_e}. \quad (5)$$

Note that, the weight space of SMoE coincides with that of the standard MoE, since the map Top-$K$ does not introduce any new trainable parameters.

## 2.3 Group Action on Weight Space of Mixture-of-Experts

We define the group $\mathcal{G}(n_e)$ as the direct product $\mathcal{G}(n_e) = \mathbb{R}^D \times \mathbb{R} \times \mathrm{S}_{n_e}$ of the groups $\mathbb{R}^D$, $\mathbb{R}$ with addition, and the permutation group $\mathrm{S}_{n_e}$. Each element $g \in \mathcal{G}(n_e)$ is of the form $g = (\gamma_W, \gamma_b, \tau)$, where $\gamma_W \in \mathbb{R}^D, \gamma_b \in \mathbb{R}$ and $\tau \in \mathrm{S}_{n_e}$. The group $\mathcal{G}(n_e)$ acts on the weight space $\Theta(n_e)$ as follows. For $g \in \mathcal{G}(n_e)$ and $\theta \in \Theta(n_e)$ presented as in Equation 4, define:

$$g\theta := \Big(\big(W^{(G,\tau(i))} + \gamma_W, b^{(G,\tau(i))} + \gamma_b\big),$$

$$\big(W^{(A,\tau(i))}, b^{(A,\tau(i))}\big), \big(W^{(B,\tau(i))}, b^{(B,\tau(i))}\big)\Big)_{i=1,\ldots,n_e}. \quad (6)$$

The result below establishes that this group action preserves the MoE map.

**Proposition 2.1** (Weight space invariance of MoE). *The MoE map is $\mathcal{G}(n_e)$-invariance under the action of $\mathcal{G}(n_e)$ on its weight space $\Theta(n_e)$, i.e. $\mathrm{MoE}(\cdot; \theta) = \mathrm{MoE}(\cdot; g\theta)$.*

A proof of Proposition 2.1 is presented in Proposition A.4. An analogous invariance result holds in the case of SMoE. However, since the Top-$K$ selection map is generally discontinuous—primarily due to tie cases in the gating scores—additional conditions are required to ensure the validity of

the invariance result. To address this, we focus on a subset of $\mathbb{R}^D$ where the Top-$K$ scores are unambiguously defined. Specifically, for $\{W^{(G,i)}, b^{(G,i)}\}_{i=1}^{n_e} \in \left(\mathbb{R}^D \times \mathbb{R}\right)^{n_e}$, we define:

$$\Omega\left(\{W^{(G,i)}, b^{(G,i)}\}_{i=1}^{n_e}\right) := \left\{x \in \mathbb{R}^D : (W^{(G,i)}x + b^{(G,i)})_{i=1}^{n_e} \text{ are pairwise distinct}\right\}. \quad (7)$$

The following result concerns the domain and the continuity properties of the SMoE map.

**Proposition 2.2.** *If $\left\{W^{(G,i)}, b^{(G,i)}\right\}$ are pairwise distinct for $i = 1, \ldots, n_e$, then $\Omega(\{W^{(G,i)}, b^{(G,i)}\}_{i=1}^{n_e})$ is an open and dense subset of $\mathbb{R}^D$. Moreover, the SMoE map, as defined in Equation 3, is continuous on $\Omega(\{W^{(G,i)}, b^{(G,i)}\}_{i=1}^{n_e})$.*

A proof of Proposition 2.2 is presented in Propositions A.1 and A.2. We now establish that the invariance property of the SMoE map holds under restriction to this domain.

**Proposition 2.3** (Weight space invariance of SMoE). *Given the SMoE map, as defined in Equation 3. Assume that $\{W^{(G,i)}, b^{(G,i)}\}$ are pairwise distinct for $i = 1, \ldots, n_e$. Then, the set $\Omega(\{W^{(G,i)}, b^{(G,i)}\}_{i=1}^{n_e})$ is invariant under the group action of $\mathcal{G}(n_e)$, i.e. for $g = (\gamma_W, \gamma_b, \tau) \in \mathcal{G}(n_e)$, we have $\Omega(\{W^{(G,i)}, b^{(G,i)}\}_{i=1}^{n_e}) = \Omega(\{W^{(G,\tau(i))} + \gamma_W, b^{(G,\tau(i))} + \gamma_b\}_{i=1}^{n_e})$. Moreover, the SMoE map, restricted to $\Omega(\{W^{(G,i)}, b^{(G,i)}\}_{i=1}^{n_e})$, is $\mathcal{G}(n_e)$-invariance under the action of $\mathcal{G}(n_e)$ on their weight space, i.e. $\mathrm{SMoE}(\cdot; \theta) = \mathrm{SMoE}(\cdot; g\theta)$ on $\Omega(\{W^{(G,i)}, b^{(G,i)}\}_{i=1}^{n_e})$.*

A proof of Proposition 2.3 is presented in Proposition A.5.

*Remark* 2.4. The invariance properties of both MoE and SMoE models in Proposition 2.1 and 2.3 stem from two fundamental characteristics: permutation invariance of the summation operator and translation invariance of the softmax function. Additionally, in the case of SMoE, these invariance properties are further supported by the permutation and translation invariance of the Top-$K$ map.

# 3 FUNCTIONAL EQUIVALENCE IN MIXTURE-OF-EXPERTS

This section is concerned with the correspondence between two sets of parameters that yield identical MoE maps. Our objective is to rigorously demonstrate that the group action induced by $\mathcal{G}(n_e)$, as defined in Equation 6, fully characterizes the symmetries inherent in the gating mechanism of MoE architectures. The dense and sparse cases will be analyzed separately due to their fundamentally distinct structural and analytical properties. Throughout the remainder of this section, we let $\theta \in \Theta(n_e)$ and $\widehat{\theta} \in \Theta(\widehat{n_e})$ denote the parameters of two models under comparison.

$$\theta = \left(\left(W^{(G,i)}, b^{(G,i)}\right), \left(W^{(A,i)}, b^{(A,i)}\right), \left(W^{(B,i)}, b^{(B,i)}\right)\right)_{i=1,\ldots,n_e}, \quad (8)$$

$$\widehat{\theta} = \left(\left(\widehat{W}^{(G,i)}, \widehat{b}^{(G,i)}\right), \left(\widehat{W}^{(A,i)}, \widehat{b}^{(A,i)}\right), \left(\widehat{W}^{(B,i)}, \widehat{b}^{(B,i)}\right)\right)_{i=1,\ldots,\widehat{n_e}}. \quad (9)$$

## 3.1 FUNCTIONAL EQUIVALENCE IN MIXTURE-OF-EXPERTS

The following result establishes a complete characterization of when $\theta$ and $\widehat{\theta}$, under certain assumptions, define the same MoE map, with particular emphasis on the behavior of the gating mechanism.

**Theorem 3.1** (Functional equivalence in MoE). *Suppose $\theta, \widehat{\theta}$ define the same MoE map, i.e. $\mathrm{MoE}(\cdot; \theta) = \mathrm{MoE}(\cdot; \widehat{\theta})$. If $\theta, \widehat{\theta}$ satisfy the following four assumptions:*

1. *$n_e$ experts $\left\{\mathrm{E}(\cdot; W^{(A,i)}, b^{(A,i)}, W^{(B,i)}, b^{(B,i)})\right\}_{i=1}^{n_e}$ are pairwise distinct functions;*

2. *$\widehat{n_e}$ experts $\{\mathrm{E}(\cdot; \widehat{W}^{(A,i)}, \widehat{b}^{(A,i)}, \widehat{W}^{(B,i)}, \widehat{b}^{(B,i)})\}_{i=1}^{\widehat{n_e}}$ are pairwise distinct functions;*

3. *$W^{(G,i)} - W^{(G,j)}$ are pairwise distinct for all $1 \leq i, j \leq n_e$ such that $i \neq j$;*

4. *$\widehat{W}^{(G,i)} - \widehat{W}^{(G,j)}$ are pairwise distinct for all $1 \leq i, j \leq \widehat{n_e}$ such that $i \neq j$;*

*then, $n_e = \widehat{n_e}$, and there exist $\tau \in \mathrm{S}_{n_e}$, $\gamma_W \in \mathbb{R}^D$, $\gamma_b \in \mathbb{R}$ such that for all $i = 1, \ldots, n_e$, we have $\widehat{W}^{(G,i)} = W^{(G,\tau(i))} + \gamma_W$, $\widehat{b}^{(G,i)} = b^{(G,\tau(i))} + \gamma_b$, and $\mathrm{E}(\cdot; W^{(A,\tau(i))}, b^{(A,\tau(i))}, W^{(B,\tau(i))}, b^{(B,\tau(i))}) = \mathrm{E}(\cdot; \widehat{W}^{(A,i)}, \widehat{b}^{(A,i)}, \widehat{W}^{(B,i)}, \widehat{b}^{(B,i)})$ on $\mathbb{R}^D$.*

A proof of Theorem 3.1 is presented in Appendix B. The proof relies on two key components: a result concerning the linear independence property of exponential functions, as stated in Lemma B.2, and an observation regarding the local affineness of ReLU networks, as discussed in Appendix B.2.

*Remark* 3.2. The four assumptions stated in Theorem 3.1 are introduced for technical reasons. At a high level, the goal in symmetry analysis is *to identify universal symmetries that are independent of specific parameter choices, while excluding singular symmetries that arise only under special configurations of the weights*. In particular, Assumptions 1 and 2 prevent degenerate cases in which two experts implement the same function and receive identical gating scores, thereby rendering their permutation inconsequential to the model's output. Assumptions 3 and 4 address a more subtle issue: they rule out configurations where linear dependencies among the gating weight vectors result in indistinguishable gating behavior across different experts. A complete justification of these assumptions, accompanied by illustrative examples, is provided in Remark B.8.

## 3.2 Functional Equivalence in Sparse Mixture-of-Experts

In the context of the sparse case, we first introduce the notion of the *strongly distinct* property. Specifically, two functions $f$ and $g$ defined on a topological space $X$ are said to be *strongly distinct* if the set $\{x \in X : f(x) \neq g(x)\}$ is dense in $X$.

*Remark* 3.3. For instance, distinct polynomials are strongly distinct, whereas distinct ReLU networks are not strongly distinct in general. A formal definition of this property, along with illustrative examples, is provided in Definition C.1 and Example C.2.

We now present a result that serves as an analogue of Theorem 3.1 in the context of SMoE for $K > 1$, formulated under a set of assumptions that are stronger than those required in the former.

**Theorem 3.4** (Functional equivalence in SMoE). *Suppose $\theta, \widehat{\theta}$ define the same SMoE maps, i.e.* $\mathrm{SMoE}(\cdot; \theta) = \mathrm{SMoE}(\cdot; \widehat{\theta})$. *If $\theta, \widehat{\theta}$ satisfy the following four assumptions:*

1. *$n_e$ experts $\{\mathrm{E}(\cdot; W^{(A,i)}, b^{(A,i)}, W^{(B,i)}, b^{(B,i)})\}_{i=1}^{n_e}$ are pairwise strongly distinct functions;*

2. *$\widehat{n_e}$ experts $\{\mathrm{E}(\cdot; \widehat{W}^{(A,i)}, \widehat{b}^{(A,i)}, \widehat{W}^{(B,i)}, \widehat{b}^{(B,i)})\}_{i=1}^{\widehat{n_e}}$ are pairwise strongly distinct functions;*

3. *$\{W^{(G,i-1)} - W^{(G,i)}\}_{i=2}^{n_e}$ is a linear independent subset of $\mathbb{R}^D$;*

4. *$\{\widehat{W}^{(G,i-1)} - \widehat{W}^{(G,i)}\}_{i=2}^{\widehat{n_e}}$ is a linear independent subset of $\mathbb{R}^D$;*

*then, $n_e = \widehat{n_e}$, and there exist $\tau \in \mathrm{S}_{n_e}$, $\gamma_W \in \mathbb{R}^D$, $\gamma_b \in \mathbb{R}$ such that for all $i = 1, \ldots, n_e$, we have $\widehat{W}^{(G,i)} = W^{(G,\tau(i))} + \gamma_W$, $\widehat{b}^{(G,i)} = b^{(G,\tau(i))} + \gamma_b$, and $\mathrm{E}(x; W^{(A,\tau(i))}, b^{(A,\tau(i))}, W^{(B,\tau(i))}, b^{(B,\tau(i))}) = \mathrm{E}(x; \widehat{W}^{(A,i)}, \widehat{b}^{(A,i)}, \widehat{W}^{(B,i)}, \widehat{b}^{(B,i)})$, for all $x \in \Omega(\{W^{(G,i)}, b^{(G,i)}\}_{i=1}^{n_e})$ such that $\tau(i) \in \mathrm{Top}\text{-}K((W^{(G,i)}x + b^{(G,i)})_{i=1}^{n_e})$.*

A proof of Theorem 3.4 is presented in Appendix C. Although Theorem 3.4 is conceptually aligned with Theorem 3.1, it is important to emphasize that *the SMoE case is significantly more challenging to establish*. The primary source of this difficulty lies in the presence of Top-$K$ operator, which introduces discontinuities by altering the set of contributing experts in a nontrivial and input-dependent manner. This behavior is notably difficult to analyze and control within the theoretical framework.

*Remark* 3.5. As previously stated, Theorem 3.4 is formulated under a stronger set of assumptions than those required in Theorem 3.1. Indeed, the assumptions of the latter directly imply those of the former. The rationale for imposing these stronger conditions stems from the observation that an expert's behavior is unconstrained on regions where it is not selected by the gating mechanism, thereby allowing arbitrary behavior in such domains. As a result, distinct collections of expert functions may yield identical overall outputs when restricted to their respective regions of activation. This ambiguity gives rise to singular symmetries, as discussed in Remark 3.2. A comprehensive justification of these assumptions, along with illustrative examples, is provided in Remark C.9.

**The case of $K = 1$.** In the special case where $K = 1$, the Top-1 gating mechanism in SMoE selects only the expert with the highest gating score, resulting in a softmax distribution that collapses to a

single entry equal to 1. Thus, the SMoE map with $K = 1$ also admits nontrivial symmetries under the action of the multiplicative group $\mathbb{R}_{>0}$. Specifically, for any $a > 0$, we have

$$\text{SMoE}\Big(x; \big\{W^{(G,i)}, b^{(G,i)}, W^{(A,i)}, b^{(A,i)}, W^{(B,i)}, b^{(B,i)}\big\}_{i=1}^{n_e}\Big)$$
$$= \text{SMoE}\Big(x; \big\{aW^{(G,i)}, ab^{(G,i)}, W^{(A,i)}, b^{(A,i)}, W^{(B,i)}, b^{(B,i)}\big\}_{i=1}^{n_e}\Big). \quad (10)$$

This invariance holds because the argmax used for expert selection is unaffected by uniform positive scaling, i.e. $\text{argmax}_{i=1,\ldots,n_e}\big(W^{(G,i)}x + b^{(G,i)}\big) = \text{argmax}_{i=1,\ldots,n_e}(aW^{(G,i)}x + ab^{(G,i)})$, for all $x \in \Omega(\{W^{(G,i)}, b^{(G,i)}\}_{i=1}^{n_e})$. Moreover, since only one expert is activated per input, no explicit interactions are formed among the expert components. This leads to a rich set of hidden symmetries within the architecture. Due to the complexity introduced by these symmetries, we choose to exclude the case $K = 1$ from our main analysis and leave its exploration to future work.

### 3.3 REMARKS ON FUNCTIONAL EQUIVALENCE IN MIXTURE-OF-EXPERTS MODELS

Theorems 3.1 and 3.4 provide a formal characterization of functional equivalence in both dense and sparse MoE architectures, with a primary focus on the role and structure of the gating mechanism. Nonetheless, these results do not exhaustively account for all symmetries inherent in the MoE and SMoE architectures as defined in Equations 2 and 3. In particular, further symmetries may exist within the internal structure of individual experts, especially when those experts are implemented as ReLU networks, as mentioned in Section 1. Since this work centers on the architectural properties of MoE, our analysis prioritizes the gating component, while abstracting expert networks by their input-output behavior rather than their internal parameterizations.

## 4 EQUIVARIANT METANETWORKS FOR MOE TRANSFORMERS

Metanetworks are neural architectures that take internal components of other models (weights, gradients, sparsity patterns, ...) as input to enable meta-level learning (Zhou et al., 2024b). A central design principle is that they operate on functions defined by parameters, not raw weights—motivating equivariance: functionally equivalent parameters should yield consistent outputs. This has led to permutation-equivariant metanetworks (Navon et al., 2023; Zhou et al., 2024b; Kofinas et al., 2024; Zhou et al., 2024c), with extensions to symmetries like scaling, sign flipping via graph message passing (Kalogeropoulos et al., 2024) and parameter sharing (Tran et al., 2024a; Vo et al., 2025).

While metanetworks have been studied in MLPs, CNNs, and Transformers, no prior work, to our knowledge, has investigated equivariant metanetworks for MoE Transformers. Using the established functional equivalence for MoE architecture, we provide a design for an equivariant metanetwork for MoE Transformers. We also release a dataset containing 179k MoE Transformer checkpoints spanning both language and vision tasks, enabling systematic analysis of their weight space.

### 4.1 EQUIVARIANT METANETWORKS FOR MOE TRANSFORMERS

Since the weight space, symmetry, and group action are the same for both MoE and SMoE, we describe the equivariant metanetwork for the MoE Transformer in this section. The construction for the SMoE Transformer is identical.

An *MoE Transformer layer* comprises a multihead attention module followed by an MoE module, where each expert in the MoE module is realized as a single hidden-layer network. Formally, an MoE Tranformer layer, denoted as MoETransformer, transforms an input sequence $X \in \mathbb{R}^{L \times D}$ to an output sequence $\text{MoETransformer}(X) \in \mathbb{R}^{L \times D}$, is defined as follows:

$$\text{MoETransformer}(X) =$$
$$\text{LayerNorm}\Big(\hat{X} + \text{MoE}\Big(\hat{X}; \big\{[W]^{(G,i)}, [b]^{(G,i)}, [W]^{(A,i)}, [b]^{(A,i)}, [W]^{(B,i)}, [b]^{(B,i)}\big\}_{i=1}^{n_e}\Big)\Big),$$
$$\text{where} \quad \hat{X} = \text{LayerNorm}\Big(X + \text{MultiHead}\Big(X; \{[W]^{(Q,i)}, [W]^{(K,i)}, [W]^{(V,i)}, [W]^{(O,i)}\}_{i=1}^{n_h}\Big)\Big).$$

Here, the MoE operator is a token-wise operator and is defined in Equation 2, and the MultiHead is defined in (Tran et al., 2025). The positive integers $n_h$ and $n_e$ represent the number of heads in the multihead attention module and the number of experts in the MoE module, respectively.

The *weight space of the MoE Transformer layer* is a direct product of the weight space of the multihead attention module (detailed description in Tran et al. (2025)) and the MoE module (refer to Section 2). In particular, the weight space $\mathcal{U}$ of the MoE Transformer layer above is defined as:

$$\mathcal{U} = \left( \mathbb{R}^{D \times D_k} \times \mathbb{R}^{D \times D_k} \times \mathbb{R}^{D \times D_v} \times \mathbb{R}^{D_v \times D} \right)^{n_h}$$

$$\times \left( \left( \mathbb{R}^D \times \mathbb{R} \right) \times \left( \mathbb{R}^{D \times D_e} \times \mathbb{R}^{1 \times D_e} \right) \times \left( \mathbb{R}^{D_e \times D} \times \mathbb{R}^{1 \times D} \right) \right)^{n_e}. \quad (11)$$

An element $U \in \mathcal{U}$ takes the form:

$$U = \left( \left( [W]^{(Q,i)}, [W]^{(K,i)}, [W]^{(V,i)}, [W]^{(O,i)} \right)_{i=1,\ldots,n_h}, \right.$$

$$\left. \left( \left( [W]^{(G,i)}, [b]^{(G,i)} \right), \left( [W]^{(A,i)}, [b]^{(A,i)} \right), \left( [W]^{(B,i)}, [b]^{(B,i)} \right) \right)_{i=1,\ldots,n_e} \right). \quad (12)$$

Here, for $i = 1, \ldots, n_h$, the matrices $[W]^{(Q,i)} \in \mathbb{R}^{D \times D_k}$, $[W]^{(K,i)} \in \mathbb{R}^{D \times D_k}$, $[W]^{(V,i)} \in \mathbb{R}^{D \times D_v}$, and $[W]^{(O,i)} \in \mathbb{R}^{D_v \times D}$ are the query, key, value, and linear projection matrices, respectively, of the $i^{\text{th}}$ head of the multihead attention. The rest of $U$ includes the parameters of the MoE component.

The *symmetry group of the weight space* $\mathcal{U}$, denoted $\mathcal{G}_{\mathcal{U}}$, is defined as the direct product of the symmetry group of the multi-head attention module and that of the MoE module, i.e.,

$$\mathcal{G}_{\mathcal{U}} = \left( \mathbf{S}_{n_h} \times \left( \mathrm{GL}_{D_k}(\mathbb{R}) \times \mathrm{GL}_{D_v}(\mathbb{R}) \right)^{n_h} \right) \times \left( \mathbb{R}^D \times \mathbb{R} \right) \times \left( \mathbf{S}_{n_e} \times \left( \mathcal{P}_{D_e} \right)^{n_e} \right). \quad (13)$$

Each element $g \in \mathcal{G}_{\mathcal{U}}$ takes the form:

$$g = \left( \left( \tau_h, \left\{ M_k^{(i)}, M_v^{(i)} \right\}_{i=1,\ldots,n_h} \right), \left\{ \gamma_W, \gamma_b \right\}, \left( \tau_e \times \left\{ \pi_e^{(i)} \right\}_{i=1,\ldots,n_e} \right) \right). \quad (14)$$

Here, the first component $\left( \tau_h, \left\{ M_k^{(i)}, M_v^{(i)} \right\}_{i=1,\ldots,n_h} \right)$ of $g$ arises from the symmetry of the multi-head attention module. The second component $\left\{ \gamma_W, \gamma_b \right\}$ corresponds to the symmetry of the gating score functions. The third component $\left( \tau_e, \left\{ \pi_e^{(i)} \right\}_{i=1,\ldots,n_e} \right)$ captures the permutation symmetry among the $n_e$ experts as well as the permutation symmetries within the hidden layers of each expert.

*The action of $\mathcal{G}_{\mathcal{U}}$ on $\mathcal{U}$* is defined to be the map $\mathcal{G}_{\mathcal{U}} \times \mathcal{U} \to \mathcal{U}$, which maps $(g, U) \in \mathcal{G}_{\mathcal{U}} \times \mathcal{U}$ to $gU \in \mathcal{U}$. Intuitively, $gU$ is obtained by independently applying the first component of $g$ to the weights of the multi-head attention module, and then applying the remaining components of $g$ to the MoE module. As a consequence of Theorems 3.1 and 3.4, the MoE Transformer is invariant under this group action. Equivalently, $U$ and $gU$ yield the same MoE Transformer maps for every $U \in \mathcal{U}$ and $g \in \mathcal{G}_{\mathcal{U}}$. Detailed formulation for $gU$ and its properties are given explicitly in Appendix D.

*Equvariant and invariant metanetwork layers* are the essential components in the construction of our equivariant metanetworks for MoE Transformer models. In particular, an equivariant metanetwork layer is a map $E \colon \mathcal{U} \to \mathcal{U}$ such that $E(gU) = gE(U)$ for all $U \in \mathcal{U}$ and $g \in \mathcal{G}_{\mathcal{U}}$. To construct $E(U)$, we follow the design of equivariant polynomial layers in Tran et al. (2025), we adopt a quadratic polynomial in the input weights $U$ with unknown coefficients. In particular, each entry of $E(U)$ is designed to be a linear combination with unknown coefficients of the entries of

- the products $[W]^{(QK,s)} = [W]^{(Q,s)} \left( [W]^{(K,s)} \right)^{\top}$, and $[W]^{(VO,s)} = [W]^{(V,s)} \left( [W]^{(O,s)} \right)^{-1}$;

- the matrices $[W]^{(Q,s)}$, $[W]^{(K,s)}$, $[W]^{(V,s)}$, and $[W]^{(O,s)}$ inside the multihead attention module;

- the matrices $[W]^{(G,s)}$ and the vector $[g]^{(Q,s)}$ in the gating functions, as well as the matrices $[W]^{(A,s)}$, $[W]^{(B,s)}$ and the vectors $[b]^{(A,s)}$, $[b]^{(B,s)}$ of the experts;

for every index $s$ and a bias term. Following the parameter-sharing technique, we solve the system of equations arising from the condition $E(gU) = gE(U)$ with all $U \in \mathcal{U}$ and $g \in \mathcal{G}_{\mathcal{U}}$ to obtain the necessary and sufficient constraints on the unknown coefficients that ensure $E$ is equivariant. The invariant layer is constructed using the same approach. The construction of both equivariant and invariant layers are quite lengthy and it is discussed in detail in Appendices F and G. A description of how to implement the equivariant and invariant layers are presented in Appendix H.

Table 1: Evaluation of model performance on the AgNews-MoEs dataset using Kendall's $\tau$ rank correlation. Error bars denote the standard error over 5 independent runs.

| | | Accuracy threshold | | | |
| | No threshold | 20% | 40% | 60% | 80% |
|---|---|---|---|---|---|
| MLP | $0.610 \pm 0.007$ | $0.610 \pm 0.001$ | $0.595 \pm 0.021$ | $0.538 \pm 0.006$ | $0.479 \pm 0.013$ |
| XGBoost (Chen & Guestrin, 2016) | $0.666 \pm 0.002$ | $0.665 \pm 0.001$ | $0.626 \pm 0.001$ | $0.619 \pm 0.003$ | $0.611 \pm 0.001$ |
| LightGBM (Ke et al., 2017) | $0.672 \pm 0.003$ | $0.673 \pm 0.001$ | $0.623 \pm 0.017$ | $0.621 \pm 0.004$ | $0.590 \pm 0.002$ |
| Random Forest (Breiman, 2001) | $0.619 \pm 0.003$ | $0.620 \pm 0.002$ | $0.583 \pm 0.002$ | $0.571 \pm 0.002$ | $0.558 \pm 0.001$ |
| Support Vector Regression (Vapnik et al., 1996) | $0.442 \pm 0.012$ | $0.407 \pm 0.019$ | $0.414 \pm 0.003$ | $0.374 \pm 0.009$ | $0.268 \pm 0.012$ |
| Transformer-NFN (Tran et al., 2025) | $0.777 \pm 0.001$ | $0.781 \pm 0.002$ | $0.732 \pm 0.002$ | $0.726 \pm 0.001$ | $0.712 \pm 0.002$ |
| MoE-NFN (ours) | $\mathbf{0.788 \pm 0.001}$ | $\mathbf{0.790 \pm 0.002}$ | $\mathbf{0.758 \pm 0.001}$ | $\mathbf{0.745 \pm 0.002}$ | $\mathbf{0.734 \pm 0.001}$ |

## 4.2 DATASET: MoE TRANSFORMER MODEL ZOOS

Mixture of Experts (MoE) Transformers have been incorporated into several recent deep learning architectures (DeepSeek-AI et al., 2025; Riquelme et al., 2021; Du et al., 2022). However, their internal weight structures remain largely unexplored from the perspective of metanetworks—partly due to the absence of suitable pretrained weight datasets. Existing datasets (Tran et al., 2024b) only provide pretrained weights for standard Transformer architectures and do not include pretrained MoE Transformer models. To address this gap, we introduce the MoE Transformer Model Zoos, which comprise two datasets: **AGNews-MoEs** and **MNIST-MoEs**. These contain small-scale MoE Transformer weights trained on text classification task using the AG News dataset (Zhang et al., 2015) and image classification task using the MNIST dataset (LeCun & Cortes, 2005), respectively.

The AGNews-MoEs dataset includes 79,220 model checkpoints, while MNIST-MoEs comprises of 100,024 checkpoints. each generated under diverse training conditions. For each checkpoint, both training and test accuracy are recorded. These datasets provide a foundation for training metanetworks aimed at predicting model generalization performance directly from its weight, without requiring access to the original test data. Comprehensive details on the structure of the pretrained weights are provided in Appendix K. We release these datasets publicly to support further research on modeling and understanding the weight space of MoE Transformer architectures.

## 4.3 EXPERIMENTAL RESULTS

To assess the effectiveness of our proposed MoE-NFN in modeling the weight space of MoE Transformers, we conduct two generalization prediction experiments on AGNews-MoEs and MNIST-MoEs. The goal is to test whether MoE-NFN can predict test accuracy directly from learned weights. For Transformer-NFN (Tran et al., 2025), which is not fully compatible with gated MoEs, we adapt inputs by averaging expert weights and removing gating. Other baselines—including MLPs, tree-based models (Chen & Guestrin, 2016; Ke et al., 2017; Breiman, 2001), and SVR (Vapnik et al., 1996)—use flattened weight vectors as input. Performance is measured with Kendall's $\tau$ rank correlation (Kendall, 1938). Full experiment details appear in Appendix L, with an additional ablation study of layer size and depth in Appendix I.

### 4.3.1 GENERALIZATION PREDICTION FOR AGNews-MoEs TRANSFORMER WEIGHTS

**Experiment Setup.** We evaluate the performance of MoE-NFN on the AGNews-MoEs dataset, which consists of pretrained language model weights. As illustrated by the accuracy distribution in Figure 1, the dataset is slightly skewed toward high-performing models. To enable a more balanced and comprehensive evaluation, we partition the dataset into five subsets based on test accuracy thresholds. The first subset includes all models without thresholding, while the remaining four contain only models with test accuracy above 20%, 40%, 60%, and 80%, respectively. This setup allows us to assess the generalization prediction performance of different models across a range of quality.

**Results.** Table 1 shows that our proposed MoE-NFN consistently achieves the highest performance across all accuracy thresholds on the AGNews-MoEs dataset. Without any threshold, MoE-NFN achieves a Kendall's $\tau$ of 0.788, compared to 0.777 for Transformer-NFN (Tran et al., 2025). This trend persists across all thresholds, where MoE-NFN consistently outperforms other baselines, and Transformer-NFN ranks second in every case. These results highlight the importance of aligning the functional network's design with the structure of the underlying pretrained models. While Transformer-NFN is tailored to standard Transformers, MoE-NFN generalizes this formulation by explicitly modeling gating and expert modularity.

Table 2: Evaluation of model performance on the MNIST-MoEs dataset using Kendall's $\tau$ rank correlation. Error bars denote the standard error over 5 independent runs.

| | | | Accuracy threshold | | |
| --- | --- | --- | --- | --- | --- |
| | No threshold | 20% | 40% | 60% | 80% |
| MLP | $0.798 \pm 0.002$ | $0.767 \pm 0.006$ | $0.708 \pm 0.001$ | $0.662 \pm 0.001$ | $0.593 \pm 0.013$ |
| XGBoost (Chen & Guestrin, 2016) | $0.781 \pm 0.002$ | $0.778 \pm 0.004$ | $0.746 \pm 0.001$ | $0.728 \pm 0.001$ | $0.659 \pm 0.002$ |
| LightGBM (Ke et al., 2017) | $0.810 \pm 0.001$ | $0.784 \pm 0.002$ | $\underline{0.765 \pm 0.001}$ | $\mathbf{0.737 \pm 0.002}$ | $\mathbf{0.681 \pm 0.004}$ |
| Random Forest (Breiman, 2001) | $0.747 \pm 0.001$ | $0.732 \pm 0.003$ | $0.697 \pm 0.002$ | $0.686 \pm 0.004$ | $0.624 \pm 0.003$ |
| SVR (Vapnik et al., 1996) | $0.442 \pm 0.012$ | $0.407 \pm 0.019$ | $0.415 \pm 0.004$ | $0.373 \pm 0.009$ | $0.268 \pm 0.012$ |
| Transformer-NFN (Tran et al., 2025) | $\underline{0.828 \pm 0.002}$ | $\underline{0.786 \pm 0.001}$ | $0.756 \pm 0.001$ | $0.686 \pm 0.001$ | $0.623 \pm 0.003$ |
| MoE-NFN (ours) | $\mathbf{0.833 \pm 0.001}$ | $\mathbf{0.790 \pm 0.001}$ | $\mathbf{0.770 \pm 0.001}$ | $\underline{0.731 \pm 0.001}$ | $\underline{0.672 \pm 0.002}$ |

Table 3: Performance measured by Kendall's $\tau$ of all models on the original and augmented test sets for MNIST-MoEs and AGNews-MoEs using the group action $\mathcal{G}_{\mathcal{U}}$.

| Dataset | Split | MoE-NFN | Transformer-NFN | MLP | SVR | LightGBM | Random Forest | XGBoost |
| --- | --- | --- | --- | --- | --- | --- | --- | --- |
| AGNews-MoEs | Original | 0.788 | 0.769 | 0.608 | 0.445 | 0.671 | 0.621 | 0.665 |
| | Augmented | 0.788 | 0.768 | 0.048 | 0.005 | 0.559 | 0.619 | 0.653 |
| | Gap | **0** | $\underline{0.001}$ | 0.560 | 0.440 | 0.112 | 0.002 | 0.012 |
| MNIST-MoEs | Original | 0.833 | 0.828 | 0.798 | 0.451 | 0.811 | 0.747 | 0.781 |
| | Augmented | 0.833 | 0.826 | 0.223 | 0.019 | 0.797 | 0.744 | 0.776 |
| | Gap | **0** | $\underline{0.002}$ | 0.575 | 0.432 | 0.014 | 0.003 | 0.005 |

### 4.3.2 GENERALIZATION PREDICTION FOR MNIST-MoEs TRANSFORMER WEIGHTS

**Experiment Setup.** We split the MNIST-MoEs dataset into five subsets based on accuracy thresholds, following the same procedure used in the AGNews-MoEs analysis. For each subset, we evaluate the ability of each metanetwork to predict generalization performance from pretrained weights. The alignment between predicted and true test accuracy is measured by Kendall's $\tau$ correlation.

**Results.** As shown in Table 2, our MoE-NFN achieves the highest Kendall's $\tau$ on most thresholds: the full test set (0.833), the 20% threshold (0.790), and the 40% threshold (0.770), while ranking second at the 60% and 80% thresholds. Interestingly, LightGBM performs well at higher thresholds, likely due to capturing strong nonlinear correlations in these high-accuracy subsets. Despite this, MoE-NFN remains competitive and consistently strong, demonstrating robustness and adaptability. It also outperforms Transformer-NFN (Tran et al., 2025) in all cases, highlighting the benefit of modeling MoE-specific structures such as expert modularity and gating.

### 4.3.3 EFFECT OF $\mathcal{G}_{\mathcal{U}}$ TRANSFORMATIONS ON GENERALIZATION PREDICTION

**Experiment Setup.** Under the group action $g \in \mathcal{G}_{\mathcal{U}}$, different parameter values can represent the same underlying function. To evaluate whether models trained on the training set are invariant to such transformations, we construct an augmented test set by applying randomly sampled elements from $\mathcal{G}_{\mathcal{U}}$ to the test set weights, producing functionally equivalent but parametrically distinct models.

**Results.** Table 3 empirically confirms that MoE-NFN is invariant under the group transformation $\mathcal{G}_{\mathcal{U}}$, showing zero performance drop across both datasets. Notably, Transformer-NFN also demonstrates strong stability, with only minor gaps of 0.002 on MNIST-MoEs and 0.001 on AGNews-MoEs. This robustness can be attributed to its design: Transformer-NFN is explicitly invariant to the subgroup $\mathrm{S}n_h \times (\mathrm{GL}D_k(\mathbb{R}) \times \mathrm{GL}D_v(\mathbb{R}))^{n_h}$, and also $(\mathbb{R}^D \times \mathbb{R})$ due to removal of gating. In contrast, other models except Random Forest show notable performance drop on augmented sets.

## 5 CONCLUSION

This paper defines a weight space for Mixture-of-Experts (MoE) models and introduces a group action that preserves functionality across dense and sparse gating. We prove that it captures all universal MoE symmetries, though the Top-1 sparse case remains open for future analysis. Building on this, we develop an equivariant metanetwork framework for pretrained MoE weights and release two benchmarks—MNIST-MoE and AGNews-MoE. Experiments and ablations show that symmetry-aware functional reasoning significantly improves metanetwork performance. These results highlight the importance of symmetry and functional equivalence for both theoretical understanding and practical model design. One limitation is the assumption of a fixed weight, leaving dynamic-weight settings as a direction for future work.

**Ethics Statement.** Given the nature of the work, we do not foresee any negative societal and ethical impacts of our work.

**Reproducibility Statement.** Source codes for our experiments are provided in the supplementary materials of the paper. The details of our experimental settings are given in Section 4 and the Appendix L. All datasets used in this paper are publicly available through an anonymous link provided in the README file of the supplementary material.

**LLM Usage.** In this paper, large language models (LLMs) were used solely as a tool to assist and refine the writing process. They helped with phrasing, clarity, and stylistic polishing, but all conceptual work, analyses, and conclusions were developed independently by the authors. The LLM served only to improve readability and presentation, without contributing to the research content itself

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

TABLE OF NOTATION

| | |
|---|---|
| $n_h$ | Number of head of Attention module |
| $n_e$ | Number of expert of MoE module |
| $D$ | Hidden dimension of the model |
| $D_k$ | Dimension of key/query vector in Attention module |
| $D_v$ | Dimension of value vector in Attention module |
| $D_A$ | Hidden dimension of the expert |
| $[W]^{Q,i}$ | Weight of query matrix of head $i$ |
| $[W]^{K,i}$ | Weight of key matrix of head $i$ |
| $[W]^{V,i}$ | Weight of value matrix of head $i$ |
| $[W]^{O,i}$ | Weight of out projection matrix of head $i$ |
| $[W]^{G,i}$ | Weight of the gating MLP corresponding to expert $i$ |
| $[W]^{A,i}$ | Weight of the first MLP of expert $i$ |
| $[W]^{B,i}$ | Weight of the second MLP of expert $i$ |
| $[b]^{G,i}$ | Bias of the gating MLP corresponding to expert $i$ |
| $[b]^{A,i}$ | Bias of the first MLP of expert $i$ |
| $[b]^{B,i}$ | Bias of the second MLP of expert $i$ |
| $\mathcal{U}$ | Weight space of Transformer MoE |
| $\mathcal{G}_{\mathcal{U}}$ | Symmetric group of the weight space |
| $\sigma()$ | Relu activation |
| $\tau_h$ | Head permutation group action in Attention module |
| $\gamma_W$ | Symmetry parameterization of the gating weight |
| $\gamma_b$ | Symmetry parameterization of the gating bias |
| $\tau_h$ | Expert permutation group action in MoE module |
| $\pi_e^{(i)}$ | Permutation group action of hidden vector of expert $i$ |
| $E()$ | Equivariant layer |
| $I()$ | Invariant layer |
| $\mathbb{R}^d$ | $d$-dimensional Euclidean space |
| $\langle \cdot, \cdot \rangle$ | standard dot product |
| $\sqcup$ | disjoint union |
| $g$ | element of group |
| $\mathrm{GL}_D(\mathbb{R})$ | General linear group of invertible $D \times D$ matrices over $\mathbb{R}$ |

# Appendix of "Equivariant Metanetworks for Mixture-of-Experts weights"

**Table of Contents**

## A    WEIGHT SPACES OF MIXTURE-OF-EXPERTS AND THEIR GROUP ACTIONS

Denote the ReLU activation as $\sigma$.

### A.1    WEIGHT SPACE OF MIXTURE-OF-EXPERTS

We recall the definition of the weight space for Mixture-of-Experts (MoE) where experts are implemented as single-hidden-layer neural networks. Let $D$ denote the input token dimension and $D_e$ the hidden layer size. We focus on expert maps E : $\mathbb{R}^D \to \mathbb{R}^D$ of the form:

$$\mathrm{E}\left(x; W^{(A)}, b^{(A)}, W^{(B)}, b^{(B)}\right) = \sigma(xW^{(A)} + b^{(A)})W^{(B)} + b^{(B)}, \tag{15}$$

with learnable parameters

$$\left(\left(W^{(A)}, b^{(A)}\right), \left(W^{(B)}, b^{(B)}\right)\right) \in \left(\mathbb{R}^{D \times D_e} \times \mathbb{R}^{1 \times D_e}\right) \times \left(\mathbb{R}^{D_e \times D} \times \mathbb{R}^{1 \times D}\right). \tag{16}$$

Given a positive integer $n_e$ denoting the number of experts, an *MoE* is the map MoE : $\mathbb{R}^D \to \mathbb{R}^D$ defined by

$$\mathrm{MoE}\left(x; \left\{W^{(G,i)}, b^{(G,i)}, W^{(A,i)}, b^{(A,i)}, W^{(B,i)}, b^{(B,i)}\right\}_{i=1}^{n_e}\right)$$

$$= \sum_{i=1}^{n_e} \mathrm{softmax}_i\left(\left\{W^{(G,i)}x + b^{(G,i)}\right\}_{i=1}^{n_e}\right) \cdot \mathrm{E}\left(x; W^{(A,i)}, b^{(A,i)}, W^{(B,i)}, b^{(B,i)}\right). \tag{17}$$

The map MoE is parameterized as $\text{MoE}(x; \theta)$ where

$$\theta = \left( \left( W^{(G,i)}, b^{(G,i)} \right), \left( W^{(A,i)}, b^{(A,i)} \right), \left( W^{(B,i)}, b^{(B,i)} \right) \right)_{i=1,\ldots,n_e}$$

$$\in \left( \left( \mathbb{R}^D \times \mathbb{R} \right) \times \left( \mathbb{R}^{D \times D_e} \times \mathbb{R}^{1 \times D_e} \right) \times \left( \mathbb{R}^{D_e \times D} \times \mathbb{R}^{1 \times D} \right) \right)^{n_e}. \quad (18)$$

Denote the weight space of an MoE with $n_e$-experts as

$$\Theta(n_e) = \left( \left( \mathbb{R}^D \times \mathbb{R} \right) \times \left( \mathbb{R}^{D \times D_e} \times \mathbb{R}^{1 \times D_e} \right) \times \left( \mathbb{R}^{D_e \times D} \times \mathbb{R}^{1 \times D} \right) \right)^{n_e}. \quad (19)$$

Varying the number of experts leads to an MoE weight space that spans across expert sets of different sizes, denoted by

$$\Theta = \bigsqcup_{n_e=1}^{\infty} \Theta(n_e) = \bigsqcup_{n_e=1}^{\infty} \left( \left( \mathbb{R}^D \times \mathbb{R} \right) \times \left( \mathbb{R}^{D \times D_e} \times \mathbb{R}^{1 \times D_e} \right) \times \left( \mathbb{R}^{D_e \times D} \times \mathbb{R}^{1 \times D} \right) \right)^{n_e}. \quad (20)$$

## A.2 Weight Space of Sparse Mixture-of-Experts

Given a positive integer $K \leq n_e$, the Top-$K$ map is defined by: for any vector $x = (x_1, \ldots, x_n) \in \mathbb{R}^n$,

$$\text{Top-}K(x) = \{i_1, \ldots, i_K\}, \quad (21)$$

where $i_1, \ldots, i_K$ are the indices corresponding to the $K$ largest components of $x$. In the event of ties, we select smaller indices first. Using this, a *Sparse Mixture-of-Experts* (SMoE) is the map $\text{SMoE}: \mathbb{R}^D \to \mathbb{R}^D$ defined by

$$\text{SMoE}\left( x; \left\{ W^{(G,i)}, b^{(G,i)}, W^{(A,i)}, b^{(A,i)}, W^{(B,i)}, b^{(B,i)} \right\}_{i=1}^{n_e} \right)$$

$$= \sum_{i \in T(x)} \text{softmax}_i \left( \left\{ W^{(G,i)} x + b^{(G,i)} \right\}_{i \in T(x)} \right) \cdot \text{E}\left( x; W^{(A,i)}, b^{(A,i)}, W^{(B,i)}, b^{(B,i)} \right), \quad (22)$$

where

$$T(x) = T\left( x; \left\{ W^{(G,i)}, b^{(G,i)} \right\}_{i=1}^{n_e} \right) = \text{Top-}K\left( \left( W^{(G,i)} x + b^{(G,i)} \right)_{i=1}^{n_e} \right). \quad (23)$$

The weight space of SMoE coincides with that of the standard MoE, since the map Top-$K$ does not introduce any new trainable parameters.

**Note on the sparse gating.** The SMoE map is generally not continuous due to the presence of the Top-$K$ operator, particularly in cases where ties occur among the gating scores. To address this, we focus on a subset of $\mathbb{R}^D$ where the top $K$ scores are unambiguously defined. Specifically, for

$$\left\{ W^{(G,i)}, b^{(G,i)} \right\}_{i=1}^{n_e} \in \left( \mathbb{R}^D \times \mathbb{R} \right)^{n_e}, \quad (24)$$

we define

$$\Omega\left( \left\{ W^{(G,i)}, b^{(G,i)} \right\}_{i=1}^{n_e} \right)$$

$$:= \left\{ x \in \mathbb{R}^D : \left( W^{(G,i)} x + b^{(G,i)} \right)_{i=1}^{n_e} \text{ are pairwise distinct} \right\}. \quad (25)$$

We present two results concerning this domain and the behavior of the SMoE map when restricted to it.

**Proposition A.1.** *If $\left\{ W^{(G,i)}, b^{(G,i)} \right\}$ are pairwise distinct for $i = 1, \ldots, n_e$, then $\Omega\left( \left\{ W^{(G,i)}, b^{(G,i)} \right\}_{i=1}^{n_e} \right)$ is an open and dense subset of $\mathbb{R}^D$.*

*Proof.* We have

$$\Omega\left(\left\{W^{(G,i)}, b^{(G,i)}\right\}_{i=1}^{n_e}\right)$$

$$= \left\{x \in \mathbb{R}^D \ : \ W^{(G,i)}x + b^{(G,i)} \text{ is pairwise distinct for all } i = 1, \dots, n_e\right\}$$

$$= \bigcap_{1 \le i,j \le n_e} \left\{x \in \mathbb{R}^D \ : \ W^{(G,i)}x + b^{(G,i)} \ne W^{(G,j)}x + b^{(G,j)}\right\}$$

$$= \bigcap_{1 \le i,j \le n_e} \left(\mathbb{R}^D \setminus \left\{x \in \mathbb{R}^D \ : \ W^{(G,i)}x + b^{(G,i)} = W^{(G,j)}x + b^{(G,j)}\right\}\right). \tag{26}$$

Note that, the set

$$\left\{x \in \mathbb{R}^D \ : \ W^{(G,i)}x + b^{(G,i)} = W^{(G,j)}x + b^{(G,j)}\right\}$$

$$= \left\{x \in \mathbb{R}^D \ : \ \left(W^{(G,i)} - W^{(G,j)}\right)x = b^{(G,j)} - b^{(G,i)}\right\}, \tag{27}$$

is either a hyperplane (when $W^{(G,i)} \ne W^{(G,j)}$) or the empty set (when $(W^{(G,i)} = W^{(G,j)}$ and $b^{(G,j)} \ne b^{(G,i)}$). In both cases, its complement in $\mathbb{R}^D$ is an open and dense subset of $\mathbb{R}^D$. By Equation 26, since the finite intersection of open and dense subsets of $\mathbb{R}^D$ is also open and dense, the set $\Omega\left(\left\{W^{(G,i)}, b^{(G,i)}\right\}_{i=1}^{n_e}\right)$ is open and dense. $\qquad\square$

**Proposition A.2.** *If $\left\{W^{(G,i)}, b^{(G,i)}\right\}$ are pairwise distinct for $i = 1, \dots, n_e$, Then the map* SMoE, *as defined in Equation 22, is continuous on* $\Omega\left(\left\{W^{(G,i)}, b^{(G,i)}\right\}_{i=1}^{n_e}\right)$.

*Proof.* Let $x \in \Omega\left(\left\{W^{(G,i)}, b^{(G,i)}\right\}_{i=1}^{n_e}\right)$. By the definition of this domain, there exists an open neighborhood $U$ of $x$ contained in $\Omega\left(\left\{W^{(G,i)}, b^{(G,i)}\right\}_{i=1}^{n_e}\right)$ such that

$$\text{Top-}K\left(\left(W^{(G,i)}x + b^{(G,i)}\right)_{i=1}^{n_e}\right) = \text{Top-}K\left(\left(W^{(G,i)}y + b^{(G,i)}\right)_{i=1}^{n_e}\right) \tag{28}$$

holds for all $y \in U$. This ensures the sparse gating mechanism in Equation 22 remains fixed within $U$, and thus the SMoE map is continuous on this domain. $\qquad\square$

*Remark* A.3. Propositions A.1 and A.2 will be key components in establishing the proof of Theorem C.5.

### A.3 GROUP ACTION ON WEIGHT SPACES

We define the group $\mathcal{G} = \mathcal{G}(n_e)$ by

$$\mathcal{G}(n_e) = \left(\mathbb{R}^D \times \mathbb{R}\right) \times \mathrm{S}_{n_e}, \tag{29}$$

which is the direct product between the group $\mathbb{R}^D$ with addition, the group $\mathbb{R}$ with addition, and the permutation group $\mathrm{S}_{n_e}$. Each element of $\mathcal{G}(n_e)$ is of the form

$$g = (\gamma_W, \gamma_b, \tau), \quad \text{where} \ \ \gamma_W \in \mathbb{R}^D, \gamma_b \in \mathbb{R}, \text{ and } \tau \in \mathrm{S}_{n_e}. \tag{30}$$

The group $\mathcal{G}(n_e)$ acts on the weight space $\Theta(n_e)$ as follows: For $g \in \mathcal{G}(n_e)$ and $\theta \in \Theta(n_e)$ presented as in Equation 18, define

$$g\theta := \left(\left(W^{(G,\tau(i))} + \gamma_W, b^{(G,\tau(i))} + \gamma_b\right),\right.$$

$$\left.\left(W^{(A,\tau(i))}, b^{(A,\tau(i))}\right), \left(W^{(B,\tau(i))}, b^{(B,\tau(i))}\right)\right)_{i=1,\dots,n_e}. \tag{31}$$

The action of $\mathcal{G}(n_e)$ on the weight space of MoE and SMoE preserves these two maps. This invariance is a consequence of two fundamental properties: the permutation invariance of the summation operator and the translation invariance of the softmax function. We start with a result concerning the invariance of MoE maps under this group action.

**Proposition A.4** (Weight space invariance of MoE). *The MoE map is $\mathcal{G}(n_e)$-invariance under the action of $\mathcal{G}(n_e)$ on their weight space, i.e.,*

$$\text{MoE}(\cdot; \theta) = \text{MoE}(\cdot; g\theta). \tag{32}$$

*Proof.* Given $g = (\gamma_W, \gamma_b, \tau) \in \mathcal{G}(n_e)$. For all $x \in \mathbb{R}^D$, we have

$$\text{MoE}(x; g\theta) = \sum_{i=1}^{n_e} \text{softmax}_i \left( \left\{ \left( W^{(G,\tau(i))} + \gamma_W \right) x + \left( b^{(G,\tau(i))} + \gamma_b \right) \right\}_{i=1}^{n_e} \right)$$
$$\cdot \text{E}\left( x; W^{(A,\tau(i))}, b^{(A,\tau(i))}, W^{(B,\tau(i))}, b^{(B,\tau(i))} \right)$$

$$= \sum_{i=1}^{n_e} \text{softmax}_i \left( \left\{ W^{(G,\tau(i))} x + b^{(G,\tau(i))} \right\}_{i=1}^{n_e} \right)$$
$$\cdot \text{E}\left( x; W^{(A,\tau(i))}, b^{(A,\tau(i))}, W^{(B,\tau(i))}, b^{(B,\tau(i))} \right)$$

$$= \sum_{i=1}^{n_e} \text{softmax}_i \left( \left\{ W^{(G,i)} x + b^{(G,i)} \right\}_{i=1}^{n_e} \right)$$
$$\cdot \text{E}\left( x; W^{(A,i)}, b^{(A,i)}, W^{(B,i)}, b^{(B,i)} \right)$$

$$= \text{MoE}(x; \theta). \tag{33}$$

Thus, the proposition is proven. $\square$

The analysis of the SMoE architecture necessitates additional assumptions, owing to the inherent discontinuity of the Top-$K$ selection operator. We now demonstrate that the SMoE map, when restricted to this region, remains invariant under the group action of $\mathcal{G}(n_e)$.

**Proposition A.5** (Weight space invariance of SMoE). *Given the map SMoE, as defined in Equation 22 Assume that $\left\{ W^{(G,i)}, b^{(G,i)} \right\}$ are pairwise distinct for $i = 1, \ldots, n_e$. Then, the set $\Omega\left( \left\{ W^{(G,i)}, b^{(G,i)} \right\}_{i=1}^{n_e} \right)$ is invariant under the group action of $\mathcal{G}(n_e)$, i.e.,*

$$\Omega\left( \left\{ W^{(G,i)}, b^{(G,i)} \right\}_{i=1}^{n_e} \right) = \Omega\left( g\left\{ W^{(G,i)}, b^{(G,i)} \right\}_{i=1}^{n_e} \right). \tag{34}$$

*Moreover, the SMoE map, restricted to*

$$\Omega\left( \left\{ W^{(G,i)}, b^{(G,i)} \right\}_{i=1}^{n_e} \right), \tag{35}$$

*is $\mathcal{G}(n_e)$-invariance under the action of $\mathcal{G}(n_e)$ on their weight space, i.e.,*

$$\text{SMoE}(\cdot; \theta) = \text{SMoE}(\cdot; g\theta) \quad on \quad \Omega\left( \left\{ W^{(G,i)}, b^{(G,i)} \right\}_{i=1}^{n_e} \right). \tag{36}$$

*Proof.* Given $g = (\gamma_W, \gamma_b, \tau) \in \mathcal{G}(n_e)$. We first verify that the group action of $\mathcal{G}(n_e)$ preserves this set. Indeed:

$$\Omega\left( \left\{ W^{(G,i)}, b^{(G,i)} \right\}_{i=1}^{n_e} \right)$$
$$= \left\{ x \in \mathbb{R}^D : W^{(G,i)} x + b^{(G,i)} \text{ is pairwise distinct for all } i = 1, \ldots, n_e \right\}$$
$$= \left\{ x \in \mathbb{R}^D : \left( W^{(G,\tau(i))} + \gamma_W \right) x + \left( b^{(G,\tau(i))} + \gamma_b \right) \text{ is pairwise distinct for all } i = 1, \ldots, n_e \right\}$$
$$= \Omega\left( g\left\{ W^{(G,i)}, b^{(G,i)} \right\}_{i=1}^{n_e} \right). \tag{37}$$

Now, denote

$$gT(x) = \text{Top-}K\left( \left( \left( W^{(G,\tau(i))} + \gamma_W \right) x + \left( b^{(G,\tau(i))} + \gamma_b \right) \right)_{i=1}^{n_e} \right). \tag{38}$$

For all $x \in \Omega\left( \left\{ W^{(G,i)}, b^{(G,i)} \right\}_{i=1}^{n_e} \right)$, we have $gT(x) = \tau(T(x))$. The proposition now can be proven in the same manner as in Proposition A.4. $\square$

*Remark* A.6. While the group action on the MoE architecture is defined as in Equation 23, it is worth noting that additional symmetries exist within the MoE architecture. For instance, each expert admits internal neuron permutations that preserve the overall network function. However, our primary focus is on the gating mechanism of MoE, and the symmetries internal to each expert are regarded as standard neural network symmetries, which have been extensively studied in prior work.

# B    FUNCTIONAL EQUIVALENCE IN MIXTURE-OF-EXPERTS

In this section, we characterize when two elements of the weight space of MoE define the same MoE map.

## B.1    AN AUXILIARY RESULT RELATED TO HOLOMORPHIC FUNCTIONS ON $\mathbb{C}^n$

A function $f \colon \mathbb{C}^n \to \mathbb{C}$ is called *holomorphic on* $\mathbb{C}^n$ if it is complex differentiable at every point of $\mathbb{C}^n$. A function is called *meromorphic on* $\mathbb{C}^n$ if it can be locally expressed as the quotient of two holomorphic functions, where the denominator is not identically zero. The set of all holomorphic functions on $\mathbb{C}^n$ forms an integral domain, denoted by $\mathcal{D}$, and the set of all meromorphic functions on $\mathbb{C}^n$ forms a field, denoted by $\mathcal{F}$. Note that $\mathcal{F}$ is the field of fractions of the integral domain $\mathcal{D}$. Let $\mathbb{C}[x] = \mathbb{C}[x_1, \ldots, x_n]$ denote the ring of polynomials in $n$ variables with complex coefficients, and let $\mathbb{C}(x) = \mathbb{C}(x_1, \ldots, x_n)$ denote the field of rational functions in $n$ variables with complex coefficients. Then $\mathbb{C}[x] \subset \mathcal{D}$ is an integral domain, and $\mathbb{C}(x) \subset \mathcal{F}$ is a field that is the field of fractions of $\mathbb{C}[x]$.

*Remark* B.1. For $p \in \mathbb{C}[x]$, one has $e^p \in \mathcal{D}$. In other words, the exponential of a polynomial is holomorphic on $\mathbb{C}^n$.

Since $\mathbb{C}(x)$ is a subfield of $\mathcal{F}$, we can regard $\mathcal{F}$ as a vector space over $\mathbb{C}(x)$. The following result concerns the linear independence of exponentials of polynomials within $\mathcal{F}$.

**Lemma B.2.** *Let* $p_1, \ldots, p_N$ *be polynomials in* $\mathbb{C}[x]$ *such that* $p_i - p_j$ *is nonconstant for every* $i \neq j$. *Then the functions* $e^{p_1}, \ldots, e^{p_N}$ *(considered as elements of* $\mathcal{F}$*) are linearly independent over the field* $\mathbb{C}(x)$.

*Proof.* We prove the lemma by induction on $N$. The case $N = 1$ is clear, since $e^p$ is nonzero for any $p \in \mathbb{C}[x]$. Assume that $N \geq 2$ and that the lemma holds for all smaller values of $N$. Let $r_1, \ldots, r_N$ be polynomials in $\mathbb{C}[x]$ such that

$$r_1 \cdot e^{p_1} + \cdots + r_N \cdot e^{p_N} = 0, \tag{39}$$

We aim to show that $r_1 = \cdots = r_N = 0$. Suppose, for contradiction, that this is not the case. Then at least one of the $r_i$ is nonzero. Without loss of generality, assume that $r_N \neq 0$. From Equation 39, it follows that

$$\frac{r_1}{r_N} \cdot e^{r_1 - r_N} + \cdots + \frac{r_{N-1}}{r_N} \cdot e^{r_{N-1} - r_N} + 1 = 0. \tag{40}$$

Differentiating both sides with respect to $x_i$ for each $i = 1, \ldots, n$, we obtain

$$\sum_{j=1}^{N-1} \left( \frac{\partial}{\partial x_i} \left( \frac{r_j}{r_N} \right) + \frac{r_j}{r_N} \cdot \frac{\partial}{\partial x_i} (p_j - p_N) \right) \cdot e^{p_j - p_N} = 0. \tag{41}$$

Observe that

$$\frac{\partial}{\partial x_i} \left( \frac{r_j}{r_N} \right) + \frac{r_j}{r_N} \cdot \frac{\partial}{\partial x_i} (p_j - p_N) \in \mathbb{C}(x). \tag{42}$$

For the $N - 1$ polynomials $p_i - p_N$ in $\mathbb{C}[x]$, where $i = 1, \ldots, N-1$, and the difference $(p_i - p_N) - (p_j - p_N) = p_i - p_j$ is nonconstant for every $i \neq j$. By the induction hypothesis, the functions $e^{p_1 - p_N}, \ldots, e^{p_{N-1} - p_N}$ are linearly independent over $\mathbb{C}(x)$. Therefore, from Equation 41, we conclude that for all $j = 1, \ldots, N-1$ and $i = 1, \ldots, n$,

$$\frac{\partial}{\partial x_i} \left( \frac{r_j}{r_N} \right) + \frac{r_j}{r_N} \cdot \frac{\partial}{\partial x_i} (p_j - p_N) = 0, \tag{43}$$

which implies that

$$\frac{\partial}{\partial x_i} \left( \frac{r_j}{r_N} \cdot e^{p_j - p_N} \right) = 0. \tag{44}$$

Hence, for all $j = 1, \ldots, N - 1$,

$$\frac{r_j}{r_N} \cdot e^{p_j - p_N} = c_j \in \mathbb{C}, \tag{45}$$

is a constant function. If $c_j \neq 0$, then $r_j \neq 0$ and $e^{p_j - p_N} = \frac{c_j r_N}{r_j}$. This holds only if both $e^{p_j - p_N}$ and $\frac{c_j r_N}{r_j}$ are constant functions. In particular, this would imply that $p_j - p_N$ is constant, contradicting the assumption. Therefore, we must have $c_j = 0$. Thus, $r_j = 0$ for all $j = 1, \ldots, N - 1$. However, this contradicts Equation 40. The lemma is therefore proved. $\square$

*Remark* B.3. This result is fundamental and will be invoked multiple times in the proofs of Theorem B.7 and Theorem C.5.

### B.2   LOCAL AFFINENESS OF RELU NEURAL NETWORKS

A polytope is a geometric object defined by flat boundaries, which may be either bounded or unbounded. We define the notion of local affineness as follows.

**Definition B.4** (Local affineness). A function $f : \mathbb{R}^D \to \mathbb{R}^{D'}$ is said to be *locally affine* if there exists a partition of $\mathbb{R}^D$ into a collection of polytopes such that, on each polytope, $f$ coincides with an affine map from $\mathbb{R}^D$ to $\mathbb{R}^{D'}$.

*Remark* B.5. It is worth noting that the term *local affineness* may carry different meanings in other contexts. However, the usage adopted in Definition B.4 is unambiguous within the scope of this work.

We investigate the local affineness property of ReLU neural networks. Consider a neural network $f \colon \mathbb{R}^{n_0} \to \mathbb{R}^{n_L}$ composed of affine transformations and ReLU activations, defined as

$$f = f_L \circ \sigma \circ f_{L-1} \circ \cdots \circ \sigma \circ f_1, \tag{46}$$

where each $f_i \colon \mathbb{R}^{n_{i-1}} \to \mathbb{R}^{n_i}$ is an affine map given by $f_i(x) = W_i x + b_i$, and $\sigma$ is the ReLU activation function applied elementwise. The composition of these affine transformations and ReLU activations partitions the input space $\mathbb{R}^{n_0}$ into a finite number of convex polytopes. Within each polytope, the activation pattern of the ReLU units—i.e., which units are active (passing their input unchanged) and which are inactive (outputting zero)—remains constant. This fixed activation pattern determines a subnetwork where each ReLU acts either as the identity map or as the zero map. Because ReLU is piecewise linear and affine transformations are closed under composition, the entire network behaves as an affine function within each region of fixed activation.

Thus, the network is locally affine:

$$f(x) = A_i x + b_i, \;\; \text{for } x \in P_i, \tag{47}$$

where $P_i$ is a polytope in the partition $\{P_i\}_{i=1}^m$ of the input space, and $A_i, b_i$ define the affine transformation in that region.

*Remark* B.6. Let $\partial P_i$ denote the boundary of the region $P_i$ in the partition $\{P_i\}_{i=1}^m$. Then the set

$$\mathbb{R}^{n_0} \setminus \bigcup_{i=1}^m \partial P_i \tag{48}$$

is clearly open and dense in $\mathbb{R}^{n_0}$. In other words, the union of the interiors of the polytopes $\{P_i\}$ forms a set that is both open and dense.

Now consider a finite collection of ReLU networks $f^{(k)}$, for $k = 1, \ldots, n$. Since the intersection of finitely many open dense sets is again open and dense, there exists a set $\Omega \subset \mathbb{R}^{n_0}$ that is open and dense, such that for every $x \in \Omega$, there exists a neighborhood of $x$ on which all functions $f^{(k)}$ are affine.

### B.3 FUNCTIONAL EQUIVALENCE IN MIXTURE-OF-EXPERTS

The following result establishes the equivalence between two sets of weights that define the same MoE map. Certain assumptions are introduced for technical reasons, and their justification is provided in Remark B.8.

**Theorem B.7** (Functional equivalence in MoE). *Let $\theta \in \Theta(n_e)$ and $\widehat{\theta} \in \Theta(\widehat{n_e})$ be given by*

$$\theta = \left( \left( W^{(G,i)}, b^{(G,i)} \right), \left( W^{(A,i)}, b^{(A,i)} \right), \left( W^{(B,i)}, b^{(B,i)} \right) \right)_{i=1,\ldots,n_e}, \tag{49}$$

$$\widehat{\theta} = \left( \left( \widehat{W}^{(G,i)}, \widehat{b}^{(G,i)} \right), \left( \widehat{W}^{(A,i)}, \widehat{b}^{(A,i)} \right), \left( \widehat{W}^{(B,i)}, \widehat{b}^{(B,i)} \right) \right)_{i=1,\ldots,\widehat{n_e}}, \tag{50}$$

*and suppose they define the same MoE map, i.e.,*

$$\mathrm{MoE}(x; \theta) = \mathrm{MoE}(x; \widehat{\theta}) \text{ for all } x \in \mathbb{R}^D. \tag{51}$$

*If $\theta$ and $\widehat{\theta}$ satisfy the four assumptions:*

1. *$n_e$ experts $\left\{ \mathrm{E}\left(\cdot; W^{(A,i)}, b^{(A,i)}, W^{(B,i)}, b^{(B,i)} \right) \right\}_{i=1}^{n_e}$ are $n_e$ pairwise distinct functions;*

2. *$\widehat{n_e}$ experts $\left\{ \mathrm{E}\left(\cdot; \widehat{W}^{(A,i)}, \widehat{b}^{(A,i)}, \widehat{W}^{(B,i)}, \widehat{b}^{(B,i)} \right) \right\}_{i=1}^{\widehat{n_e}}$ are $\widehat{n_e}$ pairwise distinct functions;*

3. *$W^{(G,i)} - W^{(G,j)}$ are pairwise distinct for all $1 \le i, j \le n_e$ such that $i \ne j$;*

4. *$\widehat{W}^{(G,i)} - \widehat{W}^{(G,j)}$ are pairwise distinct for all $1 \le i, j \le \widehat{n_e}$ such that $i \ne j$;*

*then, $n_e = \widehat{n_e}$, and there exists $\tau \in \mathrm{S}_{n_e}$, $\gamma_W \in \mathbb{R}^D$, $\gamma_b \in \mathbb{R}$ such that for all $i = 1, \ldots, n_e$,*

$$\widehat{W}^{(G,i)} = W^{(G,\tau(i))} + \gamma_W, \quad \widehat{b}^{(G,i)} = b^{(G,\tau(i))} + \gamma_b, \tag{52}$$

*and*

$$\mathrm{E}\left(\cdot; W^{(A,\tau(i))}, b^{(A,\tau(i))}, W^{(B,\tau(i))}, b^{(B,\tau(i))} \right) = \mathrm{E}\left(\cdot; \widehat{W}^{(A,i)}, \widehat{b}^{(A,i)}, \widehat{W}^{(B,i)}, \widehat{b}^{(B,i)} \right), \tag{53}$$

*Proof.* For better readability, we begin by providing a high-level outline of the upcoming proof:

1. Explicitly express the equation $\mathrm{MoE}(\cdot; \theta) = \mathrm{MoE}(\cdot; \widehat{\theta})$ and introduce simplified notation for clarity.

2. Observe that each expert can be locally identified as an affine function.

3. Show that $n_e = \widehat{n_e}$, and establish the existence of the desired permutation $\tau$ and transformation $\gamma_W$.

4. Demonstrate the equality between the two sets of experts.

5. Show that the desired transformation $\gamma_b$ exists.

We now present the derivations and proofs corresponding to each of the five steps.

**Step 1.** Since $\mathrm{MoE}(\cdot; \theta) = \mathrm{MoE}(\cdot; \widehat{\theta})$, we have

$$\sum_{i=1}^{n_e} \mathrm{softmax}_i \left( \left\{ W^{(G,i)}x + b^{(G,i)} \right\}_{i=1}^{n_e} \right) \cdot \mathrm{E}\left( x; W^{(A,i)}, b^{(A,i)}, W^{(B,i)}, b^{(B,i)} \right)$$

$$= \sum_{i=1}^{\widehat{n_e}} \mathrm{softmax}_i \left( \left\{ \widehat{W}^{(G,i)}x + \widehat{b}^{(G,i)} \right\}_{i=1}^{\widehat{n_e}} \right) \cdot \mathrm{E}\left( x; \widehat{W}^{(A,i)}, \widehat{b}^{(A,i)}, \widehat{W}^{(B,i)}, \widehat{b}^{(B,i)} \right), \tag{54}$$

for all $x \in \mathbb{R}^D$. Denote

$$\mathrm{E}_i(\cdot) = \mathrm{E}\left(\cdot; W^{(A,i)}, b^{(A,i)}, W^{(B,i)}, b^{(B,i)}\right),$$

$$\widehat{\mathrm{E}}_i(\cdot) = \mathrm{E}\left(\cdot; \widehat{W}^{(A,i)}, \widehat{b}^{(A,i)}, \widehat{W}^{(B,i)}, \widehat{b}^{(B,i)}\right), \tag{55}$$

and simplify the notation by setting $W^{(G,i)} = W^{(i)}, b^{(G,i)} = b^{(i)}, \widehat{W}^{(G,i)} = \widehat{W}^{(i)}, \widehat{b}^{(G,i)} = \widehat{b}^{(i)}$. Then, by writing out the explicit form of the softmax operator in Equation 54, we have

$$\sum_{i=1}^{n_e} \frac{e^{W^{(i)}x+b^{(i)}}}{\sum_{j=1}^{n_e} e^{W^{(j)}x+b^{(j)}}} \cdot \mathrm{E}_i(x) = \sum_{i=1}^{\widehat{n_e}} \frac{e^{\widehat{W}^{(i)}x+\widehat{b}^{(i)}}}{\sum_{j=1}^{\widehat{n_e}} e^{\widehat{W}^{(j)}x+\widehat{b}^{(j)}}} \cdot \widehat{\mathrm{E}}_i(x). \tag{56}$$

This leads to

$$\left(\sum_{j=1}^{\widehat{n_e}} e^{\widehat{W}^{(j)}x+\widehat{b}^{(j)}}\right) \cdot \left(\sum_{i=1}^{n_e} e^{W^{(i)}x+b^{(i)}} \cdot \mathrm{E}_i(x)\right)$$

$$= \left(\sum_{j=1}^{n_e} e^{W^{(j)}x+b^{(j)}}\right) \cdot \left(\sum_{i=1}^{\widehat{n_e}} e^{\widehat{W}^{(i)}x+\widehat{b}^{(i)}} \cdot \widehat{\mathrm{E}}_i(x)\right), \tag{57}$$

or

$$\sum_{i=1}^{n_e} \sum_{j=1}^{\widehat{n_e}} e^{\left(W^{(i)}+\widehat{W}^{(j)}\right)x+\left(b^{(i)}+\widehat{b}^{(j)}\right)} \cdot \left(\mathrm{E}_i(x) - \widehat{\mathrm{E}}_j(x)\right) = 0. \tag{58}$$

**Step 2.** Since the functions $\mathrm{E}_i$ and $\widehat{\mathrm{E}}_j$ are locally affine, it follows from the observation in Appendix B.2 that there exists an open set $\Omega \subset \mathbb{R}^D$, which is dense in $\mathbb{R}^D$, such that: for every point $a \in \Omega$, there exists an open neighborhood $U \subset \Omega$ of $a$ on which all $\mathrm{E}_i$ and $\widehat{\mathrm{E}}_j$ are affine. In particular, each of these functions coincides with a polynomial on $U$. In other words, there exists a collection of open sets $\{U_k\}_{k \in I}$ covering $\Omega$, i.e.,

$$\Omega = \bigcup_{k \in I} U_k, \tag{59}$$

such that for each $U = U_k$ in the collection, there exist polynomials $p_{U,i}, \hat{p}_{U,j} \in \mathbb{R}[x]$ satisfying

$$\mathrm{E}_i(x) = p_{U,i}(x), \quad \text{and} \quad \widehat{\mathrm{E}}_j(x) = \hat{p}_{U,j}(x) \quad \text{for all } x \in U. \tag{60}$$

From Equation 58, we have:

$$\sum_{i=1}^{n_e} \sum_{j=1}^{\widehat{n_e}} e^{\left(W^{(i)}+\widehat{W}^{(j)}\right)x+\left(b^{(i)}+\widehat{b}^{(j)}\right)} \cdot (p_{U,i}(x) - \hat{p}_{U,j}(x)) = 0 \quad \text{for all } x \in U. \tag{61}$$

Note that the function on the left-hand side of the equation above is holomorphic. By the Identity Theorem for Holomorphic Functions (see Ahlfors (1979); Rudin (1987); Conway (1978); Stein & Shakarchi (2003)), it follows that:

$$\sum_{i=1}^{n_e} \sum_{j=1}^{\widehat{n_e}} e^{\left(W^{(i)}+\widehat{W}^{(j)}\right)x+\left(b^{(i)}+\widehat{b}^{(j)}\right)} \cdot (p_{U,i}(x) - \hat{p}_{U,j}(x)) = 0 \quad \text{for all } x \in \mathbb{C}^D. \tag{62}$$

**Step 3.** From Assumptions 3 and 4, the sets $\{W^{(i)}\}_{i=1}^{n_e}$ and $\{\widehat{W}^{(j)}\}_{j=1}^{\widehat{n_e}}$ consist of pairwise distinct elements. Thus, there exists a direction

$$\alpha \in \mathbb{S}^{D-1} = \{x \in \mathbb{R}^D : \|x\|_2 = 1\}, \tag{63}$$

such that the projections $\{W^{(i)}\alpha\}_{i=1}^{n_e}$ and $\{\widehat{W}^{(j)}\alpha\}_{j=1}^{\widehat{n_e}}$ yield $n_e$ and $\widehat{n_e}$ distinct real numbers, respectively. Without loss of generality, we may reorder the indices so that:

$$W^{(1)}\alpha < W^{(2)}\alpha < \ldots < W^{(n_e)}\alpha \quad \text{and} \quad \widehat{W}^{(1)}\alpha < \widehat{W}^{(2)}\alpha < \ldots < \widehat{W}^{(\widehat{n_e})}\alpha. \tag{64}$$

Moreover, note that the problem, along with all the equations above, remains invariant under the addition of a constant vector to the set $\{\widehat{W}^{(j)}\}_{j=1}^{\widehat{n_e}}$. Therefore, without loss of generality, we may assume that $W^{(1)} = \widehat{W}^{(1)}$. Under this setting, we will show that $n_e = \widehat{n_e}$ and that $W^{(i)} = \widehat{W}^{(i)}$ for all $i = 1, \ldots, n_e$. To this end, we first prove that $W^{(i)} = \widehat{W}^{(i)}$ for all $i = 1, \ldots, \min\{n_e, \widehat{n_e}\}$ by mathematical induction.

*Base case.* By assumption, we have $W^{(1)} = \widehat{W}^{(1)}$, so the base case holds trivially.

*Auxiliary result for the inductive step.* For all pairs $(i, j) \neq (1, 1)$, the following inequality holds:

$$W^{(1)}\alpha + \widehat{W}^{(1)}\alpha < W^{(i)}\alpha + \widehat{W}^{(j)}\alpha. \tag{65}$$

Thus, $W^{(1)} + \widehat{W}^{(1)}$ is distinct from $W^{(i)} + \widehat{W}^{(j)}$ for all $(i, j)$ such that $(i, j) \neq (1, 1)$. From Equation 62 and Lemma B.2, it follows that

$$p_{U,1} = \hat{p}_{U,1}. \tag{66}$$

*Inductive step.* Suppose that $W^{(i)} = \widehat{W}^{(i)}$ holds for all $1 \leq i < n$, where $n$ is an integer satisfying $1 < n \leq \min\{n_e, \widehat{n_e}\}$. Assume, toward a contradiction, that $W^{(n)} \neq \widehat{W}^{(n)}$. We examine the two quantities $W^{(1)} + \widehat{W}^{(n)}$ and $W^{(n)} + \widehat{W}^{(1)}$. Given our assumption, these two expressions must be distinct. Without loss of generality, we may assume that

$$W^{(1)}\alpha + \widehat{W}^{(n)}\alpha \leq W^{(n)}\alpha + \widehat{W}^{(1)}\alpha. \tag{67}$$

- For all $(i, j)$ with $i \geq n$, we have

$$W^{(1)}\alpha + \widehat{W}^{(n)}\alpha \leq W^{(n)}\alpha + \widehat{W}^{(1)}\alpha \leq W^{(i)}\alpha + \widehat{W}^{(j)}\alpha. \tag{68}$$

  Equality holds if and only if $(i, j) = (n, 1)$. Moreover, since $W^{(1)} + \widehat{W}^{(n)}$ and $W^{(n)} + \widehat{W}^{(1)}$ are distinct, it follows that $W^{(1)} + \widehat{W}^{(n)}$ is distinct from $W^{(i)} + \widehat{W}^{(j)}$ for all $(i, j)$ with $i \geq n$.

- For all $(i, j)$ with $j \geq n$, we have

$$W^{(1)}\alpha + \widehat{W}^{(n)}\alpha \leq W^{(i)}\alpha + \widehat{W}^{(j)}\alpha. \tag{69}$$

  Equality holds if and only if $(i, j) = (1, n)$. Therefore, $W^{(1)} + \widehat{W}^{(n)}$ is distinct from $W^{(i)} + \widehat{W}^{(j)}$ for all $(i, j) \neq (1, n)$ with $j \geq n$.

- For all $(i, j)$ such that $i, j < n$, we claim that $W^{(1)} + \widehat{W}^{(n)}$ is distinct from $W^{(i)} + \widehat{W}^{(j)}$. Indeed, suppose for contradiction that

$$W^{(1)} + \widehat{W}^{(n)} = W^{(i)} + \widehat{W}^{(j)} \tag{70}$$

  for some $(i, j)$ with $i, j < n$. Then, by the induction hypothesis, it follows that

$$\widehat{W}^{(1)} + \widehat{W}^{(n)} = \widehat{W}^{(i)} + \widehat{W}^{(j)}. \tag{71}$$

  Rearranging gives

$$\widehat{W}^{(1)} - \widehat{W}^{(j)} = \widehat{W}^{(i)} - \widehat{W}^{(n)}, \tag{72}$$

  which leads to a contradiction, since $(1, j) \neq (i, n)$ and the differences are assumed to be pairwise distinct.

From the observations above, we conclude that $W^{(1)} + \widehat{W}^{(n)}$ is distinct from $W^{(i)} + \widehat{W}^{(j)}$ for all $(i, j) \neq (1, n)$. Combining this with Equation 62 and Lemma B.2, it follows that

$$p_{U,1} = \hat{p}_{U,n}. \tag{73}$$

Moreover, from Equation 66, we also have

$$\hat{p}_{U,1} = \hat{p}_{U,n}. \tag{74}$$

Hence, $\widehat{\mathrm{E}}_1 = \widehat{\mathrm{E}}_n$ on $U$. Since this holds for every open set $U \in \{U_k\}_{k \in I}$, we conclude that $\widehat{\mathrm{E}}_1 = \widehat{\mathrm{E}}_n$ on $\Omega$. Because $\Omega$ is dense in $\mathbb{R}^D$, by continuity, it follows that $\widehat{\mathrm{E}}_1 = \widehat{\mathrm{E}}_n$ on $\mathbb{R}^D$. This contradicts the assumption that the $\widehat{\mathrm{E}}_j$ are pairwise distinct. Therefore, our assumption must be false, and we conclude that $W^{(1)} + \widehat{W}^{(n)} = W^{(n)} + \widehat{W}^{(1)}$, which implies $W^{(n)} = \widehat{W}^{(n)}$.

*Conclusion.* By mathematical induction, we have shown that $W^{(i)} = \widehat{W}^{(i)}$ for all $i = 1, \ldots, \min\{n_e, \widehat{n_e}\}$. It remains to show that $n_e = \widehat{n_e}$. Assume, for contradiction, that $n_e < \widehat{n_e}$. Consider the sum $W^{(1)} + \widehat{W}^{(\widehat{n_e})}$. We claim that this sum is distinct from all $W^{(i)} + \widehat{W}^{(j)}$ for $(i, j) \neq (1, \widehat{n_e})$. Indeed, suppose

$$W^{(1)} + \widehat{W}^{(\widehat{n_e})} = W^{(i)} + \widehat{W}^{(j)} \tag{75}$$

for some $(i, j) \neq (1, \widehat{n_e})$. Then, using the inductive result $W^{(i)} = \widehat{W}^{(i)}$ for $i \leq n_e$, we obtain

$$\widehat{W}^{(1)} + \widehat{W}^{(\widehat{n_e})} = \widehat{W}^{(i)} + \widehat{W}^{(j)}, \tag{76}$$

which implies

$$\widehat{W}^{(1)} - \widehat{W}^{(j)} = \widehat{W}^{(i)} - \widehat{W}^{(\widehat{n_e})}. \tag{77}$$

This contradicts the assumption that all differences $\widehat{W}^{(i)} - \widehat{W}^{(j)}$ are pairwise distinct. Hence, $W^{(1)} + \widehat{W}^{(\widehat{n_e})}$ is distinct from all $W^{(i)} + \widehat{W}^{(j)}$ with $(i, j) \neq (1, \widehat{n_e})$. By Equation 62 and Lemma B.2, this implies

$$p_{U,1} = \hat{p}_{U,\widehat{n_e}}. \tag{78}$$

From Equation 66, we also have

$$\hat{p}_{U,1} = \hat{p}_{U,\widehat{n_e}}. \tag{79}$$

Therefore, $\widehat{\mathrm{E}}_1 = \widehat{\mathrm{E}}_{\widehat{n_e}}$ on $U$. Since this holds for every open set $U \in \{U_k\}_{k \in I}$, we conclude that $\widehat{\mathrm{E}}_1 = \widehat{\mathrm{E}}_{\widehat{n_e}}$ on $\Omega$. As $\Omega$ is dense in $\mathbb{R}^D$, by continuity, it follows that $\widehat{\mathrm{E}}_1 = \widehat{\mathrm{E}}_{\widehat{n_e}}$ on $\mathbb{R}^D$, contradicting the assumption that the experts $\widehat{\mathrm{E}}_j$ are pairwise distinct. Thus, our assumption must be false, and we conclude that $n_e = \widehat{n_e}$. Finally, the reindexing and the translation applied to the set $\{\widehat{W}^{(j)}\}_{j=1}^{\widehat{n_e}}$ throughout the proof establish the existence of a permutation $\tau \in \mathrm{S}_{n_e}$ and a shift vector $\gamma_W \in \mathbb{R}^D$.

**Step 4.** We now prove that $\mathrm{E}_i = \widehat{\mathrm{E}}_i$ on $\mathbb{R}^D$ for all $i = 1, \ldots, n_e$. From **Step 3**, we know that $n_e = \widehat{n_e}$ and $W^{(i)} = \widehat{W}^{(i)}$ for every $i = 1, \ldots, n_e$. Consider any pair $(i, j)$. If $W^{(i)} + \widehat{W}^{(j)} = W^{(i')} + \widehat{W}^{(j')}$, then $(i', j')$ must equal either $(i, j)$ or $(j, i)$. In particular, $W^{(i)} + \widehat{W}^{(i)}$ is distinct from $W^{(j)} + \widehat{W}^{(k)}$ for all $(j, k) \neq (i, i)$. Applying Equation 62 and Lemma B.2, we obtain

$$p_{U,i} = \hat{p}_{U,i}. \tag{80}$$

This mirrors the situation encountered in **Step 3**, and by a similar argument, it follows that $\mathrm{E}_i = \widehat{\mathrm{E}}_i$ on $\mathbb{R}^D$. Since this holds for all $i = 1, \ldots, n_e$, the claim is proven.

**Step 5.** We now show that there exists a constant $\gamma_b \in \mathbb{R}$ such that

$$\widehat{b}_i = b_i + \gamma_b \quad \text{for all } i = 1, \ldots, n_e. \tag{81}$$

Recall from **Step 4** that if $W^{(i)} + \widehat{W}^{(j)} = W^{(i')} + \widehat{W}^{(j')}$, then $(i', j')$ must equal either $(i, j)$ or $(j, i)$. Using this fact, along with Equation 58, Lemma B.2, and the result $\mathrm{E}_i = \widehat{\mathrm{E}}_i$ established in **Step 4**, we obtain the following identity:

$$e^{(W^{(i)} + \widehat{W}^{(j)})x + (b^{(i)} + \widehat{b}^{(j)})} \cdot (\mathrm{E}_i(x) - \mathrm{E}_j(x))$$
$$+ e^{(W^{(j)} + \widehat{W}^{(i)})x + (b^{(j)} + \widehat{b}^{(i)})} \cdot (\mathrm{E}_j(x) - \mathrm{E}_i(x)) = 0, \tag{82}$$

for all pairs $(i, j)$. Since $\mathrm{E}_i \neq \mathrm{E}_j$ for $i \neq j$, there exists some point $x_0 \in \mathbb{R}^D$ such that $\mathrm{E}_i(x_0) \neq \mathrm{E}_j(x_0)$. Substituting $x = x_0$ into Equation 82 and simplifying by canceling all common nonzero factor, we get:

$$e^{b^{(i)} + \widehat{b}^{(j)}} = e^{b^{(j)} + \widehat{b}^{(i)}}, \tag{83}$$

which implies the equality

$$b^{(i)} + \widehat{b}^{(j)} = b^{(j)} + \widehat{b}^{(i)}, \tag{84}$$

or, equivalently,

$$b^{(i)} - \widehat{b}^{(i)} = b^{(j)} - \widehat{b}^{(j)}. \tag{85}$$

This shows that the difference $b^{(i)} - \widehat{b}^{(i)}$ is constant across all $i$. Letting $\gamma_b := \widehat{b}^{(1)} - b^{(1)}$, we conclude that

$$\widehat{b}_i = b_i + \gamma_b \quad \text{for all } i = 1, \dots, n_e. \tag{86}$$

This completes the proof of Theorem B.7. $\qquad\square$

*Remark* B.8 (Rationale behind the assumptions in Theorem B.7). For a model architecture, we require the symmetry group to be intrinsic to the model as a whole, not to hinge on special choices of individual weight vectors. In other words, the group of symmetries should act universally throughout the weight space. Concretely, this leads to the following four conditions in Theorem B.7:

1. $n_e$ experts $\left\{ \mathrm{E}\left( \cdot; W^{(A,i)}, b^{(A,i)}, W^{(B,i)}, b^{(B,i)} \right) \right\}_{i=1}^{n_e}$ are $n_e$ pairwise distinct functions;

2. $\widehat{n_e}$ experts $\left\{ \mathrm{E}\left( \cdot; \widehat{W}^{(A,i)}, \widehat{b}^{(A,i)}, \widehat{W}^{(B,i)}, \widehat{b}^{(B,i)} \right) \right\}_{i=1}^{\widehat{n_e}}$ are $\widehat{n_e}$ pairwise distinct functions;

3. $W^{(G,i)} - W^{(G,j)}$ are pairwise distinct for all $1 \leq i, j \leq n_e$ such that $i \neq j$;

4. $\widehat{W}^{(G,i)} - \widehat{W}^{(G,j)}$ are pairwise distinct for all $1 \leq i, j \leq \widehat{n_e}$ such that $i \neq j$;

We examine the underlying nature of these assumptions.

*Assumption 1 and 2.* If Assumptions 1 and 2 are violated—specifically, when two experts compute the same function and are assigned identical gating scores—the resulting model behavior remains unchanged under permutations of those experts. This introduces additional, non-essential permutations into the symmetry group, which we refer to as spurious symmetries. These symmetries do not reflect fundamental structural invariances but arise only in degenerate parameter configurations—singularities in the space of model parameters.

*Assumption 3 and 4.* Assumptions 3 and 4 address a subtler issue: they exclude cases where linear dependencies among the gating weight vectors might lead to indistinguishable gating behavior across experts. While less immediately obvious than the consequences of violating Assumptions 1 and 2, such dependencies can also enlarge the symmetry group beyond its intended structure. To illustrate this more concretely, we provide the following explicit example. Let $D = D_e = 1$, $n_e = \widehat{n_e} = 3$, and consider parameter settings $\theta, \widehat{\theta}$ such that:

- $W^{(G,1)} = \widehat{W}^{(G,1)} = -1, W^{(G,2)} = \widehat{W}^{(G,2)} = 0, W^{(G,3)} = \widehat{W}^{(G,3)} = 1$,

- $W^{(A,1)}, W^{(A,2)}, W^{(A,3)}, \widehat{W}^{(A,1)}, \widehat{W}^{(A,2)}$, and $\widehat{W}^{(A,3)}$ are arbitrary.

- $b^{(A,1)}, b^{(A,2)}, b^{(A,3)}, \widehat{b}^{(A,1)}, \widehat{b}^{(A,2)}$, and $\widehat{b}^{(A,3)}$ are arbitrary.

- $W^{(B,1)} = W^{(B,2)} = W^{(B,3)} = \widehat{W}^{(B,1)} = \widehat{W}^{(B,2)} = \widehat{W}^{(B,3)} = 0$.

We now choose the bias parameters $b^{(G,i)}, b^{(B,i)}, \widehat{b}^{(G,i)}, \widehat{b}^{(B,i)}$ so that the model outputs satisfy $\mathrm{MoE}(\cdot; \theta) = \mathrm{MoE}(\cdot; \widehat{\theta})$, even though there exists no transformation of the form described in Theorem B.7 that maps $\theta$ to $\widehat{\theta}$. For each $i = 1, 2, 3$, the expert functions reduce to constant outputs:

$$\mathrm{E}\left( x; W^{(A,i)}, b^{(A,i)}, W^{(B,i)}, b^{(B,i)} \right) = b^{(B,i)},$$
$$\mathrm{E}\left( x; \widehat{W}^{(A,i)}, \widehat{b}^{(A,i)}, \widehat{W}^{(B,i)}, \widehat{b}^{(B,i)} \right) = \widehat{b}^{(B,i)}. \tag{87}$$

To simplify notation, we write $b^{(G,i)} = b^{(i)}$ and $\widehat{b}^{(G,i)} = \widehat{b}^{(i)}$. Our goal is to ensure that $\text{MoE}(\cdot; \theta) = \text{MoE}(\cdot; \widehat{\theta})$, which requires that

$$\frac{e^{-x+b^{(1)}}}{e^{-x+b^{(1)}} + e^{b^{(2)}} + e^{x+b^{(3)}}} \cdot b^{(B,1)}$$

$$+ \frac{e^{b^{(2)}}}{e^{-x+b^{(1)}} + e^{b^{(2)}} + e^{x+b^{(3)}}} \cdot b^{(B,2)}$$

$$+ \frac{e^{x+b^{(3)}}}{e^{-x+b^{(1)}} + e^{b^{(2)}} + e^{x+b^{(3)}}} \cdot b^{(B,3)}$$

$$= \frac{e^{-x+\widehat{b}^{(1)}}}{e^{-x+\widehat{b}^{(1)}} + e^{\widehat{b}^{(2)}} + e^{x+\widehat{b}^{(3)}}} \cdot \widehat{b}^{(B,1)}$$

$$+ \frac{e^{\widehat{b}^{(2)}}}{e^{-x+\widehat{b}^{(1)}} + e^{\widehat{b}^{(2)}} + e^{x+\widehat{b}^{(3)}}} \cdot \widehat{b}^{(B,2)}$$

$$+ \frac{e^{x+\widehat{b}^{(3)}}}{e^{-x+\widehat{b}^{(1)}} + e^{\widehat{b}^{(2)}} + e^{x+\widehat{b}^{(3)}}} \cdot \widehat{b}^{(B,3)}. \tag{88}$$

Again, we simplify the notation by setting

$$\begin{aligned}
e^{b^{(G,1)}} &= a_1, & e^{\widehat{b}^{(G,1)}} &= a_2, \\
e^{b^{(G,2)}} &= b_1, & e^{\widehat{b}^{(G,2)}} &= b_2, \\
e^{b^{(G,3)}} &= c_1, & e^{\widehat{b}^{(G,3)}} &= c_2, \\
b^{(B,1)} &= A_1, & \widehat{b}^{(B,1)} &= A_2, \\
b^{(B,2)} &= B_1, & \widehat{b}^{(B,2)} &= B_2, \\
b^{(B,3)} &= C_1, & \widehat{b}^{(B,3)} &= C_2.
\end{aligned} \tag{89}$$

We can now rewrite Equation 88 as

$$\frac{e^{-x}a_1}{e^{-x}a_1 + b_1 + e^x c_1} \cdot A_1 + \frac{b_1}{e^{-x}a_1 + b_1 + e^x c_1} \cdot B_1 + \frac{e^x c_1}{e^{-x}a_1 + b_1 + e^x c_1} \cdot C_1$$

$$= \frac{e^{-x}a_2}{e^{-x}a_2 + b_2 + e^x c_2} \cdot A_2 + \frac{b_2}{e^{-x}a_2 + b_2 + e^x c_2} \cdot B_2 + \frac{e^x c_2}{e^{-x}a_2 + b_2 + e^x c_2} \cdot C_2, \tag{90}$$

which is equivalent to

$$\left(e^{-x}a_1 A_1 + b_1 B_1 + e^x c_1 C_1\right)\left(e^{-x}a_2 + b_2 + e^x c_2\right)$$
$$= \left(e^{-x}a_2 A_2 + b_2 B_2 + e^x c_2 C_2\right)\left(e^{-x}a_1 + b_1 + e^x c_1\right). \tag{91}$$

By matching the coefficients of $e^{-2x}, e^{-x}, 1, e^x, e^{2x}$, we obtain

$$\begin{aligned}
e^{-2x} &: & a_1 a_2 A_1 &= a_1 a_2 A_2, \\
e^{2x} &: & c_1 c_2 C_1 &= c_1 c_2 C_2, \\
e^x &: & b_1 c_2 B_1 + c_1 b_2 C_1 &= b_1 c_2 C_2 + c_1 b_2 B_2, \\
e^{-x} &: & b_1 a_2 B_1 + a_1 b_2 A_1 &= b_1 a_2 A_2 + a_1 b_2 B_2, \\
1 &: & a_1 c_2 A_1 + c_1 a_2 C_1 + b_1 b_2 B_1 &= a_1 c_2 C_2 + c_1 a_2 A_2 + b_1 b_2 B_2.
\end{aligned} \tag{92}$$

By setting $A_1 = A_2 = A$ and $C_1 = C_2 = C$, the equations corresponding to the terms $e^{-2x}$ and $e^{2x}$ are automatically satisfied. Removing these, Equation 92 simplifies to

$$\begin{aligned}
e^x &: & b_1 c_2 B_1 + c_1 b_2 C &= b_1 c_2 C + c_1 b_2 B_2, \\
e^{-x} &: & b_1 a_2 B_1 + a_1 b_2 A &= b_1 a_2 A + a_1 b_2 B_2, \\
1 &: & a_1 c_2 A + c_1 a_2 C + b_1 b_2 B_1 &= a_1 c_2 C + c_1 a_2 A + b_1 b_2 B_2.
\end{aligned} \tag{93}$$

From the equations associated with $e^{-x}$ and $e^x$, and assuming $c_1 b_2 \neq b_1 c_2$ and $a_1 b_2 \neq b_1 a_2$, we obtain

$$
\begin{aligned}
A &= \frac{a_1 b_2 B_2 - b_1 a_2 B_1}{a_1 b_2 - b_1 a_2}, \\
C &= \frac{c_1 b_2 B_2 - b_1 c_2 B_1}{c_1 b_2 - b_1 c_2}.
\end{aligned}
\tag{94}
$$

The equation corresponding to the constant term in Equation 93 can be rewritten as

$$
b_1 b_2 (B_1 - B_2) = (C - A)(a_1 c_2 - c_1 a_2).
\tag{95}
$$

Next, we compute the difference $A - C$ as follows:

$$
\begin{aligned}
A - C &= \frac{a_1 b_2 B_2 - b_1 a_2 B_1}{a_1 b_2 - b_1 a_2} - \frac{c_1 b_2 B_2 - b_1 c_2 B_1}{c_1 b_2 - b_1 c_2} \\
&= \frac{b_1 b_2 (B_1 - B_2)(a_1 c_2 - c_1 a_2)}{(a_1 b_2 - b_1 a_2)(c_1 b_2 - b_1 c_2)}.
\end{aligned}
\tag{96}
$$

Substituting this expression for $(A - C)$ into Equation 95 yields

$$
b_1 b_2 (B_1 - B_2) = -\frac{b_1 b_2 (B_1 - B_2)(a_1 c_2 - c_1 a_2)}{(a_1 b_2 - b_1 a_2)(c_1 b_2 - b_1 c_2)}(a_1 c_2 - c_1 a_2).
\tag{97}
$$

Assuming that $B_1 \neq B_2$ and $b_1 b_2 \neq 0$, we can divide both sides of the equation by $b_1 b_2 (B_1 - B_2)$, which leads to

$$
(a_1 b_2 - b_1 a_2)(b_1 c_2 - c_1 b_2) = (a_1 c_2 - c_1 a_2)^2.
\tag{98}
$$

Although this equation can be solved explicitly, for our purposes it suffices to exhibit a single solution. In this case, we choose

$$
\begin{aligned}
(a_1, a_2) &= (1, 2), \\
(b_1, b_2) &= (3, 5), \\
(c_1, c_2) &= (2, 3).
\end{aligned}
\tag{99}
$$

With this choice, the values of $B_1$ and $B_2$ can be selected arbitrarily. These parameter assignments determine corresponding values for $\theta$ and $\widehat{\theta}$. It is straightforward to verify that no transformation of the form described in Theorem B.7 maps $\theta$ to $\widehat{\theta}$.

## C  FUNCTIONAL EQUIVALENCE IN SPARSE MIXTURE-OF-EXPERTS

In this section, we characterize when two elements of the weight space of SMoE define the same SMoE map.

### C.1  AUXILIARY RESULTS

The following definition formalizes the notion of the strongly distinct property, which is later be used in Theorem C.5.

**Definition C.1** (Strongly distinct). Two functions $f$ and $g$ from $X$ to $Y$ are called *strongly distinct* if $\{x \in X : f(x) \neq g(x)\}$ is a dense subset of $X$.

*Example* C.2. Two distinct polynomials on $\mathbb{R}^n$ or $\mathbb{C}^n$ are strongly distinct. Two distinct holomorphic functions are strongly distinct. Two distinct locally affine functions are not strongly distinct in general. Indeed:

- Consider $f_1, f_2 \colon \mathbb{R} \to \mathbb{R}$ as follows:

$$
f_1(x) = \begin{cases} 0 & \text{if } x < 0, \\ x & \text{if } x \geq 0, \end{cases} \qquad f_2(x) = 1.
\tag{100}
$$

Then $f_1$ and $f_2$ are strongly distinct.

- Consider $g_1, g_2 \colon \mathbb{R} \to \mathbb{R}$ as follows:

$$g_1(x) = \begin{cases} 0 & \text{if } x < 0, \\ x & \text{if } x \geq 0, \end{cases} \qquad g_2(x) = 0. \tag{101}$$

Then $g_1$ and $g_2$ are distinct but not strongly distinct.

We define a class of subsets of $\mathbb{R}^D$ as follows: for

$$\left\{ W^{(G,i)}, b^{(G,i)} \right\}_{i=1}^{n_e} \in \left( \mathbb{R}^D \times \mathbb{R} \right)^{n_e}, \tag{102}$$

define

$$\Omega\left( \left\{ W^{(G,i)}, b^{(G,i)} \right\}_{i=1}^{n_e} \right)$$
$$:= \left\{ x \in \mathbb{R}^D \colon W^{(G,i)}x + b^{(G,i)} \text{ is pairwise distinct for all } i = 1, \dots, n_e \right\} \tag{103}$$

The following result establishes a sufficient condition on the gating parameters under which the Top-$K$ operator is capable of selecting every possible subset of $K$ experts from the full set of experts.

**Proposition C.3.** *Assume that $\{W^{(G,i)}\}_{i=1}^{n_e}$ satisfies $\{W^{(G,i-1)} - W^{(G,i)}\}_{i=2}^{n_e}$ is a linear independent subset of $\mathbb{R}^D$. Then, for all subsets $A$ of $K$ elements of $\{1, \dots, n_e\}$, there exists $x \in \Omega\left( \{W^{(G,i)}, b^{(G,i)}\}_{i=1}^{n_e} \right)$ such that:*

$$\text{Top-}K\left( \left( W^{(G,i)}x + b^{(G,i)} \right)_{i=1}^{n_e} \right) = A. \tag{104}$$

*Proof.* Without loss of generality, assume that $A = \{1, \dots, K\}$. To show that there exists $x \in \Omega\left( \{W^{(G,i)}, b^{(G,i)}\}_{i=1}^{n_e} \right)$ such that:

$$\text{Top-}K\left( \left( W^{(G,i)}x + b^{(G,i)} \right)_{i=1}^{n_e} \right) = \{1, \dots, K\}, \tag{105}$$

it is enough to show that there exists $x \in \mathbb{R}^D$ such that

$$W^{(G,1)}x + b^{(G,1)} > W^{(G,2)}x + b^{(G,2)} > \dots > W^{(G,n_e)}x + b^{(G,n_e)}. \tag{106}$$

We simplify it even more, we find $x \in \mathbb{R}^D$ such that

$$\begin{aligned} \left( W^{(G,1)}x + b^{(G,1)} \right) - \left( W^{(G,2)}x + b^{(G,2)} \right) &= 1, \\ \left( W^{(G,2)}x + b^{(G,2)} \right) - \left( W^{(G,3)}x + b^{(G,3)} \right) &= 1, \\ &\dots \\ \left( W^{(G,n_e-1)}x + b^{(G,n_e-1)} \right) - \left( W^{(G,n_e)}x + b^{(G,n_e)} \right) &= 1. \end{aligned} \tag{107}$$

This is equivalent to

$$\begin{aligned} \left( W^{(G,1)} - W^{(G,2)} \right) x &= 1 - \left( b^{(G,1)} - b^{(G,2)} \right), \\ \left( W^{(G,2)} - W^{(G,3)} \right) x &= 1 - \left( b^{(G,2)} - b^{(G,3)} \right), \\ &\dots \\ \left( W^{(G,n_e-1)} - W^{(G,n_e)} \right) x &= 1 - \left( b^{(G,n_e-1)} - b^{(G,n_e)} \right). \end{aligned} \tag{108}$$

Since the set $\{W^{(G,i-1)} - W^{(G,i)}\}_{i=2}^{n_e}$ is linear independent, there exists $x \in \mathbb{R}^D$ satisfies Equation 108. $\square$

*Remark* C.4. Proposition C.5 will be used in Theorem C.5. A justification of the linear independence assumption is provided in Remark C.9.

## C.2 FUNCTIONAL EQUIVALENCE IN SPARSE MIXTURE-OF-EXPERTS

We present a functional equivalence result for the SMoE architecture, analogous to the one established for MoE in Theorem B.7. However, our result is restricted to the case $K > 1$, as the setting $K = 1$ introduces singularities that invalidate the general equivalence structure. A detailed justification for the exclusion of the $K = 1$ case is provided in Remark C.10.

**Theorem C.5** (Functional equivalence in SMoE). *Let $\theta \in \Theta(n_e)$ and $\widehat{\theta} \in \Theta(\widehat{n_e})$ be given by*

$$\theta = \left( \left( W^{(G,i)}, b^{(G,i)} \right), \left( W^{(A,i)}, b^{(A,i)} \right), \left( W^{(B,i)}, b^{(B,i)} \right) \right)_{i=1,\ldots,n_e}, \tag{109}$$

$$\widehat{\theta} = \left( \left( \widehat{W}^{(G,i)}, \widehat{b}^{(G,i)} \right), \left( \widehat{W}^{(A,i)}, \widehat{b}^{(A,i)} \right), \left( \widehat{W}^{(B,i)}, \widehat{b}^{(B,i)} \right) \right)_{i=1,\ldots,\widehat{n_e}}, \tag{110}$$

*and suppose they define the same SMoE map, i.e.,*

$$\mathrm{SMoE}(x;\theta) = \mathrm{SMoE}(x;\widehat{\theta}) \text{ for all } x \in \mathbb{R}^D. \tag{111}$$

*Denote the two corresponding gating maps as follows*

$$T(x) = T\left( x; \left\{ W^{(G,i)}, b^{(G,i)} \right\}_{i=1}^{n_e} \right) = \text{Top-}K\left( \left( W^{(G,i)}x + b^{(G,i)} \right)_{i=1}^{n_e} \right), \tag{112}$$

$$\widehat{T}(x) = \widehat{T}\left( x; \left\{ \widehat{W}^{(G,i)}, \widehat{b}^{(G,i)} \right\}_{i=1}^{\widehat{n_e}} \right) = \text{Top-}K\left( \left( \widehat{W}^{(G,i)}x + \widehat{b}^{(G,i)} \right)_{i=1}^{\widehat{n_e}} \right). \tag{113}$$

*If $\theta$ and $\widehat{\theta}$ satisfy the four assumptions:*

1. *$n_e$ experts $\left\{ \mathrm{E}\left( \cdot; W^{(A,i)}, b^{(A,i)}, W^{(B,i)}, b^{(B,i)} \right) \right\}_{i=1}^{n_e}$ are $n_e$ pairwise strongly distinct functions;*

2. *$\widehat{n_e}$ experts $\left\{ \mathrm{E}\left( \cdot; \widehat{W}^{(A,i)}, \widehat{b}^{(A,i)}, \widehat{W}^{(B,i)}, \widehat{b}^{(B,i)} \right) \right\}_{i=1}^{\widehat{n_e}}$ are $\widehat{n_e}$ pairwise strongly distinct functions;*

3. *$\{ W^{(G,i-1)} - W^{(G,i)} \}_{i=2}^{n_e}$ is a linear independent subset of $\mathbb{R}^D$;*

4. *$\{ \widehat{W}^{(G,i-1)} - \widehat{W}^{(G,i)} \}_{i=2}^{\widehat{n_e}}$ is a linear independent subset of $\mathbb{R}^D$;*

*then, $n_e = \widehat{n_e}$, and there exists $\tau \in \mathrm{S}_{n_e}$, $\gamma_W \in \mathbb{R}^D$, $\gamma_b \in \mathbb{R}$ such that for all $i = 1, \ldots, n_e$,*

$$\widehat{W}^{(G,i)} = W^{(G,\tau(i))} + \gamma_W, \quad \widehat{b}^{(G,i)} = b^{(G,\tau(i))} + \gamma_b, \tag{114}$$

*and*

$$\mathrm{E}\left( x; W^{(A,\tau(i))}, b^{(A,\tau(i))}, W^{(B,\tau(i))}, b^{(B,\tau(i))} \right) = \mathrm{E}\left( x; \widehat{W}^{(A,i)}, \widehat{b}^{(A,i)}, \widehat{W}^{(B,i)}, \widehat{b}^{(B,i)} \right), \tag{115}$$

*for all $x \in \Omega\left( \left\{ W^{(G,i)}, b^{(G,i)} \right\}_{i=1}^{n_e} \right)$ such that $\tau(i) \in T(x)$.*

Before we proceed to the proof of Theorem C.5, we first make two remarks.

*Remark C.6.* Note that, if $n_e = \widehat{n_e}$, and there exists $\tau \in \mathrm{S}_{n_e}$, $\gamma_W \in \mathbb{R}^D$, $\gamma_b \in \mathbb{R}$ such that for all $i = 1, \ldots, n_e$,

$$\widehat{W}^{(G,i)} = W^{(G,\tau(i))} + \gamma_W, \quad \widehat{b}^{(G,i)} = b^{(G,\tau(i))} + \gamma_b, \tag{116}$$

then the two sets $\Omega\left( \left\{ W^{(G,i)}, b^{(G,i)} \right\}_{i=1}^{n_e} \right)$ and $\Omega\left( \left\{ \widehat{W}^{(G,i)}, \widehat{b}^{(G,i)} \right\}_{i=1}^{n_e} \right)$ are equal. Moreover, for any $x$ in this set, it holds that $\tau(i) \in T(x)$ if and only if $i \in \widehat{T}(x)$.

*Remark C.7.* It is straightforward to verify that Assumptions 3 and 4 in Theorem C.5 imply Assumptions 3 and 4 in Theorem B.7.

*Proof.* For better readability, we begin by providing a high-level outline of the upcoming proof:

1. Explicitly express the equation $\text{SMoE}(\cdot; \theta) = \text{SMoE}(\cdot; \widehat{\theta})$ and introduce simplified notation for clarity.

2. Define a partition of the space into regions where the Top-$K$ map selects the same indices, and where each expert is affine.

3. Prove that the desired property holds for a fixed number of experts. The key idea is to apply the result for MoE in Theorem B.7.

4. Extend the result to show that the desired property holds for all experts.

We now present the derivations and proofs corresponding to each of the four steps.

**Step 1.** Since $\text{SMoE}(\cdot; \theta) = \text{SMoE}(\cdot; \widehat{\theta})$, we have

$$\sum_{i \in T(x)} \text{softmax}_i \left( \left\{ W^{(G,i)}x + b^{(G,i)} \right\}_{i \in T(x)} \right) \cdot \text{E}\left( x; W^{(A,i)}, b^{(A,i)}, W^{(B,i)}, b^{(B,i)} \right)$$

$$= \sum_{i \in \widehat{T}(x)} \text{softmax}_i \left( \left\{ \widehat{W}^{(G,i)}x + \widehat{b}^{(G,i)} \right\}_{i \in \widehat{T}(x)} \right) \cdot \text{E}\left( x; \widehat{W}^{(A,i)}, \widehat{b}^{(A,i)}, \widehat{W}^{(B,i)}, \widehat{b}^{(B,i)} \right), \quad (117)$$

for all $x \in \mathbb{R}^D$. Denote

$$\text{E}_i(\cdot) = \text{E}\left( \cdot; W^{(A,i)}, b^{(A,i)}, W^{(B,i)}, b^{(B,i)} \right),$$

$$\widehat{\text{E}}_i(\cdot) = \text{E}\left( \cdot; \widehat{W}^{(A,i)}, \widehat{b}^{(A,i)}, \widehat{W}^{(B,i)}, \widehat{b}^{(B,i)} \right), \quad (118)$$

and simplify the notation by setting $W^{(G,i)} = W^{(i)}, b^{(G,i)} = b^{(i)}, \widehat{W}^{(G,i)} = \widehat{W}^{(i)}, \widehat{b}^{(G,i)} = \widehat{b}^{(i)}$. We rewrite Equation 117 as follows:

$$\sum_{i \in T(x)} \text{softmax}_i \left( \left\{ W^{(i)}x + b^{(i)} \right\}_{i \in T(x)} \right) \cdot \text{E}_i(x)$$

$$= \sum_{i \in \widehat{T}(x)} \text{softmax}_i \left( \left\{ \widehat{W}^{(i)}x + \widehat{b}^{(i)} \right\}_{i \in \widehat{T}(x)} \right) \cdot \widehat{\text{E}}_i(x). \quad (119)$$

**Step 2.** We make two key observations:

- Assumptions 3 and 4 ensure that the parameter pairs $\left\{ W^{(i)}, b^{(i)} \right\}$ are pairwise distinct for $i = 1, \ldots, n_e$, and similarly, $\left\{ \widehat{W}^{(i)}, \widehat{b}^{(i)} \right\}$ are pairwise distinct for $i = 1, \ldots, \widehat{n_e}$. By Proposition A.1, the set

$$\Omega_1 = \Omega\left( \left\{ W^{(G,i)}, b^{(G,i)} \right\}_{i=1}^{n_e} \right) \cap \Omega\left( \left\{ \widehat{W}^{(G,i)}, \widehat{b}^{(G,i)} \right\}_{i=1}^{\widehat{n_e}} \right), \quad (120)$$

  is an open and dense subset of $\mathbb{R}^D$, such that for all $x \in \Omega_1$, the values $W^{(i)}x + b^{(i)}$ are pairwise distinct for $i = 1, \ldots, n_e$, and $\widehat{W}^{(i)}x + \widehat{b}^{(i)}$ are pairwise distinct for $i = 1, \ldots, \widehat{n_e}$. By construction, for every $x \in \Omega_1$, there exists a neighborhood of $x$ in $\Omega_1$ on which the functions $T(\cdot)$ and $\widehat{T}(\cdot)$ remain constant.

- From the analysis in Appendix B.2, there exists a set $\Omega_2 \subset \mathbb{R}^D$ that is open and dense, such that for every $x \in \Omega_2$, there exists a neighborhood of $x$ in $\Omega_2$ on which all expert functions $\text{E}_i$ and $\widehat{\text{E}}_j$ are affine.

By taking the intersection $\Omega = \Omega_1 \cap \Omega_2$, we obtain a set $\Omega$ that is also open and dense. Moreover, since $T(\cdot)$ and $\widehat{T}(\cdot)$ remain constant, $\text{E}_i$ and $\widehat{\text{E}}_i$ are affine in small neighborhoods around each point in $\Omega$, there exists a collection of open sets $\{U_k\}_{k \in I}$ covering $\Omega$, i.e.,

$$\Omega = \bigcup_{k \in I} U_k, \quad (121)$$

such that within each set $U_k$ in the collection, the expert functions $\mathrm{E}_i$ and $\widehat{\mathrm{E}}_j$ are affine, and the selection functions $T(\cdot)$ and $\widehat{T}(\cdot)$ are constant.

**Step 3.** Consider an arbitrary set $U$ from the cover in Equation 121. Without loss of generality, we may reindex so that $T(\cdot) = \widehat{T}(\cdot) = \{1, \ldots, K\}$ on $U$. Under this reindexing, Equation 119 simplifies to

$$\sum_{i=1}^{K} \mathrm{softmax}_i \left( \left\{ W^{(i)} x + b^{(i)} \right\}_{i=1}^{K} \right) \cdot \mathrm{E}_i(x)$$

$$= \sum_{i=1}^{K} \mathrm{softmax}_i \left( \left\{ \widehat{W}^{(i)} x + \widehat{b}^{(i)} \right\}_{i=1}^{K} \right) \cdot \widehat{\mathrm{E}}_i(x) \quad \text{for all } x \in U. \quad (122)$$

By Assumption 1, the expert functions $\mathrm{E}_i$ are strongly distinct, which implies they remain distinct over the open set $U$. The same conclusion applies to the $\widehat{\mathrm{E}}_i$ by Assumption 2. Therefore, the first four assumptions of Theorem C.5, together with Equation 122, reduce the setting to that of Theorem B.7. As a result, up to a reindexing of the experts, there exist constants $\gamma_W \in \mathbb{R}^D$ and $\gamma_b \in \mathbb{R}$ such that for all $i = 1, \ldots, K$,

$$\widehat{W}^{(i)} = W^{(i)} + \gamma_W, \quad \widehat{b}^{(i)} = b^{(i)} + \gamma_b, \quad (123)$$

and $\mathrm{E}_i = \widehat{\mathrm{E}}_i$ on $U$.

**Step 4.** Now, for any $k = 3, 4, \ldots, n_e$, we apply Proposition C.3 to choose a set $V_1$ from the cover in Equation 121 such that both indices 1 and $k$ are included in $T(V_1)$. Considering Equation 119 restricted to $V_1$ and applying Theorem C.5, we conclude that there exist indices $1 \leq t_1, s_1 \leq \widehat{n_e}$ such that

$$W_1 - W_k = \widehat{W}_{t_1} - \widehat{W}_{s_1}. \quad (124)$$

Applying the same reasoning for indices 2 and $k$, we find $1 \leq t_2, s_2 \leq \widehat{n_e}$ satisfying

$$W_2 - W_k = \widehat{W}_{t_2} - \widehat{W}_{s_2}. \quad (125)$$

Subtracting Equations 125 from 124, we obtain

$$\widehat{W}_1 - \widehat{W}_2 = W_1 - W_2 = (W_1 - W_k) - (W_2 - W_k) = (\widehat{W}_{t_1} - \widehat{W}_{s_1}) - (\widehat{W}_{t_2} - \widehat{W}_{s_2}). \quad (126)$$

By Assumption 4, which guarantees linear independence, it follows that $t_1 = 1$, $t_2 = 2$, and $s_1 = s_2$. Let us denote this common index as $\tau(k)$, i.e., $\tau(k) = s_1 = s_2$. Then, we have

$$W_1 - W_k = \widehat{W}_1 - \widehat{W}_{\tau(k)}, \quad (127)$$

which is equivalent to

$$\widehat{W}_{\tau(k)} - W_k = \widehat{W}_1 - W_1 = \gamma_W. \quad (128)$$

We also have

$$\widehat{b}_{\tau(k)} - b_k = \widehat{b}_1 - b_1 = \gamma_b. \quad (129)$$

Finally, since $k$ ranges over $\{3, 4, \ldots, n_e\}$, the values $\tau(k)$ must be distinct. Indeed, suppose there exist $k \neq k'$ such that $\tau(k) = \tau(k')$. Then it would follow that

$$W_k - W_{k'} = \widehat{W}_{\tau(k)} - \widehat{W}_{\tau(k')} = 0, \quad (130)$$

which contradicts Assumption 3. By applying a symmetric argument to the parameters of $\widehat{\mathrm{SMoE}}$, we conclude that $n_e = \widehat{n_e}$. Furthermore, up to a suitable permutation $\tau$ of the indices, we have:

$$\widehat{W}^{(G,i)} = W^{(G,\tau(i))} + \gamma_W, \quad \widehat{b}^{(G,i)} = b^{(G,\tau(i))} + \gamma_b. \quad (131)$$

Additionally, the above analysis implies the following: for any $x \in \Omega \left( \left\{ W^{(G,i)}, b^{(G,i)} \right\}_{i=1}^{n_e} \right)$ such that $\tau(i) \in T(x)$—that is, index $i$ is selected by the Top-$K$ mechanism in SMoE—we have

$$\mathrm{E}_i(x) = \widehat{\mathrm{E}}_i(x). \quad (132)$$

This completes the proof of Theorem C.5. $\qquad \square$

*Remark* C.8. Although Theorem C.5 is conceptually aligned with Theorem B.7, it is important to emphasize that the case of SMoE is significantly more challenging to establish. The primary source of this difficulty lies in the presence of the Top-$K$ operator, which introduces discontinuities by altering the set of contributing experts in a nontrivial and input-dependent manner. This behavior is notably difficult to analyze and control within the theoretical framework.

*Remark* C.9 (Rationale behind the assumptions in Theorem C.5). We begin by recalling the four assumptions stated in Theorem C.5:

1. $n_e$ experts $\left\{ \mathrm{E}\left( \cdot; W^{(A,i)}, b^{(A,i)}, W^{(B,i)}, b^{(B,i)} \right) \right\}_{i=1}^{n_e}$ are $n_e$ pairwise strongly distinct functions;

2. $\widehat{n_e}$ experts $\left\{ \mathrm{E}\left( \cdot; \widehat{W}^{(A,i)}, \widehat{b}^{(A,i)}, \widehat{W}^{(B,i)}, \widehat{b}^{(B,i)} \right) \right\}_{i=1}^{\widehat{n_e}}$ are $\widehat{n_e}$ pairwise strongly distinct functions;

3. $\{W^{(G,i-1)} - W^{(G,i)}\}_{i=2}^{n_e}$ is a linear independent subset of $\mathbb{R}^D$;

4. $\{\widehat{W}^{(G,i-1)} - \widehat{W}^{(G,i)}\}_{i=2}^{\widehat{n_e}}$ is a linear independent subset of $\mathbb{R}^D$;

The set of assumptions in Theorem C.5 is strictly stronger than that of Theorem B.7. We analyze them as follows.

*Assumptions 1 and 2.* Assumptions 1 and 2 primarily arise due to the use of the Top-$K$ operator, which induces input-dependent expert selection. As a result, an expert's behavior is unconstrained in regions where it is not selected by the gating mechanism, allowing it to behave arbitrarily in those domains. Therefore, if we only assume that the experts are pairwise distinct—rather than pairwise strongly distinct—it is possible for different sets of expert functions, when restricted to their respective activated regions, to yield the same overall function. This ambiguity underscores the necessity of strong distinctness to ensure identifiability in the SMoE architecture.

*Assumptions 3 and 4.* In practical scenarios, the number of experts $n_e$ is typically much smaller than the token dimension $D$. Consequently, the sets $\{W^{(G,i-1)} - W^{(G,i)}\}_{i=2}^{n_e}$ and $\{\widehat{W}^{(G,i-1)} - \widehat{W}^{(G,i)}\}_{i=2}^{\widehat{n_e}}$ are generally linearly independent. However, when this condition fails, certain pairs of experts may never be selected simultaneously by the gating mechanism for any input. This limitation gives rise to singular symmetries, wherein different parameter configurations result in identical functional outputs, yet cannot be transformed into one another via the equivalence described in Theorem C.5.

To elucidate the implications of this behavior, we present a concrete example illustrating how such symmetries can manifest within the SMoE architecture. Consider the case with $n_e = 4$ and $K = 2$, and let $E_1, E_2, E_3, E_4$ be arbitrary experts. Define two MoE functions $f_1$ and $f_2$ with gating logits given by $(-2x, -x, x, 2x)$ and $(-3x, -2x, 2x, 3x)$, respectively. The explicit forms of $f_1$ and $f_2$ are:

$$f_1(x) = \begin{cases} \mathrm{softmax}_1(-2x, -x) \cdot E_1(x) + \mathrm{softmax}_2(-2x, -x) \cdot E_2(x) & \text{if } x < 0, \\ \mathrm{softmax}_1(x, 2x) \cdot E_3(x) + \mathrm{softmax}_2(x, 2x) \cdot E_4(x) & \text{if } x > 0, \end{cases} \quad (133)$$

and,

$$f_2(x) = \begin{cases} \mathrm{softmax}_1(-3x, -2x) \cdot E_1(x) + \mathrm{softmax}_2(-3x, -2x) \cdot E_2(x) & \text{if } x < 0, \\ \mathrm{softmax}_1(2x, 3x) \cdot E_3(x) + \mathrm{softmax}_2(2x, 3x) \cdot E_4(x) & \text{if } x > 0. \end{cases} \quad (134)$$

It is evident that $f_1(x) = f_2(x)$ for all $x \in \mathbb{R} \setminus 0$, where the gating scores are pairwise distinct and the Top-$K$ selection is stable. However, there exists no transformation of the form described in Theorem C.5 that maps one function to the other, highlighting the presence of singular symmetries in the SMoE architecture for some sets of parameters.

*Remark* C.10 (The case of $K = 1$). In the special case where $K = 1$, the SMoE function from Equation 22 simplifies as follows:

$$\mathrm{SMoE}\left( x; \left\{ W^{(G,i)}, b^{(G,i)}, W^{(A,i)}, b^{(A,i)}, W^{(B,i)}, b^{(B,i)} \right\}_{i=1}^{n_e} \right)$$
$$= \mathrm{E}\left( x; W^{(A,i)}, b^{(A,i)}, W^{(B,i)}, b^{(B,i)} \right), \quad (135)$$

where the index $i$ is given by

$$i = \operatorname*{argmax}_{i=1,\ldots,n_e} \left( W^{(G,i)} x + b^{(G,i)} \right). \tag{136}$$

Here, the Top-1 routing mechanism selects only the expert with the highest gating score, resulting in a softmax distribution that collapses to a single entry equal to 1. In addition to the group $\mathcal{G}(n_e)$ acting on the expert parameters, the SMoE mapping with $K = 1$ also admits a nontrivial and nonsingular symmetry under the action of the multiplicative group $\mathbb{R}_{>0}$. Specifically, for any $a > 0$, we have:

$$\mathrm{SMoE}\left( x; \left\{ W^{(G,i)}, b^{(G,i)}, W^{(A,i)}, b^{(A,i)}, W^{(B,i)}, b^{(B,i)} \right\}_{i=1}^{n_e} \right)$$
$$= \mathrm{SMoE}\left( x; \left\{ aW^{(G,i)}, ab^{(G,i)}, W^{(A,i)}, b^{(A,i)}, W^{(B,i)}, b^{(B,i)} \right\}_{i=1}^{n_e} \right). \tag{137}$$

This invariance holds because the argmax used for expert selection is unaffected by uniform positive scaling:

$$\operatorname*{argmax}_{i=1,\ldots,n_e} \left( W^{(G,i)} x + b^{(G,i)} \right) = \operatorname*{argmax}_{i=1,\ldots,n_e} \left( aW^{(G,i)} x + ab^{(G,i)} \right), \tag{138}$$

for all $x \in \Omega\left( \{ W^{(G,i)}, b^{(G,i)} \}_{i=1}^{n_e} \right)$. Moreover, since only one expert is activated per input, no explicit interactions are formed among the expert components. This leads to a rich set of hidden symmetries within the architecture. Due to the complexity introduced by these symmetries, we choose to exclude the case $K = 1$ from our main analysis and leave its exploration to future work.

# D   WEIGHT SPACES OF MOE TRANSFORMER AND ITS GROUP ACTION

Since the weight space, symmetry, and group action are the same for both MoE and SMoE, we will describe the equivariant metanetwork for the MoE Transformer in this section. The construction for the SMoE Transformer is identical.

An MoE Transformer layer comprises a multihead attention module followed by an MoE module, where each expert in the MoE module is realized as a single hidden-layer network. Formally, an MoE Tranformer layer, which will be denoted by $\mathrm{MoETransformer}$, transforms an input sequence $X \in \mathbb{R}^{L \times D}$ to an output sequence $\mathrm{MoETransformer}(X) \in \mathbb{R}^{L \times D}$ defined as follows:

$$\mathrm{MoETransformer}(X) = \mathrm{LayerNorm}\left( \mathrm{MoE}\left( \hat{X}; \left\{ [W]^{(G,i)}, [b]^{(G,i)}, [W]^{(A,i)}, [b]^{(A,i)}, [W]^{(B,i)}, [b]^{(B,i)} \right\}_{i=1}^{n_e} \right) \right),$$
$$\hat{X} = \mathrm{LayerNorm}\left( \mathrm{MultiHead}\left( X; \{ [W]^{(Q,i)}, [W]^{(K,i)}, [W]^{(V,i)}, [W]^{(O,i)} \}_{i=1}^{n_h} \right) \right),$$

where the $\mathrm{MoE}$ operator is a token-wise operator and is defined in Equation 2. While the $\mathrm{MultiHead}$ is defined in (Tran et al., 2025) as

$$\mathrm{MultiHead}\left( X; W^{(O)}, \left\{ W^{(Q,i)}, W^{(K,i)}, W^{(V,i)} \right\}_{i=1}^{h} \right)$$
$$= \left( \bigoplus_{i=1}^{h} \mathrm{Head}\left( X; W^{(Q,i)}, W^{(K,i)}, W^{(V,i)} \right) \right) W^{(O)}$$
$$= \sum_{i=1}^{h} \mathrm{Head}\left( X; W^{(Q,i)}, W^{(K,i)}, W^{(V,i)} \right) W^{(O,i)}$$
$$= \sum_{i=1}^{h} \mathrm{softmax}\left( X \cdot \left( \frac{W^{(Q,i)} \cdot \left( W^{(K,i)} \right)^{\top}}{\sqrt{D_k}} \right) \cdot X^{\top} \right) \cdot X \cdot \left( W^{(V,i)} \cdot W^{(O,i)} \right),$$

where $W^{(O)} = \left( W^{(O,1)}, \ldots, W^{(O,h)} \right)$ with each $W^{(O,i)} \in \mathbb{R}^{D_v \times D}$. The positive integers $n_h$ and $n_e$ represent the number of heads in the multihead attention module and the number of experts in the MoE module, respectively.

Accordingly, the *weight space* $\mathcal{U}$ of an MoE Transformer layer with $n_e$ experts is defined as the vector space:

$$\mathcal{U} = \left( \mathbb{R}^{D \times D_k} \times \mathbb{R}^{D \times D_k} \times \mathbb{R}^{D \times D_v} \times \mathbb{R}^{D_v \times D} \right)^{n_h}$$

$$\times \left( \left( \mathbb{R}^D \times \mathbb{R} \right) \times \left( \mathbb{R}^{D \times D_e} \times \mathbb{R}^{1 \times D_e} \right) \times \left( \mathbb{R}^{D_e \times D} \times \mathbb{R}^{1 \times D} \right) \right)^{n_e}. \quad (139)$$

An element $U \in \mathcal{U}$ takes the form:

$$U = \left( \left( [W]^{(Q,i)}, [W]^{(K,i)}, [W]^{(V,i)}, [W]^{(O,i)} \right)_{i=1,\ldots,n_h}, \right.$$

$$\left. \left( \left( [W]^{(G,i)}, [b]^{(G,i)} \right), \left( [W]^{(A,i)}, [b]^{(A,i)} \right), \left( [W]^{(B,i)}, [b]^{(B,i)} \right) \right)_{i=1,\ldots,n_e} \right). \quad (140)$$

Define the group

$$\mathcal{G}_\mathcal{U} = \left( S_{n_h} \times \left( \mathrm{GL}_{D_k}(\mathbb{R}) \times \mathrm{GL}_{D_v}(\mathbb{R}) \right)^{n_h} \right) \times \left( \mathbb{R}^D \times \mathbb{R} \right) \times \left( S_{n_e} \times \left( \mathcal{P}_{D_e} \right)^{n_e} \right). \quad (141)$$

Each element $g \in \mathcal{G}_\mathcal{U}$ takes the form:

$$g = \left( \left( \tau_h, \left\{ M_k^{(i)}, M_v^{(i)} \right\}_{i=1,\ldots,n_h} \right), \{ \gamma_W, \gamma_b \}, \left( \tau_e \times \left\{ \pi_e^{(i)} \right\}_{i=1,\ldots,n_e} \right) \right). \quad (142)$$

The action of $\mathcal{G}_\mathcal{U}$ on $\mathcal{U}$ is defined to be $\mathcal{G}_\mathcal{U} \times \mathcal{U} \to \mathcal{U}$, which maps $(g, U) \in \mathcal{G}_\mathcal{U} \times \mathcal{U}$ to:

$$gU = \left( \left( [gW]^{(Q,i)}, [gW]^{(K,i)}, [gW]^{(V,i)}, [gW]^{(O,i)} \right)_{i=1,\ldots,n_h}, \right.$$

$$\left. \left( \left( [gW]^{(G,i)}, [gb]^{(G,i)} \right), \left( [gW]^{(A,i)}, [gb]^{(A,i)} \right), \left( [gW]^{(B,i)}, [gb]^{(B,i)} \right) \right)_{i=1,\ldots,n_e} \right), \quad (143)$$

where

$$\begin{aligned}
[gW]^{(Q,i)} &:= [W]^{(Q,\tau_h(i))} \cdot \left( M_k^{(\tau_h(i))} \right)^\top, \\
[gW]^{(K,i)} &:= [W]^{(K,\tau_h(i))} \cdot \left( M_k^{(\tau_h(i))} \right)^{-1}, \\
[gW]^{(V,i)} &:= [W]^{(V,\tau_h(i))} \cdot M_v^{(\tau_h(i))}, \\
[gW]^{(O,i)} &:= \left( M_v^{(\tau_h(i))} \right)^{-1} \cdot [W]^{(O,\tau_h(i))}, \\
[gW]^{(QK,i)} &:= [W]^{(QK,\tau_h(i))}, \\
[gW]^{(VO,i)} &:= [W]^{(VO,\tau_h(i))}, \\
[gW]^{(G,i)} &:= [W]^{(G,\tau_e(i))} + \gamma_W, \\
[gb]^{(G,i)} &:= [b]^{(G,\tau_e(i))} + \gamma_b, \\
[gW]^{(A,i)} &:= [W]^{(A,\tau_e(i))} \cdot P_{\pi_e^{(\tau_e(i))}}, \\
[gb]^{(A,i)} &:= [b]^{(A,\tau_e(i))} \cdot P_{\pi_e^{(\tau_e(i))}}, \\
[gW]^{(B,i)} &:= \left( P_{\pi_e^{(\tau_e(i))}} \right)^{-1} \cdot [W]^{(B,\tau_e(i))}, \\
[gb]^{(B,i)} &:= [b]^{(B,\tau_e(i))}.
\end{aligned} \quad (144)$$

When express the set of Equations 144 in terms of individual entries, this takes the form:

$$[gW]_{j,k}^{(Q,i)} := \left[ [W]^{(Q,\tau_h(i))} \cdot \left( M_k^{(\tau_h(i))} \right)^\top \right]_{j,k},$$

$$[gW]_{j,k}^{(K,i)} := \left[ [W]^{(K,\tau_h(i))} \cdot \left( M_k^{(\tau_h(i))} \right)^{-1} \right]_{j,k},$$

$$[gW]_{j,k}^{(V,i)} := \left[ [W]^{(V,\tau_h(i))} \cdot M_v^{(\tau_h(i))} \right]_{j,k},$$

$$[gW]_{j,k}^{(O,i)} := \left[ \left( M_v^{(\tau_h(i))} \right)^{-1} \cdot [W]^{(O,\tau_h(i))} \right]_{j,k},$$

$$[gW]_{j,k}^{(QK,i)} := \left[ [W]^{(QK,\tau_h(i))} \right]_{j,k},$$

$$[gW]_{j,k}^{(VO,i)} := \left[ [W]^{(VO,\tau_h(i))} \right]_{j,k}, \tag{145}$$

$$[gW]_{j}^{(G,i)} := [W]_{j}^{(G,\tau_e(i))} + (\gamma_W)_j,$$

$$[gb]^{(G,i)} := [b]^{(G,\tau_e(i))} + \gamma_b,$$

$$[gW]_{j,k}^{(A,i)} := [W]_{j,\pi_e^{(\tau_e(i))}(k)}^{(A,\tau_e(i))},$$

$$[gb]_{j}^{(A,i)} := [b]_{\pi_e^{(\tau_e(i))}(j)}^{(A,\tau_e(i))},$$

$$[gW]_{j,k}^{(B,i)} := [W]_{\pi_e^{(\tau_e(i))}(j),k}^{(B,\tau_e(i))},$$

$$[gb]_{j}^{(B,i)} := [b]_{j}^{(B,\tau_e(i))}.$$

# E    METANETWORK FOR MOE TRANSFORMERS: A POLYNOMIAL LAYER AND NOTATIONS

Our objective is twofold:

1. to construct a network mapping from $\mathcal{U}^d$ to $\mathcal{U}^{d'}$ that is $\mathcal{G}_{\mathcal{U}}$-equivariant;

2. to construct a network mapping from $\mathcal{U}^d$ to $\mathcal{U}^{d'}$ that is $\mathcal{G}_{\mathcal{U}}$-invariant,

where $d$ and $d'$ represent the input and output dimensions, respectively.

To this end, we design equivariant and invariant layers with respect to the group action induced by $\mathcal{G}_{\mathcal{U}}$. These layers adopt a quadratic polynomial in the input weights with unknown coefficients, in line with recent developments of metanetworks for Transformers in Tran et al. (2025). Rather than providing explicit functional expressions for each layer, we offer an illustrative and structured description in Tables 4, 5, 6 and 7. Each table includes visual cues and concrete examples to facilitate understanding.

1. Table 4 presents each layer as an affine transformation, with parameters denoted by expressions of the form $\Phi_-^-$. The superscript and subscript indices respectively indicate the output and input positions of the parameters. Importantly, the index notation is constructed so that one can unambiguously determine the dependency between inputs and outputs. Throughout, the indices $i, j, k$ refer to output components, while $s, p, q$ correspond to input components. With the exception of the symbol $1$, which denotes the bias term, all other components are defined in Appendix D.

2. Table 5 is a color-annotated version of Table 4. Elements related to the output are highlighted in blue, while those associated with the input are shown in red, including their corresponding indices.

3. Table 6 provides a detailed breakdown of the parameter notation $\Phi_-^-$. Each parameter entry corresponds to the output indicated by its column and the input indicated by its row. For instance:

- The term $\Phi_{(V,s):p,q}^{(G,i):j}$ denotes the parameter connecting $[W]_{p,q}^{(V,s)} \to [W]_j^{(G,i)}$.

- The term $\Phi_{(O,s):p,q}^{(B,i):j,k}$ denotes the parameter connecting $[W]_{p,q}^{(O,s)} \to [W]_{j,k}^{(B,i)}$.

- The term $\Phi_{(A,s):p,q}^{(G,i)}$ denotes the parameter connecting $[W]_{p,q}^{(A,s)} \to [b]^{(G,i)}$.

- The term $\Phi_{(A,s):p}^{(V,i):j,k}$ denotes the parameter connecting $[b]_p^{(A,s)} \to [W]_{j,k}^{(V,i)}$.

4. The output is computed as follows. In Table 7, for each output entry, we take a "dot product" between the corresponding column indicating the output and the final column representing the input. The summation is carried out over all indices that are compatible according to the indexing scheme. For example:

- The output $[W]_{j,k}^{(V,i)}$ is computed as:

$$
\begin{aligned}
[W]_{j,k}^{(V,i)} = \quad & \sum_{s=1}^{n_h}\sum_{p=1}^{D}\sum_{q=1}^{D_k} \Phi_{(Q,s):p,q}^{(V,i):j,k}[W]_{p,q}^{(Q,s)} + \sum_{s=1}^{n_h}\sum_{p=1}^{D}\sum_{q=1}^{D_k} \Phi_{(K,s):p,q}^{(V,i):j,k}[W]_{p,q}^{(K,s)} \\
+ & \sum_{s=1}^{n_h}\sum_{p=1}^{D}\sum_{q=1}^{D_v} \Phi_{(V,s):p,q}^{(V,i):j,k}[W]_{p,q}^{(V,s)} + \sum_{s=1}^{n_h}\sum_{p=1}^{D_v}\sum_{q=1}^{D} \Phi_{(O,s):p,q}^{(V,i):j,k}[W]_{p,q}^{(O,s)} \\
+ & \sum_{s=1}^{n_h}\sum_{p=1}^{D}\sum_{q=1}^{D} \Phi_{(QK,s):p,q}^{(V,i):j,k}[W]_{p,q}^{(QK,s)} + \sum_{s=1}^{n_h}\sum_{p=1}^{D}\sum_{q=1}^{D} \Phi_{(VO,s):p,q}^{(V,i):j,k}[W]_{p,q}^{(VO,s)} \\
+ & \sum_{s=1}^{n_e}\sum_{p=1}^{D} \Phi_{(G,s):p}^{(V,i):j,k}[W]_p^{(G,s)} + \sum_{s=1}^{n_e} \Phi_{(G,s)}^{(V,i):j,k}[b]^{(G,s)} \\
+ & \sum_{s=1}^{n_e}\sum_{p=1}^{D}\sum_{q=1}^{D_e} \Phi_{(A,s):p,q}^{(V,i):j,k}[W]_{p,q}^{(A,s)} + \sum_{s=1}^{n_e}\sum_{p=1}^{D_e} \Phi_{(A,s):p}^{(V,i):j,k}[b]_p^{(A,s)} \\
+ & \sum_{s=1}^{n_e}\sum_{p=1}^{D_e}\sum_{q=1}^{D} \Phi_{(B,s):p,q}^{(V,i):j,k}[W]_{p,q}^{(B,s)} + \sum_{s=1}^{n_e}\sum_{p=1}^{D} \Phi_{(B,s):p}^{(V,i):j,k}[b]_p^{(B,s)} \\
+ & \Phi_1^{(V,i):j,k}
\end{aligned}
\tag{146}
$$

- The output $[W]_{j,k}^{(A,i)}$ is computed as:

$$
\begin{aligned}
[W]_{j,k}^{(A,i)} = \quad & \sum_{s=1}^{n_h}\sum_{p=1}^{D}\sum_{q=1}^{D_k} \Phi_{(Q,s):p,q}^{(A,i):j,k}[W]_{p,q}^{(Q,s)} + \sum_{s=1}^{n_h}\sum_{p=1}^{D}\sum_{q=1}^{D_k} \Phi_{(K,s):p,q}^{(A,i):j,k}[W]_{p,q}^{(K,s)} \\
+ & \sum_{s=1}^{n_h}\sum_{p=1}^{D}\sum_{q=1}^{D_v} \Phi_{(V,s):p,q}^{(A,i):j,k}[W]_{p,q}^{(V,s)} + \sum_{s=1}^{n_h}\sum_{p=1}^{D_v}\sum_{q=1}^{D} \Phi_{(O,s):p,q}^{(A,i):j,k}[W]_{p,q}^{(O,s)} \\
+ & \sum_{s=1}^{n_h}\sum_{p=1}^{D}\sum_{q=1}^{D} \Phi_{(QK,s):p,q}^{(A,i):j,k}[W]_{p,q}^{(QK,s)} + \sum_{s=1}^{n_h}\sum_{p=1}^{D}\sum_{q=1}^{D} \Phi_{(VO,s):p,q}^{(A,i):j,k}[W]_{p,q}^{(VO,s)} \\
+ & \sum_{s=1}^{n_e}\sum_{p=1}^{D} \Phi_{(G,s):p}^{(A,i):j,k}[W]_p^{(G,s)} + \sum_{s=1}^{n_e} \Phi_{(G,s)}^{(A,i):j,k}[b]^{(G,s)} \\
+ & \sum_{s=1}^{n_e}\sum_{p=1}^{D}\sum_{q=1}^{D_e} \Phi_{(A,s):p,q}^{(V,i):j,k}[W]_{p,q}^{(A,s)} + \sum_{s=1}^{n_e}\sum_{p=1}^{D_e} \Phi_{(A,s):p}^{(A,i):j,k}[b]_p^{(A,s)} \\
+ & \sum_{s=1}^{n_e}\sum_{p=1}^{D_e}\sum_{q=1}^{D} \Phi_{(B,s):p,q}^{(A,i):j,k}[W]_{p,q}^{(B,s)} + \sum_{s=1}^{n_e}\sum_{p=1}^{D} \Phi_{(B,s):p}^{(A,i):j,k}[b]_p^{(B,s)} \\
+ & \Phi_1^{(A,i):j,k}
\end{aligned}
\tag{147}
$$

Table 4: This table presents each layer as an affine transformation, with parameters denoted by expressions of the form $\Phi^{-}_{-}$. The superscript and subscript indices respectively indicate the output and input positions of the parameters. Importantly, the index notation is constructed so that one can unambiguously determine the dependency between inputs and outputs. Throughout, the indices $i, j, k$ refer to output components, while $s, p, q$ correspond to input components. With the exception of the symbol 1, which denotes the bias term, all other components are defined in Appendix D.

|  | $[W]^{(Q,i)}_{j,k}$ | $[W]^{(K,i)}_{j,k}$ | $[W]^{(V,i)}_{j,k}$ | $[W]^{(O,i)}_{j,k}$ | $[W]^{(G,i)}_j$ | $[b]^{(G,i)}$ | $[W]^{(A,i)}_{j,k}$ | $[b]^{(A,i)}_j$ | $[W]^{(B,i)}_{j,k}$ | $[b]^{(B,i)}_j$ |  |
|---|---|---|---|---|---|---|---|---|---|---|---|
| $\Phi_{(Q,s):p,q}$ | $\Phi^{(Q,i):j,k}_{(Q,s):p,q}$ | $\Phi^{(K,i):j,k}_{(Q,s):p,q}$ | $\Phi^{(V,i):j,k}_{(Q,s):p,q}$ | $\Phi^{(O,i):j,k}_{(Q,s):p,q}$ | $\Phi^{(G,i):j}_{(Q,s):p,q}$ | $\Phi^{(G,i)}_{(Q,s):p,q}$ | $\Phi^{(A,i):j,k}_{(Q,s):p,q}$ | $\Phi^{(A,i):j}_{(Q,s):p,q}$ | $\Phi^{(B,i):j,k}_{(Q,s):p,q}$ | $\Phi^{(B,i):j}_{(Q,s):p,q}$ | $[W]^{(Q,s)}_{p,q}$ |
| $\Phi_{(K,s):p,q}$ | $\Phi^{(Q,i):j,k}_{(K,s):p,q}$ | $\Phi^{(K,i):j,k}_{(K,s):p,q}$ | $\Phi^{(V,i):j,k}_{(K,s):p,q}$ | $\Phi^{(O,i):j,k}_{(K,s):p,q}$ | $\Phi^{(G,i):j}_{(K,s):p,q}$ | $\Phi^{(G,i)}_{(K,s):p,q}$ | $\Phi^{(A,i):j,k}_{(K,s):p,q}$ | $\Phi^{(A,i):j}_{(K,s):p,q}$ | $\Phi^{(B,i):j,k}_{(K,s):p,q}$ | $\Phi^{(B,i):j}_{(K,s):p,q}$ | $[W]^{(K,s)}_{p,q}$ |
| $\Phi_{(V,s):p,q}$ | $\Phi^{(Q,i):j,k}_{(V,s):p,q}$ | $\Phi^{(K,i):j,k}_{(V,s):p,q}$ | $\Phi^{(V,i):j,k}_{(V,s):p,q}$ | $\Phi^{(O,i):j,k}_{(V,s):p,q}$ | $\Phi^{(G,i):j}_{(V,s):p,q}$ | $\Phi^{(G,i)}_{(V,s):p,q}$ | $\Phi^{(A,i):j,k}_{(V,s):p,q}$ | $\Phi^{(A,i):j}_{(V,s):p,q}$ | $\Phi^{(B,i):j,k}_{(V,s):p,q}$ | $\Phi^{(B,i):j}_{(V,s):p,q}$ | $[W]^{(V,s)}_{p,q}$ |
| $\Phi_{(O,s):p,q}$ | $\Phi^{(Q,i):j,k}_{(O,s):p,q}$ | $\Phi^{(K,i):j,k}_{(O,s):p,q}$ | $\Phi^{(V,i):j,k}_{(O,s):p,q}$ | $\Phi^{(O,i):j,k}_{(O,s):p,q}$ | $\Phi^{(G,i):j}_{(O,s):p,q}$ | $\Phi^{(G,i)}_{(O,s):p,q}$ | $\Phi^{(A,i):j,k}_{(O,s):p,q}$ | $\Phi^{(A,i):j}_{(O,s):p,q}$ | $\Phi^{(B,i):j,k}_{(O,s):p,q}$ | $\Phi^{(B,i):j}_{(O,s):p,q}$ | $[W]^{(O,s)}_{p,q}$ |
| $\Phi_{(QK,s):p,q}$ | $\Phi^{(Q,i):j,k}_{(QK,s):p,q}$ | $\Phi^{(K,i):j,k}_{(QK,s):p,q}$ | $\Phi^{(V,i):j,k}_{(QK,s):p,q}$ | $\Phi^{(O,i):j,k}_{(QK,s):p,q}$ | $\Phi^{(G,i):j}_{(QK,s):p,q}$ | $\Phi^{(G,i)}_{(QK,s):p,q}$ | $\Phi^{(A,i):j,k}_{(QK,s):p,q}$ | $\Phi^{(A,i):j}_{(QK,s):p,q}$ | $\Phi^{(B,i):j,k}_{(QK,s):p,q}$ | $\Phi^{(B,i):j}_{(QK,s):p,q}$ | $[W]^{(QK,s)}_{p,q}$ |
| $\Phi_{(VO,s):p,q}$ | $\Phi^{(Q,i):j,k}_{(VO,s):p,q}$ | $\Phi^{(K,i):j,k}_{(VO,s):p,q}$ | $\Phi^{(V,i):j,k}_{(VO,s):p,q}$ | $\Phi^{(O,i):j,k}_{(VO,s):p,q}$ | $\Phi^{(G,i):j}_{(VO,s):p,q}$ | $\Phi^{(G,i)}_{(VO,s):p,q}$ | $\Phi^{(A,i):j,k}_{(VO,s):p,q}$ | $\Phi^{(A,i):j}_{(VO,s):p,q}$ | $\Phi^{(B,i):j,k}_{(VO,s):p,q}$ | $\Phi^{(B,i):j}_{(VO,s):p,q}$ | $[W]^{(VO,s)}_{p,q}$ |
| $\Phi_{(G,s):p}$ | $\Phi^{(Q,i):j,k}_{(G,s):p}$ | $\Phi^{(K,i):j,k}_{(G,s):p}$ | $\Phi^{(V,i):j,k}_{(G,s):p}$ | $\Phi^{(O,i):j,k}_{(G,s):p}$ | $\Phi^{(G,i):j}_{(G,s):p}$ | $\Phi^{(G,i)}_{(G,s):p}$ | $\Phi^{(A,i):j,k}_{(G,s):p}$ | $\Phi^{(A,i):j}_{(G,s):p}$ | $\Phi^{(B,i):j,k}_{(G,s):p}$ | $\Phi^{(B,i):j}_{(G,s):p}$ | $[W]^{(G,s)}_p$ |
| $\Phi_{(G,s)}$ | $\Phi^{(Q,i):j,k}_{(G,s)}$ | $\Phi^{(K,i):j,k}_{(G,s)}$ | $\Phi^{(V,i):j,k}_{(G,s)}$ | $\Phi^{(O,i):j,k}_{(G,s)}$ | $\Phi^{(G,i):j}_{(G,s)}$ | $\Phi^{(G,i)}_{(G,s)}$ | $\Phi^{(A,i):j,k}_{(G,s)}$ | $\Phi^{(A,i):j}_{(G,s)}$ | $\Phi^{(B,i):j,k}_{(G,s)}$ | $\Phi^{(B,i):j}_{(G,s)}$ | $[b]^{(G,s)}$ |
| $\Phi_{(A,s):p,q}$ | $\Phi^{(Q,i):j,k}_{(A,s):p,q}$ | $\Phi^{(K,i):j,k}_{(A,s):p,q}$ | $\Phi^{(V,i):j,k}_{(A,s):p,q}$ | $\Phi^{(O,i):j,k}_{(A,s):p,q}$ | $\Phi^{(G,i):j}_{(A,s):p,q}$ | $\Phi^{(G,i)}_{(A,s):p,q}$ | $\Phi^{(A,i):j,k}_{(A,s):p,q}$ | $\Phi^{(A,i):j}_{(A,s):p,q}$ | $\Phi^{(B,i):j,k}_{(A,s):p,q}$ | $\Phi^{(B,i):j}_{(A,s):p,q}$ | $[W]^{(A,s)}_{p,q}$ |
| $\Phi_{(A,s):p}$ | $\Phi^{(Q,i):j,k}_{(A,s):p}$ | $\Phi^{(K,i):j,k}_{(A,s):p}$ | $\Phi^{(V,i):j,k}_{(A,s):p}$ | $\Phi^{(O,i):j,k}_{(A,s):p}$ | $\Phi^{(G,i):j}_{(A,s):p}$ | $\Phi^{(G,i)}_{(A,s):p}$ | $\Phi^{(A,i):j,k}_{(A,s):p}$ | $\Phi^{(A,i):j}_{(A,s):p}$ | $\Phi^{(B,i):j,k}_{(A,s):p}$ | $\Phi^{(B,i):j}_{(A,s):p}$ | $[b]^{(A,s)}_p$ |
| $\Phi_{(B,s):p,q}$ | $\Phi^{(Q,i):j,k}_{(B,s):p,q}$ | $\Phi^{(K,i):j,k}_{(B,s):p,q}$ | $\Phi^{(V,i):j,k}_{(B,s):p,q}$ | $\Phi^{(O,i):j,k}_{(B,s):p,q}$ | $\Phi^{(G,i):j}_{(B,s):p,q}$ | $\Phi^{(G,i)}_{(B,s):p,q}$ | $\Phi^{(A,i):j,k}_{(B,s):p,q}$ | $\Phi^{(A,i):j}_{(B,s):p,q}$ | $\Phi^{(B,i):j,k}_{(B,s):p,q}$ | $\Phi^{(B,i):j}_{(B,s):p,q}$ | $[W]^{(B,s)}_{p,q}$ |
| $\Phi_{(B,s):p}$ | $\Phi^{(Q,i):j,k}_{(B,s):p}$ | $\Phi^{(K,i):j,k}_{(B,s):p}$ | $\Phi^{(V,i):j,k}_{(B,s):p}$ | $\Phi^{(O,i):j,k}_{(B,s):p}$ | $\Phi^{(G,i):j}_{(B,s):p}$ | $\Phi^{(G,i)}_{(B,s):p}$ | $\Phi^{(A,i):j,k}_{(B,s):p}$ | $\Phi^{(A,i):j}_{(B,s):p}$ | $\Phi^{(B,i):j,k}_{(B,s):p}$ | $\Phi^{(B,i):j}_{(B,s):p}$ | $[b]^{(B,s)}_p$ |
| $\Phi_1$ | $\Phi^{(Q,i):j,k}_1$ | $\Phi^{(K,i):j,k}_1$ | $\Phi^{(V,i):j,k}_1$ | $\Phi^{(O,i):j,k}_1$ | $\Phi^{(G,i):j}_1$ | $\Phi^{(G,i)}_1$ | $\Phi^{(A,i):j,k}_1$ | $\Phi^{(A,i):j}_1$ | $\Phi^{(B,i):j,k}_1$ | $\Phi^{(B,i):j}_1$ | 1 |
|  | $\Phi^{(Q,i):j,k}$ | $\Phi^{(K,i):j,k}$ | $\Phi^{(V,i):j,k}$ | $\Phi^{(O,i):j,k}$ | $\Phi^{(G,i):j}$ | $\Phi^{(G,i)}$ | $\Phi^{(A,i):j,k}$ | $\Phi^{(A,i):j}$ | $\Phi^{(B,i):j,k}$ | $\Phi^{(B,i):j}$ |  |

Table 5: This table is a color-annotated version of Table 4. Elements related to the output are highlighted in blue, while those associated with the input are shown in red, including their corresponding indices.

|  | $[W]^{(Q,i)}_{j,k}$ | $[W]^{(K,i)}_{j,k}$ | $[W]^{(V,i)}_{j,k}$ | $[W]^{(O,i)}_{j,k}$ | $[W]^{(G,i)}_{j,k}$ | $[b]^{(G,i)}$ | $[W]^{(A,i)}_{j,k}$ | $[b]^{(A,i)}_j$ | $[W]^{(B,i)}_{j,k}$ | $[b]^{(B,i)}_j$ |  |
|---|---|---|---|---|---|---|---|---|---|---|---|
| $\Phi_{(Q,s):p,q}$ | $\Phi^{(Q,i):j,k}_{(Q,s):p,q}$ | $\Phi^{(K,i):j,k}_{(Q,s):p,q}$ | $\Phi^{(V,i):j,k}_{(Q,s):p,q}$ | $\Phi^{(O,i):j,k}_{(Q,s):p,q}$ | $\Phi^{(G,i):j}_{(Q,s):p,q}$ | $\Phi^{(G,i)}_{(Q,s):p,q}$ | $\Phi^{(A,i):j,k}_{(Q,s):p,q}$ | $\Phi^{(A,i):j}_{(Q,s):p,q}$ | $\Phi^{(B,i):j,k}_{(Q,s):p,q}$ | $\Phi^{(B,i):j}_{(Q,s):p,q}$ | $[W]^{(Q,s)}_{p,q}$ |
| $\Phi_{(K,s):p,q}$ | $\Phi^{(Q,i):j,k}_{(K,s):p,q}$ | $\Phi^{(K,i):j,k}_{(K,s):p,q}$ | $\Phi^{(V,i):j,k}_{(K,s):p,q}$ | $\Phi^{(O,i):j,k}_{(K,s):p,q}$ | $\Phi^{(G,i):j}_{(K,s):p,q}$ | $\Phi^{(G,i)}_{(K,s):p,q}$ | $\Phi^{(A,i):j,k}_{(K,s):p,q}$ | $\Phi^{(A,i):j}_{(K,s):p,q}$ | $\Phi^{(B,i):j,k}_{(K,s):p,q}$ | $\Phi^{(B,i):j}_{(K,s):p,q}$ | $[W]^{(K,s)}_{p,q}$ |
| $\Phi_{(V,s):p,q}$ | $\Phi^{(Q,i):j,k}_{(V,s):p,q}$ | $\Phi^{(K,i):j,k}_{(V,s):p,q}$ | $\Phi^{(V,i):j,k}_{(V,s):p,q}$ | $\Phi^{(O,i):j,k}_{(V,s):p,q}$ | $\Phi^{(G,i):j}_{(V,s):p,q}$ | $\Phi^{(G,i)}_{(V,s):p,q}$ | $\Phi^{(A,i):j,k}_{(V,s):p,q}$ | $\Phi^{(A,i):j}_{(V,s):p,q}$ | $\Phi^{(B,i):j,k}_{(V,s):p,q}$ | $\Phi^{(B,i):j}_{(V,s):p,q}$ | $[W]^{(V,s)}_{p,q}$ |
| $\Phi_{(O,s):p,q}$ | $\Phi^{(Q,i):j,k}_{(O,s):p,q}$ | $\Phi^{(K,i):j,k}_{(O,s):p,q}$ | $\Phi^{(V,i):j,k}_{(O,s):p,q}$ | $\Phi^{(O,i):j,k}_{(O,s):p,q}$ | $\Phi^{(G,i):j}_{(O,s):p,q}$ | $\Phi^{(G,i)}_{(O,s):p,q}$ | $\Phi^{(A,i):j,k}_{(O,s):p,q}$ | $\Phi^{(A,i):j}_{(O,s):p,q}$ | $\Phi^{(B,i):j,k}_{(O,s):p,q}$ | $\Phi^{(B,i):j}_{(O,s):p,q}$ | $[W]^{(O,s)}_{p,q}$ |
| $\Phi_{(QK,s):p,q}$ | $\Phi^{(Q,i):j,k}_{(QK,s):p,q}$ | $\Phi^{(K,i):j,k}_{(QK,s):p,q}$ | $\Phi^{(V,i):j,k}_{(QK,s):p,q}$ | $\Phi^{(O,i):j,k}_{(QK,s):p,q}$ | $\Phi^{(G,i):j}_{(QK,s):p,q}$ | $\Phi^{(G,i)}_{(QK,s):p,q}$ | $\Phi^{(A,i):j,k}_{(QK,s):p,q}$ | $\Phi^{(A,i):j}_{(QK,s):p,q}$ | $\Phi^{(B,i):j,k}_{(QK,s):p,q}$ | $\Phi^{(B,i):j}_{(QK,s):p,q}$ | $[W]^{(QK,s)}_{p,q}$ |
| $\Phi_{(VO,s):p,q}$ | $\Phi^{(Q,i):j,k}_{(VO,s):p,q}$ | $\Phi^{(K,i):j,k}_{(VO,s):p,q}$ | $\Phi^{(V,i):j,k}_{(VO,s):p,q}$ | $\Phi^{(O,i):j,k}_{(VO,s):p,q}$ | $\Phi^{(G,i):j}_{(VO,s):p,q}$ | $\Phi^{(G,i)}_{(VO,s):p,q}$ | $\Phi^{(A,i):j,k}_{(VO,s):p,q}$ | $\Phi^{(A,i):j}_{(VO,s):p,q}$ | $\Phi^{(B,i):j,k}_{(VO,s):p,q}$ | $\Phi^{(B,i):j}_{(VO,s):p,q}$ | $[W]^{(VO,s)}_{p,q}$ |
| $\Phi_{(G,s):p}$ | $\Phi^{(Q,i):j,k}_{(G,s):p}$ | $\Phi^{(K,i):j,k}_{(G,s):p}$ | $\Phi^{(V,i):j,k}_{(G,s):p}$ | $\Phi^{(O,i):j,k}_{(G,s):p}$ | $\Phi^{(G,i):j}_{(G,s):p}$ | $\Phi^{(G,i)}_{(G,s):p}$ | $\Phi^{(A,i):j,k}_{(G,s):p}$ | $\Phi^{(A,i):j}_{(G,s):p}$ | $\Phi^{(B,i):j,k}_{(G,s):p}$ | $\Phi^{(B,i):j}_{(G,s):p}$ | $[W]^{(G,s)}_p$ |
| $\Phi_{(G,s)}$ | $\Phi^{(Q,i):j,k}_{(G,s)}$ | $\Phi^{(K,i):j,k}_{(G,s)}$ | $\Phi^{(V,i):j,k}_{(G,s)}$ | $\Phi^{(O,i):j,k}_{(G,s)}$ | $\Phi^{(G,i):j}_{(G,s)}$ | $\Phi^{(G,i)}_{(G,s)}$ | $\Phi^{(A,i):j,k}_{(G,s)}$ | $\Phi^{(A,i):j}_{(G,s)}$ | $\Phi^{(B,i):j,k}_{(G,s)}$ | $\Phi^{(B,i):j}_{(G,s)}$ | $[b]^{(G,s)}$ |
| $\Phi_{(A,s):p,q}$ | $\Phi^{(Q,i):j,k}_{(A,s):p,q}$ | $\Phi^{(K,i):j,k}_{(A,s):p,q}$ | $\Phi^{(V,i):j,k}_{(A,s):p,q}$ | $\Phi^{(O,i):j,k}_{(A,s):p,q}$ | $\Phi^{(G,i):j}_{(A,s):p,q}$ | $\Phi^{(G,i)}_{(A,s):p,q}$ | $\Phi^{(A,i):j,k}_{(A,s):p,q}$ | $\Phi^{(A,i):j}_{(A,s):p,q}$ | $\Phi^{(B,i):j,k}_{(A,s):p,q}$ | $\Phi^{(B,i):j}_{(A,s):p,q}$ | $[W]^{(A,s)}_{p,q}$ |
| $\Phi_{(A,s):p}$ | $\Phi^{(Q,i):j,k}_{(A,s):p}$ | $\Phi^{(K,i):j,k}_{(A,s):p}$ | $\Phi^{(V,i):j,k}_{(A,s):p}$ | $\Phi^{(O,i):j,k}_{(A,s):p}$ | $\Phi^{(G,i):j}_{(A,s):p}$ | $\Phi^{(G,i)}_{(A,s):p}$ | $\Phi^{(A,i):j,k}_{(A,s):p}$ | $\Phi^{(A,i):j}_{(A,s):p}$ | $\Phi^{(B,i):j,k}_{(A,s):p}$ | $\Phi^{(B,i):j}_{(A,s):p}$ | $[b]^{(A,s)}_p$ |
| $\Phi_{(B,s):p,q}$ | $\Phi^{(Q,i):j,k}_{(B,s):p,q}$ | $\Phi^{(K,i):j,k}_{(B,s):p,q}$ | $\Phi^{(V,i):j,k}_{(B,s):p,q}$ | $\Phi^{(O,i):j,k}_{(B,s):p,q}$ | $\Phi^{(G,i):j}_{(B,s):p,q}$ | $\Phi^{(G,i)}_{(B,s):p,q}$ | $\Phi^{(A,i):j,k}_{(B,s):p,q}$ | $\Phi^{(A,i):j}_{(B,s):p,q}$ | $\Phi^{(B,i):j,k}_{(B,s):p,q}$ | $\Phi^{(B,i):j}_{(B,s):p,q}$ | $[W]^{(B,s)}_{p,q}$ |
| $\Phi_{(B,s):p}$ | $\Phi^{(Q,i):j,k}_{(B,s):p}$ | $\Phi^{(K,i):j,k}_{(B,s):p}$ | $\Phi^{(V,i):j,k}_{(B,s):p}$ | $\Phi^{(O,i):j,k}_{(B,s):p}$ | $\Phi^{(G,i):j}_{(B,s):p}$ | $\Phi^{(G,i)}_{(B,s):p}$ | $\Phi^{(A,i):j,k}_{(B,s):p}$ | $\Phi^{(A,i):j}_{(B,s):p}$ | $\Phi^{(B,i):j,k}_{(B,s):p}$ | $\Phi^{(B,i):j}_{(B,s):p}$ | $[b]^{(B,s)}_p$ |
| $\Phi_1$ | $\Phi^{(Q,i):j,k}_1$ | $\Phi^{(K,i):j,k}_1$ | $\Phi^{(V,i):j,k}_1$ | $\Phi^{(O,i):j,k}_1$ | $\Phi^{(G,i):j}_1$ | $\Phi^{(G,i)}_1$ | $\Phi^{(A,i):j,k}_1$ | $\Phi^{(A,i):j}_1$ | $\Phi^{(B,i):j,k}_1$ | $\Phi^{(B,i):j}_1$ | 1 |
|  | $\Phi^{(Q,i):j,k}$ | $\Phi^{(K,i):j,k}$ | $\Phi^{(V,i):j,k}$ | $\Phi^{(O,i):j,k}$ | $\Phi^{(G,i):j}$ | $\Phi^{(G,i)}$ | $\Phi^{(A,i):j,k}$ | $\Phi^{(A,i):j}$ | $\Phi^{(B,i):j,k}$ | $\Phi^{(B,i):j}$ |  |

Table 6: This table provides a detailed breakdown of the parameter notation $\Phi^-_-$. Each parameter entry corresponds to the output indicated by its column and the input indicated by its row.

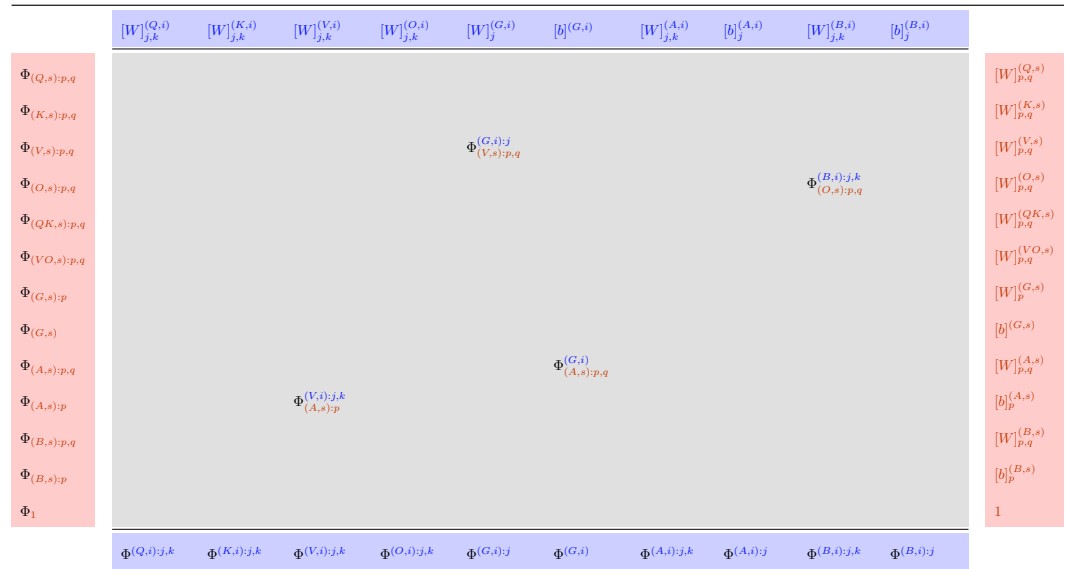

Table 7: The output is computed as follows. For each output entry, we take a "dot product" between the corresponding column indicating the output and the final column representing the input. The summation is carried out over all indices that are compatible according to the indexing scheme.

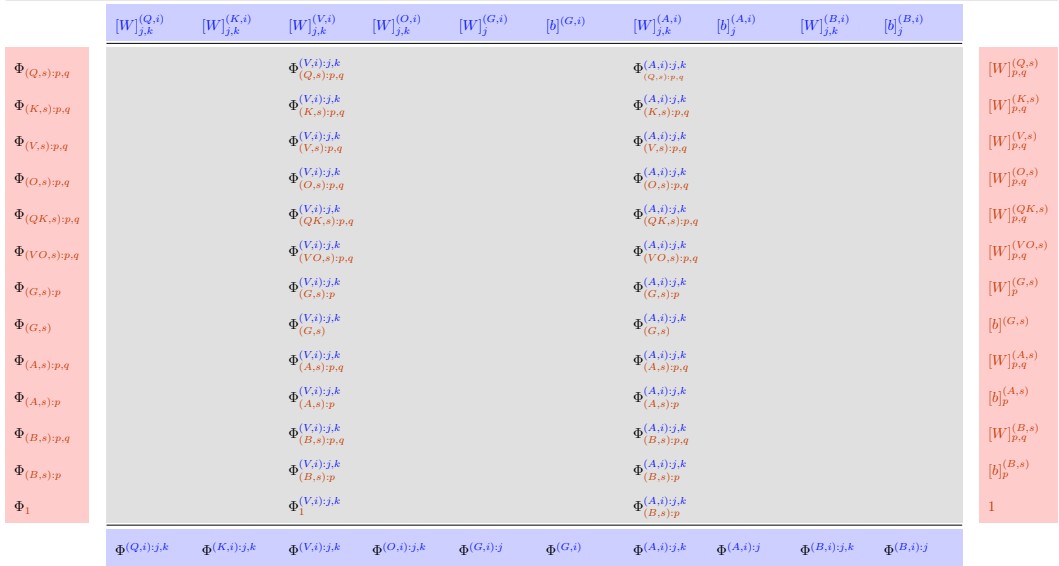

## F  EQUIVARIANT LAYER

In this section, we provide a detailed computation of $E(U)$. To construct $E(U)$, following the design of equivariant polynomial layers in Tran et al. (2025), we adopt a quadratic polynomial in the input weights $U$ with unknown coefficients, as described in the previous section, and use a parameter-sharing technique to determine the constraints on these coefficients that ensure $E$ is equivariant. We begin with the formulation of $E(U)$ below:

$$
E(U) = \left( \left( [E(W)]^{(Q,i)}, [E(W)]^{(K,i)}, [E(W)]^{(V,i)}, [E(W)]^{(O,i)} \right)_{i=1,\ldots,n_h}, \right.
$$
$$
\left( \left( [E(W)]^{(G,i)}, [E(b)]^{(G,i)} \right), \right.
$$
$$
\left. \left. \left( [E(W)]^{(AE,i)}, [E(b)]^{(A,i)} \right), \left( [E(W)]^{(B,i)}, [E(b)]^{(B,i)} \right) \right)_{i=1,\ldots,n_e} \right).
$$
$$(148)$$

### F.1  COMPUTING $E(gU)$

We borrow the following lemmas from Tran et al. (2025).

**Lemma F.1** (See (Tran et al., 2025, Section D.2))*. Assume that $E\colon \mathcal{U} \to \mathcal{U}$ is a function defined as in Equation 148 for some coefficients $\Phi_-^-$. If $E(U) = 0$ for all $U \in \mathcal{U}$, then all coefficients are equal to zero.*

**Lemma F.2** (See (Tran et al., 2025, Section D.2))*. Let $h$ and $D$ be positive integers. Let $f_s^{(1)}, f_s^{(2)}\colon \mathbb{R}^{D\times D} \to \mathbb{R}$ be $\mathbb{R}$-linear functions for each $s = 1,\ldots,h$. Assume that there exists a constant $\lambda \in \mathbb{R}$ such that*

$$
\sum_{s=1}^{h} f_s^{(1)}\left(M^{(s)}\right) + f_s^{(2)}\left(\left(M^{(s)}\right)^{-1}\right) = \lambda, \tag{149}
$$

*for all $\left(M^{(1)},\ldots,M^{(h)}\right) \in \mathrm{GL}_D(\mathbb{R})^h$. Then*

$$
f_s^{(1)}(M) = f_s^{(2)}(M) = \lambda = 0
$$

*for all $s = 1,\ldots,h$ and $M \in \mathrm{GL}_D(\mathbb{R})$.*

We now return to the computation of $E(U)$. A detailed explanation of the $E(U)$ layer and its associated computations is provided in Section E.

By Equation 143, we have:

$$
E(gU) = \left( \left( [E(gW)]^{(Q,i)}, [E(gW)]^{(K,i)}, [E(gW)]^{(V,i)}, [E(gW)]^{(O,i)} \right)_{i=1,\ldots,n_h}, \right.
$$
$$
\left. \left( \left( [E(gW)]^{(G,i)}, [E(gb)]^{(G,i)} \right), \left( [E(gW)]^{(A,i)}, [E(gb)]^{(A,i)} \right), \left( [E(gW)]^{(B,i)}, [E(gb)]^{(B,i)} \right) \right)_{i=1,\ldots,n_e} \right),
$$
$$(150)$$

where

$$
[E(gW)]_{j,k}^{(Q,i)} = \sum_{s=1}^{h}\sum_{p=1}^{D}\sum_{q=1}^{D} \Phi_{(QK,s):p,q}^{(Q,i):j,k}[gWgW]_{p,q}^{(QK,s)}
$$

$$
+ \sum_{s=1}^{h} \sum_{p=1}^{D} \sum_{q=1}^{D} \Phi_{(VO,s):p,q}^{(Q,i):j,k} [gWgW]_{p,q}^{(VO,s)}
$$

$$
+ \sum_{s=1}^{h} \sum_{p=1}^{D} \sum_{q=1}^{D_k} \Phi_{(Q,s):p,q}^{(Q,i):j,k} [gW]_{p,q}^{(Q,s)} + \sum_{s=1}^{h} \sum_{p=1}^{D} \sum_{q=1}^{D_k} \Phi_{(K,s):p,q}^{(Q,i):j,k} [gW]_{p,q}^{(K,s)}
$$

$$
+ \sum_{s=1}^{h} \sum_{p=1}^{D} \sum_{q=1}^{D_v} \Phi_{(V,s):p,q}^{(Q,i):j,k} [gW]_{p,q}^{(V,s)} + \sum_{s=1}^{h} \sum_{p=1}^{D_v} \sum_{q=1}^{D} \Phi_{(O,s):p,q}^{(Q,i):j,k} [gW]_{p,q}^{(O,s)}
$$

$$
+ \sum_{s=1}^{n_e} \sum_{p=1}^{D} \Phi_{(G,s):p}^{(Q,i):j,k} [gW]_{p}^{(G,s)} + \sum_{s=1}^{n_e} \sum_{p=1}^{D} \sum_{q=1}^{D_A} \Phi_{(A,s):p,q}^{(Q,i):j,k} [gW]_{p,q}^{(A,s)} + \sum_{s=1}^{n_e} \sum_{p=1}^{D_A} \sum_{q=1}^{D} \Phi_{(B,s):p,q}^{(Q,i):j,k} [gW]_{p,q}^{(B,s)}
$$

$$
\sum_{s=1}^{n_e} \Phi_{(G,s)}^{(Q,i):j,k} [gb]^{(G,s)} + \sum_{s=1}^{n_e} \sum_{q=1}^{D_A} \Phi_{(A,s):q}^{(Q,i):j,k} [gb]_{q}^{(A,s)} + \sum_{s=1}^{n_e} \sum_{q=1}^{D} \Phi_{(B,s):q}^{(Q,i):j,k} [gb]_{q}^{(B,s)} + \Phi_{1}^{(Q,i):j,k},
$$

$$(151)$$

$$
[E(gW)]_{j,k}^{(K,i)} = \sum_{s=1}^{h} \sum_{p=1}^{D} \sum_{q=1}^{D} \Phi_{(QK,s):p,q}^{(K,i):j,k} [gWgW]_{p,q}^{(QK,s)}
$$

$$
+ \sum_{s=1}^{h} \sum_{p=1}^{D} \sum_{q=1}^{D} \Phi_{(VO,s):p,q}^{(K,i):j,k} [gWgW]_{p,q}^{(VO,s)}
$$

$$
+ \sum_{s=1}^{h} \sum_{p=1}^{D} \sum_{q=1}^{D_k} \Phi_{(Q,s):p,q}^{(K,i):j,k} [gW]_{p,q}^{(Q,s)} + \sum_{s=1}^{h} \sum_{p=1}^{D} \sum_{q=1}^{D_k} \Phi_{(K,s):p,q}^{(K,i):j,k} [gW]_{p,q}^{(K,s)}
$$

$$
+ \sum_{s=1}^{h} \sum_{p=1}^{D} \sum_{q=1}^{D_v} \Phi_{(V,s):p,q}^{(K,i):j,k} [gW]_{p,q}^{(V,s)} + \sum_{s=1}^{h} \sum_{p=1}^{D_v} \sum_{q=1}^{D} \Phi_{(O,s):p,q}^{(K,i):j,k} [gW]_{p,q}^{(O,s)}
$$

$$
+ \sum_{s=1}^{n_e} \sum_{p=1}^{D} \Phi_{(G,s):p}^{(K,i):j,k} [gW]_{p}^{(G,s)} + \sum_{s=1}^{n_e} \sum_{p=1}^{D} \sum_{q=1}^{D_A} \Phi_{(A,s):p,q}^{(K,i):j,k} [gW]_{p,q}^{(A,s)} + \sum_{s=1}^{n_e} \sum_{p=1}^{D_A} \sum_{q=1}^{D} \Phi_{(B,s):p,q}^{(K,i):j,k} [gW]_{p,q}^{(B,s)}
$$

$$
\sum_{s=1}^{n_e} \Phi_{(G,s)}^{(K,i):j,k} [gb]^{(G,s)} + \sum_{s=1}^{n_e} \sum_{q=1}^{D_A} \Phi_{(A,s):q}^{(K,i):j,k} [gb]_{q}^{(A,s)} + \sum_{s=1}^{n_e} \sum_{q=1}^{D} \Phi_{(B,s):q}^{(K,i):j,k} [gb]_{q}^{(B,s)} + \Phi_{1}^{(K,i):j,k},
$$

$$(152)$$

$$
[E(gW)]_{j,k}^{(V,i)} = \sum_{s=1}^{h} \sum_{p=1}^{D} \sum_{q=1}^{D} \Phi_{(QK,s):p,q}^{(V,i):j,k} [gWgW]_{p,q}^{(QK,s)}
$$

$$
+ \sum_{s=1}^{h} \sum_{p=1}^{D} \sum_{q=1}^{D} \Phi_{(VO,s):p,q}^{(V,i):j,k} [gWgW]_{p,q}^{(VO,s)}
$$

$$
+ \sum_{s=1}^{h} \sum_{p=1}^{D} \sum_{q=1}^{D_k} \Phi_{(Q,s):p,q}^{(V,i):j,k} [gW]_{p,q}^{(Q,s)} + \sum_{s=1}^{h} \sum_{p=1}^{D} \sum_{q=1}^{D_k} \Phi_{(K,s):p,q}^{(V,i):j,k} [gW]_{p,q}^{(K,s)}
$$

$$
+ \sum_{s=1}^{h} \sum_{p=1}^{D} \sum_{q=1}^{D_v} \Phi_{(V,s):p,q}^{(V,i):j,k} [gW]_{p,q}^{(V,s)} + \sum_{s=1}^{h} \sum_{p=1}^{D_v} \sum_{q=1}^{D} \Phi_{(O,s):p,q}^{(V,i):j,k} [gW]_{p,q}^{(O,s)}
$$

$$
+ \sum_{s=1}^{n_e} \sum_{p=1}^{D} \Phi_{(G,s):p}^{(V,i):j,k} [gW]_{p}^{(G,s)} + \sum_{s=1}^{n_e} \sum_{p=1}^{D} \sum_{q=1}^{D_A} \Phi_{(A,s):p,q}^{(V,i):j,k} [gW]_{p,q}^{(A,s)} + \sum_{s=1}^{n_e} \sum_{p=1}^{D_A} \sum_{q=1}^{D} \Phi_{(B,s):p,q}^{(V,i):j,k} [gW]_{p,q}^{(B,s)}
$$

$$
\sum_{s=1}^{n_e} \Phi_{(G,s)}^{(V,i):j,k} [gb]^{(G,s)} + \sum_{s=1}^{n_e} \sum_{q=1}^{D_A} \Phi_{(A,s):q}^{(V,i):j,k} [gb]_{q}^{(A,s)} + \sum_{s=1}^{n_e} \sum_{q=1}^{D} \Phi_{(B,s):q}^{(V,i):j,k} [gb]_{q}^{(B,s)} + \Phi_{1}^{(V,i):j,k},
$$

$$(153)$$

$$[E(gW)]_{j,k}^{(O,i)} = \sum_{s=1}^{h}\sum_{p=1}^{D}\sum_{q=1}^{D}\Phi_{(QK,s):p,q}^{(O,i):j,k}[gWgW]_{p,q}^{(QK,s)}$$

$$+ \sum_{s=1}^{h}\sum_{p=1}^{D}\sum_{q=1}^{D}\Phi_{(VO,s):p,q}^{(O,i):j,k}[gWgW]_{p,q}^{(VO,s)}$$

$$+ \sum_{s=1}^{h}\sum_{p=1}^{D}\sum_{q=1}^{D_k}\Phi_{(Q,s):p,q}^{(O,i):j,k}[gW]_{p,q}^{(Q,s)} + \sum_{s=1}^{h}\sum_{p=1}^{D}\sum_{q=1}^{D_k}\Phi_{(K,s):p,q}^{(O,i):j,k}[gW]_{p,q}^{(K,s)}$$

$$+ \sum_{s=1}^{h}\sum_{p=1}^{D}\sum_{q=1}^{D_v}\Phi_{(V,s):p,q}^{(O,i):j,k}[gW]_{p,q}^{(V,s)} + \sum_{s=1}^{h}\sum_{p=1}^{D_v}\sum_{q=1}^{D}\Phi_{(O,s):p,q}^{(O,i):j,k}[gW]_{p,q}^{(O,s)}$$

$$+ \sum_{s=1}^{n_e}\sum_{p=1}^{D}\Phi_{(G,s):p}^{(O,i):j,k}[gW]_{p}^{(G,s)} + \sum_{s=1}^{n_e}\sum_{p=1}^{D}\sum_{q=1}^{D_A}\Phi_{(A,s):p,q}^{(O,i):j,k}[gW]_{p,q}^{(A,s)} + \sum_{s=1}^{n_e}\sum_{p=1}^{D_A}\sum_{q=1}^{D}\Phi_{(B,s):p,q}^{(O,i):j,k}[gW]_{p,q}^{(B,s)}$$

$$\sum_{s=1}^{n_e}\Phi_{(G,s)}^{(O,i):j,k}[gb]^{(G,s)} + \sum_{s=1}^{n_e}\sum_{q=1}^{D_A}\Phi_{(A,s):q}^{(O,i):j,k}[gb]_{q}^{(A,s)} + \sum_{s=1}^{n_e}\sum_{q=1}^{D}\Phi_{(B,s):q}^{(O,i):j,k}[gb]_{q}^{(B,s)} + \Phi_1^{(O,i):j,k},$$

$$(154)$$

$$[E(gW)]_{j}^{(G,i)} = \sum_{s=1}^{h}\sum_{p=1}^{D}\sum_{q=1}^{D}\Phi_{(QK,s):p,q}^{(G,i):j}[gWgW]_{p,q}^{(QK,s)}$$

$$+ \sum_{s=1}^{h}\sum_{p=1}^{D}\sum_{q=1}^{D}\Phi_{(VO,s):p,q}^{(G,i):j}[gWgW]_{p,q}^{(VO,s)}$$

$$+ \sum_{s=1}^{h}\sum_{p=1}^{D}\sum_{q=1}^{D_k}\Phi_{(Q,s):p,q}^{(G,i):j}[gW]_{p,q}^{(Q,s)} + \sum_{s=1}^{h}\sum_{p=1}^{D}\sum_{q=1}^{D_k}\Phi_{(K,s):p,q}^{(G,i):j}[gW]_{p,q}^{(K,s)}$$

$$+ \sum_{s=1}^{h}\sum_{p=1}^{D}\sum_{q=1}^{D_v}\Phi_{(V,s):p,q}^{(G,i):j}[gW]_{p,q}^{(V,s)} + \sum_{s=1}^{h}\sum_{p=1}^{D_v}\sum_{q=1}^{D}\Phi_{(O,s):p,q}^{(G,i):j}[gW]_{p,q}^{(O,s)}$$

$$+ \sum_{s=1}^{n_e}\sum_{p=1}^{D}\Phi_{(G,s):p}^{(G,i):j}[gW]_{p}^{(G,s)} + \sum_{s=1}^{n_e}\sum_{p=1}^{D}\sum_{q=1}^{D_A}\Phi_{(A,s):p,q}^{(G,i):j}[gW]_{p,q}^{(A,s)} + \sum_{s=1}^{n_e}\sum_{p=1}^{D_A}\sum_{q=1}^{D}\Phi_{(B,s):p,q}^{(G,i):j}[gW]_{p,q}^{(B,s)}$$

$$\sum_{s=1}^{n_e}\Phi_{(G,s)}^{(G,i):j}[gb]^{(G,s)} + \sum_{s=1}^{n_e}\sum_{q=1}^{D_A}\Phi_{(A,s):q}^{(G,i):j}[gb]_{q}^{(A,s)} + \sum_{s=1}^{n_e}\sum_{q=1}^{D}\Phi_{(B,s):q}^{(G,i):j}[gb]_{q}^{(B,s)} + \Phi_1^{(G,i):j},$$

$$(155)$$

$$[E(gW)]_{j,k}^{(A,i)} = \sum_{s=1}^{h}\sum_{p=1}^{D}\sum_{q=1}^{D}\Phi_{(QK,s):p,q}^{(A,i):j,k}[gWgW]_{p,q}^{(QK,s)}$$

$$+ \sum_{s=1}^{h}\sum_{p=1}^{D}\sum_{q=1}^{D}\Phi_{(VO,s):p,q}^{(A,i):j,k}[gWgW]_{p,q}^{(VO,s)}$$

$$+ \sum_{s=1}^{h}\sum_{p=1}^{D}\sum_{q=1}^{D_k}\Phi_{(Q,s):p,q}^{(A,i):j,k}[gW]_{p,q}^{(Q,s)} + \sum_{s=1}^{h}\sum_{p=1}^{D}\sum_{q=1}^{D_k}\Phi_{(K,s):p,q}^{(A,i):j,k}[gW]_{p,q}^{(K,s)}$$

$$+ \sum_{s=1}^{h}\sum_{p=1}^{D}\sum_{q=1}^{D_v}\Phi_{(V,s):p,q}^{(A,i):j,k}[gW]_{p,q}^{(V,s)} + \sum_{s=1}^{h}\sum_{p=1}^{D_v}\sum_{q=1}^{D}\Phi_{(O,s):p,q}^{(A,i):j,k}[gW]_{p,q}^{(O,s)}$$

$$+ \sum_{s=1}^{n_e}\sum_{p=1}^{D}\Phi_{(G,s):p}^{(A,i):j,k}[gW]_{p}^{(G,s)} + \sum_{s=1}^{n_e}\sum_{p=1}^{D}\sum_{q=1}^{D_A}\Phi_{(A,s):p,q}^{(A,i):j,k}[gW]_{p,q}^{(A,s)} + \sum_{s=1}^{n_e}\sum_{p=1}^{D_A}\sum_{q=1}^{D}\Phi_{(B,s):p,q}^{(A,i):j,k}[gW]_{p,q}^{(B,s)}$$

$$\sum_{s=1}^{n_e} \Phi_{(G,s)}^{(A,i):j,k}[gb]^{(G,s)} + \sum_{s=1}^{n_e}\sum_{q=1}^{D_A} \Phi_{(A,s):q}^{(A,i):j,k}[gb]_q^{(A,s)} + \sum_{s=1}^{n_e}\sum_{q=1}^{D} \Phi_{(B,s):q}^{(A,i):j,k}[gb]_q^{(B,s)} + \Phi_1^{(A,i):j,k},$$

$$(156)$$

$$[E(gW)]_{j,k}^{(B,i)} = \sum_{s=1}^{h}\sum_{p=1}^{D}\sum_{q=1}^{D} \Phi_{(QK,s):p,q}^{(B,i):j,k}[gWgW]_{p,q}^{(QK,s)}$$

$$+ \sum_{s=1}^{h}\sum_{p=1}^{D}\sum_{q=1}^{D} \Phi_{(VO,s):p,q}^{(B,i):j,k}[gWgW]_{p,q}^{(VO,s)}$$

$$+ \sum_{s=1}^{h}\sum_{p=1}^{D}\sum_{q=1}^{D_k} \Phi_{(Q,s):p,q}^{(B,i):j,k}[gW]_{p,q}^{(Q,s)} + \sum_{s=1}^{h}\sum_{p=1}^{D}\sum_{q=1}^{D_k} \Phi_{(K,s):p,q}^{(B,i):j,k}[gW]_{p,q}^{(K,s)}$$

$$+ \sum_{s=1}^{h}\sum_{p=1}^{D}\sum_{q=1}^{D_v} \Phi_{(V,s):p,q}^{(B,i):j,k}[gW]_{p,q}^{(V,s)} + \sum_{s=1}^{h}\sum_{p=1}^{D_v}\sum_{q=1}^{D} \Phi_{(O,s):p,q}^{(B,i):j,k}[gW]_{p,q}^{(O,s)}$$

$$+ \sum_{s=1}^{n_e}\sum_{p=1}^{D} \Phi_{(G,s):p}^{(B,i):j,k}[gW]_p^{(G,s)} + \sum_{s=1}^{n_e}\sum_{p=1}^{D}\sum_{q=1}^{D_A} \Phi_{(A,s):p,q}^{(B,i):j,k}[gW]_{p,q}^{(A,s)} + \sum_{s=1}^{n_e}\sum_{p=1}^{D_A}\sum_{q=1}^{D} \Phi_{(B,s):p,q}^{(B,i):j,k}[gW]_{p,q}^{(B,s)}$$

$$\sum_{s=1}^{n_e} \Phi_{(G,s)}^{(B,i):j,k}[gb]^{(G,s)} + \sum_{s=1}^{n_e}\sum_{q=1}^{D_A} \Phi_{(A,s):q}^{(B,i):j,k}[gb]_q^{(A,s)} + \sum_{s=1}^{n_e}\sum_{q=1}^{D} \Phi_{(B,s):q}^{(B,i):j,k}[gb]_q^{(B,s)} + \Phi_1^{(B,i):j,k},$$

$$(157)$$

$$[E(gb)]^{(G,i)} = \sum_{s=1}^{h}\sum_{p=1}^{D}\sum_{q=1}^{D} \Phi_{(QK,s):p,q}^{(G,i)}[gWgW]_{p,q}^{(QK,s)}$$

$$+ \sum_{s=1}^{h}\sum_{p=1}^{D}\sum_{q=1}^{D} \Phi_{(VO,s):p,q}^{(G,i)}[gWgW]_{p,q}^{(VO,s)}$$

$$+ \sum_{s=1}^{h}\sum_{p=1}^{D}\sum_{q=1}^{D_k} \Phi_{(Q,s):p,q}^{(G,i)}[gW]_{p,q}^{(Q,s)} + \sum_{s=1}^{h}\sum_{p=1}^{D}\sum_{q=1}^{D_k} \Phi_{(K,s):p,q}^{(G,i)}[gW]_{p,q}^{(K,s)}$$

$$+ \sum_{s=1}^{h}\sum_{p=1}^{D}\sum_{q=1}^{D_v} \Phi_{(V,s):p,q}^{(G,i)}[gW]_{p,q}^{(V,s)} + \sum_{s=1}^{h}\sum_{p=1}^{D_v}\sum_{q=1}^{D} \Phi_{(O,s):p,q}^{(G,i)}[gW]_{p,q}^{(O,s)}$$

$$+ \sum_{s=1}^{n_e}\sum_{p=1}^{D} \Phi_{(G,s):p}^{(G,i)}[gW]_p^{(G,s)} + \sum_{s=1}^{n_e}\sum_{p=1}^{D}\sum_{q=1}^{D_A} \Phi_{(A,s):p,q}^{(G,i)}[gW]_{p,q}^{(A,s)} + \sum_{s=1}^{n_e}\sum_{p=1}^{D_A}\sum_{q=1}^{D} \Phi_{(B,s):p,q}^{(G,i)}[gW]_{p,q}^{(B,s)}$$

$$\sum_{s=1}^{n_e} \Phi_{(G,s)}^{(G,i)}[gb]^{(G,s)} + \sum_{s=1}^{n_e}\sum_{q=1}^{D_A} \Phi_{(A,s):q}^{(G,i)}[gb]_q^{(A,s)} + \sum_{s=1}^{n_e}\sum_{q=1}^{D} \Phi_{(B,s):q}^{(G,i)}[gb]_q^{(B,s)} + \Phi_1^{(G,i)},$$

$$(158)$$

$$[E(gb)]_j^{(A,i)} = \sum_{s=1}^{h}\sum_{p=1}^{D}\sum_{q=1}^{D} \Phi_{(QK,s):p,q}^{(A,i):j}[gWgW]_{p,q}^{(QK,s)}$$

$$+ \sum_{s=1}^{h}\sum_{p=1}^{D}\sum_{q=1}^{D} \Phi_{(VO,s):p,q}^{(A,i):j}[gWgW]_{p,q}^{(VO,s)}$$

$$+ \sum_{s=1}^{h}\sum_{p=1}^{D}\sum_{q=1}^{D_k} \Phi_{(Q,s):p,q}^{(A,i):j}[gW]_{p,q}^{(Q,s)} + \sum_{s=1}^{h}\sum_{p=1}^{D}\sum_{q=1}^{D_k} \Phi_{(K,s):p,q}^{(A,i):j}[gW]_{p,q}^{(K,s)}$$

$$+ \sum_{s=1}^{h}\sum_{p=1}^{D}\sum_{q=1}^{D_v} \Phi_{(V,s):p,q}^{(A,i):j}[gW]_{p,q}^{(V,s)} + \sum_{s=1}^{h}\sum_{p=1}^{D_v}\sum_{q=1}^{D} \Phi_{(O,s):p,q}^{(A,i):j}[gW]_{p,q}^{(O,s)}$$

$$+ \sum_{s=1}^{n_e} \sum_{p=1}^{D} \Phi_{(G,s):p}^{(A,i):j}[gW]_p^{(G,s)} + \sum_{s=1}^{n_e} \sum_{p=1}^{D} \sum_{q=1}^{D_A} \Phi_{(A,s):p,q}^{(A,i):j}[gW]_{p,q}^{(A,s)} + \sum_{s=1}^{n_e} \sum_{p=1}^{D_A} \sum_{q=1}^{D} \Phi_{(B,s):p,q}^{(A,i):j}[gW]_{p,q}^{(B,s)}$$

$$\sum_{s=1}^{n_e} \Phi_{(G,s)}^{(A,i):j}[gb]^{(G,s)} + \sum_{s=1}^{n_e} \sum_{q=1}^{D_A} \Phi_{(A,s):q}^{(A,i):j}[gb]_q^{(A,s)} + \sum_{s=1}^{n_e} \sum_{q=1}^{D} \Phi_{(B,s):q}^{(A,i):j}[gb]_q^{(B,s)} + \Phi_1^{(A,i):j},$$

$$(159)$$

$$[E(gb)]_j^{(B,i)} = \sum_{s=1}^{h} \sum_{p=1}^{D} \sum_{q=1}^{D} \Phi_{(QK,s):p,q}^{(B,i):j}[gWgW]_{p,q}^{(QK,s)}$$

$$+ \sum_{s=1}^{h} \sum_{p=1}^{D} \sum_{q=1}^{D} \Phi_{(VO,s):p,q}^{(B,i):j}[gWgW]_{p,q}^{(VO,s)}$$

$$+ \sum_{s=1}^{h} \sum_{p=1}^{D} \sum_{q=1}^{D_k} \Phi_{(Q,s):p,q}^{(B,i):j}[gW]_{p,q}^{(Q,s)} + \sum_{s=1}^{h} \sum_{p=1}^{D} \sum_{q=1}^{D_k} \Phi_{(K,s):p,q}^{(B,i):j}[gW]_{p,q}^{(K,s)}$$

$$+ \sum_{s=1}^{h} \sum_{p=1}^{D} \sum_{q=1}^{D_v} \Phi_{(V,s):p,q}^{(B,i):j}[gW]_{p,q}^{(V,s)} + \sum_{s=1}^{h} \sum_{p=1}^{D_v} \sum_{q=1}^{D} \Phi_{(O,s):p,q}^{(B,i):j}[gW]_{p,q}^{(O,s)}$$

$$+ \sum_{s=1}^{n_e} \sum_{p=1}^{D} \Phi_{(G,s):p}^{(B,i):j}[gW]_p^{(G,s)} + \sum_{s=1}^{n_e} \sum_{p=1}^{D} \sum_{q=1}^{D_A} \Phi_{(A,s):p,q}^{(B,i):j}[gW]_{p,q}^{(A,s)} + \sum_{s=1}^{n_e} \sum_{p=1}^{D_A} \sum_{q=1}^{D} \Phi_{(B,s):p,q}^{(B,i):j}[gW]_{p,q}^{(B,s)}$$

$$\sum_{s=1}^{n_e} \Phi_{(G,s)}^{(B,i):j}[gb]^{(G,s)} + \sum_{s=1}^{n_e} \sum_{q=1}^{D_A} \Phi_{(A,s):q}^{(B,i):j}[gb]_q^{(A,s)} + \sum_{s=1}^{n_e} \sum_{q=1}^{D} \Phi_{(B,s):q}^{(B,i):j}[gb]_q^{(B,s)} + \Phi_1^{(B,i):j}.$$

$$(160)$$

Plugging the transformation for each index defined in Equation 145, we obtain:

$$[E(gW)]_{j,k}^{(Q,i)} = \sum_{s=1}^{h} \sum_{p=1}^{D} \sum_{q=1}^{D} \Phi_{(QK,s):p,q}^{(Q,i):j,k}[WW]_{p,q}^{(QK,\tau_h(s))}$$

$$+ \sum_{s=1}^{h} \sum_{p=1}^{D} \sum_{q=1}^{D} \Phi_{(VO,s):p,q}^{(Q,i):j,k}[WW]_{p,q}^{(VO,\tau_h(s))}$$

$$+ \sum_{s=1}^{h} \sum_{p=1}^{D} \sum_{q=1}^{D_k} \Phi_{(Q,s):p,q}^{(Q,i):j,k}\left[[W]^{(Q,\tau_h(s))} \cdot \left(M_k^{(\tau_h(s))}\right)^{\top}\right]_{p,q}$$

$$+ \sum_{s=1}^{h} \sum_{p=1}^{D} \sum_{q=1}^{D_k} \Phi_{(K,s):p,q}^{(Q,i):j,k}\left[[W]^{(K,\tau_h(s))} \cdot \left(M_k^{(\tau_h(s))}\right)^{-1}\right]_{p,q}$$

$$+ \sum_{s=1}^{h} \sum_{p=1}^{D} \sum_{q=1}^{D_v} \Phi_{(V,s):p,q}^{(Q,i):j,k}\left[[W]^{(V,\tau_h(s))} \cdot M_v^{(\tau_h(s))}\right]_{p,q}$$

$$+ \sum_{s=1}^{h} \sum_{p=1}^{D_v} \sum_{q=1}^{D} \Phi_{(O,s):p,q}^{(Q,i):j,k}\left[\left(M_v^{(\tau_h(s))}\right)^{-1} \cdot [W]^{(O,\tau_h(s))}\right]_{p,q}$$

$$+ \sum_{s=1}^{n_e} \sum_{p=1}^{D} \Phi_{(G,s):p}^{(Q,i):j,k}\left[[W]^{(G,\tau_e(s))} + \gamma_W\right]_p$$

$$+ \sum_{s=1}^{n_e} \sum_{p=1}^{D} \sum_{q=1}^{D_A} \Phi_{(A,s):p,q}^{(Q,i):j,k}\left[[W]^{(A,\tau_e(s))} \cdot P_{\pi_e^{(\tau_e(s))}}\right]_{p,q}$$

$$+ \sum_{s=1}^{n_e} \sum_{p=1}^{D_A} \sum_{q=1}^{D} \Phi_{(B,s):p,q}^{(Q,i):j,k} \left[ \left( P_{\pi_e^{(\tau_e(s))}} \right)^{-1} \cdot [W]^{(B,\tau_e(s))} \right]_{p,q}$$

$$+ \sum_{s=1}^{n_e} \Phi_{(G,s)}^{(Q,i):j,k} \left( [b]^{(G,\tau_e(s))} + \gamma_b \right)$$

$$+ \sum_{s=1}^{n_e} \sum_{q=1}^{D_A} \Phi_{(A,s):q}^{(Q,i):j,k} \left[ [b]^{(A,\tau_e(s))} \cdot P_{\pi_e^{(\tau_e(s))}} \right]_q$$

$$+ \sum_{s=1}^{n_e} \sum_{q=1}^{D} \Phi_{(B,s):q}^{(Q,i):j,k} \left[ [b]^{(B,\tau_e(s))} \right]_q$$

$$+ \Phi_1^{(Q,i):j,k}. \tag{161}$$

We observe that

$$[gE(W)]_{j,k}^{(Q,i)} = \left[ [E(W)]^{(Q,\tau_h(i))} \cdot \left( M_k^{(\tau_h(i))} \right)^\top \right]_{j,k} \tag{162}$$

is an $\mathbb{R}$-linear function of $M_k^{(\tau_h(i))}$. Therefore, by equating

$$[E(gW)]_{j,k}^{(Q,i)} = [gE(W)]_{j,k}^{(Q,i)} \tag{163}$$

and applying Lemma F.2, we conclude that the only nonzero parameters $\Phi$ in the expression must correspond to terms that are $\mathbb{R}$-linear functions of $M_k^{(\tau_h(i))}$. Consequently, only the coefficients $\Phi_{(Q,s):p,q}^{(Q,i):j,k}$ can remain nonzero. Thus, we can rewrite the expression for $Q$ component as:

$$[E(gW)]_{j,k}^{(Q,i)} = \sum_{s=1}^{h} \sum_{p=1}^{D} \sum_{q=1}^{D_k} \Phi_{(Q,s):p,q}^{(Q,i):j,k} \left[ W^{(Q,\tau_h(s))} \cdot \left( M_k^{(\tau_h(s))} \right)^\top \right]_{p,q}.$$

Combining the result for the $Q$ component with analogous reasoning applied to $K$, $V$, and $O$, we obtain the following expressions:

$$[E(gW)]_{j,k}^{(Q,i)} = \sum_{s=1}^{h} \sum_{p=1}^{D} \sum_{q=1}^{D_k} \Phi_{(Q,s):p,q}^{(Q,i):j,k} \left[ W^{(Q,\tau_h(s))} \cdot \left( M_k^{(\tau_h(s))} \right)^\top \right]_{p,q},$$

$$[E(gW)]_{j,k}^{(K,i)} = \sum_{s=1}^{h} \sum_{p=1}^{D} \sum_{q=1}^{D_k} \Phi_{(K,s):p,q}^{(K,i):j,k} \left[ [W]^{(K,\tau_h(s))} \cdot \left( M_k^{(\tau_h(s))} \right)^{-1} \right]_{p,q},$$

$$[E(gW)]_{j,k}^{(V,i)} = \sum_{s=1}^{h} \sum_{p=1}^{D} \sum_{q=1}^{D_v} \Phi_{(V,s):p,q}^{(V,i):j,k} \left[ [W]^{(V,\tau_h(s))} \cdot M_v^{(\tau_h(s))} \right]_{p,q},$$

$$[E(gW)]_{j,k}^{(O,i)} = \sum_{s=1}^{h} \sum_{p=1}^{D_v} \sum_{q=1}^{D} \Phi_{(O,s):p,q}^{(O,i):j,k} \left[ \left( M_v^{(\tau_h(s))} \right)^{-1} \cdot [W]^{(O,\tau_h(s))} \right]_{p,q}.$$

Using symmetry of the indices, we obtain:

$$[E(gW)]_{j,k}^{(Q,i)} = \sum_{s=1}^{h} \sum_{p=1}^{D} \sum_{q=1}^{D_k} \Phi_{(Q,\tau_h^{-1}(s)):p,q}^{(Q,i):j,k} \left[ W^{(Q,s)} \cdot \left( M_k^{(s)} \right)^\top \right]_{p,q},$$

$$[E(gW)]_{j,k}^{(K,i)} = \sum_{s=1}^{h} \sum_{p=1}^{D} \sum_{q=1}^{D_k} \Phi_{(K,\tau_h^{-1}(s)):p,q}^{(K,i):j,k} \left[ [W]^{(K,s)} \cdot \left( M_k^{(s)} \right)^{-1} \right]_{p,q},$$

$$[E(gW)]_{j,k}^{(V,i)} = \sum_{s=1}^{h} \sum_{p=1}^{D} \sum_{q=1}^{D_v} \Phi_{(V,\tau_h^{-1}(s)):p,q}^{(V,i):j,k} \left[ [W]^{(V,s)} \cdot M_v^{(s)} \right]_{p,q},$$

$$[E(gW)]_{j,k}^{(O,i)} = \sum_{s=1}^{h} \sum_{p=1}^{D_v} \sum_{q=1}^{D} \Phi_{(O,\tau_h^{-1}(s)):p,q}^{(O,i):j,k} \left[ \left(M_v^{(s)}\right)^{-1} \cdot [W]^{(O,s)} \right]_{p,q}.$$

Now consider the equivariant component corresponding to the gate component. By using the expression of the equivariant layer and plugging in Equation 145, we obtain:

$$\begin{aligned}
[E(gW)]_j^{(G,i)} &= \sum_{s=1}^{h} \sum_{p=1}^{D} \sum_{q=1}^{D} \Phi_{(QK,s):p,q}^{(G,i):j} [WW]_{p,q}^{(QK,\tau_h(s))} \\
&+ \sum_{s=1}^{h} \sum_{p=1}^{D} \sum_{q=1}^{D} \Phi_{(VO,s):p,q}^{(G,i):j} [WW]_{p,q}^{(VO,\tau_h(s))} \\
&+ \sum_{s=1}^{h} \sum_{p=1}^{D} \sum_{q=1}^{D_k} \Phi_{(Q,s):p,q}^{(G,i):j} \left[ [W]^{(Q,\tau_h(s))} \cdot \left(M_k^{(\tau_h(s))}\right)^{\top} \right]_{p,q} \\
&+ \sum_{s=1}^{h} \sum_{p=1}^{D} \sum_{q=1}^{D_k} \Phi_{(K,s):p,q}^{(G,i):j} \left[ [W]^{(K,\tau_h(s))} \cdot \left(M_k^{(\tau_h(s))}\right)^{-1} \right]_{p,q} \\
&+ \sum_{s=1}^{h} \sum_{p=1}^{D} \sum_{q=1}^{D_v} \Phi_{(V,s):p,q}^{(G,i):j} \left[ [W]^{(V,\tau_h(s))} \cdot M_v^{(\tau_h(s))} \right]_{p,q} \\
&+ \sum_{s=1}^{h} \sum_{p=1}^{D_v} \sum_{q=1}^{D} \Phi_{(O,s):p,q}^{(G,i):j} \left[ \left(M_v^{(\tau_h(s))}\right)^{-1} \cdot [W]^{(O,\tau_h(s))} \right]_{p,q} \\
&+ \sum_{s=1}^{n_e} \sum_{p=1}^{D} \Phi_{(G,s):p}^{(G,i):j} \left[ [W]^{(G,\tau_e(s))} + \gamma_W \right]_p \\
&+ \sum_{s=1}^{n_e} \sum_{p=1}^{D} \sum_{q=1}^{D_A} \Phi_{(A,s):p,q}^{(G,i):j} \left[ [W]^{(A,\tau_e(s))} \cdot P_{\pi_e^{(\tau_e(s))}} \right]_{p,q} \\
&+ \sum_{s=1}^{n_e} \sum_{p=1}^{D_A} \sum_{q=1}^{D} \Phi_{(B,s):p,q}^{(G,i):j} \left[ \left(P_{\pi_e^{(\tau_e(s))}}\right)^{-1} \cdot [W]^{(B,\tau_e(s))} \right]_{p,q} \\
&+ \sum_{s=1}^{n_e} \Phi_{(G,s)}^{(G,i):j} \left( [b]^{(G,\tau_e(s))} + \gamma_b \right) \\
&+ \sum_{s=1}^{n_e} \sum_{q=1}^{D_A} \Phi_{(A,s):q}^{(G,i):j} \left[ [b]^{(A,\tau_e(s))} \cdot P_{\pi_e^{(\tau_e(s))}} \right]_q \\
&+ \sum_{s=1}^{n_e} \sum_{q=1}^{D} \Phi_{(B,s):q}^{(G,i):j} \left[ [b]^{(B,\tau_e(s))} \right]_q \\
&+ \Phi_1^{(G,i):j}.
\end{aligned} \tag{164}$$

From Equation 145, we observe that:

$$[gE(W)]_j^{(G,i)} = \left[ [E(W)]^{(G,\tau_e(i))} + \gamma_W \right]_j. \tag{165}$$

When equating $[gE(W)]_j^{(G,i)} = [E(gW)]_j^{(G,i)}$, we notice that all components involving $\mathbb{R}$-linear functions of $M_k^{(\tau_h(s))}$, $(M_k^{(\tau_h(s))})^{-1}$, $M_v^{(\tau_h(s))}$, $(M_v^{(\tau_h(s))})^{-1}$ appear exclusively in $[E(gW)]_j^{(G,i)}$ and not in $[gE(W)]_j^{(G,i)}$. Consequently, the corresponding $\Phi$-parameters corresponding to the inputs from $W_q, W_k, W_v, W_o$ must vanish. This allows us to express the $G$ component of the equivariant layer as:

$$[E(gW)]_j^{(G,i)} = \sum_{s=1}^{h}\sum_{p=1}^{D}\sum_{q=1}^{D} \Phi_{(QK,s):p,q}^{(G,i):j}[WW]_{p,q}^{(QK,\tau_h(s))}$$

$$+ \sum_{s=1}^{h}\sum_{p=1}^{D}\sum_{q=1}^{D} \Phi_{(VO,s):p,q}^{(G,i):j}[WW]_{p,q}^{(VO,\tau_h(s))}$$

$$+ \sum_{s=1}^{n_e}\sum_{p=1}^{D} \Phi_{(G,s):p}^{(G,i):j} \left[[W]^{(G,\tau_e(s))} + \gamma_W\right]_p$$

$$+ \sum_{s=1}^{n_e}\sum_{p=1}^{D}\sum_{q=1}^{D_A} \Phi_{(A,s):p,q}^{(G,i):j} \left[[W]^{(A,\tau_e(s))} \cdot P_{\pi_e^{(\tau_e(s))}}\right]_{p,q}$$

$$+ \sum_{s=1}^{n_e}\sum_{p=1}^{D_A}\sum_{q=1}^{D} \Phi_{(B,s):p,q}^{(G,i):j} \left[\left(P_{\pi_e^{(\tau_e(s))}}\right)^{-1} \cdot [W]^{(B,\tau_e(s))}\right]_{p,q}$$

$$+ \sum_{s=1}^{n_e} \Phi_{(G,s)}^{(G,i):j} \left([b]^{(G,\tau_e(s))} + \gamma_b\right)$$

$$+ \sum_{s=1}^{n_e}\sum_{q=1}^{D_A} \Phi_{(A,s):q}^{(G,i):j} \left[[b]^{(A,\tau_e(s))} \cdot P_{\pi_e^{(\tau_e(s))}}\right]_q$$

$$+ \sum_{s=1}^{n_e}\sum_{q=1}^{D} \Phi_{(B,s):q}^{(G,i):j} \left[[b]^{(B,\tau_e(s))}\right]_q$$

$$+ \Phi_1^{(G,i):j}. \tag{166}$$

Applying the same reasoning to $A$ and $B$ components and combining them with the expression for $G$, we obtain the set of equations:

$$[E(gW)]_j^{(G,i)} = \sum_{s=1}^{h}\sum_{p=1}^{D}\sum_{q=1}^{D} \Phi_{(QK,s):p,q}^{(G,i):j}[WW]_{p,q}^{(QK,\tau_h(s))}$$

$$+ \sum_{s=1}^{h}\sum_{p=1}^{D}\sum_{q=1}^{D} \Phi_{(VO,s):p,q}^{(G,i):j}[WW]_{p,q}^{(VO,\tau_h(s))}$$

$$+ \sum_{s=1}^{n_e}\sum_{p=1}^{D} \Phi_{(G,s):p}^{(G,i):j} \left[[W]^{(G,\tau_e(s))} + \gamma_W\right]_p$$

$$+ \sum_{s=1}^{n_e}\sum_{p=1}^{D}\sum_{q=1}^{D_A} \Phi_{(A,s):p,q}^{(G,i):j} \left[[W]^{(A,\tau_e(s))}\right]_{p,\pi_e^{(\tau_e(s))}(q)}$$

$$+ \sum_{s=1}^{n_e}\sum_{p=1}^{D_A}\sum_{q=1}^{D} \Phi_{(B,s):p,q}^{(G,i):j} \left[[W]^{(B,\tau_e(s))}\right]_{\pi_e^{(\tau_e(s))}(p),q}$$

$$+ \sum_{s=1}^{n_e} \Phi_{(G,s)}^{(G,i):j} \left([b]^{(G,\tau_e(s))} + \gamma_b\right)$$

$$+ \sum_{s=1}^{n_e}\sum_{q=1}^{D_A} \Phi_{(A,s):q}^{(G,i):j} \left[[b]^{(A,\tau_e(s))}\right]_{\pi_e^{(\tau_e(s))}(q)}$$

$$+ \sum_{s=1}^{n_e}\sum_{q=1}^{D} \Phi_{(B,s):q}^{(G,i):j} \left[[b]^{(B,\tau_e(s))}\right]_q$$

$$+ \Phi_1^{(G,i):j}, \tag{167}$$

$$[E(gW)]_{j,k}^{(A,i)} = \sum_{s=1}^{h}\sum_{p=1}^{D}\sum_{q=1}^{D} \Phi_{(QK,s):p,q}^{(A,i):j,k}[WW]_{p,q}^{(QK,\tau_h(s))}$$

$$+ \sum_{s=1}^{h}\sum_{p=1}^{D}\sum_{q=1}^{D} \Phi_{(VO,s):p,q}^{(A,i):j,k}[WW]_{p,q}^{(VO,\tau_h(s))}$$

$$+ \sum_{s=1}^{n_e}\sum_{p=1}^{D} \Phi_{(G,s):p}^{(A,i):j,k} \left[[W]^{(G,\tau_e(s))} + \gamma_W\right]_p$$

$$+ \sum_{s=1}^{n_e}\sum_{p=1}^{D}\sum_{q=1}^{D_A} \Phi_{(A,s):p,q}^{(A,i):j,k} \left[[W]^{(A,\tau_e(s))}\right]_{p,\pi_e^{(\tau_e(s))}(q)}$$

$$+ \sum_{s=1}^{n_e}\sum_{p=1}^{D_A}\sum_{q=1}^{D} \Phi_{(B,s):p,q}^{(A,i):j,k} \left[[W]^{(B,\tau_e(s))}\right]_{\pi_e^{(\tau_e(s))}(p),q}$$

$$+ \sum_{s=1}^{n_e} \Phi_{(G,s)}^{(A,i):j,k} \left([b]^{(G,\tau_e(s))} + \gamma_b\right)$$

$$+ \sum_{s=1}^{n_e}\sum_{q=1}^{D_A} \Phi_{(A,s):q}^{(A,i):j,k} \left[[b]^{(A,\tau_e(s))}\right]_{\pi_e^{(\tau_e(s))}(q)}$$

$$+ \sum_{s=1}^{n_e}\sum_{q=1}^{D} \Phi_{(B,s):q}^{(A,i):j,k} \left[[b]^{(B,\tau_e(s))}\right]_q$$

$$+ \Phi_1^{(A,i):j,k}, \tag{168}$$

$$[E(gW)]_{j,k}^{(B,i)} = \sum_{s=1}^{h}\sum_{p=1}^{D}\sum_{q=1}^{D} \Phi_{(QK,s):p,q}^{(B,i):j,k}[WW]_{p,q}^{(QK,\tau_h(s))}$$

$$+ \sum_{s=1}^{h}\sum_{p=1}^{D}\sum_{q=1}^{D} \Phi_{(VO,s):p,q}^{(B,i):j,k}[WW]_{p,q}^{(VO,\tau_h(s))}$$

$$+ \sum_{s=1}^{n_e}\sum_{p=1}^{D} \Phi_{(G,s):p}^{(B,i):j,k} \left[[W]^{(G,\tau_e(s))} + \gamma_W\right]_p$$

$$+ \sum_{s=1}^{n_e}\sum_{p=1}^{D}\sum_{q=1}^{D_A} \Phi_{(A,s):p,q}^{(B,i):j,k} \left[[W]^{(A,\tau_e(s))}\right]_{p,\pi_e^{(\tau_e(s))}(q)}$$

$$+ \sum_{s=1}^{n_e}\sum_{p=1}^{D_A}\sum_{q=1}^{D} \Phi_{(B,s):p,q}^{(B,i):j,k} \left[[W]^{(B,\tau_e(s))}\right]_{\pi_e^{(\tau_e(s))}(p),q}$$

$$+ \sum_{s=1}^{n_e} \Phi_{(G,s)}^{(B,i):j,k} \left([b]^{(G,\tau_e(s))} + \gamma_b\right)$$

$$+ \sum_{s=1}^{n_e}\sum_{q=1}^{D_A} \Phi_{(A,s):q}^{(B,i):j,k} \left[[b]^{(A,\tau_e(s))}\right]_{\pi_e^{(\tau_e(s))}(q)}$$

$$+ \sum_{s=1}^{n_e}\sum_{q=1}^{D} \Phi_{(B,s):q}^{(B,i):j,k} \left[[b]^{(B,\tau_e(s))}\right]_q$$

$$+ \Phi_1^{(B,i):j,k}, \tag{169}$$

$$[E(gb)]^{(G,i)} = \sum_{s=1}^{h}\sum_{p=1}^{D}\sum_{q=1}^{D}\Phi_{(QK,s):p,q}^{(G,i)}[WW]_{p,q}^{(QK,\tau_h(s))}$$

$$+ \sum_{s=1}^{h}\sum_{p=1}^{D}\sum_{q=1}^{D}\Phi_{(VO,s):p,q}^{(G,i)}[WW]_{p,q}^{(VO,\tau_h(s))}$$

$$+ \sum_{s=1}^{n_e}\sum_{p=1}^{D}\Phi_{(G,s):p}^{(G,i)}\left[[W]^{(G,\tau_e(s))} + \gamma_W\right]_p$$

$$+ \sum_{s=1}^{n_e}\sum_{p=1}^{D}\sum_{q=1}^{D_A}\Phi_{(A,s):p,q}^{(G,i)}\left[[W]^{(A,\tau_e(s))}\right]_{p,\pi_e^{(\tau_e(s))}(q)}$$

$$+ \sum_{s=1}^{n_e}\sum_{p=1}^{D_A}\sum_{q=1}^{D}\Phi_{(B,s):p,q}^{(G,i)}\left[[W]^{(B,\tau_e(s))}\right]_{\pi_e^{(\tau_e(s))}(p),q}$$

$$+ \sum_{s=1}^{n_e}\Phi_{(G,s)}^{(G,i)}\left([b]^{(G,\tau_e(s))} + \gamma_b\right)$$

$$+ \sum_{s=1}^{n_e}\sum_{q=1}^{D_A}\Phi_{(A,s):q}^{(G,i)}\left[[b]^{(A,\tau_e(s))}\right]_{\pi_e^{(\tau_e(s))}(q)}$$

$$+ \sum_{s=1}^{n_e}\sum_{q=1}^{D}\Phi_{(B,s):q}^{(G,i)}\left[[b]^{(B,\tau_e(s))}\right]_q$$

$$+ \Phi_1^{(G,i)}, \tag{170}$$

$$[E(gb)]_j^{(A,i)} = \sum_{s=1}^{h}\sum_{p=1}^{D}\sum_{q=1}^{D}\Phi_{(QK,s):p,q}^{(A,i):j}[WW]_{p,q}^{(QK,\tau_h(s))}$$

$$+ \sum_{s=1}^{h}\sum_{p=1}^{D}\sum_{q=1}^{D}\Phi_{(VO,s):p,q}^{(A,i):j}[WW]_{p,q}^{(VO,\tau_h(s))}$$

$$+ \sum_{s=1}^{n_e}\sum_{p=1}^{D}\Phi_{(G,s):p}^{(A,i):j}\left[[W]^{(G,\tau_e(s))} + \gamma_W\right]_p$$

$$+ \sum_{s=1}^{n_e}\sum_{p=1}^{D}\sum_{q=1}^{D_A}\Phi_{(A,s):p,q}^{(A,i):j}\left[[W]^{(A,\tau_e(s))}\right]_{p,\pi_e^{(\tau_e(s))}(q)}$$

$$+ \sum_{s=1}^{n_e}\sum_{p=1}^{D_A}\sum_{q=1}^{D}\Phi_{(B,s):p,q}^{(A,i):j}\left[[W]^{(B,\tau_e(s))}\right]_{\pi_e^{(\tau_e(s))}(p),q}$$

$$+ \sum_{s=1}^{n_e}\Phi_{(G,s)}^{(A,i):j}\left([b]^{(G,\tau_e(s))} + \gamma_b\right)$$

$$+ \sum_{s=1}^{n_e}\sum_{q=1}^{D_A}\Phi_{(A,s):q}^{(A,i):j}\left[[b]^{(A,\tau_e(s))}\right]_{\pi_e^{(\tau_e(s))}(q)}$$

$$+ \sum_{s=1}^{n_e}\sum_{q=1}^{D}\Phi_{(B,s):q}^{(A,i):j}\left[[b]^{(B,\tau_e(s))}\right]_q$$

$$+ \Phi_1^{(A,i):j}, \tag{171}$$

$$[E(gb)]_j^{(B,i)} = \sum_{s=1}^{h}\sum_{p=1}^{D}\sum_{q=1}^{D}\Phi_{(QK,s):p,q}^{(B,i):j}[WW]_{p,q}^{(QK,\tau_h(s))}$$

$$+ \sum_{s=1}^{h} \sum_{p=1}^{D} \sum_{q=1}^{D} \Phi_{(VO,s):p,q}^{(B,i):j} [WW]_{p,q}^{(VO,\tau_h(s))}$$

$$+ \sum_{s=1}^{n_e} \sum_{p=1}^{D} \Phi_{(G,s):p}^{(B,i):j} \left[ [W]^{(G,\tau_e(s))} + \gamma_W \right]_p$$

$$+ \sum_{s=1}^{n_e} \sum_{p=1}^{D} \sum_{q=1}^{D_A} \Phi_{(A,s):p,q}^{(B,i):j} \left[ [W]^{(A,\tau_e(s))} \right]_{p,\pi_e^{(\tau_e(s))}(q)}$$

$$+ \sum_{s=1}^{n_e} \sum_{p=1}^{D_A} \sum_{q=1}^{D} \Phi_{(B,s):p,q}^{(B,i):j} \left[ [W]^{(B,\tau_e(s))} \right]_{\pi_e^{(\tau_e(s))}(p),q}$$

$$+ \sum_{s=1}^{n_e} \Phi_{(G,s)}^{(B,i):j} \left( [b]^{(G,\tau_e(s))} + \gamma_b \right)$$

$$+ \sum_{s=1}^{n_e} \sum_{q=1}^{D_A} \Phi_{(A,s):q}^{(B,i):j} \left[ [b]^{(A,\tau_e(s))} \right]_{\pi_e^{(\tau_e(s))}(q)}$$

$$+ \sum_{s=1}^{n_e} \sum_{q=1}^{D} \Phi_{(B,s):q}^{(B,i):j} \left[ [b]^{(B,\tau_e(s))} \right]_q$$

$$+ \Phi_1^{(B,i):j}. \tag{172}$$

By making use of index symmetry in the summation:

$$[E(gW)]_j^{(G,i)} = \sum_{s=1}^{h} \sum_{p=1}^{D} \sum_{q=1}^{D} \Phi_{(QK,\tau_h^{-1}(s)):p,q}^{(G,i):j} [WW]_{p,q}^{(QK,s)}$$

$$+ \sum_{s=1}^{h} \sum_{p=1}^{D} \sum_{q=1}^{D} \Phi_{(VO,\tau_h^{-1}(s)):p,q}^{(G,i):j} [WW]_{p,q}^{(VO,s)}$$

$$+ \sum_{s=1}^{n_e} \sum_{p=1}^{D} \Phi_{(G,\tau_e^{-1}(s)):p}^{(G,i):j} \left[ [W]^{(G,s)} + \gamma_W \right]_p$$

$$+ \sum_{s=1}^{n_e} \sum_{p=1}^{D} \sum_{q=1}^{D_A} \Phi_{(A,\tau_e^{-1}(s)):p,(\pi_e^{(s)})^{-1}(q)}^{(G,i):j} \left[ [W]^{(A,s)} \right]_{p,q}$$

$$+ \sum_{s=1}^{n_e} \sum_{p=1}^{D_A} \sum_{q=1}^{D} \Phi_{(B,\tau_e^{-1}(s)):(\pi_e^{(s)})^{-1}(p),q}^{(G,i):j} \left[ [W]^{(B,s)} \right]_{p,q}$$

$$+ \sum_{s=1}^{n_e} \Phi_{(G,\tau_e^{-1}(s))}^{(G,i):j} \left( [b]^{(G,s)} + \gamma_b \right)$$

$$+ \sum_{s=1}^{n_e} \sum_{q=1}^{D_A} \Phi_{(A,\tau_e^{-1}(s)):(\pi_e^{(s)})^{-1}(q)}^{(G,i):j} \left[ [b]^{(A,s)} \right]_q$$

$$+ \sum_{s=1}^{n_e} \sum_{q=1}^{D} \Phi_{(B,\tau_e^{-1}(s)):q}^{(G,i):j} \left[ [b]^{(B,s)} \right]_q$$

$$+ \Phi_1^{(G,i):j}, \tag{173}$$

$$[E(gW)]_{j,k}^{(A,i)} = \sum_{s=1}^{h} \sum_{p=1}^{D} \sum_{q=1}^{D} \Phi_{(QK,\tau_h^{-1}(s)):p,q}^{(A,i):j,k} [WW]_{p,q}^{(QK,s)}$$

$$
+ \sum_{s=1}^{h} \sum_{p=1}^{D} \sum_{q=1}^{D} \Phi^{(A,i):j,k}_{(VO,\tau_h^{-1}(s)):p,q} [WW]^{(VO,s)}_{p,q}
$$

$$
+ \sum_{s=1}^{n_e} \sum_{p=1}^{D} \Phi^{(A,i):j,k}_{(G,\tau_e^{-1}(s)):p} \left[ [W]^{(G,s)} + \gamma_W \right]_p
$$

$$
+ \sum_{s=1}^{n_e} \sum_{p=1}^{D} \sum_{q=1}^{D_A} \Phi^{(A,i):j,k}_{(A,\tau_e^{-1}(s)):p,(\pi_e^{(s)})^{-1}(q)} \left[ [W]^{(A,s)} \right]_{p,q}
$$

$$
+ \sum_{s=1}^{n_e} \sum_{p=1}^{D_A} \sum_{q=1}^{D} \Phi^{(A,i):j,k}_{(B,\tau_e^{-1}(s)):(\pi_e^{(s)})^{-1}(p),q} \left[ [W]^{(B,s)} \right]_{p,q}
$$

$$
+ \sum_{s=1}^{n_e} \Phi^{(A,i):j,k}_{(G,\tau_e^{-1}(s))} \left( [b]^{(G,s)} + \gamma_b \right)
$$

$$
+ \sum_{s=1}^{n_e} \sum_{q=1}^{D_A} \Phi^{(A,i):j,k}_{(A,\tau_e^{-1}(s)):(\pi_e^{(s)})^{-1}(q)} \left[ [b]^{(A,s)} \right]_q
$$

$$
+ \sum_{s=1}^{n_e} \sum_{q=1}^{D} \Phi^{(A,i):j,k}_{(B,\tau_e^{-1}(s)):q} \left[ [b]^{(B,s)} \right]_q
$$

$$
+ \Phi^{(A,i):j,k}_1, \tag{174}
$$

$$
[E(gW)]^{(B,i)}_{j,k} = \sum_{s=1}^{h} \sum_{p=1}^{D} \sum_{q=1}^{D} \Phi^{(B,i):j,k}_{(QK,\tau_h^{-1}(s)):p,q} [WW]^{(QK,s)}_{p,q}
$$

$$
+ \sum_{s=1}^{h} \sum_{p=1}^{D} \sum_{q=1}^{D} \Phi^{(B,i):j,k}_{(VO,\tau_h^{-1}(s)):p,q} [WW]^{(VO,s)}_{p,q}
$$

$$
+ \sum_{s=1}^{n_e} \sum_{p=1}^{D} \Phi^{(B,i):j,k}_{(G,\tau_e^{-1}(s)):p} \left[ [W]^{(G,s)} + \gamma_W \right]_p
$$

$$
+ \sum_{s=1}^{n_e} \sum_{p=1}^{D} \sum_{q=1}^{D_A} \Phi^{(B,i):j,k}_{(A,\tau_e^{-1}(s)):p,(\pi_e^{(s)})^{-1}(q)} \left[ [W]^{(A,s)} \right]_{p,q}
$$

$$
+ \sum_{s=1}^{n_e} \sum_{p=1}^{D_A} \sum_{q=1}^{D} \Phi^{(B,i):j,k}_{(B,\tau_e^{-1}(s)):(\pi_e^{(s)})^{-1}(p),q} \left[ [W]^{(B,s)} \right]_{p,q}
$$

$$
+ \sum_{s=1}^{n_e} \Phi^{(B,i):j,k}_{(G,\tau_e^{-1}(s))} \left( [b]^{(G,s)} + \gamma_b \right)
$$

$$
+ \sum_{s=1}^{n_e} \sum_{q=1}^{D_A} \Phi^{(B,i):j,k}_{(A,\tau_e^{-1}(s)):(\pi_e^{(s)})^{-1}(q)} \left[ [b]^{(A,s)} \right]_q
$$

$$
+ \sum_{s=1}^{n_e} \sum_{q=1}^{D} \Phi^{(B,i):j,k}_{(B,\tau_e^{-1}(s)):q} \left[ [b]^{(B,s)} \right]_q
$$

$$
+ \Phi^{(B,i):j,k}_1, \tag{175}
$$

$$
[E(gb)]^{(G,i)} = \sum_{s=1}^{h} \sum_{p=1}^{D} \sum_{q=1}^{D} \Phi^{(G,i)}_{(QK,\tau_h^{-1}(s)):p,q} [WW]^{(QK,s)}_{p,q}
$$

$$
+ \sum_{s=1}^{h} \sum_{p=1}^{D} \sum_{q=1}^{D} \Phi^{(G,i)}_{(VO,\tau_h^{-1}(s)):p,q} [WW]^{(VO,s)}_{p,q}
$$

$$+ \sum_{s=1}^{n_e} \sum_{p=1}^{D} \Phi^{(G,i)}_{(G,\tau_e^{-1}(s)):p} \left[ [W]^{(G,s)} + \gamma_W \right]_p$$

$$+ \sum_{s=1}^{n_e} \sum_{p=1}^{D} \sum_{q=1}^{D_A} \Phi^{(G,i)}_{(A,\tau_e^{-1}(s)):p,(\pi_e^{(s)})^{-1}(q)} \left[ [W]^{(A,s)} \right]_{p,q}$$

$$+ \sum_{s=1}^{n_e} \sum_{p=1}^{D_A} \sum_{q=1}^{D} \Phi^{(G,i)}_{(B,\tau_e^{-1}(s)):(\pi_e^{(s)})^{-1}(p),q} \left[ [W]^{(B,s)} \right]_{p,q}$$

$$+ \sum_{s=1}^{n_e} \Phi^{(G,i)}_{(G,\tau_e^{-1}(s))} \left( [b]^{(G,s)} + \gamma_b \right)$$

$$+ \sum_{s=1}^{n_e} \sum_{q=1}^{D_A} \Phi^{(G,i)}_{(A,\tau_e^{-1}(s)):(\pi_e^{(s)})^{-1}(q)} \left[ [b]^{(A,s)} \right]_q$$

$$+ \sum_{s=1}^{n_e} \sum_{q=1}^{D} \Phi^{(G,i)}_{(B,\tau_e^{-1}(s)):q} \left[ [b]^{(B,s)} \right]_q$$

$$+ \Phi^{(G,i)}_1, \tag{176}$$

$$[E(gb)]^{(A,i)}_j = \sum_{s=1}^{h} \sum_{p=1}^{D} \sum_{q=1}^{D} \Phi^{(A,i):j}_{(QK,\tau_h^{-1}(s)):p,q} [WW]^{(QK,s)}_{p,q}$$

$$+ \sum_{s=1}^{h} \sum_{p=1}^{D} \sum_{q=1}^{D} \Phi^{(A,i):j}_{(VO,\tau_h^{-1}(s)):p,q} [WW]^{(VO,s)}_{p,q}$$

$$+ \sum_{s=1}^{n_e} \sum_{p=1}^{D} \Phi^{(A,i):j}_{(G,\tau_e^{-1}(s)):p} \left[ [W]^{(G,s)} + \gamma_W \right]_p$$

$$+ \sum_{s=1}^{n_e} \sum_{p=1}^{D} \sum_{q=1}^{D_A} \Phi^{(A,i):j}_{(A,\tau_e^{-1}(s)):p,(\pi_e^{(s)})^{-1}(q)} \left[ [W]^{(A,s)} \right]_{p,q}$$

$$+ \sum_{s=1}^{n_e} \sum_{p=1}^{D_A} \sum_{q=1}^{D} \Phi^{(A,i):j}_{(B,\tau_e^{-1}(s)):(\pi_e^{(s)})^{-1}(p),q} \left[ [W]^{(B,s)} \right]_{p,q}$$

$$+ \sum_{s=1}^{n_e} \Phi^{(A,i):j}_{(G,\tau_e^{-1}(s))} \left( [b]^{(G,s)} + \gamma_b \right)$$

$$+ \sum_{s=1}^{n_e} \sum_{q=1}^{D_A} \Phi^{(A,i):j}_{(A,\tau_e^{-1}(s)):(\pi_e^{(s)})^{-1}(q)} \left[ [b]^{(A,s)} \right]_q$$

$$+ \sum_{s=1}^{n_e} \sum_{q=1}^{D} \Phi^{(A,i):j}_{(B,\tau_e^{-1}(s)):q} \left[ [b]^{(B,s)} \right]_q$$

$$+ \Phi^{(A,i):j}_1, \tag{177}$$

$$[E(gb)]^{(B,i)}_j = \sum_{s=1}^{h} \sum_{p=1}^{D} \sum_{q=1}^{D} \Phi^{(B,i):j}_{(QK,\tau_h^{-1}(s)):p,q} [WW]^{(QK,s)}_{p,q}$$

$$+ \sum_{s=1}^{h} \sum_{p=1}^{D} \sum_{q=1}^{D} \Phi^{(B,i):j}_{(VO,\tau_h^{-1}(s)):p,q} [WW]^{(VO,s)}_{p,q}$$

$$+ \sum_{s=1}^{n_e} \sum_{p=1}^{D} \Phi^{(B,i):j}_{(G,\tau_e^{-1}(s)):p} \left[ [W]^{(G,s)} + \gamma_W \right]_p$$

$$+ \sum_{s=1}^{n_e} \sum_{p=1}^{D} \sum_{q=1}^{D_A} \Phi^{(B,i):j}_{(A,\tau_e^{-1}(s)):p,(\pi_e^{(s)})^{-1}(q)} \left[ [W]^{(A,s)} \right]_{p,q}$$

$$+ \sum_{s=1}^{n_e} \sum_{p=1}^{D_A} \sum_{q=1}^{D} \Phi^{(B,i):j}_{(B,\tau_e^{-1}(s)):(\pi_e^{(s)})^{-1}(p),q} \left[ [W]^{(B,s)} \right]_{p,q}$$

$$+ \sum_{s=1}^{n_e} \Phi^{(B,i):j}_{(G,\tau_e^{-1}(s))} \left( [b]^{(G,s)} + \gamma_b \right)$$

$$+ \sum_{s=1}^{n_e} \sum_{q=1}^{D_A} \Phi^{(B,i):j}_{(A,\tau_e^{-1}(s)):(\pi_e^{(s)})^{-1}(q)} \left[ [b]^{(A,s)} \right]_{q}$$

$$+ \sum_{s=1}^{n_e} \sum_{q=1}^{D} \Phi^{(B,i):j}_{(B,\tau_e^{-1}(s)):q} \left[ [b]^{(B,s)} \right]_{q}$$

$$+ \Phi^{(B,i):j}_1. \tag{178}$$

### F.2 COMPUTING $gE(U)$

Using Equation 143:

$$gE(U) = \left( \left( [gE(W)]^{(Q,i)}, [gE(W)]^{(K,i)}, [gE(W)]^{(V,i)}, [gE(W)]^{(O,i)} \right)_{i=1,\dots,n_h}, \right.$$

$$\left. \left( \left( [gE(W)]^{(G,i)}, [gE(b)]^{(G,i)} \right), \left( [gE(W)]^{(A,i)}, [gE(b)]^{(A,i)} \right), \left( [gE(W)]^{(B,i)}, [gE(b)]^{(B,i)} \right) \right)_{i=1,\dots,n_e} \right) \tag{179}$$

Using Equation 145 to rewrite the group transformation in index-wise form, we obtain:

$$[gE(W)]^{(Q,i)}_{j,k} = \left[ [E(W)]^{(Q,\tau_h(i))} \cdot \left( M_k^{(\tau_h(i))} \right)^{\top} \right]_{j,k}$$

$$= \sum_{l=1}^{D_k} [E(W)]^{(Q,\tau(i))}_{j,l} \cdot \left( M^{(\tau(i))} \right)^{\top}_{l,k}$$

$$= \sum_{l=1}^{D_k} M^{(\tau(i))}_{k,l} \cdot \sum_{s=1}^{h} \sum_{p=1}^{D} \sum_{q=1}^{D_k} \Phi^{(Q,\tau(i)):j,l}_{(Q,s):p,q} [W]^{(Q,s)}_{p,q}, \tag{180}$$

$$[gE(W)]^{(K,i)}_{j,k} = \left[ [E(W)]^{(K,\tau_h(i))} \cdot \left( M_k^{(\tau_h(i))} \right)^{-1} \right]_{j,k}$$

$$= \sum_{l=1}^{D_k} [E(W)]^{(K,\tau(i))}_{j,l} \cdot \left( M^{(\tau(i))} \right)^{-1}_{l,k}$$

$$= \sum_{l=1}^{D_k} \left( M^{(\tau(i))} \right)^{-1}_{l,k} \cdot \sum_{s=1}^{h} \sum_{p=1}^{D} \sum_{q=1}^{D_k} \Phi^{(K,\tau(i)):j,l}_{(K,s):p,q} [W]^{(K,s)}_{p,q}, \tag{181}$$

$$[gE(W)]^{(V,i)}_{j,k} = \left[ [E(W)]^{(V,\tau_h(i))} \cdot M_v^{(\tau_h(i))} \right]_{j,k}$$

$$= \sum_{l=1}^{D_v} [E(W)]^{(V,\tau_h(i))}_{j,l} \cdot (M_v^{(\tau_h(i))})_{l,k}$$

$$= \sum_{l=1}^{D_v} (M_v^{(\tau_h(i))})_{l,k} \cdot \sum_{s=1}^{h} \sum_{p=1}^{D} \sum_{q=1}^{D_k} \Phi_{(V,s):p,q}^{(V,\tau(i)):j,l} [W]_{p,q}^{(V,s)}, \tag{182}$$

$$[gE(W)]_{j,k}^{(O,i)} = \left[ \left( M_v^{(\tau_h(i))} \right)^{-1} \cdot [E(W)]^{(O,\tau_h(i))} \right]_{j,k}$$

$$= \sum_{l=1}^{D_v} \left( \left( M_v^{(\tau_h(i))} \right)^{-1} \right)_{j,l} \cdot [E(W)]_{l,k}^{(O,\tau_h(i))}$$

$$= \sum_{l=1}^{D_v} \left( \left( M_v^{(\tau_h(i))} \right)^{-1} \right)_{j,l} \cdot \sum_{s=1}^{h} \sum_{p=1}^{D_v} \sum_{q=1}^{D} \Phi_{(O,s):p,q}^{(O,\tau(i)):l,k} [W]_{p,q}^{(V,s)}, \tag{183}$$

$$[gE(W)]_j^{(G,i)} = [E(W)]_j^{(G,\tau_e(i))} + (\gamma_W)_j$$

$$= \sum_{s=1}^{h} \sum_{p=1}^{D} \sum_{q=1}^{D} \Phi_{(QK,s):p,q}^{(G,\tau_e(i)):j} [WW]_{p,q}^{(QK,s)} + \sum_{s=1}^{h} \sum_{p=1}^{D} \sum_{q=1}^{D} \Phi_{(VO,s):p,q}^{(G,\tau_e(i)):j} [WW]_{p,q}^{(VO,s)}$$

$$+ \sum_{s=1}^{n_e} \sum_{p=1}^{D} \Phi_{(G,s):p}^{(G,\tau_e(i)):j} [W]_p^{(G,s)} + \sum_{s=1}^{n_e} \sum_{p=1}^{D} \sum_{q=1}^{D_A} \Phi_{(A,s):p,q}^{(G,\tau_e(i)):j} [W]_{p,q}^{(A,s)} + \sum_{s=1}^{n_e} \sum_{p=1}^{D_A} \sum_{q=1}^{D} \Phi_{(B,s):p,q}^{(G,\tau_e(i)):j} [W]_{p,q}^{(B,s)}$$

$$+ \sum_{s=1}^{n_e} \Phi_{(G,s)}^{(G,\tau_e(i)):j} [b]^{(G,s)} + \sum_{s=1}^{n_e} \sum_{q=1}^{D_A} \Phi_{(A,s):q}^{(G,\tau_e(i)):j} [b]_q^{(A,s)} + \sum_{s=1}^{n_e} \sum_{q=1}^{D} \Phi_{(B,s):q}^{(G,\tau_e(i)):j} [b]_q^{(B,s)} + \Phi_1^{(G,\tau_e(i)):j} + (\gamma_W)_j, \tag{184}$$

$$[gE(W)]_{j,k}^{(A,i)} = [E(W)]_{j,\pi_e^{(\tau_e(i))}(k)}^{(A,\tau_e(i))}$$

$$= \sum_{s=1}^{h} \sum_{p=1}^{D} \sum_{q=1}^{D} \Phi_{(QK,s):p,q}^{(A,\tau_e(i)):j,\pi_e^{(\tau_e(i))}(k)} [WW]_{p,q}^{(QK,s)} + \sum_{s=1}^{h} \sum_{p=1}^{D} \sum_{q=1}^{D} \Phi_{(VO,s):p,q}^{(A,\tau_e(i)):j,\pi_e^{(\tau_e(i))}(k)} [WW]_{p,q}^{(VO,s)}$$

$$+ \sum_{s=1}^{n_e} \sum_{p=1}^{D} \Phi_{(G,s):p}^{(A,\tau_e(i)):j,\pi_e^{(\tau_e(i))}(k)} [W]_p^{(G,s)} + \sum_{s=1}^{n_e} \sum_{p=1}^{D} \sum_{q=1}^{D_A} \Phi_{(A,s):p,q}^{(A,\tau_e(i)):j,\pi_e^{(\tau_e(i))}(k)} [W]_{p,q}^{(A,s)}$$

$$+ \sum_{s=1}^{n_e} \sum_{p=1}^{D_A} \sum_{q=1}^{D} \Phi_{(B,s):p,q}^{(A,\tau_e(i)):j,\pi_e^{(\tau_e(i))}(k)} [W]_{p,q}^{(B,s)} + \sum_{s=1}^{n_e} \Phi_{(G,s)}^{(A,\tau_e(i)):j,\pi_e^{(\tau_e(i))}(k)} [b]^{(G,s)} + \sum_{s=1}^{n_e} \sum_{q=1}^{D_A} \Phi_{(A,s):q}^{(A,\tau_e(i)):j,\pi_e^{(\tau_e(i))}(k)} [b]_q^{(A,s)}$$

$$+ \sum_{s=1}^{n_e} \sum_{q=1}^{D} \Phi_{(B,s):q}^{(A,\tau_e(i)):j,\pi_e^{(\tau_e(i))}(k)} [b]_q^{(B,s)} + \Phi_1^{(A,\tau_e(i)):j,\pi_e^{(\tau_e(i))}(k)}, \tag{185}$$

$$[gE(W)]_{j,k}^{(B,i)} = [E(W)]_{\pi_e^{(\tau_e(i))}(j),k}^{(B,\tau_e(i))}$$

$$= \sum_{s=1}^{h} \sum_{p=1}^{D} \sum_{q=1}^{D} \Phi_{(QK,s):p,q}^{(B,\tau_e(i)):\pi_e^{(\tau_e(i))}(j),k} [WW]_{p,q}^{(QK,s)} + \sum_{s=1}^{h} \sum_{p=1}^{D} \sum_{q=1}^{D} \Phi_{(VO,s):p,q}^{(B,\tau_e(i)):\pi_e^{(\tau_e(i))}(j),k} [WW]_{p,q}^{(VO,s)}$$

$$+ \sum_{s=1}^{n_e} \sum_{p=1}^{D} \Phi_{(G,s):p}^{(B,\tau_e(i)):\pi_e^{(\tau_e(i))}(j),k} [W]_p^{(G,s)} + \sum_{s=1}^{n_e} \sum_{p=1}^{D} \sum_{q=1}^{D_A} \Phi_{(A,s):p,q}^{(B,\tau_e(i)):\pi_e^{(\tau_e(i))}(j),k} [W]_{p,q}^{(A,s)}$$

$$+ \sum_{s=1}^{n_e} \sum_{p=1}^{D_A} \sum_{q=1}^{D} \Phi_{(B,s):p,q}^{(B,\tau_e(i)):\pi_e^{(\tau_e(i))}(j),k} [W]_{p,q}^{(B,s)} + \sum_{s=1}^{n_e} \Phi_{(G,s)}^{(B,\tau_e(i)):\pi_e^{(\tau_e(i))}(j),k} [b]^{(G,s)} + \sum_{s=1}^{n_e} \sum_{q=1}^{D_A} \Phi_{(A,s):q}^{(B,\tau_e(i)):\pi_e^{(\tau_e(i))}(j),k} [b]_q^{(A,s)}$$

$$+ \sum_{s=1}^{n_e} \sum_{q=1}^{D} \Phi_{(B,s):q}^{(B,\tau_e(i)):\pi_e^{(\tau_e(i))}(j),k} [b]_q^{(B,s)} + \Phi_1^{(B,\tau_e(i)):\pi_e^{(\tau_e(i))}(j),k}, \tag{186}$$

$$[gE(b)]^{(G,i)} = [E(b)]^{(G,\tau_e(i))} + \gamma_b$$

$$= \sum_{s=1}^{h} \sum_{p=1}^{D} \sum_{q=1}^{D} \Phi_{(QK,s):p,q}^{(G,\tau_e(i))} [WW]_{p,q}^{(QK,s)} + \sum_{s=1}^{h} \sum_{p=1}^{D} \sum_{q=1}^{D} \Phi_{(VO,s):p,q}^{(G,\tau_e(i))} [WW]_{p,q}^{(VO,s)}$$

$$+ \sum_{s=1}^{n_e} \sum_{p=1}^{D} \Phi_{(G,s):p}^{(G,\tau_e(i))} [W]_p^{(G,s)} + \sum_{s=1}^{n_e} \sum_{p=1}^{D} \sum_{q=1}^{D_A} \Phi_{(A,s):p,q}^{(G,\tau_e(i))} [W]_{p,q}^{(A,s)} + \sum_{s=1}^{n_e} \sum_{p=1}^{D_A} \sum_{q=1}^{D} \Phi_{(B,s):p,q}^{(G,\tau_e(i))} [W]_{p,q}^{(B,s)}$$

$$+ \sum_{s=1}^{n_e} \Phi_{(G,s)}^{(G,\tau_e(i))} [b]^{(G,s)} + \sum_{s=1}^{n_e} \sum_{q=1}^{D_A} \Phi_{(A,s):q}^{(G,\tau_e(i))} [b]_q^{(A,s)} + \sum_{s=1}^{n_e} \sum_{q=1}^{D} \Phi_{(B,s):q}^{(G,\tau_e(i))} [b]_q^{(B,s)} + \Phi_1^{(G,\tau_e(i))} + \gamma_b,$$

$$(187)$$

$$[gE(b)]_j^{(A,i)} = [E(b)]_{\pi_e^{(\tau_e(i))}(j)}^{(A,\tau_e(i))}$$

$$= \sum_{s=1}^{h} \sum_{p=1}^{D} \sum_{q=1}^{D} \Phi_{(QK,s):p,q}^{(A,\tau_e(i)):\pi_e^{(\tau_e(i))}(j)} [WW]_{p,q}^{(QK,s)} + \sum_{s=1}^{h} \sum_{p=1}^{D} \sum_{q=1}^{D} \Phi_{(VO,s):p,q}^{(A,\tau_e(i)):\pi_e^{(\tau_e(i))}(j)} [WW]_{p,q}^{(VO,s)}$$

$$+ \sum_{s=1}^{n_e} \sum_{p=1}^{D} \Phi_{(G,s):p}^{(A,\tau_e(i)):\pi_e^{(\tau_e(i))}(j)} [W]_p^{(G,s)} + \sum_{s=1}^{n_e} \sum_{p=1}^{D} \sum_{q=1}^{D_A} \Phi_{(A,s):p,q}^{(A,\tau_e(i)):\pi_e^{(\tau_e(i))}(j)} [W]_{p,q}^{(A,s)}$$

$$+ \sum_{s=1}^{n_e} \sum_{p=1}^{D_A} \sum_{q=1}^{D} \Phi_{(B,s):p,q}^{(A,\tau_e(i)):\pi_e^{(\tau_e(i))}(j)} [W]_{p,q}^{(B,s)} + \sum_{s=1}^{n_e} \Phi_{(G,s)}^{(A,\tau_e(i)):\pi_e^{(\tau_e(i))}(j)} [b]^{(G,s)}$$

$$+ \sum_{s=1}^{n_e} \sum_{q=1}^{D_A} \Phi_{(A,s):q}^{(A,\tau_e(i)):\pi_e^{(\tau_e(i))}(j)} [b]_q^{(A,s)} + \sum_{s=1}^{n_e} \sum_{q=1}^{D} \Phi_{(B,s):q}^{(A,\tau_e(i)):\pi_e^{(\tau_e(i))}(j)} [b]_q^{(B,s)} + \Phi_1^{(A,\tau_e(i)):\pi_e^{(\tau_e(i))}(j)},$$

$$(188)$$

$$[gE(b)]_j^{(B,i)} = [E(b)]_j^{(B,\tau_e(i))}$$

$$= \sum_{s=1}^{h} \sum_{p=1}^{D} \sum_{q=1}^{D} \Phi_{(QK,s):p,q}^{(B,\tau_e(i)):j} [WW]_{p,q}^{(QK,s)} + \sum_{s=1}^{h} \sum_{p=1}^{D} \sum_{q=1}^{D} \Phi_{(VO,s):p,q}^{(B,\tau_e(i)):j} [WW]_{p,q}^{(VO,s)}$$

$$+ \sum_{s=1}^{n_e} \sum_{p=1}^{D} \Phi_{(G,s):p}^{(B,\tau_e(i)):j} [W]_p^{(G,s)} + \sum_{s=1}^{n_e} \sum_{p=1}^{D} \sum_{q=1}^{D_A} \Phi_{(A,s):p,q}^{(B,\tau_e(i)):j} [W]_{p,q}^{(A,s)} + \sum_{s=1}^{n_e} \sum_{p=1}^{D_A} \sum_{q=1}^{D} \Phi_{(B,s):p,q}^{(B,\tau_e(i)):j} [W]_{p,q}^{(B,s)}$$

$$\sum_{s=1}^{n_e} \Phi_{(G,s)}^{(B,\tau_e(i)):j} [b]^{(G,s)} + \sum_{s=1}^{n_e} \sum_{q=1}^{D_A} \Phi_{(A,s):q}^{(B,\tau_e(i)):j} [b]_q^{(A,s)} + \sum_{s=1}^{n_e} \sum_{q=1}^{D} \Phi_{(B,s):q}^{(B,\tau_e(i)):j} [b]_q^{(B,s)} + \Phi_1^{(B,\tau_e(i)):j}.$$

$$(189)$$

## F.3 COMPARE COEFFICIENTS FROM EQUATION $E(gU) = gE(U)$

To enforce equivariance property, we solve the following equalities to identify the constraints on the parameters $\Phi$:

$$[E(gW)]_{j,k}^{(Q,i)} = [gE(W)]_{j,k}^{(Q,i)},$$
$$[E(gW)]_{j,k}^{(K,i)} = [gE(W)]_{j,k}^{(K,i)},$$
$$[E(gW)]_{j,k}^{(V,i)} = [gE(W)]_{j,k}^{(V,i)},$$
$$[E(gW)]_{j,k}^{(O,i)} = [gE(W)]_{j,k}^{(O,i)},$$
$$[E(gW)]_{j}^{(G,i)} = [gE(W)]_{j}^{(G,i)},$$

$$[E(gW)]_{j,k}^{(A,i)} = [gE(W)]_{j,k}^{(A,i)},$$

$$[E(gW)]_{j,k}^{(B,i)} = [gE(W)]_{j,k}^{(B,i)},$$

$$[E(gb)]^{(G,i)} = [gE(b)]^{(A,i)},$$

$$[E(gb)]_{j}^{(A,i)} = [gE(b)]_{j}^{(A,i)},$$

$$[E(gb)]_{j}^{(B,i)} = [gE(b)]_{j}^{(B,i)}.$$

We break the process into multiple steps to solve each constraint as follows.

**Step 1. Solving $[E(gW)]_{j,k}^{(Q,i)} = [gE(W)]_{j,k}^{(Q,i)}$.**
For this equality, by following the same argument in (Tran et al., 2025, Appendix D.3.3), we see that

$$\Phi_{(Q,i):p,k}^{(Q,i):j,k} = \Phi_{(Q,\tau(i)):p,k'}^{(Q,\tau(i)):j,k'}. \tag{190}$$

**Step 2. Solving $[E(gW)]_{j,k}^{(K,i)} = [gE(W)]_{j,k}^{(K,i)}$.**
For this equality, by following the same argument in (Tran et al., 2025, Appendix D.3.3), we see that

$$\Phi_{(K,i):p,k}^{(K,i):j,k} = \Phi_{(K,\tau(i)):p,k'}^{(K,\tau(i)):j,k'}. \tag{191}$$

**Step 3. Solving $[E(gW)]_{j,k}^{(V,i)} = [gE(W)]_{j,k}^{(V,i)}$.**
For this equality, by following the same argument in (Tran et al., 2025, Appendix D.3.3), we see that

$$\Phi_{(V,i):p,k}^{(V,i):j,k} = \Phi_{(V,\tau(i)):p,k'}^{(V,\tau(i)):j,k'}. \tag{192}$$

**Step 4. Solving $[E(gW)]_{j,k}^{(O,i)} = [gE(W)]_{j,k}^{(O,i)}$.**
For this equality, by following the same argument in (Tran et al., 2025, Appendix D.3.3), we see that

$$\Phi_{(O,i):j,q}^{(O,i):j,k} = \Phi_{(O,\tau(i)):j',q}^{(O,\tau(i)):j',k}. \tag{193}$$

and all other indices equal to 0.

**Step 5. Solving $[E(gW)]_{j}^{(G,i)} = [gE(W)]_{j}^{(G,i)}$.**
To solve the constraint for this equation, we expand both sides in full and apply the index-wise group action defined in Equation 145, which yields:

$$\sum_{s=1}^{h}\sum_{p=1}^{D}\sum_{q=1}^{D}\Phi_{(QK,\tau_h^{-1}(s)):p,q}^{(G,i):j}[WW]_{p,q}^{(QK,s)} + \sum_{s=1}^{h}\sum_{p=1}^{D}\sum_{q=1}^{D}\Phi_{(VO,\tau_h^{-1}(s)):p,q}^{(G,i):j}[WW]_{p,q}^{(VO,s)}$$

$$+ \sum_{s=1}^{n_e}\sum_{p=1}^{D}\Phi_{(G,\tau_e^{-1}(s)):p}^{(G,i):j}\left[[W]^{(G,s)} + \gamma_W\right]_p + \sum_{s=1}^{n_e}\sum_{p=1}^{D}\sum_{q=1}^{D_A}\Phi_{(A,\tau_e^{-1}(s)):p,(\pi_e^{(s)})^{-1}(q)}^{(G,i):j}\left[[W]^{(A,s)}\right]_{p,q}$$

$$+ \sum_{s=1}^{n_e}\sum_{p=1}^{D_A}\sum_{q=1}^{D}\Phi_{(B,\tau_e^{-1}(s)):(\pi_e^{(s)})^{-1}(p),q}^{(G,i):j}\left[[W]^{(B,s)}\right]_{p,q} + \sum_{s=1}^{n_e}\Phi_{(G,\tau_e^{-1}(s))}^{(G,i):j}\left([b]^{(G,s)} + \gamma_b\right)$$

$$+ \sum_{s=1}^{n_e}\sum_{q=1}^{D_A}\Phi_{(A,\tau_e^{-1}(s)):(\pi_e^{(s)})^{-1}(q)}^{(G,i):j}\left[[b]^{(A,s)}\right]_q + \sum_{s=1}^{n_e}\sum_{q=1}^{D}\Phi_{(B,\tau_e^{-1}(s)):q}^{(G,i):j}\left[[b]^{(B,s)}\right]_q + \Phi_1^{(G,i):j}$$

$$= \sum_{s=1}^{h}\sum_{p=1}^{D}\sum_{q=1}^{D}\Phi_{(QK,s):p,q}^{(G,\tau_e(i)):j}[WW]_{p,q}^{(QK,s)} + \sum_{s=1}^{h}\sum_{p=1}^{D}\sum_{q=1}^{D}\Phi_{(VO,s):p,q}^{(G,\tau_e(i)):j}[WW]_{p,q}^{(VO,s)}$$

$$+ \sum_{s=1}^{n_e} \sum_{p=1}^{D} \Phi_{(G,s):p}^{(G,\tau_e(i)):j} [W]_p^{(G,s)} + \sum_{s=1}^{n_e} \sum_{p=1}^{D} \sum_{q=1}^{D_A} \Phi_{(A,s):p,q}^{(G,\tau_e(i)):j} [W]_{p,q}^{(A,s)}$$

$$+ \sum_{s=1}^{n_e} \sum_{p=1}^{D_A} \sum_{q=1}^{D} \Phi_{(B,s):p,q}^{(G,\tau_e(i)):j} [W]_{p,q}^{(B,s)} + \sum_{s=1}^{n_e} \Phi_{(G,s)}^{(G,\tau_e(i)):j} [b]^{(G,s)}$$

$$+ \sum_{s=1}^{n_e} \sum_{q=1}^{D_A} \Phi_{(A,s):q}^{(G,\tau_e(i)):j} [b]_q^{(A,s)} + \sum_{s=1}^{n_e} \sum_{q=1}^{D} \Phi_{(B,s):q}^{(G,\tau_e(i)):j} [b]_q^{(B,s)} + \Phi_1^{(G,\tau_e(i)):j} + (\gamma_W)_j. \tag{194}$$

Using lemma F.1, we obtain the constraints:

$$\Phi_{(QK,\tau_h^{-1}(s)):p,q}^{(G,i):j} = \Phi_{(QK,s):p,q}^{(G,\tau_e(i)):j},$$

$$\Phi_{(VO,\tau_h^{-1}(s)):p,q}^{(G,i):j} = \Phi_{(VO,s):p,q}^{(G,\tau_e(i)):j},$$

$$\Phi_{(G,\tau_e^{-1}(s)):p}^{(G,i):j} = \Phi_{(G,s):p}^{(G,\tau_e(i)):j},$$

$$\Phi_{(A,\tau_e^{-1}(s)):p,(\pi_e^{(s)})^{-1}(q)}^{(G,i):j} = \Phi_{(A,s):p,q}^{(G,\tau_e(i)):j},$$

$$\Phi_{(B,\tau_e^{-1}(s)):(\pi_e^{(s)})^{-1}(p),q}^{(G,i):j} = \Phi_{(B,s):p,q}^{(G,\tau_e(i)):j},$$

$$\Phi_{(G,\tau_e^{-1}(s))}^{(G,i):j} = \Phi_{(G,s)}^{(G,\tau_e(i)):j},$$

$$\Phi_{(A,\tau_e^{-1}(s)):(\pi_e^{(s)})^{-1}(q)}^{(G,i):j} = \Phi_{(A,s):q}^{(G,\tau_e(i)):j},$$

$$\Phi_{(B,\tau_e^{-1}(s)):q}^{(G,i):j} = \Phi_{(B,s):q}^{(G,\tau_e(i)):j},$$

$$\Phi_1^{(G,i):j} = \Phi_1^{(G,\tau_e(i)):j},$$

$$\sum_{s=1}^{n_e} \sum_{p=1}^{D} \Phi_{(G,\tau_e^{-1}(s)):p}^{(G,i):j} [\gamma_W]_p = (\gamma_W)_j,$$

$$\sum_{s=1}^{n_e} \Phi_{(G,s)}^{(G,i):j} = 0.$$

By a change of indexes, we obtain:

$$\Phi_{(QK,s):p,q}^{(G,i):j} = \Phi_{(QK,\tau_h(s)):p,q}^{(G,\tau_e(i)):j},$$

$$\Phi_{(VO,s):p,q}^{(G,i):j} = \Phi_{(VO,\tau_h(s)):p,q}^{(G,\tau_e(i)):j},$$

$$\Phi_{(G,s):p}^{(G,i):j} = \Phi_{(G,\tau_e(s)):p}^{(G,\tau_e(i)):j},$$

$$\Phi_{(A,s):p,q}^{(G,i):j} = \Phi_{(A,\tau_e(s)):p,\pi_e^{(\tau_e(s))}(q)}^{(G,\tau_e(i)):j},$$

$$\Phi_{(B,s):p,q}^{(G,i):j} = \Phi_{(B,\tau_e(s)):\pi_e^{(\tau_e(s))}(p),q}^{(G,\tau_e(i)):j},$$

$$\Phi_{(G,s)}^{(G,i):j} = \Phi_{(G,\tau_e(s))}^{(G,\tau_e(i)):j},$$

$$\Phi_{(A,s):q}^{(G,i):j} = \Phi_{(A,\tau_e(s)):\pi_e^{(\tau_e(s))}(q)}^{(G,\tau_e(i)):j},$$

$$\Phi_{(B,s):q}^{(G,i):j} = \Phi_{(B,\tau_e(s)):q}^{(G,\tau_e(i)):j},$$

$$\Phi_1^{(G,i):j} = \Phi_1^{(G,\tau_e(i)):j},$$

$$\sum_{s=1}^{n_e} \sum_{p=1}^{D} \Phi_{(G,s):p}^{(G,i):j} [\gamma_W]_p = (\gamma_W)_j,$$

$$\sum_{s=1}^{n_e} \Phi_{(G,s)}^{(G,i):j} = 0.$$

As a consequence, we have:

$$\Phi_{(QK,s):p,q}^{(G,i):j} = \Phi_{(QK,\tau_h(s)):p,q}^{(G,\tau_e(i)):j},$$

$$\Phi_{(VO,s):p,q}^{(G,i):j} = \Phi_{(VO,\tau_h(s)):p,q}^{(G,\tau_e(i)):j},$$

$$\Phi_{(G,s):p}^{(G,i):j} = \Phi_{(G,\tau_e(s)):p}^{(G,\tau_e(i)):j},$$

$$\Phi_{(A,s):p,q}^{(G,i):j} = \Phi_{(A,\tau_e(s)):p,\pi_e^{(\tau_e(s))}(q)}^{(G,\tau_e(i)):j},$$

$$\Phi_{(B,s):p,q}^{(G,i):j} = \Phi_{(B,\tau_e(s)):\pi_e^{(\tau_e(s))}(p),q}^{(G,\tau_e(i)):j},$$

$$\Phi_{(G,s)}^{(G,i):j} = \Phi_{(G,\tau_e(s))}^{(G,\tau_e(i)):j},$$

$$\Phi_{(A,s):q}^{(G,i):j} = \Phi_{(A,\tau_e(s)):\pi_e^{(\tau_e(s))}(q)}^{(G,\tau_e(i)):j},$$

$$\Phi_{(B,s):q}^{(G,i):j} = \Phi_{(B,\tau_e(s)):q}^{(G,\tau_e(i)):j},$$

$$\Phi_1^{(G,i):j} = \Phi_1^{(G,\tau_e(i)):j},$$

$$\sum_{s=1}^{n_e} \Phi_{(G,s):p}^{(G,i):j} = 0 \quad (p \neq j),$$

$$\sum_{s=1}^{n_e} \Phi_{(G,s):j}^{(G,i):j} = 1,$$

$$\sum_{s=1}^{n_e} \Phi_{(G,s)}^{(G,i):j} = 0. \tag{195}$$

**Step 6. Solving** $[E(gW)]_{j,k}^{(A,i)} = [gE(W)]_{j,k}^{(A,i)}$.

For this equation, we proceed as follow:

$$\sum_{s=1}^{h}\sum_{p=1}^{D}\sum_{q=1}^{D} \Phi_{(QK,\tau_h^{-1}(s)):p,q}^{(A,i):j,k}[WW]_{p,q}^{(QK,s)} + \sum_{s=1}^{h}\sum_{p=1}^{D}\sum_{q=1}^{D} \Phi_{(VO,\tau_h^{-1}(s)):p,q}^{(A,i):j,k}[WW]_{p,q}^{(VO,s)}$$

$$+ \sum_{s=1}^{n_e}\sum_{p=1}^{D} \Phi_{(G,\tau_e^{-1}(s)):p}^{(A,i):j,k} \left[[W]^{(G,s)} + \gamma_W\right]_p + \sum_{s=1}^{n_e}\sum_{p=1}^{D}\sum_{q=1}^{D_A} \Phi_{(A,\tau_e^{-1}(s)):p,(\pi_e^{(s)})^{-1}(q)}^{(A,i):j,k} \left[[W]^{(A,s)}\right]_{p,q}$$

$$+ \sum_{s=1}^{n_e}\sum_{p=1}^{D_A}\sum_{q=1}^{D} \Phi_{(B,\tau_e^{-1}(s)):(\pi_e^{(s)})^{-1}(p),q}^{(A,i):j,k} \left[[W]^{(B,s)}\right]_{p,q} + \sum_{s=1}^{n_e} \Phi_{(G,\tau_e^{-1}(s))}^{(A,i):j,k} \left([b]^{(G,s)} + \gamma_b\right)$$

$$+ \sum_{s=1}^{n_e}\sum_{q=1}^{D_A} \Phi_{(A,\tau_e^{-1}(s)):(\pi_e^{(s)})^{-1}(q)}^{(A,i):j,k} \left[[b]^{(A,s)}\right]_q + \sum_{s=1}^{n_e}\sum_{q=1}^{D} \Phi_{(B,\tau_e^{-1}(s)):q}^{(A,i):j,k} \left[[b]^{(B,s)}\right]_q + \Phi_1^{(A,i):j,k}$$

$$= \sum_{s=1}^{h}\sum_{p=1}^{D}\sum_{q=1}^{D} \Phi_{(QK,s):p,q}^{(A,\tau_e(i)):j,\pi_e^{(\tau_e(i))}(k)}[WW]_{p,q}^{(QK,s)} + \sum_{s=1}^{h}\sum_{p=1}^{D}\sum_{q=1}^{D} \Phi_{(VO,s):p,q}^{(A,\tau_e(i)):j,\pi_e^{(\tau_e(i))}(k)}[WW]_{p,q}^{(VO,s)}$$

$$+ \sum_{s=1}^{n_e}\sum_{p=1}^{D} \Phi_{(G,s):p}^{(A,\tau_e(i)):j,\pi_e^{(\tau_e(i))}(k)}[W]_p^{(G,s)} + \sum_{s=1}^{n_e}\sum_{p=1}^{D}\sum_{q=1}^{D_A} \Phi_{(A,s):p,q}^{(A,\tau_e(i)):j,\pi_e^{(\tau_e(i))}(k)}[W]_{p,q}^{(A,s)}$$

$$+ \sum_{s=1}^{n_e}\sum_{p=1}^{D_A}\sum_{q=1}^{D} \Phi_{(B,s):p,q}^{(A,\tau_e(i)):j,\pi_e^{(\tau_e(i))}(k)}[W]_{p,q}^{(B,s)} + \sum_{s=1}^{n_e} \Phi_{(G,s)}^{(A,\tau_e(i)):j,\pi_e^{(\tau_e(i))}(k)}[b]^{(G,s)}$$

$$+ \sum_{s=1}^{n_e}\sum_{q=1}^{D_A} \Phi_{(A,s):q}^{(A,\tau_e(i)):j,\pi_e^{(\tau_e(i))}(k)}[b]_q^{(A,s)} + \sum_{s=1}^{n_e}\sum_{q=1}^{D} \Phi_{(B,s):q}^{(A,\tau_e(i)):j,\pi_e^{(\tau_e(i))}(k)}[b]_q^{(B,s)} + \Phi_1^{(A,\tau_e(i)):j,\pi_e^{(\tau_e(i))}(k)}.$$

$$\tag{196}$$

Using lemma F.1, we obtain the constraints:

$$\Phi^{(A,i):j,k}_{(QK,\tau_h^{-1}(s)):p,q} = \Phi^{(A,\tau_e(i)):j,\pi_e^{(\tau_e(i))}(k)}_{(QK,s):p,q},$$

$$\Phi^{(A,i):j,k}_{(VO,\tau_h^{-1}(s)):p,q} = \Phi^{(A,\tau_e(i)):j,\pi_e^{(\tau_e(i))}(k)}_{(VO,s):p,q},$$

$$\Phi^{(A,i):j,k}_{(G,\tau_e^{-1}(s)):p} = \Phi^{(A,\tau_e(i)):j,\pi_e^{(\tau_e(i))}(k)}_{(G,s):p},$$

$$\Phi^{(A,i):j,k}_{(A,\tau_e^{-1}(s)):p,(\pi_e^{(s)})^{-1}(q)} = \Phi^{(A,\tau_e(i)):j,\pi_e^{(\tau_e(i))}(k)}_{(A,s):p,q},$$

$$\Phi^{(A,i):j,k}_{(B,\tau_e^{-1}(s)):(\pi_e^{(s)})^{-1}(p),q} = \Phi^{(A,\tau_e(i)):j,\pi_e^{(\tau_e(i))}(k)}_{(B,s):p,q},$$

$$\Phi^{(A,i):j,k}_{(G,\tau_e^{-1}(s))} = \Phi^{(A,\tau_e(i)):j,\pi_e^{(\tau_e(i))}(k)}_{(G,s)},$$

$$\Phi^{(A,i):j,k}_{(A,\tau_e^{-1}(s)):(\pi_e^{(s)})^{-1}(q)} = \Phi^{(A,\tau_e(i)):j,\pi_e^{(\tau_e(i))}(k)}_{(A,s):q},$$

$$\Phi^{(A,i):j,k}_{(B,\tau_e^{-1}(s)):q} = \Phi^{(A,\tau_e(i)):j,\pi_e^{(\tau_e(i))}(k)}_{(B,s):q},$$

$$\Phi^{(A,i):j,k}_1 = \Phi^{(A,\tau_e(i)):j,\pi_e^{(\tau_e(i))}(k)}_1,$$

$$\sum_{s=1}^{n_e} \Phi^{(A,i):j,k}_{(G,\tau_e^{-1}(s)):p} = 0,$$

$$\sum_{s=1}^{n_e} \Phi^{(A,i):j,k}_{(G,s)} = 0.$$

Therefore,

$$\Phi^{(A,i):j,k}_{(QK,s):p,q} = \Phi^{(A,\tau_e(i)):j,\pi_e^{(\tau_e(i))}(k)}_{(QK,\tau_h(s)):p,q},$$

$$\Phi^{(A,i):j,k}_{(VO,s):p,q} = \Phi^{(A,\tau_e(i)):j,\pi_e^{(\tau_e(i))}(k)}_{(VO,\tau_h(s)):p,q},$$

$$\Phi^{(A,i):j,k}_{(G,s):p} = \Phi^{(A,\tau_e(i)):j,\pi_e^{(\tau_e(i))}(k)}_{(G,\tau_e(s)):p},$$

$$\Phi^{(A,i):j,k}_{(A,s):p,q} = \Phi^{(A,\tau_e(i)):j,\pi_e^{(\tau_e(i))}(k)}_{(A,\tau_e(s)):p,\pi_e^{(\tau_e(s))}(q)},$$

$$\Phi^{(A,i):j,k}_{(B,s):p,q} = \Phi^{(A,\tau_e(i)):j,\pi_e^{(\tau_e(i))}(k)}_{(B,\tau_e(s)):\pi_e^{(\tau_e(s))}(p),q},$$

$$\Phi^{(A,i):j,k}_{(G,s)} = \Phi^{(A,\tau_e(i)):j,\pi_e^{(\tau_e(i))}(k)}_{(G,\tau_e(s))}, \tag{197}$$

$$\Phi^{(A,i):j,k}_{(A,s):q} = \Phi^{(A,\tau_e(i)):j,\pi_e^{(\tau_e(i))}(k)}_{(A,\tau_e(s)):\pi_e^{(\tau_e(s))}(q)},$$

$$\Phi^{(A,i):j,k}_{(B,s):q} = \Phi^{(A,\tau_e(i)):j,\pi_e^{(\tau_e(i))}(k)}_{(B,\tau_e(s)):q},$$

$$\Phi^{(A,i):j,k}_1 = \Phi^{(A,\tau_e(i)):j,\pi_e^{(\tau_e(i))}(k)}_1,$$

$$\sum_{s=1}^{n_e} \Phi^{(A,i):j,k}_{(G,s):p} = 0,$$

$$\sum_{s=1}^{n_e} \Phi^{(A,i):j,k}_{(G,s)} = 0.$$

**Step 7. Solving** $[E(gW)]^{(B,i)}_{j,k} = [gE(W)]^{(B,i)}_{j,k}$.

For this equation, we proceed as follow:

$$
\sum_{s=1}^{h}\sum_{p=1}^{D}\sum_{q=1}^{D}\Phi_{(QK,\tau_h^{-1}(s)):p,q}^{(B,i):j,k}[WW]_{p,q}^{(QK,s)} + \sum_{s=1}^{h}\sum_{p=1}^{D}\sum_{q=1}^{D}\Phi_{(VO,\tau_h^{-1}(s)):p,q}^{(B,i):j,k}[WW]_{p,q}^{(VO,s)}
$$

$$
+ \sum_{s=1}^{n_e}\sum_{p=1}^{D}\Phi_{(G,\tau_e^{-1}(s)):p}^{(B,i):j,k}\left[[W]^{(G,s)}+\gamma_W\right]_p + \sum_{s=1}^{n_e}\sum_{p=1}^{D}\sum_{q=1}^{D_A}\Phi_{(A,\tau_e^{-1}(s)):p,(\pi_e^{(s)})^{-1}(q)}^{(B,i):j,k}\left[[W]^{(A,s)}\right]_{p,q}
$$

$$
+ \sum_{s=1}^{n_e}\sum_{p=1}^{D_A}\sum_{q=1}^{D}\Phi_{(B,\tau_e^{-1}(s)):(\pi_e^{(s)})^{-1}(p),q}^{(B,i):j,k}\left[[W]^{(B,s)}\right]_{p,q} + \sum_{s=1}^{n_e}\Phi_{(G,\tau_e^{-1}(s))}^{(B,i):j,k}\left([b]^{(G,s)}+\gamma_b\right)
$$

$$
+ \sum_{s=1}^{n_e}\sum_{q=1}^{D_A}\Phi_{(A,\tau_e^{-1}(s)):(\pi_e^{(s)})^{-1}(q)}^{(B,i):j,k}\left[[b]^{(A,s)}\right]_q + \sum_{s=1}^{n_e}\sum_{q=1}^{D}\Phi_{(B,\tau_e^{-1}(s)):q}^{(B,i):j,k}\left[[b]^{(B,s)}\right]_q + \Phi_1^{(B,i):j,k}
$$

$$
= \sum_{s=1}^{h}\sum_{p=1}^{D}\sum_{q=1}^{D}\Phi_{(QK,s):p,q}^{(B,\tau_e(i)):\pi_e^{(\tau_e(i))}(j),k}[WW]_{p,q}^{(QK,s)} + \sum_{s=1}^{h}\sum_{p=1}^{D}\sum_{q=1}^{D}\Phi_{(VO,s):p,q}^{(B,\tau_e(i)):\pi_e^{(\tau_e(i))}(j),k}[WW]_{p,q}^{(VO,s)}
$$

$$
+ \sum_{s=1}^{n_e}\sum_{p=1}^{D}\Phi_{(G,s):p}^{(B,\tau_e(i)):\pi_e^{(\tau_e(i))}(j),k}[W]_p^{(G,s)} + \sum_{s=1}^{n_e}\sum_{p=1}^{D}\sum_{q=1}^{D_A}\Phi_{(A,s):p,q}^{(B,\tau_e(i)):\pi_e^{(\tau_e(i))}(j),k}[W]_{p,q}^{(A,s)}
$$

$$
+ \sum_{s=1}^{n_e}\sum_{p=1}^{D_A}\sum_{q=1}^{D}\Phi_{(B,s):p,q}^{(B,\tau_e(i)):\pi_e^{(\tau_e(i))}(j),k}[W]_{p,q}^{(B,s)} + \sum_{s=1}^{n_e}\Phi_{(G,s)}^{(B,\tau_e(i)):\pi_e^{(\tau_e(i))}(j),k}[b]^{(G,s)}
$$

$$
+ \sum_{s=1}^{n_e}\sum_{q=1}^{D_A}\Phi_{(A,s):q}^{(B,\tau_e(i)):\pi_e^{(\tau_e(i))}(j),k}[b]_q^{(A,s)} + \sum_{s=1}^{n_e}\sum_{q=1}^{D}\Phi_{(B,s):q}^{(B,\tau_e(i)):\pi_e^{(\tau_e(i))}(j),k}[b]_q^{(B,s)} + \Phi_1^{(B,\tau_e(i)):\pi_e^{(\tau_e(i))}(j),k}.
$$

$$(198)$$

Using lemma F.1, we obtain the constraints:

$$
\Phi_{(QK,\tau_h^{-1}(s)):p,q}^{(B,i):j,k} = \Phi_{(QK,s):p,q}^{(B,\tau_e(i)):\pi_e^{(\tau_e(i))}(j),k},
$$

$$
\Phi_{(VO,\tau_h^{-1}(s)):p,q}^{(B,i):j,k} = \Phi_{(VO,s):p,q}^{(B,\tau_e(i)):\pi_e^{(\tau_e(i))}(j),k},
$$

$$
\Phi_{(G,\tau_e^{-1}(s)):p}^{(B,i):j,k} = \Phi_{(G,s):p}^{(B,\tau_e(i)):\pi_e^{(\tau_e(i))}(j),k},
$$

$$
\Phi_{(A,\tau_e^{-1}(s)):p,(\pi_e^{(s)})^{-1}(q)}^{(B,i):j,k} = \Phi_{(A,s):p,q}^{(B,\tau_e(i)):\pi_e^{(\tau_e(i))}(j),k},
$$

$$
\Phi_{(B,\tau_e^{-1}(s)):(\pi_e^{(s)})^{-1}(p),q}^{(B,i):j,k} = \Phi_{(B,s):p,q}^{(B,\tau_e(i)):\pi_e^{(\tau_e(i))}(j),k},
$$

$$
\Phi_{(G,\tau_e^{-1}(s))}^{(B,i):j,k} = \Phi_{(G,s)}^{(B,\tau_e(i)):\pi_e^{(\tau_e(i))}(j),k},
$$

$$
\Phi_{(A,\tau_e^{-1}(s)):(\pi_e^{(s)})^{-1}(q)}^{(B,i):j,k} = \Phi_{(A,s):q}^{(B,\tau_e(i)):\pi_e^{(\tau_e(i))}(j),k},
$$

$$
\Phi_{(B,\tau_e^{-1}(s)):q}^{(B,i):j,k} = \Phi_{(B,s):q}^{(B,\tau_e(i)):\pi_e^{(\tau_e(i))}(j),k},
$$

$$
\Phi_1^{(B,i):j,k} = \Phi_1^{(B,\tau_e(i)):\pi_e^{(\tau_e(i))}(j),k},
$$

$$
\sum_{s=1}^{n_e}\Phi_{(G,\tau_e^{-1}(s)):p}^{(B,i):j,k} = 0,
$$

$$
\sum_{s=1}^{n_e}\Phi_{(G,s)}^{(B,i):j,k} = 0.
$$

Therefore:

$$
\Phi^{(B,i):j,k}_{(QK,s):p,q} = \Phi^{(B,\tau_e(i)):\pi_e^{(\tau_e(i))}(j),k}_{(QK,\tau_h(s)):p,q},
$$

$$
\Phi^{(B,i):j,k}_{(VO,s):p,q} = \Phi^{(B,\tau_e(i)):\pi_e^{(\tau_e(i))}(j),k}_{(VO,\tau_h(s)):p,q},
$$

$$
\Phi^{(B,i):j,k}_{(G,s):p} = \Phi^{(B,\tau_e(i)):\pi_e^{(\tau_e(i))}(j),k}_{(G,\tau_e(s)):p},
$$

$$
\Phi^{(B,i):j,k}_{(A,s):p,q} = \Phi^{(B,\tau_e(i)):\pi_e^{(\tau_e(i))}(j),k}_{(A,\tau_e(s)):p,\pi_e^{(\tau_e(s)}(q)},
$$

$$
\Phi^{(B,i):j,k}_{(B,s):p,q} = \Phi^{(B,\tau_e(i)):\pi_e^{(\tau_e(i))}(j),k}_{(B,\tau_e(s)):\pi_e^{(\tau_e(s))}(p),q},
$$

$$
\Phi^{(B,i):j,k}_{(G,s)} = \Phi^{(B,\tau_e(i)):\pi_e^{(\tau_e(i))}(j),k}_{(G,\tau_e(s))}, \tag{199}
$$

$$
\Phi^{(B,i):j,k}_{(A,s):q} = \Phi^{(B,\tau_e(i)):\pi_e^{(\tau_e(i))}(j),k}_{(A,\tau_e(s)):\pi_e^{(\tau_e(s))}(q)},
$$

$$
\Phi^{(B,i):j,k}_{(B,s):q} = \Phi^{(B,\tau_e(i)):\pi_e^{(\tau_e(i))}(j),k}_{(B,\tau_e(s)):q},
$$

$$
\Phi^{(B,i):j,k}_{1} = \Phi^{(B,\tau_e(i)):\pi_e^{(\tau_e(i))}(j),k}_{1},
$$

$$
\sum_{s=1}^{n_e} \Phi^{(B,i):j,k}_{(G,s):p} = 0,
$$

$$
\sum_{s=1}^{n_e} \Phi^{(B,i):j,k}_{(G,s)} = 0.
$$

**Step 8. Solving** $[E(gb)]^{(G,i)} = [gE(b)]^{(G,i)}$.

For this equation, we proceed as follow:

$$
\sum_{s=1}^{h}\sum_{p=1}^{D}\sum_{q=1}^{D}\Phi^{(G,i)}_{(QK,\tau_h^{-1}(s)):p,q}[WW]^{(QK,s)}_{p,q} + \sum_{s=1}^{h}\sum_{p=1}^{D}\sum_{q=1}^{D}\Phi^{(G,i)}_{(VO,\tau_h^{-1}(s)):p,q}[WW]^{(VO,s)}_{p,q}
$$

$$
+ \sum_{s=1}^{n_e}\sum_{p=1}^{D}\Phi^{(G,i)}_{(G,\tau_e^{-1}(s)):p}\left[[W]^{(G,s)} + \gamma_W\right]_p + \sum_{s=1}^{n_e}\sum_{p=1}^{D}\sum_{q=1}^{D_A}\Phi^{(G,i)}_{(A,\tau_e^{-1}(s)):p,(\pi_e^{(s)})^{-1}(q)}\left[[W]^{(A,s)}\right]_{p,q}
$$

$$
+ \sum_{s=1}^{n_e}\sum_{p=1}^{D_A}\sum_{q=1}^{D}\Phi^{(G,i)}_{(B,\tau_e^{-1}(s)):(\pi_e^{(s)})^{-1}(p),q}\left[[W]^{(B,s)}\right]_{p,q} + \sum_{s=1}^{n_e}\Phi^{(G,i)}_{(G,\tau_e^{-1}(s))}\left([b]^{(G,s)} + \gamma_b\right)
$$

$$
+ \sum_{s=1}^{n_e}\sum_{q=1}^{D_A}\Phi^{(G,i)}_{(A,\tau_e^{-1}(s)):(\pi_e^{(s)})^{-1}(q)}\left[[b]^{(A,s)}\right]_q + \sum_{s=1}^{n_e}\sum_{q=1}^{D}\Phi^{(G,i)}_{(B,\tau_e^{-1}(s)):q}\left[[b]^{(B,s)}\right]_q + \Phi^{(G,i)}_{1}
$$

$$
= \sum_{s=1}^{h}\sum_{p=1}^{D}\sum_{q=1}^{D}\Phi^{(G,\tau_e(i))}_{(QK,s):p,q}[WW]^{(QK,s)}_{p,q} + \sum_{s=1}^{h}\sum_{p=1}^{D}\sum_{q=1}^{D}\Phi^{(G,\tau_e(i))}_{(VO,s):p,q}[WW]^{(VO,s)}_{p,q}
$$

$$
+ \sum_{s=1}^{n_e}\sum_{p=1}^{D}\Phi^{(G,\tau_e(i))}_{(G,s):p}[W]^{(G,s)}_p + \sum_{s=1}^{n_e}\sum_{p=1}^{D}\sum_{q=1}^{D_A}\Phi^{(G,\tau_e(i))}_{(A,s):p,q}[W]^{(A,s)}_{p,q}
$$

$$
+ \sum_{s=1}^{n_e}\sum_{p=1}^{D_A}\sum_{q=1}^{D}\Phi^{(G,\tau_e(i))}_{(B,s):p,q}[W]^{(B,s)}_{p,q} + \sum_{s=1}^{n_e}\Phi^{(G,\tau_e(i))}_{(G,s)}[b]^{(G,s)}
$$

$$
+ \sum_{s=1}^{n_e}\sum_{q=1}^{D_A}\Phi^{(G,\tau_e(i))}_{(A,s):q}[b]^{(A,s)}_q + \sum_{s=1}^{n_e}\sum_{q=1}^{D}\Phi^{(G,\tau_e(i))}_{(B,s):q}[b]^{(B,s)}_q + \Phi^{(G,\tau_e(i))}_{1} + \gamma_b. \tag{200}
$$

Using lemma F.1, we obtain the constraints:

$$\Phi^{(G,i)}_{(QK,\tau_h^{-1}(s)):p,q} = \Phi^{(G,\tau_e(i))}_{(QK,s):p,q},$$

$$\Phi^{(G,i)}_{(VO,\tau_h^{-1}(s)):p,q} = \Phi^{(G,\tau_e(i))}_{(VO,s):p,q},$$

$$\Phi^{(G,i)}_{(G,\tau_e^{-1}(s)):p} = \Phi^{(G,\tau_e(i))}_{(G,s):p},$$

$$\Phi^{(G,i)}_{(A,\tau_e^{-1}(s)):p,(\pi_e^{(s)})^{-1}(q)} = \Phi^{(G,\tau_e(i))}_{(A,s):p,q},$$

$$\Phi^{(G,i)}_{(B,\tau_e^{-1}(s)):(\pi_e^{(s)})^{-1}(p),q} = \Phi^{(G,\tau_e(i))}_{(B,s):p,q},$$

$$\Phi^{(G,i)}_{(G,\tau_e^{-1}(s))} = \Phi^{(G,\tau_e(i))}_{(G,s)},$$

$$\Phi^{(G,i)}_{(A,\tau_e^{-1}(s)):(\pi_e^{(s)})^{-1}(q)} = \Phi^{(G,\tau_e(i))}_{(A,s):q},$$

$$\Phi^{(G,i)}_{(B,\tau_e^{-1}(s)):q} = \Phi^{(G,\tau_e(i))}_{(B,s):q},$$

$$\Phi^{(G,i)}_1 = \Phi^{(G,\tau_e(i))}_1,$$

$$\sum_{s=1}^{n_e} \Phi^{(G,i)}_{(G,\tau_e^{-1}(s)):p} = 0,$$

$$\sum_{s=1}^{n_e} \Phi^{(G,i)}_{(G,s)} = 1.$$

Therefore:

$$\Phi^{(G,i)}_{(QK,s):p,q} = \Phi^{(G,\tau_e(i))}_{(QK,\tau_h(s)):p,q},$$

$$\Phi^{(G,i)}_{(VO,s):p,q} = \Phi^{(G,\tau_e(i))}_{(VO,\tau_h(s)):p,q},$$

$$\Phi^{(G,i)}_{(G,s):p} = \Phi^{(G,\tau_e(i))}_{(G,\tau_e(s)):p},$$

$$\Phi^{(G,i)}_{(A,s):p,q} = \Phi^{(G,\tau_e(i))}_{(A,\tau_e(s)):p,\pi_e^{(\tau_e(s))}(q)},$$

$$\Phi^{(G,i)}_{(B,s):p,q} = \Phi^{(G,\tau_e(i))}_{(B,\tau_e(s)):\pi_e^{(\tau_e(s))}(p),q},$$

$$\Phi^{(G,i)}_{(G,s)} = \Phi^{(G,\tau_e(i))}_{(G,\tau_e(s))}, \tag{201}$$

$$\Phi^{(G,i)}_{(A,s):q} = \Phi^{(G,\tau_e(i))}_{(A,\tau_e(s)):\pi_e^{(\tau_e(s))}(q)},$$

$$\Phi^{(G,i)}_{(B,s):q} = \Phi^{(G,\tau_e(i))}_{(B,\tau_e(s)):q},$$

$$\Phi^{(G,i)}_1 = \Phi^{(G,\tau_e(i))}_1,$$

$$\sum_{s=1}^{n_e} \Phi^{(G,i)}_{(G,s):p} = 0,$$

$$\sum_{s=1}^{n_e} \Phi^{(G,i)}_{(G,s)} = 1.$$

**Step 9. Solving** $[E(gb)]^{(A,i)}_j = [gE(b)]^{(A,i)}_j$.

For this equation, we proceed as follow:

$$\sum_{s=1}^{h}\sum_{p=1}^{D}\sum_{q=1}^{D} \Phi^{(A,i):j}_{(QK,\tau_h^{-1}(s)):p,q}[WW]^{(QK,s)}_{p,q} + \sum_{s=1}^{h}\sum_{p=1}^{D}\sum_{q=1}^{D} \Phi^{(A,i):j}_{(VO,\tau_h^{-1}(s)):p,q}[WW]^{(VO,s)}_{p,q}$$

$$+ \sum_{s=1}^{n_e}\sum_{p=1}^{D} \Phi^{(A,i):j}_{(G,\tau_e^{-1}(s)):p} \left[[W]^{(G,s)} + \gamma_W\right]_p + \sum_{s=1}^{n_e}\sum_{p=1}^{D}\sum_{q=1}^{D_A} \Phi^{(A,i):j}_{(A,\tau_e^{-1}(s)):p,(\pi_e^{(s)})^{-1}(q)} \left[[W]^{(A,s)}\right]_{p,q}$$

$$+ \sum_{s=1}^{n_e} \sum_{p=1}^{D_A} \sum_{q=1}^{D} \Phi^{(A,i):j}_{(B,\tau_e^{-1}(s)):(\pi_e^{(s)})^{-1}(p),q} \left[ [W]^{(B,s)} \right]_{p,q} + \sum_{s=1}^{n_e} \Phi^{(A,i):j}_{(G,\tau_e^{-1}(s))} \left( [b]^{(G,s)} + \gamma_b \right)$$

$$+ \sum_{s=1}^{n_e} \sum_{q=1}^{D_A} \Phi^{(A,i):j}_{(A,\tau_e^{-1}(s)):(\pi_e^{(s)})^{-1}(q)} \left[ [b]^{(A,s)} \right]_q + \sum_{s=1}^{n_e} \sum_{q=1}^{D} \Phi^{(A,i):j}_{(B,\tau_e^{-1}(s)):q} \left[ [b]^{(B,s)} \right]_q + \Phi^{(A,i):j}_1,$$

$$= \sum_{s=1}^{h} \sum_{p=1}^{D} \sum_{q=1}^{D} \Phi^{(A,\tau_e(i)):\pi_e^{(\tau_e(i))}(j)}_{(QK,s):p,q} [WW]^{(QK,s)}_{p,q} + \sum_{s=1}^{h} \sum_{p=1}^{D} \sum_{q=1}^{D} \Phi^{(A,\tau_e(i)):\pi_e^{(\tau_e(i))}(j)}_{(VO,s):p,q} [WW]^{(VO,s)}_{p,q}$$

$$+ \sum_{s=1}^{n_e} \sum_{p=1}^{D} \Phi^{(A,\tau_e(i)):\pi_e^{(\tau_e(i))}(j)}_{(G,s):p} [W]^{(G,s)}_p + \sum_{s=1}^{n_e} \sum_{p=1}^{D} \sum_{q=1}^{D_A} \Phi^{(A,\tau_e(i)):\pi_e^{(\tau_e(i))}(j)}_{(A,s):p,q} [W]^{(A,s)}_{p,q}$$

$$+ \sum_{s=1}^{n_e} \sum_{p=1}^{D_A} \sum_{q=1}^{D} \Phi^{(A,\tau_e(i)):\pi_e^{(\tau_e(i))}(j)}_{(B,s):p,q} [W]^{(B,s)}_{p,q} + \sum_{s=1}^{n_e} \Phi^{(A,\tau_e(i)):\pi_e^{(\tau_e(i))}(j)}_{(G,s)} [b]^{(G,s)}$$

$$+ \sum_{s=1}^{n_e} \sum_{q=1}^{D_A} \Phi^{(A,\tau_e(i)):\pi_e^{(\tau_e(i))}(j)}_{(A,s):q} [b]^{(A,s)}_q + \sum_{s=1}^{n_e} \sum_{q=1}^{D} \Phi^{(A,\tau_e(i)):\pi_e^{(\tau_e(i))}(j)}_{(B,s):q} [b]^{(B,s)}_q + \Phi^{(A,\tau_e(i)):\pi_e^{(\tau_e(i))}(j)}_1.$$

$$(202)$$

Using lemma F.1, we obtain the constraints:

$$\Phi^{(A,i):j}_{(QK,\tau_h^{-1}(s)):p,q} = \Phi^{(A,\tau_e(i)):\pi_e^{(\tau_e(i))}(j)}_{(QK,s):p,q},$$

$$\Phi^{(A,i):j}_{(VO,\tau_h^{-1}(s)):p,q} = \Phi^{(A,\tau_e(i)):\pi_e^{(\tau_e(i))}(j)}_{(VO,s):p,q},$$

$$\Phi^{(A,i):j}_{(G,\tau_e^{-1}(s)):p} = \Phi^{(A,\tau_e(i)):\pi_e^{(\tau_e(i))}(j)}_{(G,s):p},$$

$$\Phi^{(A,i):j}_{(A,\tau_e^{-1}(s)):p,(\pi_e^{(s)})^{-1}(q)} = \Phi^{(A,\tau_e(i)):\pi_e^{(\tau_e(i))}(j)}_{(A,s):p,q},$$

$$\Phi^{(A,i):j}_{(B,\tau_e^{-1}(s)):(\pi_e^{(s)})^{-1}(p),q} = \Phi^{(A,\tau_e(i)):\pi_e^{(\tau_e(i))}(j)}_{(B,s):p,q},$$

$$\Phi^{(A,i):j}_{(G,\tau_e^{-1}(s))} = \Phi^{(A,\tau_e(i)):\pi_e^{(\tau_e(i))}(j)}_{(G,s)},$$

$$\Phi^{(A,i):j}_{(A,\tau_e^{-1}(s)):(\pi_e^{(s)})^{-1}(q)} = \Phi^{(A,\tau_e(i)):\pi_e^{(\tau_e(i))}(j)}_{(A,s):q},$$

$$\Phi^{(A,i):j}_{(B,\tau_e^{-1}(s)):q} = \Phi^{(A,\tau_e(i)):\pi_e^{(\tau_e(i))}(j)}_{(B,s):q},$$

$$\Phi^{(A,i):j}_1 = \Phi^{(A,\tau_e(i)):\pi_e^{(\tau_e(i))}(j)}_1,$$

$$\sum_{s=1}^{n_e} \Phi^{(A,i):j}_{(G,\tau_e^{-1}(s)):p} = 0,$$

$$\sum_{s=1}^{n_e} \Phi^{(A,i):j}_{(G,s)} = 0.$$

Therefore:

$$\Phi^{(A,i):j}_{(QK,s):p,q} = \Phi^{(A,\tau_e(i)):\pi_e^{(\tau_e(i))}(j)}_{(QK,\tau_h(s)):p,q},$$

$$\Phi^{(A,i):j}_{(VO,s):p,q} = \Phi^{(A,\tau_e(i)):\pi_e^{(\tau_e(i))}(j)}_{(VO,\tau_h(s)):p,q},$$

$$\Phi^{(A,i):j}_{(G,s):p} = \Phi^{(A,\tau_e(i)):\pi_e^{(\tau_e(i))}(j)}_{(G,\tau_e(s)):p},$$

$$\Phi^{(A,i):j}_{(A,s):p,q} = \Phi^{(A,\tau_e(i)):\pi_e^{(\tau_e(i))}(j)}_{(A,\tau_e(s)):p,\pi_e^{(\tau_e(s))}(q)},$$

$$\Phi^{(A,i):j}_{(B,s):p,q} = \Phi^{(A,\tau_e(i)):\pi_e^{(\tau_e(i))}(j)}_{(B,\tau_e(s)):\pi_e^{(\tau_e(s))}(p),q},$$

$$\Phi^{(A,i):j}_{(G,s)} = \Phi^{(A,\tau_e(i)):\pi_e^{(\tau_e(i))}(j)}_{(G,\tau_e(s))},$$

$$\Phi^{(A,i):j}_{(A,s):q} = \Phi^{(A,\tau_e(i)):\pi_e^{(\tau_e(i))}(j)}_{(A,\tau_e(s)):\pi_e^{(\tau_e(s))}(q)},$$

$$\Phi^{(A,i):j}_{(B,s):q} = \Phi^{(A,\tau_e(i)):\pi_e^{(\tau_e(i))}(j)}_{(B,\tau_e(s)):q},$$

$$\Phi^{(A,i):j}_1 = \Phi^{(A,\tau_e(i)):\pi_e^{(\tau_e(i))}(j)}_1,$$

$$\sum_{s=1}^{n_e} \Phi^{(A,i):j}_{(G,s):p} = 0,$$

$$\sum_{s=1}^{n_e} \Phi^{(A,i):j}_{(G,s)} = 0. \tag{203}$$

**Step 10. Solving $[E(gb)]^{(B,i)}_j = [gE(b)]^{(B,i)}_j$.**

For this equation, we proceed as follow:

$$\sum_{s=1}^{h}\sum_{p=1}^{D}\sum_{q=1}^{D} \Phi^{(B,i):j}_{(QK,\tau_h^{-1}(s)):p,q}[WW]^{(QK,s)}_{p,q} + \sum_{s=1}^{h}\sum_{p=1}^{D}\sum_{q=1}^{D} \Phi^{(B,i):j}_{(VO,\tau_h^{-1}(s)):p,q}[WW]^{(VO,s)}_{p,q}$$

$$+ \sum_{s=1}^{n_e}\sum_{p=1}^{D} \Phi^{(B,i):j}_{(G,\tau_e^{-1}(s)):p}\left[[W]^{(G,s)} + \gamma_W\right]_p + \sum_{s=1}^{n_e}\sum_{p=1}^{D}\sum_{q=1}^{D_A} \Phi^{(B,i):j}_{(A,\tau_e^{-1}(s)):p,(\pi_e^{(s)})^{-1}(q)}\left[[W]^{(A,s)}\right]_{p,q}$$

$$+ \sum_{s=1}^{n_e}\sum_{p=1}^{D_A}\sum_{q=1}^{D} \Phi^{(B,i):j}_{(B,\tau_e^{-1}(s)):(\pi_e^{(s)})^{-1}(p),q}\left[[W]^{(B,s)}\right]_{p,q} + \sum_{s=1}^{n_e} \Phi^{(B,i):j}_{(G,\tau_e^{-1}(s))}\left([b]^{(G,s)} + \gamma_b\right)$$

$$+ \sum_{s=1}^{n_e}\sum_{q=1}^{D_A} \Phi^{(B,i):j}_{(A,\tau_e^{-1}(s)):(\pi_e^{(s)})^{-1}(q)}\left[[b]^{(A,s)}\right]_q + \sum_{s=1}^{n_e}\sum_{q=1}^{D} \Phi^{(B,i):j}_{(B,\tau_e^{-1}(s)):q}\left[[b]^{(B,s)}\right]_q + \Phi^{(B,i):j}_1$$

$$= \sum_{s=1}^{h}\sum_{p=1}^{D}\sum_{q=1}^{D} \Phi^{(B,\tau_e(i)):j}_{(QK,s):p,q}[WW]^{(QK,s)}_{p,q} + \sum_{s=1}^{h}\sum_{p=1}^{D}\sum_{q=1}^{D} \Phi^{(B,\tau_e(i)):j}_{(VO,s):p,q}[WW]^{(VO,s)}_{p,q}$$

$$+ \sum_{s=1}^{n_e}\sum_{p=1}^{D} \Phi^{(B,\tau_e(i)):j}_{(G,s):p}[W]^{(G,s)}_p + \sum_{s=1}^{n_e}\sum_{p=1}^{D}\sum_{q=1}^{D_A} \Phi^{(B,\tau_e(i)):j}_{(A,s):p,q}[W]^{(A,s)}_{p,q}$$

$$+ \sum_{s=1}^{n_e}\sum_{p=1}^{D_A}\sum_{q=1}^{D} \Phi^{(B,\tau_e(i)):j}_{(B,s):p,q}[W]^{(B,s)}_{p,q} + \sum_{s=1}^{n_e} \Phi^{(B,\tau_e(i)):j}_{(G,s)}[b]^{(G,s)}$$

$$+ \sum_{s=1}^{n_e}\sum_{q=1}^{D_A} \Phi^{(B,\tau_e(i)):j}_{(A,s):q}[b]^{(A,s)}_q + \sum_{s=1}^{n_e}\sum_{q=1}^{D} \Phi^{(B,\tau_e(i)):j}_{(B,s):q}[b]^{(B,s)}_q + \Phi^{(B,\tau_e(i)):j}_1. \tag{204}$$

Using Lemma F.1, we obtain the constraints:

$$\Phi^{(B,i):j}_{(QK,\tau_h^{-1}(s)):p,q} = \Phi^{(B,\tau_e(i)):j}_{(QK,s):p,q},$$

$$\Phi^{(B,i):j}_{(VO,\tau_h^{-1}(s)):p,q} = \Phi^{(B,\tau_e(i)):j}_{(VO,s):p,q},$$

$$\Phi^{(B,i):j}_{(G,\tau_e^{-1}(s)):p} = \Phi^{(B,\tau_e(i)):j}_{(G,s):p},$$

$$\Phi^{(B,i):j}_{(A,\tau_e^{-1}(s)):p,(\pi_e^{(s)})^{-1}(q)} = \Phi^{(B,\tau_e(i)):j}_{(A,s):p,q},$$

$$\Phi^{(B,i):j}_{(B,\tau_e^{-1}(s)):(\pi_e^{(s)})^{-1}(p),q} = \Phi^{(B,\tau_e(i)):j}_{(B,s):p,q},$$

$$\Phi^{(B,i):j}_{(G,\tau_e^{-1}(s))} = \Phi^{(B,\tau_e(i)):j}_{(G,s)},$$

$$\Phi^{(B,i):j}_{(A,\tau_e^{-1}(s)):(\pi_e^{(s)})^{-1}(q)} = \Phi^{(B,\tau_e(i)):j}_{(A,s):q},$$

$$\Phi^{(B,i):j}_{(B,\tau_e^{-1}(s)):q} = \Phi^{(B,\tau_e(i)):j}_{(B,s):q},$$

$$\Phi^{(B,i):j}_1 = \Phi^{(B,\tau_e(i)):j}_1,$$

$$\sum_{s=1}^{n_e} \Phi^{(B,i):j}_{(G,\tau_e^{-1}(s)):p} = 0,$$

$$\sum_{s=1}^{n_e} \Phi^{(B,i):j}_{(G,s)} = 0.$$

Therefore:

$$\Phi^{(B,i):j}_{(QK,s):p,q} = \Phi^{(B,\tau_e(i)):j}_{(QK,\tau_h(s)):p,q},$$

$$\Phi^{(B,i):j}_{(VO,s):p,q} = \Phi^{(B,\tau_e(i)):j}_{(VO,\tau_h(s)):p,q},$$

$$\Phi^{(B,i):j}_{(G,s):p} = \Phi^{(B,\tau_e(i)):j}_{(G,\tau_e(s)):p},$$

$$\Phi^{(B,i):j}_{(A,s):p,q} = \Phi^{(B,\tau_e(i)):j}_{(A,\tau_e(s)):p,\pi_e^{(\tau_e(s))}(q)},$$

$$\Phi^{(B,i):j}_{(B,s):p,q} = \Phi^{(B,\tau_e(i)):j}_{(B,\tau_e(s)):\pi_e^{(\tau_e(s))}(p),q},$$

$$\Phi^{(B,i):j}_{(G,s)} = \Phi^{(B,\tau_e(i)):j}_{(G,\tau_e(s))},$$

$$\Phi^{(B,i):j}_{(A,s):q} = \Phi^{(B,\tau_e(i)):j}_{(A,\tau_e(s)):\pi_e^{(\tau_e(s))}(q)},$$

$$\Phi^{(B,i):j}_{(B,s):q} = \Phi^{(B,\tau_e(i)):j}_{(B,\tau_e(s)):q},$$

$$\Phi^{(B,i):j}_1 = \Phi^{(B,\tau_e(i)):j}_1,$$

$$\sum_{s=1}^{n_e} \Phi^{(B,i):j}_{(G,s):p} = 0,$$

$$\sum_{s=1}^{n_e} \Phi^{(B,i):j}_{(G,s)} = 0.$$

(205)

### F.4 FINAL FORM OF THE EQUIVARIANT POLYNOMIAL LAYER

The final form of $E(U)$ after solving all constraints are given below for each entries:

1. $[E(W)]^{(Q,i)}_{j,k}$ is given by

$$[E(W)]^{(Q,i)}_{j,k} = \sum_{p=1}^{D} \Phi^{(Q,i):j,k}_{(Q,i):p,k} [W]^{(Q,i)}_{p,k},$$

with constraints

$$\Phi^{(Q,i):j,k}_{(Q,i):p,k} = \Phi^{(Q,\tau(i)):j,k'}_{(Q,\tau(i)):p,k'}. \tag{206}$$

2. $[E(W)]_{j,k}^{(K,i)}$ is given by

$$[E(W)]_{j,k}^{(K,i)} = \sum_{p=1}^{D} \Phi_{(K,i):p,k}^{(K,i):j,k} [W]_{p,k}^{(K,i)},$$

with constraints

$$\Phi_{(K,i):p,k}^{(K,i):j,k} = \Phi_{(K,\tau(i)):p,k'}^{(K,\tau(i)):j,k'}. \tag{207}$$

3. $[E(W)]_{j,k}^{(V,i)}$ is given by

$$[E(W)]_{j,k}^{(V,i)} = \sum_{p=1}^{D} \Phi_{(V,i):p,k}^{(V,i):j,k} [W]_{p,k}^{(V,i)},$$

with constraints

$$\Phi_{(V,i):p,k}^{(V,i):j,k} = \Phi_{(V,\tau(i)):p,k'}^{(V,\tau(i)):j,k'}. \tag{208}$$

4. $[E(W)]_{j,k}^{(O,i)}$ is given by

$$[E(W)]_{j,k}^{(O,i)} = \sum_{p=1}^{D_k} \Phi_{(O,i):j,q}^{(O,i):j,k} [W]_{p,k}^{(O,i)},$$

with constraints

$$\Phi_{(O,i):j',q}^{(O,i):j',k} = \Phi_{(O,\tau(i)):j',q}^{(O,\tau(i)):j,k}. \tag{209}$$

5. $[E(W)]_{j}^{(G,i)}$ is given by

$$[E(W)]_{j}^{(G,i)} = \sum_{s=1}^{h}\sum_{p=1}^{D}\sum_{q=1}^{D} \Phi_{(QK,s):p,q}^{(G,i):j} [WW]_{p,q}^{(QK,s)} + \sum_{s=1}^{h}\sum_{p=1}^{D}\sum_{q=1}^{D} \Phi_{(VO,s):p,q}^{(G,i):j} [WW]_{p,q}^{(VO,s)}$$

$$+ \sum_{s=1}^{n_e}\sum_{p=1}^{D} \Phi_{(G,s):p}^{(G,i):j} [W]_{p}^{(G,s)} + \sum_{s=1}^{n_e}\sum_{p=1}^{D}\sum_{q=1}^{D_A} \Phi_{(A,s):p,q}^{(G,i):j} [W]_{p,q}^{(A,s)} + \sum_{s=1}^{n_e}\sum_{p=1}^{D_A}\sum_{q=1}^{D} \Phi_{(B,s):p,q}^{(G,i):j} [W]_{p,q}^{(B,s)}$$

$$+ \sum_{s=1}^{n_e} \Phi_{(G,s)}^{(G,i):j} [b]^{(G,s)} + \sum_{s=1}^{n_e}\sum_{q=1}^{D_A} \Phi_{(A,s):q}^{(G,i):j} [b]_{q}^{(A,s)} + \sum_{s=1}^{n_e}\sum_{q=1}^{D} \Phi_{(B,s):q}^{(G,i):j} [b]_{q}^{(B,s)} + \Phi_{1}^{(G,i):j}$$

with constraints

$$\Phi^{(G,i):j}_{(QK,s):p,q} = \Phi^{(G,\tau_e(i)):j}_{(QK,\tau_h(s)):p,q},$$

$$\Phi^{(G,i):j}_{(VO,s):p,q} = \Phi^{(G,\tau_e(i)):j}_{(VO,\tau_h(s)):p,q},$$

$$\Phi^{(G,i):j}_{(G,s):p} = \Phi^{(G,\tau_e(i)):j}_{(G,\tau_e(s)):p},$$

$$\Phi^{(G,i):j}_{(A,s):p,q} = \Phi^{(G,\tau_e(i)):j}_{(A,\tau_e(s)):p,\pi_e^{(\tau_e(s))}(q)},$$

$$\Phi^{(G,i):j}_{(B,s):p,q} = \Phi^{(G,\tau_e(i)):j}_{(B,\tau_e(s)):\pi_e^{(\tau_e(s))}(p),q},$$

$$\Phi^{(G,i):j}_{(G,s)} = \Phi^{(G,\tau_e(i)):j}_{(G,\tau_e(s))},$$

$$\Phi^{(G,i):j}_{(A,s):q} = \Phi^{(G,\tau_e(i)):j}_{(A,\tau_e(s)):\pi_e^{(\tau_e(s))}(q)},$$

$$\Phi^{(G,i):j}_{(B,s):q} = \Phi^{(G,\tau_e(i)):j}_{(B,\tau_e(s)):q},$$

$$\Phi^{(G,i):j}_1 = \Phi^{(G,\tau_e(i)):j}_1,$$

$$\sum_{s=1}^{n_e} \Phi^{(G,i):j}_{(G,s):p} = 0 \quad (p \neq j),$$

$$\sum_{s=1}^{n_e} \Phi^{(G,i):j}_{(G,s):j} = 1,$$

$$\sum_{s=1}^{n_e} \Phi^{(G,i):j}_{(G,s)} = 0. \tag{210}$$

6. $[E(W)]^{(A,i)}_{j,k}$ is given by

$$[E(W)]^{(A,i)}_{j,k} = \sum_{s=1}^{h}\sum_{p=1}^{D}\sum_{q=1}^{D} \Phi^{(A,i):j,k}_{(QK,s):p,q}[WW]^{(QK,s)}_{p,q} + \sum_{s=1}^{h}\sum_{p=1}^{D}\sum_{q=1}^{D} \Phi^{(A,i):j,k}_{(VO,s):p,q}[WW]^{(VO,s)}_{p,q}$$

$$+ \sum_{s=1}^{n_e}\sum_{p=1}^{D} \Phi^{(A,i):j,k}_{(G,s):p}[W]^{(G,s)}_p + \sum_{s=1}^{n_e}\sum_{p=1}^{D}\sum_{q=1}^{D_A} \Phi^{(A,i):j,k}_{(A,s):p,q}[W]^{(A,s)}_{p,q} + \sum_{s=1}^{n_e}\sum_{p=1}^{D_A}\sum_{q=1}^{D} \Phi^{(A,i):j,k}_{(B,s):p,q}[W]^{(B,s)}_{p,q}$$

$$\sum_{s=1}^{n_e} \Phi^{(A,i):j,k}_{(G,s)}[b]^{(G,s)} + \sum_{s=1}^{n_e}\sum_{q=1}^{D_A} \Phi^{(A,i):j,k}_{(A,s):q}[b]^{(A,s)}_q + \sum_{s=1}^{n_e}\sum_{q=1}^{D} \Phi^{(A,i):j,k}_{(B,s):q}[b]^{(B,s)}_q + \Phi^{(A,i):j,k}_1$$

with constraints

$$
\Phi^{(A,i):j,k}_{(QK,s):p,q} = \Phi^{(A,\tau_e(i)):j,\pi_e^{(\tau_e(i))}(k)}_{(QK,\tau_h(s)):p,q},
$$

$$
\Phi^{(A,i):j,k}_{(VO,s):p,q} = \Phi^{(A,\tau_e(i)):j,\pi_e^{(\tau_e(i))}(k)}_{(VO,\tau_h(s)):p,q},
$$

$$
\Phi^{(A,i):j,k}_{(G,s):p} = \Phi^{(A,\tau_e(i)):j,\pi_e^{(\tau_e(i))}(k)}_{(G,\tau_e(s)):p},
$$

$$
\Phi^{(A,i):j,k}_{(A,s):p,q} = \Phi^{(A,\tau_e(i)):j,\pi_e^{(\tau_e(i))}(k)}_{(A,\tau_e(s)):p,\pi_e^{(\tau_e(s))}(q)},
$$

$$
\Phi^{(A,i):j,k}_{(B,s):p,q} = \Phi^{(A,\tau_e(i)):j,\pi_e^{(\tau_e(i))}(k)}_{(B,\tau_e(s)):\pi_e^{(\tau_e(s))}(p),q},
$$

$$
\Phi^{(A,i):j,k}_{(G,s)} = \Phi^{(A,\tau_e(i)):j,\pi_e^{(\tau_e(i))}(k)}_{(G,\tau_e(s))},
$$

$$
\Phi^{(A,i):j,k}_{(A,s):q} = \Phi^{(A,\tau_e(i)):j,\pi_e^{(\tau_e(i))}(k)}_{(A,\tau_e(s)):\pi_e^{(\tau_e(s))}(q)},
$$

$$
\Phi^{(A,i):j,k}_{(B,s):q} = \Phi^{(A,\tau_e(i)):j,\pi_e^{(\tau_e(i))}(k)}_{(B,\tau_e(s)):q},
$$

$$
\Phi^{(A,i):j,k}_1 = \Phi^{(A,\tau_e(i)):j,\pi_e^{(\tau_e(i))}(k)}_1,
$$

$$
\sum_{s=1}^{n_e} \Phi^{(A,i):j,k}_{(G,s):p} = 0,
$$

$$
\sum_{s=1}^{n_e} \Phi^{(A,i):j,k}_{(G,s)} = 0. \tag{211}
$$

7. $[E(W)]^{(B,i)}_{j,k}$ is given by

$$
[E(W)]^{(B,i)}_{j,k} = \sum_{s=1}^{h}\sum_{p=1}^{D}\sum_{q=1}^{D} \Phi^{(B,i):j,k}_{(QK,s):p,q}[WW]^{(QK,s)}_{p,q} + \sum_{s=1}^{h}\sum_{p=1}^{D}\sum_{q=1}^{D} \Phi^{(B,i):j,k}_{(VO,s):p,q}[WW]^{(VO,s)}_{p,q}
$$

$$
+ \sum_{s=1}^{n_e}\sum_{p=1}^{D} \Phi^{(B,i):j,k}_{(G,s):p}[W]^{(G,s)}_p + \sum_{s=1}^{n_e}\sum_{p=1}^{D}\sum_{q=1}^{D_A} \Phi^{(B,i):j,k}_{(A,s):p,q}[W]^{(A,s)}_{p,q} + \sum_{s=1}^{n_e}\sum_{p=1}^{D_A}\sum_{q=1}^{D} \Phi^{(B,i):j,k}_{(B,s):p,q}[W]^{(B,s)}_{p,q}
$$

$$
\sum_{s=1}^{n_e} \Phi^{(B,i):j,k}_{(G,s)}[b]^{(G,s)} + \sum_{s=1}^{n_e}\sum_{q=1}^{D_A} \Phi^{(B,i):j,k}_{(A,s):q}[b]^{(A,s)}_q + \sum_{s=1}^{n_e}\sum_{q=1}^{D} \Phi^{(B,i):j,k}_{(B,s):q}[b]^{(B,s)}_q + \Phi^{(B,i):j,k}_1
$$

with constraints

$$\Phi^{(B,i):j,k}_{(QK,s):p,q} = \Phi^{(B,\tau_e(i)):\pi_e^{(\tau_e(i))}(j),k}_{(QK,\tau_h(s)):p,q},$$

$$\Phi^{(B,i):j,k}_{(VO,s):p,q} = \Phi^{(B,\tau_e(i)):\pi_e^{(\tau_e(i))}(j),k}_{(VO,\tau_h(s)):p,q},$$

$$\Phi^{(B,i):j,k}_{(G,s):p} = \Phi^{(B,\tau_e(i)):\pi_e^{(\tau_e(i))}(j),k}_{(G,\tau_e(s)):p},$$

$$\Phi^{(B,i):j,k}_{(A,s):p,q} = \Phi^{(B,\tau_e(i)):\pi_e^{(\tau_e(i))}(j),k}_{(A,\tau_e(s)):p,\pi_e^{(\tau_e(s))}(q)},$$

$$\Phi^{(B,i):j,k}_{(B,s):p,q} = \Phi^{(B,\tau_e(i)):\pi_e^{(\tau_e(i))}(j),k}_{(B,\tau_e(s)):\pi_e^{(\tau_e(s))}(p),q},$$

$$\Phi^{(B,i):j,k}_{(G,s)} = \Phi^{(B,\tau_e(i)):\pi_e^{(\tau_e(i))}(j),k}_{(G,s)},$$

$$\Phi^{(B,i):j,k}_{(A,s):q} = \Phi^{(B,\tau_e(i)):\pi_e^{(\tau_e(i))}(j),k}_{(A,\tau_e(s)):\pi_e^{(\tau_e(s))}(q)},$$

$$\Phi^{(B,i):j,k}_{(B,s):q} = \Phi^{(B,\tau_e(i)):\pi_e^{(\tau_e(i))}(j),k}_{(B,\tau_e(s)):q},$$

$$\Phi^{(B,i):j,k}_{1} = \Phi^{(B,\tau_e(i)):\pi_e^{(\tau_e(i))}(j),k}_{1},$$

$$\sum_{s=1}^{n_e} \Phi^{(B,i):j,k}_{(G,s):p} = 0,$$

$$\sum_{s=1}^{n_e} \Phi^{(B,i):j,k}_{(G,s)} = 0. \tag{212}$$

8. $[E(b)]^{(G,i)}$ is given by

$$[E(b)]^{(G,i)} = \sum_{s=1}^{h}\sum_{p=1}^{D}\sum_{q=1}^{D} \Phi^{(G,i)}_{(QK,s):p,q}[WW]^{(QK,s)}_{p,q} + \sum_{s=1}^{h}\sum_{p=1}^{D}\sum_{q=1}^{D} \Phi^{(G,i)}_{(VO,s):p,q}[WW]^{(VO,s)}_{p,q}$$

$$+ \sum_{s=1}^{n_e}\sum_{p=1}^{D} \Phi^{(G,i)}_{(G,s):p}[W]^{(G,s)}_{p} + \sum_{s=1}^{n_e}\sum_{p=1}^{D}\sum_{q=1}^{D_A} \Phi^{(G,i)}_{(A,s):p,q}[W]^{(A,s)}_{p,q} + \sum_{s=1}^{n_e}\sum_{p=1}^{D_A}\sum_{q=1}^{D} \Phi^{(G,i)}_{(B,s):p,q}[W]^{(B,s)}_{p,q}$$

$$\sum_{s=1}^{n_e} \Phi^{(G,i)}_{(G,s)}[b]^{(G,s)} + \sum_{s=1}^{n_e}\sum_{q=1}^{D_A} \Phi^{(G,i)}_{(A,s):q}[b]^{(A,s)}_{q} + \sum_{s=1}^{n_e}\sum_{q=1}^{D} \Phi^{(G,i)}_{(B,s):q}[b]^{(B,s)}_{q} + \Phi^{(G,i)}_{1}$$

with constraints

$$\Phi^{(G,i)}_{(QK,s):p,q} = \Phi^{(G,\tau_e(i))}_{(QK,\tau_h(s)):p,q},$$

$$\Phi^{(G,i)}_{(VO,s):p,q} = \Phi^{(G,\tau_e(i))}_{(VO,\tau_h(s)):p,q},$$

$$\Phi^{(G,i)}_{(G,s):p} = \Phi^{(G,\tau_e(i))}_{(G,\tau_e(s)):p},$$

$$\Phi^{(G,i)}_{(A,s):p,q} = \Phi^{(G,\tau_e(i))}_{(A,\tau_e(s)):p,\pi_e^{(\tau_e(s))}(q)},$$

$$\Phi^{(G,i)}_{(B,s):p,q} = \Phi^{(G,\tau_e(i))}_{(B,\tau_e(s)):\pi_e^{(\tau_e(s))}(p),q},$$

$$\Phi^{(G,i)}_{(G,s)} = \Phi^{(G,\tau_e(i))}_{(G,\tau_e(s))},$$

$$\Phi^{(G,i)}_{(A,s):q} = \Phi^{(G,\tau_e(i))}_{(A,\tau_e(s)):\pi_e^{(\tau_e(s))}(q)},$$

$$\Phi^{(G,i)}_{(B,s):q} = \Phi^{(G,\tau_e(i))}_{(B,\tau_e(s)):q},$$

$$\Phi^{(G,i)}_{1} = \Phi^{(G,\tau_e(i))}_{1},$$

$$\sum_{s=1}^{n_e} \Phi^{(G,i)}_{(G,s):p} = 0,$$

$$\sum_{s=1}^{n_e} \Phi^{(G,i)}_{(G,s)} = 1. \tag{213}$$

9. $[E(b)]_j^{(A,i)}$ is given by

$$[E(b)]_j^{(A,i)} = \sum_{s=1}^{h}\sum_{p=1}^{D}\sum_{q=1}^{D}\Phi_{(QK,s):p,q}^{(A,i):j}[WW]_{p,q}^{(QK,s)} + \sum_{s=1}^{h}\sum_{p=1}^{D}\sum_{q=1}^{D}\Phi_{(VO,s):p,q}^{(A,i):j}[WW]_{p,q}^{(VO,s)}$$

$$+ \sum_{s=1}^{n_e}\sum_{p=1}^{D}\Phi_{(G,s):p}^{(A,i):j}[W]_p^{(G,s)} + \sum_{s=1}^{n_e}\sum_{p=1}^{D}\sum_{q=1}^{D_A}\Phi_{(A,s):p,q}^{(A,i):j}[W]_{p,q}^{(A,s)} + \sum_{s=1}^{n_e}\sum_{p=1}^{D_A}\sum_{q=1}^{D}\Phi_{(B,s):p,q}^{(A,i):j}[W]_{p,q}^{(B,s)}$$

$$\sum_{s=1}^{n_e}\Phi_{(G,s)}^{(A,i):j}[b]^{(G,s)} + \sum_{s=1}^{n_e}\sum_{q=1}^{D_A}\Phi_{(A,s):q}^{(A,i):j}[b]_q^{(A,s)} + \sum_{s=1}^{n_e}\sum_{q=1}^{D}\Phi_{(B,s):q}^{(A,i):j}[b]_q^{(B,s)} + \Phi_1^{(A,i):j}$$

with constraints

$$\begin{aligned}
\Phi_{(QK,s):p,q}^{(A,i):j} &= \Phi_{(QK,\tau_h(s)):p,q}^{(A,\tau_e(i)):\pi_e^{(\tau_e(i))}(j)}, \\
\Phi_{(VO,s):p,q}^{(A,i):j} &= \Phi_{(VO,\tau_h(s)):p,q}^{(A,\tau_e(i)):\pi_e^{(\tau_e(i))}(j)}, \\
\Phi_{(G,s):p}^{(A,i):j} &= \Phi_{(G,\tau_e(s)):p}^{(A,\tau_e(i)):\pi_e^{(\tau_e(i))}(j)}, \\
\Phi_{(A,s):p,q}^{(A,i):j} &= \Phi_{(A,\tau_e(s)):p,\pi_e^{(\tau_e(s))}(q)}^{(A,\tau_e(i)):\pi_e^{(\tau_e(i))}(j)}, \\
\Phi_{(B,s):p,q}^{(A,i):j} &= \Phi_{(B,\tau_e(s)):\pi_e^{(\tau_e(s))}(p),q}^{(A,\tau_e(i)):\pi_e^{(\tau_e(i))}(j)}, \\
\Phi_{(G,s)}^{(A,i):j} &= \Phi_{(G,\tau_e(s))}^{(A,\tau_e(i)):\pi_e^{(\tau_e(i))}(j)}, \\
\Phi_{(A,s):q}^{(A,i):j} &= \Phi_{(A,\tau_e(s)):\pi_e^{(\tau_e(s))}(q)}^{(A,\tau_e(i)):\pi_e^{(\tau_e(i))}(j)}, \\
\Phi_{(B,s):q}^{(A,i):j} &= \Phi_{(B,\tau_e(s)):q}^{(A,\tau_e(i)):\pi_e^{(\tau_e(i))}(j)}, \\
\Phi_1^{(A,i):j} &= \Phi_1^{(A,\tau_e(i)):\pi_e^{(\tau_e(i))}(j)}, \\
\sum_{s=1}^{n_e}\Phi_{(G,s):p}^{(A,i):j} &= 0, \\
\sum_{s=1}^{n_e}\Phi_{(G,s)}^{(A,i):j} &= 0.
\end{aligned} \tag{214}$$

10. $[E(b)]_j^{(B,i)}$ is given by

$$[E(b)]_j^{(B,i)} = \sum_{s=1}^{h}\sum_{p=1}^{D}\sum_{q=1}^{D}\Phi_{(QK,s):p,q}^{(B,i):j}[WW]_{p,q}^{(QK,s)} + \sum_{s=1}^{h}\sum_{p=1}^{D}\sum_{q=1}^{D}\Phi_{(VO,s):p,q}^{(B,i):j}[WW]_{p,q}^{(VO,s)}$$

$$+ \sum_{s=1}^{n_e}\sum_{p=1}^{D}\Phi_{(G,s):p}^{(B,i):j}[W]_p^{(G,s)} + \sum_{s=1}^{n_e}\sum_{p=1}^{D}\sum_{q=1}^{D_A}\Phi_{(A,s):p,q}^{(B,i):j}[W]_{p,q}^{(A,s)} + \sum_{s=1}^{n_e}\sum_{p=1}^{D_A}\sum_{q=1}^{D}\Phi_{(B,s):p,q}^{(B,i):j}[W]_{p,q}^{(B,s)}$$

$$\sum_{s=1}^{n_e}\Phi_{(G,s)}^{(B,i):j}[b]^{(G,s)} + \sum_{s=1}^{n_e}\sum_{q=1}^{D_A}\Phi_{(A,s):q}^{(B,i):j}[b]_q^{(A,s)} + \sum_{s=1}^{n_e}\sum_{q=1}^{D}\Phi_{(B,s):q}^{(B,i):j}[b]_q^{(B,s)} + \Phi_1^{(B,i):j},$$

with constraints

$$\Phi_{(QK,s):p,q}^{(B,i):j} = \Phi_{(QK,\tau_h(s)):p,q}^{(B,\tau_e(i)):j},$$

$$\Phi_{(VO,s):p,q}^{(B,i):j} = \Phi_{(VO,\tau_h(s)):p,q}^{(B,\tau_e(i)):j},$$

$$\Phi_{(G,s):p}^{(B,i):j} = \Phi_{(G,\tau_e(s)):p}^{(B,\tau_e(i)):j},$$

$$\Phi_{(A,s):p,q}^{(B,i):j} = \Phi_{(A,\tau_e(s)):p,\pi_e^{(\tau_e(s))}(q)}^{(B,\tau_e(i)):j},$$

$$\Phi_{(B,s):p,q}^{(B,i):j} = \Phi_{(B,\tau_e(s)):\pi_e^{(\tau_e(s))}(p),q}^{(B,\tau_e(i)):j},$$

$$\Phi_{(G,s)}^{(B,i):j} = \Phi_{(G,\tau_e(s))}^{(B,\tau_e(i)):j},$$

$$\Phi_{(A,s):q}^{(B,i):j} = \Phi_{(A,\tau_e(s)):\pi_e^{(\tau_e(s))}(q)}^{(B,\tau_e(i)):j},$$

$$\Phi_{(B,s):q}^{(B,i):j} = \Phi_{(B,\tau_e(s)):q}^{(B,\tau_e(i)):j},$$

$$\Phi_{1}^{(B,i):j} = \Phi_{1}^{(B,\tau_e(i)):j},$$

$$\sum_{s=1}^{n_e} \Phi_{(G,s):p}^{(B,i):j} = 0,$$

$$\sum_{s=1}^{n_e} \Phi_{(G,s)}^{(B,i):j} = 0. \tag{215}$$

## G INVARIANT LAYER

In this section, we provide a detailed computation of the invariant layer $I(U)$ following the parameter-sharing technique as the computation of equivariant layer above. We begin with the formulation of $I(U)$ below:

$$
\begin{aligned}
I(U)_i &= \sum_{s=1}^{h}\sum_{p=1}^{D}\sum_{q=1}^{D} \Phi_{(QK,s):p,q}^{i}[WW]_{p,q}^{(QK,s)} \\
&+ \sum_{s=1}^{h}\sum_{p=1}^{D}\sum_{q=1}^{D} \Phi_{(VO,s):p,q}^{i}[WW]_{p,q}^{(VO,s)} \\
&+ \sum_{s=1}^{h}\sum_{p=1}^{D}\sum_{q=1}^{D_k} \Phi_{(Q,s):p,q}^{i}[W]_{p,q}^{(Q,s)} + \sum_{s=1}^{h}\sum_{p=1}^{D}\sum_{q=1}^{D_k} \Phi_{(K,s):p,q}^{i}[W]_{p,q}^{(K,s)} \\
&+ \sum_{s=1}^{h}\sum_{p=1}^{D}\sum_{q=1}^{D_v} \Phi_{(V,s):p,q}^{i}[W]_{p,q}^{(V,s)} + \sum_{s=1}^{h}\sum_{p=1}^{D_v}\sum_{q=1}^{D} \Phi_{(O,s):p,q}^{i}[W]_{p,q}^{(O,s)} \\
&+ \sum_{s=1}^{n_e}\sum_{p=1}^{D} \Phi_{(G,s):p}^{i}[W]_{p}^{(G,s)} + \sum_{s=1}^{n_e}\sum_{p=1}^{D}\sum_{q=1}^{D_A} \Phi_{(A,s):p,q}^{i}[W]_{p,q}^{(A,s)} + \sum_{s=1}^{n_e}\sum_{p=1}^{D_A}\sum_{q=1}^{D} \Phi_{(B,s):p,q}^{i}[W]_{p,q}^{(B,s)} \\
&\quad \sum_{s=1}^{n_e} \Phi_{(G,s)}^{i}[b]^{(G,s)} + \sum_{s=1}^{n_e}\sum_{q=1}^{D_A} \Phi_{(A,s):q}^{i}[b]_{q}^{(A,s)} + \sum_{s=1}^{n_e}\sum_{q=1}^{D} \Phi_{(B,s):q}^{i}[b]_{q}^{(B,s)} + \Phi_{1}^{i}.
\end{aligned}
\tag{216}
$$

### G.1 COMPUTING $I(gU)$

Plugging entry-wise group action 145 into Equation 216, we obtain the following expression:

$$I(gU)_i = \sum_{s=1}^{h}\sum_{p=1}^{D}\sum_{q=1}^{D} \Phi_{(QK,\tau_h^{-1}(s)):p,q}^{i}[WW]_{p,q}^{(QK,s)}$$

$$+ \sum_{s=1}^{h} \sum_{p=1}^{D} \sum_{q=1}^{D} \Phi^i_{(VO,\tau_h^{-1}(s)):p,q}[WW]^{(VO,s)}_{p,q}$$

$$+ \sum_{s=1}^{h} \sum_{p=1}^{D} \sum_{q=1}^{D_k} \Phi^i_{(Q,\tau_h^{-1}(s)):p,q} \left[ [W]^{(Q,s)} \cdot \left( M_k^{(s)} \right)^\top \right]_{p,q}$$

$$+ \sum_{s=1}^{h} \sum_{p=1}^{D} \sum_{q=1}^{D_k} \Phi^i_{(K,\tau_h^{-1}(s)):p,q} \left[ [W]^{(K,s)} \cdot \left( M_k^{(s)} \right)^{-1} \right]_{p,q}$$

$$+ \sum_{s=1}^{h} \sum_{p=1}^{D} \sum_{q=1}^{D_v} \Phi^i_{(V,\tau_h^{-1}(s)):p,q} \left[ [W]^{(V,s)} \cdot M_v^{(s)} \right]_{p,q}$$

$$+ \sum_{s=1}^{h} \sum_{p=1}^{D_v} \sum_{q=1}^{D} \Phi^i_{(O,s):p,q} \left[ \left( M_v^{(\tau_h(s))} \right)^{-1} \cdot [W]^{(O,\tau_h(s))} \right]_{p,q}$$

$$+ \sum_{s=1}^{n_e} \sum_{p=1}^{D} \Phi^i_{(G,\tau_e^{-1}(s)):p} \left[ [W]^{(G,s)} + \gamma_W \right]_p$$

$$+ \sum_{s=1}^{n_e} \sum_{p=1}^{D} \sum_{q=1}^{D_A} \Phi^i_{(A,\tau_e^{-1}(s)):p,(\pi_e^{(s)})^{-1}(q)} \left[ [W]^{(A,s)} \right]_{p,q}$$

$$+ \sum_{s=1}^{n_e} \sum_{p=1}^{D_A} \sum_{q=1}^{D} \Phi^i_{(B,\tau_e^{-1}(s)):(\pi_e^{(s)})^{-1}(p),q} \left[ [W]^{(B,s)} \right]_{p,q}$$

$$+ \sum_{s=1}^{n_e} \Phi^i_{(G,\tau_e^{-1}(s))} \left( [b]^{(G,s)} + \gamma_b \right)$$

$$+ \sum_{s=1}^{n_e} \sum_{q=1}^{D_A} \Phi^i_{(A,\tau_e^{-1}(s)):(\pi_e^{(s)})^{-1}(q)} \left[ [b]^{(A,s)} \right]_q$$

$$+ \sum_{s=1}^{n_e} \sum_{q=1}^{D} \Phi^i_{(B,\tau_e^{-1}(s)):q} \left[ [b]^{(B,s)} \right]_q$$

$$+ \Phi^i_1. \tag{217}$$

$$\tag{218}$$

## G.2 COMPARE COEFFICIENTS FROM EQUATION $I(gU) = I(U)$

In the following, we solve the equation $I(gU) = I(U)$ for all $U \in \mathcal{U}$ and $g \in \mathcal{G}_{\mathcal{U}}$ to determine the constraints for the unknown coefficients $\Phi$.

**Solving for $I(U)_i = I(gU)_i$.**

$$\sum_{s=1}^{h} \sum_{p=1}^{D} \sum_{q=1}^{D} \Phi^i_{(QK,s):p,q}[WW]^{(QK,s)}_{p,q} + \sum_{s=1}^{h} \sum_{p=1}^{D} \sum_{q=1}^{D} \Phi^i_{(VO,s):p,q}[WW]^{(VO,s)}_{p,q}$$

$$+ \sum_{s=1}^{h} \sum_{p=1}^{D} \sum_{q=1}^{D_k} \Phi^i_{(Q,s):p,q}[W]^{(Q,s)}_{p,q} + \sum_{s=1}^{h} \sum_{p=1}^{D} \sum_{q=1}^{D_k} \Phi^i_{(K,s):p,q}[W]^{(K,s)}_{p,q}$$

$$+ \sum_{s=1}^{h} \sum_{p=1}^{D} \sum_{q=1}^{D_v} \Phi^i_{(V,s):p,q}[W]^{(V,s)}_{p,q} + \sum_{s=1}^{h} \sum_{p=1}^{D_v} \sum_{q=1}^{D} \Phi^i_{(O,s):p,q}[W]^{(O,s)}_{p,q}$$

$$+ \sum_{s=1}^{n_e} \sum_{p=1}^{D} \Phi^i_{(G,s):p}[W]^{(G,s)}_p + \sum_{s=1}^{n_e} \sum_{p=1}^{D} \sum_{q=1}^{D_A} \Phi^i_{(A,s):p,q}[W]^{(A,s)}_{p,q} + \sum_{s=1}^{n_e} \sum_{p=1}^{D_A} \sum_{q=1}^{D} \Phi^i_{(B,s):p,q}[W]^{(B,s)}_{p,q}$$

$$\sum_{s=1}^{n_e} \Phi^i_{(G,s)}[b]^{(G,s)} + \sum_{s=1}^{n_e}\sum_{q=1}^{D_A} \Phi^i_{(A,s):q}[b]^{(A,s)}_q + \sum_{s=1}^{n_e}\sum_{q=1}^{D} \Phi^i_{(B,s):q}[b]^{(B,s)}_q + \Phi^i_1,$$

$$= \sum_{s=1}^{h}\sum_{p=1}^{D}\sum_{q=1}^{D} \Phi^i_{(QK,\tau_h^{-1}(s)):p,q}[WW]^{(QK,s)}_{p,q} + \sum_{s=1}^{h}\sum_{p=1}^{D}\sum_{q=1}^{D} \Phi^i_{(VO,\tau_h^{-1}(s)):p,q}[WW]^{(VO,s)}_{p,q}$$

$$+ \sum_{s=1}^{h}\sum_{p=1}^{D}\sum_{q=1}^{D_k} \Phi^i_{(Q,\tau_h^{-1}(s)):p,q}\left[[W]^{(Q,s)} \cdot \left(M_k^{(s)}\right)^\top\right]_{p,q} + \sum_{s=1}^{h}\sum_{p=1}^{D}\sum_{q=1}^{D_k} \Phi^i_{(K,\tau_h^{-1}(s)):p,q}\left[[W]^{(K,s)} \cdot \left(M_k^{(s)}\right)^{-1}\right]_{p,q}$$

$$+ \sum_{s=1}^{h}\sum_{p=1}^{D}\sum_{q=1}^{D_v} \Phi^i_{(V,\tau_h^{-1}(s)):p,q}\left[[W]^{(V,s)} \cdot M_v^{(s)}\right]_{p,q} + \sum_{s=1}^{h}\sum_{p=1}^{D_v}\sum_{q=1}^{D} \Phi^i_{(O,s):p,q}\left[\left(M_v^{(\tau_h(s))}\right)^{-1} \cdot [W]^{(O,\tau_h(s))}\right]_{p,q}$$

$$+ \sum_{s=1}^{n_e}\sum_{p=1}^{D} \Phi^i_{(G,\tau_e^{-1}(s)):p}\left[[W]^{(G,s)} + \gamma_W\right]_p + \sum_{s=1}^{n_e}\sum_{p=1}^{D}\sum_{q=1}^{D_A} \Phi^i_{(A,\tau_e^{-1}(s)):p,(\pi_e^{(s)})^{-1}(q)}\left[[W]^{(A,s)}\right]_{p,q}$$

$$+ \sum_{s=1}^{n_e}\sum_{p=1}^{D_A}\sum_{q=1}^{D} \Phi^i_{(B,\tau_e^{-1}(s)):(\pi_e^{(s)})^{-1}(p),q}\left[[W]^{(B,s)}\right]_{p,q} + \sum_{s=1}^{n_e} \Phi^i_{(G,\tau_e^{-1}(s))}\left([b]^{(G,s)} + \gamma_b\right)$$

$$+ \sum_{s=1}^{n_e}\sum_{q=1}^{D_A} \Phi^i_{(A,\tau_e^{-1}(s)):(\pi_e^{(s)})^{-1}(q)}\left[[b]^{(A,s)}\right]_q + \sum_{s=1}^{n_e}\sum_{q=1}^{D} \Phi^i_{(B,\tau_e^{-1}(s)):q}\left[[b]^{(B,s)}\right]_q + \Phi^i_1.$$

Using lemma F.1, we obtain the constraints:

$$\Phi^i_{(QK,s):p,q} = \Phi^i_{(QK,\tau_h^{-1}(s)):p,q},$$
$$\Phi^i_{(VO,s):p,q} = \Phi^i_{(VO,\tau_h^{-1}(s)):p,q},$$
$$\Phi^i_{(Q,\tau_h^{-1}(s)):p,q} = 0,$$
$$\Phi^i_{(K,\tau_h^{-1}(s)):p,q} = 0,$$
$$\Phi^i_{(V,\tau_h^{-1}(s)):p,q} = 0,$$
$$\Phi^i_{(O,\tau_h^{-1}(s)):p,q} = 0,$$
$$\Phi^i_{(G,s):p} = \Phi^i_{(G,\tau_e^{-1}(s)):p},$$
$$\Phi^i_{(A,s):p,q} = \Phi^i_{(A,\tau_e^{-1}(s)):p,(\pi_e^{(s)})^{-1}(q)},$$
$$\Phi^i_{(B,s):p,q} = \Phi^i_{(B,\tau_e^{-1}(s)):(\pi_e^{(s)})^{-1}(p),q},$$
$$\Phi^i_{(G,s)} = \Phi^i_{(G,\tau_e^{-1}(s))},$$
$$\Phi^i_{(A,s):q} = \Phi^i_{(A,\tau_e^{-1}(s)):(\pi_e^{(s)})^{-1}(q)},$$
$$\Phi^i_{(B,s):q} = \Phi^i_{(B,\tau_e^{-1}(s)):q},$$
$$\Phi^i_1 = \Phi^i_1,$$
$$\sum_{s=1}^{n_e} \Phi^i_{(G,\tau_e^{-1}(s)):p} = 0,$$
$$\sum_{s=1}^{n_e} \Phi^i_{(G,\tau_e^{-1}(s))} = 0.$$

Therefore:

$$\Phi^i_{(QK,s):p,q} = \Phi^i_{(QK,\tau_h(s)):p,q},$$

$$\Phi^i_{(VO,s):p,q} = \Phi^i_{(VO,\tau_h(s)):p,q},$$

$$\Phi^i_{(Q,s):p,q} = 0,$$

$$\Phi^i_{(K,s):p,q} = 0,$$

$$\Phi^i_{(V,s):p,q} = 0,$$

$$\Phi^i_{(O,s):p,q} = 0,$$

$$\Phi^i_{(G,s):p} = \Phi^i_{(G,\tau_e(s)):p},$$

$$\Phi^i_{(A,s):p,q} = \Phi^i_{(A,\tau_e(s)):p,\pi_e^{(s)}(q)}, \qquad (219)$$

$$\Phi^i_{(B,s):p,q} = \Phi^i_{(B,\tau_e(s)):\pi_e^{(s)}(p),q},$$

$$\Phi^i_{(G,s)} = \Phi^i_{(G,\tau_e(s))},$$

$$\Phi^i_{(A,s):q} = \Phi^i_{(A,\tau_e(s)):\pi_e^{(s)}(q)},$$

$$\Phi^i_{(B,s):q} = \Phi^i_{(B,\tau_e(s)):q},$$

$$\sum_{s=1}^{n_e} \Phi^i_{(G,s):p} = 0,$$

$$\sum_{s=1}^{n_e} \Phi^i_{(G,s)} = 0.$$

## H   IMPLEMENTATION DETAILS OF THE EQUIVARIANT AND INVARIANT LAYER

In this section, we provide implementation details for the equivariant and invariant layers described in the previous sections. The bullet notation $\bullet$ is used to indicate index-wise equality. For example, $x_{i,\bullet}$ denotes that all values along the second index are equal, i.e., $x_{i,j} = x_{i,j'}$ for all pairs $(j, j')$.

Based on the constraints derived in Section F.4, we express all formulations using bullet notation, which provides a more practical and concise format for implementation. This notation not only streamlines the empirical realization of the constraints but also clearly highlights the underlying parameter-sharing structure. Each summation written in bullet notation is implemented using PyTorch's `einsum`, as detailed in Section H.4. For certain parameterization constraints that are not straightforward, we rely on Propositions H.1, H.4, and Corollaries H.3, H.5 from Section H.1 to present them in bullet notation.

### H.1   EQUIVARIANT CONSTRAINT REDUCTION TO BULLET FORM

**Proposition H.1.** *Under the parameter sharing constraint:*

$$\begin{cases} \Phi^{(G,i):j}_{(G,s):p} & = \Phi^{(G,\tau_e(i)):j}_{(G,\tau_e(s)):p}, \\ \sum_{s=1}^{n_e} \Phi^{(G,i):j}_{(G,s):j} & = 1, \\ \sum_{s=1}^{n_e} \Phi^{(G,i):j}_{(G,s):p} & = 0, \quad for\ p \neq j, \end{cases}$$

*we can write the summation*

$$\sum_{s=1}^{n_e} \sum_{p=1}^{D} \Phi^{(G,i):j}_{(G,s):p} [W]^{(G,s)}_p$$

$$= [W]^{(G,i)}_j + \sum_{s=1}^{n_e} \sum_{p=1}^{D} \left( \Phi^{(G,\bullet):j}_{(G,\bullet):p} \right)_1 [W]^{(G,s)}_p - n_e \sum_{p=1}^{D} \left( \Phi^{(G,\bullet):j}_{(G,\bullet):p} \right)_1 [W]^{(G,i)}_p.$$

*Proof.* From the constraint $\Phi^{(G,i):j}_{(G,s):p} = \Phi^{(G,\tau_e(i)):j}_{(G,\tau_e(s)):p}$, we obtain:

$$\Phi^{(G,i):j}_{(G,s):p} = \begin{cases} (\varphi_1)^j_p & \text{if } i \neq s, \\ (\varphi_2)^j_p & \text{if } i = s. \end{cases}$$

To determine the constraints on $(\varphi_1)^j_p$ and $(\varphi_2)^j_p$, we examine two cases:

**Case 1:** $p = j$

From the constraint

$$\sum_{s=1}^{n_e} \Phi^{(G,i):j}_{(G,s):j} = 1,$$

we substitute the expression for $\Phi$ and obtain:

$$(\varphi_2)^j_j + (n_e - 1)(\varphi_1)^j_j = 1, \quad \Rightarrow \quad (\varphi_2)^j_j = 1 - (n_e - 1)(\varphi_1)^j_j.$$

**Case 2:** $p \neq j$

From the constraint

$$\sum_{s=1}^{n_e} \Phi^{(G,i):j}_{(G,s):p} = 0,$$

we similarly obtain:

$$(\varphi_2)^j_p + (n_e - 1)(\varphi_1)^j_p = 0, \quad \Rightarrow \quad (\varphi_2)^j_p = -(n_e - 1)(\varphi_1)^j_p.$$

Combining both cases, we conclude with the following expressions ($i \neq s, p \neq j$):

$$\begin{aligned}
\Phi^{(G,i):j}_{(G,i):j} &= 1 - (n_e - 1)(\varphi_1)^j_j, \\
\Phi^{(G,i):j}_{(G,s):j} &= (\varphi_1)^j_j, \\
\Phi^{(G,i):j}_{(G,i):p} &= -(n_e - 1)(\varphi_1)^j_p, \\
\Phi^{(G,i):j}_{(G,s):p} &= (\varphi_1)^j_p.
\end{aligned} \tag{220}$$

We have the following chain of reduction:

$$\sum_{s=1}^{n_e} \sum_{p=1}^{D} \Phi^{(G,i):j}_{(G,s):p} [W]^{(G,s)}_p$$

$$= \sum_{s=1}^{n_e} \left( \sum_{p \neq j} \Phi^{(G,i):j}_{(G,s):p} [W]^{(G,s)}_p + \Phi^{(G,i):j}_{(G,s):j} [W]^{(G,s)}_j \right)$$

$$= \sum_{s \neq i} \sum_{p \neq j} \Phi^{(G,i):j}_{(G,s):p} [W]^{(G,s)}_p + \sum_{p \neq j} \Phi^{(G,i):j}_{(G,i):p} [W]^{(G,i)}_p + \sum_{s \neq i} \Phi^{(G,i):j}_{(G,s):j} [W]^{(G,s)}_j + \Phi^{(G,i):j}_{(G,i):j} [W]^{(G,i)}_j.$$

Plugging Equation 220 into the expression, we obtain:

$$\sum_{s=1}^{n_e} \sum_{p=1}^{D} \Phi^{(G,i):j}_{(G,s):p} [W]^{(G,s)}_p$$

$$= \sum_{s \neq i} \sum_{p \neq j} (\varphi_1)^j_p - (n_e - 1) \sum_{p \neq j} (\varphi_1)^j_p [W]^{(G,i)}_p + (\varphi_1)^j_j \sum_{s \neq i} [W]^{(G,s)}_j + (1 - (n_e - 1)(\varphi_1)^j_j)[W]^{(G,i)}_j$$

$$= [W]^{(G,i)}_j - (n_e - 1) \left( \sum_{p \neq j} (\varphi_1)^j_p [W]^{(G,i)}_p + (\varphi_1)^j_j [W]^{(G,i)}_j \right) + \sum_{s \neq i} \sum_{p \neq j} (\varphi_1)^j_p [W]^{(G,s)}_p + (\varphi_1)^j_j \sum_{s \neq i} [W]^{(G,s)}_j$$

$$= [W]_j^{(G,i)} - n_e \sum_{p=1}^{D} (\varphi_1)_p^j [W]_p^{(G,i)} + \left( \sum_{p=1}^{D} (\varphi_1)_p^j [W]_p^{(G,i)} + \sum_{s \neq i} \sum_{p \neq j} (\varphi_1)_p^j [W]_p^{(G,s)} + (\varphi_1)_j^j \sum_{s \neq i} [W]_j^{(G,s)} \right)$$

$$= [W]_j^{(G,i)} - n_e \sum_{p=1}^{D} (\varphi_1)_p^j [W]_p^{(G,i)} + \left( \sum_{s \neq i} \sum_{p=1}^{D} (\varphi_1)_p^j [W]_p^{(G,s)} + (\varphi_1)_j^j \sum_{s \neq i} [W]_j^{(G,s)} \right)$$

$$= [W]_j^{(G,i)} - n_e \sum_{p=1}^{D} (\varphi_1)_p^j [W]_p^{(G,i)} + \sum_{s=1}^{n_e} \sum_{p=1}^{D} (\varphi_1)_p^j [W]_p^{(G,s)}.$$

Define $(\varphi_i)_p^j = \left( \Phi_{(G,\bullet):p}^{(G,\bullet):j} \right)_1$. This concludes the proof of the proposition.

$\square$

*Remark* H.2. As shown in Proposition H.1, the equivariant layer for the $W_G$ component naturally introduces a skip connection $[W]_j^{(G,i)}$. This behavior is absent in equivariant layers defined under the symmetry group of standard Transformers and arises specifically from the group structure associated with MoE Transformers. Thus, it highlights a distinctive feature of the MoE-specific equivariant formulation.

**Corollary H.3.** *Under the parameter sharing constraint:*

$$\begin{cases} \Phi_{(G,s)}^{(G,i)} & = \Phi_{(G,\tau_e(s))}^{(G,\tau_e(i))}, \\ \sum_{s=1}^{n_e} \Phi_{(G,s)}^{(G,i)} & = 1, \end{cases}$$

*we can write the summation*

$$\sum_{s=1}^{n_e} \Phi_{(G,s)}^{(G,i)} [W]^{(G,s)} = [W]^{(G,i)} + \sum_{s=1}^{n_e} \left( \Phi_{(G,\bullet)}^{(G,\bullet)} \right)_1 [W]^{(G,s)} - n_e \left( \Phi_{(G,\bullet)}^{(G,\bullet)} \right)_1 [W]^{(G,i)}.$$

*Proof.* Applying Proposition H.1 with $D = 1$ and renaming the index, we obtain the desired result and thus conclude the proof of Corollary H.3. $\square$

**Proposition H.4.** *Under the parameter sharing constraint:*

$$\begin{cases} \Phi_{(G,s):p}^{(A,i):j,k} & = \Phi_{(G,\tau_e(s)):p}^{(A,\tau_e(i)):j,\pi_e^{(\tau_e(i))}(k)}, \\ \sum_{s=1}^{n_e} \Phi_{(G,s):p}^{(A,i):j,k} & = 0, \end{cases}$$

*we can write the summation*

$$\sum_{s=1}^{n_e} \sum_{p=1}^{D} \Phi_{(G,s):p}^{(A,i):j,k} [W]_p^{(G,s)} = \sum_{s=1}^{n_e} \sum_{p=1}^{D} \left( \Phi_{(G,\bullet):p}^{(A,\bullet):j,\bullet} \right)_1 [W]_p^{(G,s)} - \sum_{p=1}^{D} n_e \left( \Phi_{(G,\bullet):p}^{(A,\bullet):j,\bullet} \right)_1 [W]_p^{(G,i)}.$$

*Proof.* From the constraint $\Phi_{(G,s):p}^{(A,i):j,k} = \Phi_{(G,\tau_e(s)):p}^{(A,\tau_e(i)):j,\pi_e^{(\tau_e(i))}(k)}$, we obtain:

$$\Phi_{(G,s):p}^{(A,i):j,k} = \begin{cases} (\varphi_1)_p^j & \text{if } i \neq s, \\ (\varphi_2)_p^j & \text{if } i = s. \end{cases}$$

From the constraint

$$\sum_{s=1}^{n_e} \Phi_{(G,s):p}^{(A,i):j,k} = 0,$$

we substitute the expression for $\Phi$ to obtain:

$$(n_e - 1)(\varphi_1)_p^j + (\varphi_2)_p^j = 0 \Rightarrow (\varphi_2)_p^j = -(n_e - 1)(\varphi_1)_p^j.$$

We conclude the following expressions ($i \neq s$):

$$\Phi^{(A,i):j,k}_{(G,s):p} = (\varphi_1)^j_p,$$

$$\Phi^{(A,i):j,k}_{(G,i):p} = -(n_e - 1)(\varphi_1)^j_p.$$

We consider the following chain of reduction:

$$\sum_{s=1}^{n_e} \sum_{p=1}^{D} \Phi^{(A,i):j,k}_{(G,s):p} [W]^{(G,s)}_p = \sum_{s \neq i} \sum_{p=1}^{D} \Phi^{(A,i):j,k}_{(G,s):p} [W]^{(G,s)}_p + \sum_{p=1}^{D} \Phi^{(A,i):j,k}_{(G,i):p} [W]^{(G,i)}_p$$

$$= \sum_{s \neq i} \sum_{p=1}^{D} (\varphi_1)^j_p [W]^{(G,s)}_p - (n_e - 1) \sum_{p=1}^{D} (\varphi_1)^j_p [W]^{(G,i)}_p$$

$$= \left( \sum_{s \neq i} \sum_{p=1}^{D} (\varphi_1)^j_p [W]^{(G,s)}_p + \sum_{p=1}^{D} (\varphi_1)^j_p [W]^{(G,i)}_p \right) - n_e \sum_{p=1}^{D} (\varphi_1)^j_p [W]^{(G,i)}_p$$

$$= \sum_{s=1}^{n_e} \sum_{p=1}^{D} (\varphi_1)^j_p [W]^{(G,s)}_p - n_e \sum_{p=1}^{D} (\varphi_1)^j_p [W]^{(G,i)}_p.$$

Define $(\varphi_1)^j_p = \left( \Phi^{(A,\bullet):j,\bullet}_{(G,\bullet):p} \right)_1$, this concludes the proof of the proposition. $\square$

**Corollary H.5.** *Under the parameter sharing constraint:*

$$\begin{cases} \sum_{s=1}^{n_e} \Phi^{(G,i):j}_{(G,s)} &= 0, \\ \Phi^{(G,i):j}_{(G,s)} &= \Phi^{(G,\tau_e(i)):j}_{(G,\tau_e(s))}, \end{cases}$$

*we can write the summation*

$$\sum_{s=1}^{n_e} \Phi^{(G,i):j}_{(G,s)} [b]^{(G,s)} = \sum_{s=1}^{n_e} \left( \Phi^{(G,\bullet):j}_{(G,\bullet)} \right)_1 [b]^{(G,s)} - n_e \left( \Phi^{(G,\bullet):j}_{(G,\bullet)} \right)_1 [b]^{(G,i)}.$$

*Proof.* Applying Proposition H.4 with $D = 1$ and renaming the index, we conclude the proof of the Corollary. $\square$

## H.2 EQUIVARIANT LAYERS WITH BULLET NOTATION

1. Weight sharing form for $[E(W)]^{(Q,i)}_{j,k}$.
   From Equation 206:

$$\Phi^{(Q,i):j,k}_{(Q,i):p,k} = \Phi^{(Q,\tau(i)):j,k'}_{(Q,\tau(i)):p,k'}. \tag{221}$$

   Since the constraint is satisfied with any $\tau$, we obtain the following weight sharing form:

$$[E(W)]^{(Q,i)}_{j,k} = \sum_{p=1}^{D} \Phi^{(Q,\bullet):j,\bullet}_{(Q,\bullet):p,\bullet} [W]^{(Q,i)}_{p,k}.$$

2. Weight sharing form for $[E(W)]^{(K,i)}_{j,k}$.
   From Equation 207:

$$\Phi^{(K,i):j,k}_{(K,i):p,k} = \Phi^{(K,\tau(i)):j,k'}_{(K,\tau(i)):p,k'}. \tag{222}$$

   Similarly, we obtain the weight sharing form:

$$[E(W)]^{(K,i)}_{j,k} = \sum_{p=1}^{D} \Phi^{(K,\bullet):j,\bullet}_{(K,\bullet):p,\bullet} [W]^{(K,i)}_{p,k}.$$

3. Weight sharing form for $[E(W)]_{j,k}^{(V,i)}$.

   From Equation 208:

$$\Phi_{(V,i):p,k}^{(V,i):j,k} = \Phi_{(V,\tau(i)):p,k'}^{(V,\tau(i)):j,k'}. \tag{223}$$

   We obtain the weight sharing form:

$$[E(W)]_{j,k}^{(V,i)} = \sum_{p=1}^{D} \Phi_{(V,\bullet):p,\bullet}^{(V,\bullet):j,\bullet} [W]_{p,k}^{(V,i)}.$$

4. Weight sharing form for $[E(W)]_{j,k}^{(O,i)}$.

   From Equation 209:

$$\Phi_{(O,i):j',q}^{(O,i):j',k} = \Phi_{(O,\tau(i)):j',q}^{(O,\tau(i)):j,k}. \tag{224}$$

   We obtain the weight sharing form:

$$[E(W)]_{j,k}^{(O,i)} = \sum_{q=1}^{D} \Phi_{(O,\bullet):\bullet,q}^{(O,\bullet):\bullet,k} [W]_{j,k}^{(O,i)}.$$

5. Weight sharing form for $[E(W)]_{j}^{(G,i)}$.

   From Equation 210:

$$\Phi_{(QK,s):p,q}^{(G,i):j} = \Phi_{(QK,\tau_h(s)):p,q}^{(G,\tau_e(i)):j},$$

$$\Phi_{(VO,s):p,q}^{(G,i):j} = \Phi_{(VO,\tau_h(s)):p,q}^{(G,\tau_e(i)):j},$$

$$\Phi_{(A,s):p,q}^{(G,i):j} = \Phi_{(A,\tau_e(s)):p,\pi_e^{(\tau_e(s))}(q)}^{(G,\tau_e(i)):j},$$

$$\Phi_{(B,s):p,q}^{(G,i):j} = \Phi_{(B,\tau_e(s)):\pi_e^{(\tau_e(s))}(p),q}^{(G,\tau_e(i)):j},$$

$$\Phi_{(A,s):q}^{(G,i):j} = \Phi_{(A,\tau_e(s)):\pi_e^{(\tau_e(s))}(q)}^{(G,\tau_e(i)):j},$$

$$\Phi_{(B,s):q}^{(G,i):j} = \Phi_{(B,\tau_e(s)):q}^{(G,\tau_e(i)):j},$$

$$\Phi_{1}^{(G,i):j} = \Phi_{1}^{(G,\tau_e(i)):j},$$

$$\begin{cases} \Phi_{(G,s):p}^{(G,i):j} = \Phi_{(G,\tau_e(s)):p}^{(G,\tau_e(i)):j}, \\ \sum_{s=1}^{n_e} \Phi_{(G,s):p}^{(G,i):j} = 0 \quad (p \neq j), \\ \sum_{s=1}^{n_e} \Phi_{(G,s):j}^{(G,i):j} = 1, \end{cases}$$

$$\begin{cases} \sum_{s=1}^{n_e} \Phi_{(G,s)}^{(G,i):j} = 0, \\ \Phi_{(G,s)}^{(G,i):j} = \Phi_{(G,\tau_e(s))}^{(G,\tau_e(i)):j}. \end{cases}$$

   Using Proposition H.1 and Corollary H.5, we obtain the weight sharing form:

$$[E(W)]_{j}^{(G,i)} = \sum_{s=1}^{h} \sum_{p=1}^{D} \sum_{q=1}^{D} \Phi_{(QK,\bullet):p,q}^{(G,\bullet):j} [WW]_{p,q}^{(QK,s)}$$

$$+ \sum_{s=1}^{h} \sum_{p=1}^{D} \sum_{q=1}^{D} \Phi_{(VO,\bullet):p,q}^{(G,\bullet):j} [WW]_{p,q}^{(VO,s)}$$

$$+ [W]_{j}^{(G,i)} + \sum_{s=1}^{n_e} \sum_{p=1}^{D} \left( \Phi_{(G,\bullet):p}^{(G,\bullet):j} \right)_{1} [W]_{p}^{(G,s)} - n_e \sum_{p=1}^{D} \left( \Phi_{(G,\bullet):p}^{(G,\bullet):j} \right)_{1} [W]_{p}^{(G,i)}$$

$$+ \sum_{s=1}^{n_e} \sum_{p=1}^{D} \sum_{q=1}^{D_A} \left( \Phi_{(A,\bullet):p,\bullet}^{(G,\bullet):j} \right)_1 [W]_{p,q}^{(A,s)} + \sum_{p=1}^{D} \sum_{q=1}^{D_A} \left( \Phi_{(A,\bullet):p,\bullet}^{(G,\bullet):j} \right)_2 [W]_{p,q}^{(A,i)}$$

$$+ \sum_{s=1}^{n_e} \sum_{p=1}^{D_A} \sum_{q=1}^{D} \left( \Phi_{(B,\bullet):\bullet,q}^{(G,\bullet):j} \right)_1 [W]_{p,q}^{(B,s)} + \sum_{p=1}^{D_A} \sum_{q=1}^{D} \left( \Phi_{(B,\bullet):\bullet,q}^{(G,\bullet):j} \right)_2 [W]_{p,q}^{(B,i)}$$

$$+ \sum_{s=1}^{n_e} \left( \Phi_{(G,\bullet)}^{(G,\bullet):j} \right)_1 [b]^{(G,s)} - n_e \left( \Phi_{(G,\bullet)}^{(G,\bullet):j} \right)_1 [b]^{(G,i)}$$

$$+ \sum_{s=1}^{n_e} \sum_{q=1}^{D_A} \left( \Phi_{(A,\bullet):\bullet}^{(G,\bullet):j} \right)_1 [b]_q^{(A,s)} + \sum_{q=1}^{D_A} \left( \Phi_{(A,\bullet):\bullet}^{(G,\bullet):j} \right)_2 [b]_q^{(A,i)}$$

$$+ \sum_{s=1}^{n_e} \sum_{q=1}^{D} \left( \Phi_{(B,\bullet):q}^{(G,\bullet):j} \right)_1 [b]_q^{(B,s)} + \sum_{q=1}^{D} \left( \Phi_{(B,\bullet):q}^{(G,\bullet):j} \right)_1 [b]_q^{(B,i)}$$

$$+ \Phi_1^{(G,\bullet):j}.$$

6. Weight sharing form for $[E(W)]_{j,k}^{(A,i)}$.

From Equation 211:

$$\Phi_{(QK,s):p,q}^{(A,i):j,k} = \Phi_{(QK,\tau_h(s)):p,q}^{(A,\tau_e(i)):j,\pi_e^{(\tau_e(i))}(k)},$$

$$\Phi_{(VO,s):p,q}^{(A,i):j,k} = \Phi_{(VO,\tau_h(s)):p,q}^{(A,\tau_e(i)):j,\pi_e^{(\tau_e(i))}(k)},$$

$$\Phi_{(A,s):p,q}^{(A,i):j,k} = \Phi_{(A,\tau_e(s)):p,\pi_e^{(\tau_e(s))}(q)}^{(A,\tau_e(i)):j,\pi_e^{(\tau_e(i))}(k)},$$

$$\Phi_{(B,s):p,q}^{(A,i):j,k} = \Phi_{(B,\tau_e(s)):\pi_e^{(\tau_e(s))}(p),q}^{(A,\tau_e(i)):j,\pi_e^{(\tau_e(i))}(k)},$$

$$\Phi_{(A,s):q}^{(A,i):j,k} = \Phi_{(A,\tau_e(s)):\pi_e^{(\tau_e(s))}(q)}^{(A,\tau_e(i)):j,\pi_e^{(\tau_e(i))}(k)},$$

$$\Phi_{(B,s):q}^{(A,i):j,k} = \Phi_{(B,\tau_e(s)):q}^{(A,\tau_e(i)):j,\pi_e^{(\tau_e(i))}(k)},$$

$$\Phi_1^{(A,i):j,k} = \Phi_1^{(A,\tau_e(i)):j,\pi_e^{(\tau_e(i))}(k)},$$

$$\begin{cases} \Phi_{(G,s):p}^{(A,i):j,k} = \Phi_{(G,\tau_e(s)):p}^{(A,\tau_e(i)):j,\pi_e^{(\tau_e(i))}(k)}, \\ \sum_{s=1}^{n_e} \Phi_{(G,s):p}^{(A,i):j,k} = 0, \end{cases}$$

$$\begin{cases} \Phi_{(G,s)}^{(A,i):j,k} = \Phi_{(G,\tau_e(s))}^{(A,\tau_e(i)):j,\pi_e^{(\tau_e(i))}(k)}, \\ \sum_{s=1}^{n_e} \Phi_{(G,s)}^{(A,i):j,k} = 0. \end{cases}$$

Using Proposition H.4 and Corollary H.5, we obtain the weight sharing form:

$$[E(W)]_{j,k}^{(A,i)} = \sum_{s=1}^{h} \sum_{p=1}^{D} \sum_{q=1}^{D} \Phi_{(QK,\bullet):p,q}^{(A,\bullet):j,\bullet} [WW]_{p,q}^{(QK,s)}$$

$$+ \sum_{s=1}^{h} \sum_{p=1}^{D} \sum_{q=1}^{D} \Phi_{(VO,\bullet):p,q}^{(A,\bullet):j,\bullet} [WW]_{p,q}^{(VO,s)}$$

$$+ \sum_{s=1}^{n_e} \sum_{p=1}^{D} \left( \Phi_{(G,\bullet):p}^{(A,\bullet):j,\bullet} \right)_1 [W]_p^{(G,s)} - \sum_{p=1}^{D} n_e \left( \Phi_{(G,\bullet):p}^{(A,\bullet):j,\bullet} \right)_1 [W]_p^{(G,i)}$$

$$+ \sum_{s=1}^{n_e} \sum_{p=1}^{D} \sum_{q=1}^{D_A} \left( \Phi_{(A,\bullet):p,\bullet}^{(A,\bullet):j,\bullet} \right)_1 [W]_{p,q}^{(A,s)} + \sum_{p=1}^{D} \sum_{q=1}^{D_A} \left( \Phi_{(A,\bullet):p,\bullet}^{(A,\bullet):j,\bullet} \right)_2 [W]_{p,q}^{(A,i)}$$

$$+ \sum_{s=1}^{n_e} \sum_{p=1}^{D} \left( \Phi_{(A,\bullet):p,\bullet}^{(A,\bullet):j,\bullet} \right)_3 [W]_{p,k}^{(A,s)} + \sum_{p=1}^{D} \left( \Phi_{(A,\bullet):p,\bullet}^{(A,\bullet):j,\bullet} \right)_4 [W]_{p,k}^{(A,i)}$$

$$+ \sum_{s=1}^{n_e} \sum_{p=1}^{D_A} \sum_{q=1}^{D} \left( \Phi_{(B,\bullet):\bullet,q}^{(A,\bullet):j,\bullet} \right)_1 [W]_{p,q}^{(B,s)} + \sum_{p=1}^{D_A} \sum_{q=1}^{D} \left( \Phi_{(B,\bullet):\bullet,q}^{(A,\bullet):j,\bullet} \right)_2 [W]_{p,q}^{(B,i)}$$

$$+ \sum_{s=1}^{n_e} \sum_{q=1}^{D} \left( \Phi_{(B,\bullet):\bullet,q}^{(A,\bullet):j,\bullet} \right)_3 [W]_{k,q}^{(B,s)} + \sum_{q=1}^{D} \left( \Phi_{(B,\bullet):\bullet,q}^{(A,\bullet):j,\bullet} \right)_4 [W]_{k,q}^{(B,i)}$$

$$+ \sum_{s=1}^{n_e} \left( \Phi_{(G,\bullet)}^{(A,\bullet):j,\bullet} \right)_1 [b]^{(G,s)} - n_e \left( \Phi_{(G,\bullet)}^{(A,\bullet):j,\bullet} \right)_1 [b]^{(G,i)}$$

$$+ \sum_{s=1}^{n_e} \sum_{q=1}^{D_A} \left( \Phi_{(A,\bullet):\bullet}^{(A,\bullet):j,\bullet} \right)_1 [b]_q^{(A,s)} + \sum_{q=1}^{D_A} \left( \Phi_{(A,\bullet):\bullet}^{(A,\bullet):j,\bullet} \right)_2 [b]_q^{(A,i)}$$

$$+ \sum_{s=1}^{n_e} \left( \Phi_{(A,\bullet):\bullet}^{(A,\bullet):j,\bullet} \right)_3 [b]_k^{(A,s)} + \left( \Phi_{(A,\bullet):\bullet}^{(A,\bullet):j,\bullet} \right)_4 [b]_k^{(A,i)}$$

$$+ \sum_{s=1}^{n_e} \sum_{q=1}^{D} \left( \Phi_{(B,\bullet):q}^{(A,\bullet):j,\bullet} \right)_1 [b]_q^{(B,s)} + \sum_{q=1}^{D} \left( \Phi_{(B,\bullet):q}^{(A,\bullet):j,\bullet} \right)_2 [b]_q^{(B,i)}$$

$$+ \Phi_1^{(A,\bullet):j,\bullet}.$$

7. Weight sharing form for $[E(W)]_{j,k}^{(B,i)}$.

   From Equation 212:

$$\Phi_{(QK,s):p,q}^{(B,i):j,k} = \Phi_{(QK,\tau_h(s)):p,q}^{(B,\tau_e(i)):\pi_e^{(\tau_e(i))}(j),k},$$

$$\Phi_{(VO,s):p,q}^{(B,i):j,k} = \Phi_{(VO,\tau_h(s)):p,q}^{(B,\tau_e(i)):\pi_e^{(\tau_e(i))}(j),k},$$

$$\Phi_{(A,s):p,q}^{(B,i):j,k} = \Phi_{(A,\tau_e(s)):p,\pi_e^{(\tau_e(s))}(q)}^{(B,\tau_e(i)):\pi_e^{(\tau_e(i))}(j),k},$$

$$\Phi_{(B,s):p,q}^{(B,i):j,k} = \Phi_{(B,\tau_e(s)):\pi_e^{(\tau_e(s))}(p),q}^{(B,\tau_e(i)):\pi_e^{(\tau_e(i))}(j),k},$$

$$\Phi_{(A,s):q}^{(B,i):j,k} = \Phi_{(A,\tau_e(s)):\pi_e^{(\tau_e(s))}(q)}^{(B,\tau_e(i)):\pi_e^{(\tau_e(i))}(j),k},$$

$$\Phi_1^{(B,i):j,k} = \Phi_1^{(B,\tau_e(i)):\pi_e^{(\tau_e(i))}(j),k},$$

$$\begin{cases} \Phi_{(G,s):p}^{(B,i):j,k} = \Phi_{(G,\tau_e(s)):p}^{(B,\tau_e(i)):\pi_e^{(\tau_e(i))}(j),k}, \\ \sum_{s=1}^{n_e} \Phi_{(G,s):p}^{(B,i):j,k} = 0, \end{cases}$$

$$\begin{cases} \Phi_{(G,s)}^{(B,i):j,k} = \Phi_{(G,\tau_e(s))}^{(B,\tau_e(i)):\pi_e^{(\tau_e(i))}(j),k}, \\ \sum_{s=1}^{n_e} \Phi_{(G,s)}^{(B,i):j,k} = 0. \end{cases}$$

Using Corollary H.5, we obtain the weight sharing form:

$$[E(W)]_{j,k}^{(B,i)} = \sum_{s=1}^{h} \sum_{p=1}^{D} \sum_{q=1}^{D} \Phi_{(QK,\bullet):p,q}^{(B,\bullet):\bullet,k} [WW]_{p,q}^{(QK,s)}$$

$$+ \sum_{s=1}^{h} \sum_{p=1}^{D} \sum_{q=1}^{D} \Phi_{(VO,\bullet):p,q}^{(B,\bullet):\bullet,k} [WW]_{p,q}^{(VO,s)}$$

$$+ \sum_{s=1}^{n_e} \sum_{p=1}^{D} \left( \Phi_{(G,\bullet):p}^{(B,\bullet):\bullet,k} \right)_1 [W]_p^{(G,s)} - \sum_{p=1}^{D} n_e \left( \Phi_{(G,\bullet):p}^{(B,\bullet):\bullet,k} \right)_1 [W]_p^{(G,i)}$$

$$+ \sum_{s=1}^{n_e} \sum_{p=1}^{D} \sum_{q=1}^{D_A} \left( \Phi_{(A,\bullet):p,\bullet}^{(B,\bullet):\bullet,k} \right)_1 [W]_{p,q}^{(A,s)} + \sum_{p=1}^{D} \sum_{q=1}^{D_A} \left( \Phi_{(A,\bullet):p,\bullet}^{(B,\bullet):\bullet,k} \right)_2 [W]_{p,q}^{(A,i)}$$

$$+ \sum_{s=1}^{n_e} \sum_{p=1}^{D} \left( \Phi_{(A,\bullet):p,\bullet}^{(B,\bullet):\bullet,k} \right)_3 [W]_{p,j}^{(A,s)} + \sum_{p=1}^{D} \left( \Phi_{(A,\bullet):p,\bullet}^{(B,\bullet):\bullet,k} \right)_4 [W]_{p,j}^{(A,i)}$$

$$+ \sum_{s=1}^{n_e} \sum_{p=1}^{D_A} \sum_{q=1}^{D} \left( \Phi_{(B,\bullet):\bullet,q}^{(B,\bullet):\bullet,k} \right)_1 [W]_{p,q}^{(B,s)} + \sum_{p=1}^{D_A} \sum_{q=1}^{D} \left( \Phi_{(B,\bullet):\bullet,q}^{(B,\bullet):\bullet,k} \right)_2 [W]_{p,q}^{(B,i)}$$

$$+ \sum_{s=1}^{n_e} \sum_{q=1}^{D} \left( \Phi_{(B,\bullet):\bullet,q}^{(B,\bullet):\bullet,k} \right)_3 [W]_{j,q}^{(B,s)} + \sum_{q=1}^{D} \left( \Phi_{(B,\bullet):\bullet,q}^{(B,\bullet):\bullet,k} \right)_4 [W]_{j,q}^{(B,i)}$$

$$+ \sum_{s=1}^{n_e} \left( \Phi_{(G,\bullet)}^{(B,\bullet):\bullet,k} \right)_1 [b]^{(G,s)} - n_e \left( \Phi_{(G,\bullet)}^{(B,\bullet):\bullet,k} \right)_1 [b]^{(G,i)}$$

$$+ \sum_{s=1}^{n_e} \sum_{q=1}^{D_A} \left( \Phi_{(A,\bullet):\bullet}^{(B,\bullet):\bullet,k} \right)_1 [b]_q^{(A,s)} + \sum_{q=1}^{D_A} \left( \Phi_{(A,\bullet):\bullet}^{(B,\bullet):\bullet,k} \right)_2 [b]_q^{(A,i)}$$

$$+ \sum_{s=1}^{n_e} \left( \Phi_{(A,\bullet):\bullet}^{(B,\bullet):\bullet,k} \right)_3 [b]_j^{(A,s)} + \left( \Phi_{(A,\bullet):\bullet}^{(B,\bullet):\bullet,k} \right)_4 [b]_j^{(A,j)}$$

$$+ \sum_{s=1}^{n_e} \sum_{q=1}^{D} (\Phi_{(B,\bullet):q}^{(B,\bullet):\bullet,k})_1 [b]_q^{(B,s)} + \sum_{q=1}^{D} (\Phi_{(B,\bullet):q}^{(B,\bullet):\bullet,k})_2 [b]_q^{(B,i)}$$

$$+ \Phi_1^{(B,\bullet):\bullet,k}.$$

8. Weight sharing form for $[E(b)]^{(G,i)}$.

   From Equation 213:

$$\Phi_{(QK,s):p,q}^{(G,i)} = \Phi_{(QK,\tau_h(s)):p,q}^{(G,\tau_e(i))},$$

$$\Phi_{(VO,s):p,q}^{(G,i)} = \Phi_{(VO,\tau_h(s)):p,q}^{(G,\tau_e(i))},$$

$$\Phi_{(A,s):p,q}^{(G,i)} = \Phi_{(A,\tau_e(s)):p,\pi_e^{(\tau_e(s))}(q)}^{(G,\tau_e(i))},$$

$$\Phi_{(B,s):p,q}^{(G,i)} = \Phi_{(B,\tau_e(s)):\pi_e^{(\tau_e(s))}(p),q}^{(G,\tau_e(i))},$$

$$\Phi_{(A,s):q}^{(G,i)} = \Phi_{(A,\tau_e(s)):\pi_e^{(\tau_e(s))}(q)}^{(G,\tau_e(i))},$$

$$\Phi_{(B,s):q}^{(G,i)} = \Phi_{(B,\tau_e(s)):q}^{(G,\tau_e(i))},$$

$$\Phi_1^{(G,i)} = \Phi_1^{(G,\tau_e(i))},$$

$$\begin{cases} \Phi_{(G,s):p}^{(G,i)} = \Phi_{(G,\tau_e(s)):p}^{(G,\tau_e(i))}, \\ \sum_{s=1}^{n_e} \Phi_{(G,s):p}^{(G,i)} = 0, \end{cases}$$

$$\begin{cases} \Phi_{(G,s)}^{(G,i)} = \Phi_{(G,\tau_e(s))}^{(G,\tau_e(i))}, \\ \sum_{s=1}^{n_e} \Phi_{(G,s)}^{(G,i)} = 1. \end{cases}$$

   Using Proposition H.4 and Corollary H.3, we obtain the weight sharing form:

$$[E(b)]^{(G,i)} = \sum_{s=1}^{h} \sum_{p=1}^{D} \sum_{q=1}^{D} \Phi_{(QK,\bullet):p,q}^{(G,\bullet)} [WW]_{p,q}^{(QK,s)}$$

$$+ \sum_{s=1}^{h} \sum_{p=1}^{D} \sum_{q=1}^{D} \Phi_{(VO,\bullet):p,q}^{(G,\bullet)} [WW]_{p,q}^{(VO,s)}$$

$$+ \sum_{s=1}^{n_e} \sum_{p=1}^{D} \left( \Phi_{(G,\bullet):p}^{(G,\bullet)} \right)_1 [W]_p^{(G,s)} - \sum_{p=1}^{D} n_e \left( \Phi_{(G,\bullet):p}^{(G,\bullet)} \right)_1 [W]_p^{(G,i)}$$

$$+ \sum_{s=1}^{n_e} \sum_{p=1}^{D} \sum_{q=1}^{D_A} \Phi_{(A,\bullet):p,\bullet}^{(G,\bullet)} [W]_{p,q}^{(A,s)} + \sum_{p=1}^{D} \sum_{q=1}^{D_A} \Phi_{(A,\bullet):p,\bullet}^{(G,\bullet)} [W]_{p,q}^{(A,i)}$$

$$+ \sum_{s=1}^{n_e} \sum_{p=1}^{D_A} \sum_{q=1}^{D} \left( \Phi_{(B,\bullet):\bullet,q}^{(G,\bullet)} \right) [W]_{p,q}^{(B,s)} + \sum_{p=1}^{D_A} \sum_{q=1}^{D} \left( \Phi_{(B,\bullet):\bullet,q}^{(G,\bullet)} \right) [W]_{p,q}^{(B,i)}$$

$$+ [b]^{(G,i)} + \sum_{s=1}^{n_e} \left( \Phi_{(G,\bullet)}^{(G,\bullet)} \right)_1 [b]^{(G,s)} - n_e \left( \Phi_{(G,\bullet)}^{(G,\bullet)} \right)_1 [b]^{(G,i)}$$

$$+ \sum_{s=1}^{n_e} \sum_{q=1}^{D_A} \left( \Phi_{(A,\bullet):\bullet}^{(G,\bullet)} \right)_1 [b]_q^{(A,s)} + \sum_{q=1}^{D_A} \left( \Phi_{(A,\bullet):\bullet}^{(G,\bullet)} \right)_2 [b]_q^{(A,i)}$$

$$+ \sum_{s=1}^{n_e} \sum_{q=1}^{D} \left( \Phi_{(B,\bullet):q}^{(G,\bullet)} \right)_1 [b]_q^{(B,s)} + \sum_{q=1}^{D} \left( \Phi_{(B,\bullet):q}^{(G,\bullet)} \right)_2 [b]_q^{(B,i)}$$

$$+ \Phi_1^{(G,\bullet)}.$$

9. Weight sharing form for $[E(b)]_j^{(A,i)}$.

From Equation 214:

$$\Phi_{(QK,s):p,q}^{(A,i):j} = \Phi_{(QK,\tau_h(s)):p,q}^{(A,\tau_e(i)):\pi_e^{(\tau_e(i))}(j)},$$

$$\Phi_{(VO,s):p,q}^{(A,i):j} = \Phi_{(VO,\tau_h(s)):p,q}^{(A,\tau_e(i)):\pi_e^{(\tau_e(i))}(j)},$$

$$\Phi_{(A,s):p,q}^{(A,i):j} = \Phi_{(A,\tau_e(s)):p,\pi_e^{(\tau_e(s))}(q)}^{(A,\tau_e(i)):\pi_e^{(\tau_e(i))}(j)},$$

$$\Phi_{(B,s):p,q}^{(A,i):j} = \Phi_{(B,\tau_e(s)):\pi_e^{(\tau_e(s))}(p),q}^{(A,\tau_e(i)):\pi_e^{(\tau_e(i))}(j)},$$

$$\Phi_{(A,s):q}^{(A,i):j} = \Phi_{(A,\tau_e(s)):\pi_e^{(\tau_e(s))}(q)}^{(A,\tau_e(i)):\pi_e^{(\tau_e(i))}(j)},$$

$$\Phi_{(B,s):q}^{(A,i):j} = \Phi_{(B,\tau_e(s)):q}^{(A,\tau_e(i)):\pi_e^{(\tau_e(i))}(j)},$$

$$\Phi_1^{(A,i):j} = \Phi_1^{(A,\tau_e(i)):\pi_e^{(\tau_e(i))}(j)},$$

$$\begin{cases} \Phi_{(G,s):p}^{(A,i):j} = \Phi_{(G,\tau_e(s)):p}^{(A,\tau_e(i)):\pi_e^{(\tau_e(i))}(j)}, \\ \sum_{s=1}^{n_e} \Phi_{(G,s):p}^{(A,i):j} = 0, \end{cases}$$

$$\begin{cases} \Phi_{(G,s)}^{(A,i):j} = \Phi_{(G,\tau_e(s))}^{(A,\tau_e(i)):\pi_e^{(\tau_e(i))}(j)}, \\ \sum_{s=1}^{n_e} \Phi_{(G,s)}^{(A,i):j} = 0. \end{cases}$$

Using Corollary H.5, we obtain the weight sharing form:

$$[E(b)]_j^{(A,i)} = \sum_{s=1}^{h} \sum_{p=1}^{D} \sum_{q=1}^{D} \Phi_{(QK,\bullet):p,q}^{(A,\bullet):\bullet} [WW]_{p,q}^{(QK,s)}$$

$$+ \sum_{s=1}^{h} \sum_{p=1}^{D} \sum_{q=1}^{D} \Phi_{(VO,\bullet):p,q}^{(A,\bullet):\bullet} [WW]_{p,q}^{(VO,s)}$$

$$+ \sum_{s=1}^{n_e} \sum_{p=1}^{D} \left( \Phi_{(G,\bullet):p}^{(A,\bullet):\bullet} \right)_1 [W]_p^{(G,s)} - \sum_{p=1}^{D} n_e \left( \Phi_{(G,\bullet):p}^{(A,\bullet):\bullet} \right)_1 [W]_p^{(G,i)}$$

$$+ \sum_{s=1}^{n_e} \sum_{p=1}^{D} \sum_{q=1}^{D_A} \left( \Phi^{(A,\bullet):\bullet}_{(A,\bullet):p,\bullet} \right)_1 [W]^{(A,s)}_{p,q} + \sum_{p=1}^{D} \sum_{q=1}^{D_A} \left( \Phi^{(A,\bullet):\bullet}_{(A,\bullet):p,\bullet} \right)_2 [W]^{(A,i)}_{p,q}$$

$$+ \sum_{s=1}^{n_e} \sum_{p=1}^{D} \left( \Phi^{(A,\bullet):\bullet}_{(A,\bullet):p,\bullet} \right)_3 [W]^{(A,s)}_{p,j} + \sum_{p=1}^{D} \left( \Phi^{(A,\bullet):\bullet}_{(A,\bullet):p,\bullet} \right)_4 [W]^{(A,i)}_{p,j}$$

$$+ \sum_{s=1}^{n_e} \sum_{p=1}^{D_A} \sum_{q=1}^{D} \left( \Phi^{(A,\bullet):\bullet}_{(B,\bullet):\bullet,q} \right)_1 [W]^{(B,s)}_{p,q} + \sum_{p=1}^{D_A} \sum_{q=1}^{D} \left( \Phi^{(A,\bullet):\bullet}_{(B,\bullet):\bullet,q} \right)_2 [W]^{(B,i)}_{p,q}$$

$$+ \sum_{s=1}^{n_e} \sum_{q=1}^{D} \left( \Phi^{(A,\bullet):\bullet}_{(B,\bullet):\bullet,q} \right)_3 [W]^{(B,s)}_{j,q} + \sum_{q=1}^{D} \left( \Phi^{(A,\bullet):\bullet}_{(B,\bullet):\bullet,q} \right)_4 [W]^{(B,i)}_{j,q}$$

$$+ \sum_{s=1}^{n_e} \left( \Phi^{(A\bullet):\bullet}_{(G,\bullet)} \right)_1 [b]^{(G,s)} - n_e \left( \Phi^{(A,\bullet):\bullet}_{(G,\bullet)} \right)_1 [b]^{(G,i)}$$

$$+ \sum_{s=1}^{n_e} \sum_{q=1}^{D_A} \left( \Phi^{(A,\bullet):\bullet}_{(A,\bullet):\bullet} \right)_1 [b]^{(A,s)}_q + \sum_{q=1}^{D_A} \left( \Phi^{(A,\bullet):\bullet}_{(A,\bullet):\bullet} \right)_2 [b]^{(A,i)}_q$$

$$+ \sum_{s=1}^{n_e} \left( \Phi^{(A,\bullet):\bullet}_{(A,\bullet):\bullet} \right)_3 [b]^{(A,s)}_j + \left( \Phi^{(A,\bullet):\bullet}_{(A,\bullet):\bullet} \right)_4 [b]^{(A,i)}_j$$

$$+ \sum_{s=1}^{n_e} \sum_{q=1}^{D} \left( \Phi^{(A,\bullet):\bullet}_{(B,\bullet):q} \right)_1 [b]^{(B,s)}_q + \sum_{q=1}^{D} \left( \Phi^{(A,\bullet):\bullet}_{(B,\bullet):q} \right)_2 [b]^{(B,i)}_q$$

$$+ \Phi^{(A,\bullet):\bullet}_1.$$

10. Weight sharing form for $[E(b)]^{(B,i)}_j$.

    From Equation 215:

$$\Phi^{(B,i):j}_{(QK,s):p,q} = \Phi^{(B,\tau_e(i)):j}_{(QK,\tau_h(s)):p,q},$$

$$\Phi^{(B,i):j}_{(VO,s):p,q} = \Phi^{(B,\tau_e(i)):j}_{(VO,\tau_h(s)):p,q},$$

$$\Phi^{(B,i):j}_{(A,s):p,q} = \Phi^{(B,\tau_e(i)):j}_{(A,\tau_e(s)):p,\pi_e^{(\tau_e(s))}(q)},$$

$$\Phi^{(B,i):j}_{(B,s):p,q} = \Phi^{(B,\tau_e(i)):j}_{(B,\tau_e(s)):\pi_e^{(\tau_e(s))}(p),q},$$

$$\Phi^{(B,i):j}_{(A,s):q} = \Phi^{(B,\tau_e(i)):j}_{(A,\tau_e(s)):\pi_e^{(\tau_e(s))}(q)},$$

$$\Phi^{(B,i):j}_{(B,s):q} = \Phi^{(B,\tau_e(i)):j}_{(B,\tau_e(s)):q},$$

$$\Phi^{(B,i):j}_1 = \Phi^{(B,\tau_e(i)):j}_1,$$

$$\begin{cases} \Phi^{(B,i):j}_{(G,s):p} = \Phi^{(B,\tau_e(i)):j}_{(G,\tau_e(s)):p}, \\ \sum_{s=1}^{n_e} \Phi^{(B,i):j}_{(G,s):p} = 0, \end{cases}$$

$$\begin{cases} \Phi^{(B,i):j}_{(G,s)} = \Phi^{(B,\tau_e(i)):j}_{(G,\tau_e(s))}, \\ \sum_{s=1}^{n_e} \Phi^{(B,i):j}_{(G,s)} = 0. \end{cases}$$

Using Corrolary H.5, we obtain the weight sharing form:

$$[E(b)]^{(B,i)}_j = \sum_{s=1}^{h} \sum_{p=1}^{D} \sum_{q=1}^{D} \Phi^{(B,\bullet):j}_{(QK,\bullet):p,q} [WW]^{(QK,s)}_{p,q}$$

$$+ \sum_{s=1}^{h} \sum_{p=1}^{D} \sum_{q=1}^{D} \Phi^{(B,\bullet):j}_{(VO,\bullet):p,q} [WW]^{(VO,s)}_{p,q}$$

$$+ \sum_{s=1}^{n_e} \sum_{p=1}^{D} \left( \Phi_{(G,\bullet):p}^{(B,\bullet):j} \right)_1 [W]_p^{(G,s)} - \sum_{p=1}^{D} n_e \left( \Phi_{(G,\bullet):p}^{(B,\bullet):j} \right)_1 [W]_p^{(G,i)}$$

$$+ \sum_{s=1}^{n_e} \sum_{p=1}^{D} \sum_{q=1}^{D_A} \left( \Phi_{(A,\bullet):p,\bullet}^{(B,\bullet):j} \right)_1 [W]_{p,q}^{(A,s)} + \sum_{p=1}^{D} \sum_{q=1}^{D_A} \left( \Phi_{(A,\bullet):p,\bullet}^{(B,\bullet):j} \right)_2 [W]_{p,q}^{(A,i)}$$

$$+ \sum_{s=1}^{n_e} \sum_{p=1}^{D_A} \sum_{q=1}^{D} \left( \Phi_{(B,\bullet):\bullet,q}^{(B,\bullet):j} \right)_1 [W]_{p,q}^{(B,s)} + \sum_{p=1}^{D_A} \sum_{q=1}^{D} \left( \Phi_{(B,\bullet):\bullet,q}^{(B,\bullet):j} \right)_2 [W]_{p,q}^{(B,i)}$$

$$+ \left( \Phi_{(G,\bullet)}^{(B,\bullet):j} \right)_1 [b]^{(G,s)} - n_e \left( \Phi_{(G,\bullet)}^{(B,\bullet):j} \right)_1 [b]^{(G,i)}$$

$$+ \sum_{s=1}^{n_e} \sum_{q=1}^{D_A} \left( \Phi_{(A,\bullet):\bullet}^{(B,\bullet):j} \right)_1 [b]_q^{(A,s)} + \sum_{q=1}^{D_A} \left( \Phi_{(A,\bullet):\bullet}^{(B,\bullet):j} \right)_2 [b]_q^{(A,i)}$$

$$+ \sum_{s=1}^{n_e} \sum_{q=1}^{D} \left( \Phi_{(B,\bullet):q}^{(B,\bullet):j} \right)_1 [b]_q^{(B,s)} + \sum_{q=1}^{D} \left( \Phi_{(B,\bullet):q}^{(B,\bullet):j} \right)_2 [b]_q^{(B,i)}$$

$$+ \Phi_1^{(B,\bullet):j}.$$

## H.3 INVARIANT LAYERS WITH BULLET NOTATION

From Equation 219:

$$\Phi_{(QK,s):p,q}^i = \Phi_{(QK,\tau_h(s)):p,q}^i,$$

$$\Phi_{(VO,s):p,q}^i = \Phi_{(VO,\tau_h(s)):p,q}^i,$$

$$\Phi_{(Q,s):p,q}^i = 0,$$

$$\Phi_{(K,s):p,q}^i = 0,$$

$$\Phi_{(V,s):p,q}^i = 0,$$

$$\Phi_{(O,s):p,q}^i = 0,$$

$$\Phi_{(A,s):p,q}^i = \Phi_{(A,\tau_e(s)):\pi_e^{(s)}(p),\pi_e^{(s)}(q)}^i,$$

$$\Phi_{(B,s):p,q}^i = \Phi_{(B,\tau_e(s)):\pi_e^{(s)}(p),q}^i,$$

$$\Phi_{(A,s):q}^i = \Phi_{(A,\tau_e(s)):\pi_e^{(s)}(q)}^i,$$

$$\Phi_{(B,s):q}^i = \Phi_{(B,\tau_e(s)):q}^i,$$

$$\begin{cases} \Phi_{(G,s):p}^i = \Phi_{(G,\tau_e(s)):p}^i, \\ \sum_{s=1}^{n_e} \Phi_{(G,s):p}^i = 0, \end{cases}$$

$$\begin{cases} \Phi_{(G,s)}^i = \Phi_{(G,\tau_e(s))}^i, \\ \sum_{s=1}^{n_e} \Phi_{(G,s)}^i = 0. \end{cases}$$

Which results in the weight sharing form:

$$I(U)_i = \sum_{s=1}^{h} \sum_{p=1}^{D} \sum_{q=1}^{D} \Phi_{(QK,\bullet):p,q}^i [WW]_{p,q}^{(QK,s)}$$

$$+ \sum_{s=1}^{h} \sum_{p=1}^{D} \sum_{q=1}^{D} \Phi_{(VO,\bullet):p,q}^i [WW]_{p,q}^{(VO,s)}$$

Table 8: Summary of key dimensions involved in the implementation

| Symbol | Description |
|---|---|
| $d$ | Number of input channels for the equivariant and invariant layer |
| $e$ | Number of output channels for the equivariant and invariant layer |
| $D$ | Embedding dimension of the input and output sequences of the transformer block |
| $D_k = D_q$ | Embedding dimension for key and query vectors in the transformer block |
| $D_v$ | Embedding dimension for value vectors in the transformer block |
| $D_e$ | MoE hidden dimension |
| $h$ | Number of attention heads in the transformer block |
| $b$ | Batch size |
| $n_e$ | Number of experts in MoE layer |
| $D'$ | Embedding dimension of the invariant layer's output |

Table 9: Shapes of input terms used in the implementation

| Term | Shape |
|---|---|
| $[W]_{p,q}^{(Q,i)}$ | $[b, d, h, D, D_q]$ |
| $[W]_{p,q}^{(K,i)}$ | $[b, d, h, D, D_k]$ |
| $[W]_{p,q}^{(V,i)}$ | $[b, d, h, D, D_v]$ |
| $[W]_{p,q}^{(O,i)}$ | $[b, d, h, D_v, D]$ |
| $[W]_{p}^{(G,s)}$ | $[b, d, n_e, D]$ |
| $[b]^{(G,s)}$ | $[b, d, n_e]$ |
| $[WW]_{p,q}^{(QK,i)}$ | $[b, d, h, D, D]$ |
| $[WW]_{p,q}^{(VO,i)}$ | $[b, d, h, D, D]$ |
| $[W]_{p,q}^{(A,s)}$ | $[b, d, n_e, D, D_e]$ |
| $[b]_{q}^{(A,s)}$ | $[b, d, n_e, D_e]$ |
| $[W]_{p,q}^{(B,s)}$ | $[b, d, n_e, D_e, D]$ |
| $[b]_{q}^{(B,s)}$ | $[b, d, n_e, D]$ |

$$
+ \sum_{s=1}^{n_e} \sum_{p=1}^{D} \left( \Phi_{(G,\bullet):p}^i - \frac{1}{n_e} \sum_{s=1}^{n_e} \Phi_{(G,\bullet):p}^i \right) [W]_p^{(G,s)}
$$

$$
+ \sum_{s=1}^{n_e} \sum_{p=1}^{D} \sum_{q=1}^{D_A} \Phi_{(A,\bullet):p,\bullet}^i [W]_{p,q}^{(A,s)}
$$

$$
+ \sum_{s=1}^{n_e} \sum_{p=1}^{D_A} \sum_{q=1}^{D} \Phi_{(B,\bullet):\bullet,q}^i [W]_{p,q}^{(B,s)}
$$

$$
+ \sum_{s=1}^{n_e} \left( \Phi_{(G,\bullet)}^i - \frac{1}{n_e} \sum_{s=1}^{n_e} \Phi_{(G,\bullet)}^i \right) [b]^{(G,s)}
$$

$$
+ \sum_{s=1}^{n_e} \sum_{q=1}^{D_A} \Phi_{(A,\bullet):\bullet}^i [b]_q^{(A,s)}
$$

$$
+ \sum_{s=1}^{n_e} \sum_{q=1}^{D} \Phi_{(B,\bullet):q}^i [b]_q^{(B,s)}
$$

$$
+ \Phi_1^i.
$$

## H.4 EQUIVARIANT LAYERS PSEUDOCODE

### H.4.1 $[E(W)]_{j,k}^{(Q,i)}$ PSEUDOCODE

$$[E(W)]_{j,k}^{(Q,i)} = \sum_{p=1}^{D} \Phi_{(Q,\bullet):p,\bullet}^{(Q,\bullet):j,\bullet}[W]_{p,k}^{(Q,i)}.$$

- $\Phi_{(Q,\bullet):p,\bullet}^{(Q,\bullet):j,\bullet}[W]_{p,k}^{(Q,i)}$
  **Shapes**:

$$[W]_{p,k}^{(Q,i)} : [b, d, h, D, D]$$
$$\Phi_{(Q,\bullet):p,\bullet}^{(Q,\bullet):j,\bullet} : [e, d, D, D].$$

  **Pseudocode**: einsum($bdhpk, edjp \to behjk$)

### H.4.2 $[E(W)]_{j,k}^{(K,i)}$ PSEUDOCODE

$$[E(W)]_{j,k}^{(K,i)} = \sum_{p=1}^{D} \Phi_{(K,\bullet):p,\bullet}^{(K,\bullet):j,\bullet}[W]_{p,k}^{(K,i)}.$$

- $\Phi_{(K,\bullet):p,\bullet}^{(K,\bullet):j,\bullet}[W]_{p,k}^{(K,i)}$
  **Shapes**:

$$[W]_{p,k}^{(K,i)} : [b, d, h, D, D]$$
$$\Phi_{(K,\bullet):p,\bullet}^{(K,\bullet):j,\bullet} : [e, d, D, D]$$

  **Pseudocode**: einsum($bdhpk, edjp \to behjk$)

### H.4.3 $[E(W)]_{j,k}^{(V,i)}$ PSEUDOCODE

$$[E(W)]_{j,k}^{(V,i)} = \sum_{p=1}^{D} \Phi_{(V,\bullet):p,\bullet}^{(V,\bullet):j,\bullet}[W]_{p,k}^{(V,i)}.$$

- $\Phi_{(V,\bullet):p,\bullet}^{(V,\bullet):j,\bullet}[W]_{p,k}^{(V,i)}$
  **Shapes**:

$$[W]_{p,k}^{(V,i)} : [b, d, h, D, D]$$
$$\Phi_{(V,\bullet):p,\bullet}^{(V,\bullet):j,\bullet} : [e, d, D, D]$$

  **Pseudocode**: einsum($bdhpk, edjp \to behjk$)

### H.4.4 $[E(W)]_{j,k}^{(O:i)}$ PSEUDOCODE

$$[E(W)]_{j,k}^{(O,i)} = \sum_{q=1}^{D} \Phi_{(O,\bullet):\bullet,q}^{(O,\bullet):\bullet,k}[W]_{j,k}^{(O,i)}.$$

- $\Phi_{(O,\bullet):p,\bullet}^{(O,\bullet):j,\bullet}[W]_{j,k}^{(O,i)}$
  **Shapes**:

$$[W]_{j,k}^{(O,i)} : [b, d, h, D_v, D]$$
$$\Phi_{(O,\bullet):\bullet,q}^{(O,\bullet):\bullet,k} : [e, d, D, D].$$

  **Pseudocode**: einsum($bdhpk, edkq \to behkq$)

### H.4.5 $[E(W)]_j^{(G,i)}$ PSEUDOCODE

$$
[E(W)]_j^{(G,i)} = \sum_{p=1}^{D}\sum_{q=1}^{D}\sum_{s=1}^{h} \Phi_{(QK,\bullet):p,q}^{(G,\bullet):j}[WW]_{p,q}^{(QK,s)}
$$

$$
+ \sum_{p=1}^{D}\sum_{q=1}^{D}\sum_{s=1}^{h} \Phi_{(VO,\bullet):p,q}^{(G,\bullet):j}[WW]_{p,q}^{(VO,s)}
$$

$$
+ [W]_j^{(G,i)} + \sum_{s=1}^{n_e}\sum_{p=1}^{D} \left(\Phi_{(G,\bullet):p}^{(G,\bullet):j}\right)_1 [W]_p^{(G,s)} - n_e \sum_{p=1}^{D} \left(\Phi_{(G,\bullet):p}^{(G,\bullet):j}\right)_1 [W]_p^{(G,i)}
$$

$$
+ \sum_{s=1}^{n_e}\sum_{p=1}^{D}\sum_{q=1}^{D_A} \left(\Phi_{(A,\bullet):\bullet,q}^{(G,\bullet):j}\right)_1 [W]_{p,q}^{(A,s)} + \sum_{p=1}^{D}\sum_{q=1}^{D_A} \left(\Phi_{(A,\bullet):\bullet,q}^{(G,\bullet):j}\right)_2 [W]_{p,q}^{(A,i)}
$$

$$
+ \sum_{s=1}^{n_e}\sum_{p=1}^{D_A}\sum_{q=1}^{D} \left(\Phi_{(B,\bullet):\bullet,q}^{(G,\bullet):j}\right)_1 [W]_{p,q}^{(B,s)} + \sum_{p=1}^{D_A}\sum_{q=1}^{D} \left(\Phi_{(B,\bullet):\bullet,q}^{(G,\bullet):j}\right)_2 [W]_{p,q}^{(B,i)}
$$

$$
+ \sum_{s=1}^{n_e} \left(\Phi_{(G,\bullet)}^{(G,\bullet):j}\right)_1 [b]^{(G,s)} - n_e \left(\Phi_{(G,\bullet)}^{(G,\bullet):j}\right)_1 [b]^{(G,i)}
$$

$$
+ \sum_{s=1}^{n_e}\sum_{q=1}^{D_A} \left(\Phi_{(A,\bullet):\bullet}^{(G,\bullet):j}\right)_1 [b]_q^{(A,s)} + \sum_{q=1}^{D_A} \left(\Phi_{(A,\bullet):\bullet}^{(G,\bullet):j}\right)_2 [b]_q^{(A,i)}
$$

$$
+ \sum_{s=1}^{n_e}\sum_{q=1}^{D} \left(\Phi_{(B,\bullet):q}^{(G,\bullet):j}\right)_1 [b]_q^{(B,s)} + \sum_{q=1}^{D} \left(\Phi_{(B,\bullet):q}^{(G,\bullet):j}\right)_1 [b]_q^{(B,i)}
$$

$$
+ \Phi_1^{(G,\bullet):j}.
$$

**Shapes and pseudocode:** See Table 10.

### H.4.6 $[E(W)]_{j,k}^{(A,i)}$ PSEUDOCODE

$$
[E(W)]_{j,k}^{(A,i)} = \sum_{s=1}^{h}\sum_{p=1}^{D}\sum_{q=1}^{D} \Phi_{(QK,\bullet):p,q}^{(A,\bullet):j,\bullet}[WW]_{p,q}^{(QK,s)}
$$

$$
+ \sum_{s=1}^{h}\sum_{p=1}^{D}\sum_{q=1}^{D} \Phi_{(VO,\bullet):p,q}^{(A,\bullet):j,\bullet}[WW]_{p,q}^{(VO,s)}
$$

$$
+ \sum_{s=1}^{n_e}\sum_{p=1}^{D} \left(\Phi_{(G,\bullet):p}^{(A,\bullet):j,\bullet}\right)_1 [W]_p^{(G,s)} - \sum_{p=1}^{D} n_e \left(\Phi_{(G,\bullet):p}^{(A,\bullet):j,\bullet}\right)_1 [W]_p^{(G,i)}
$$

$$
+ \sum_{s=1}^{n_e}\sum_{p=1}^{D}\sum_{q=1}^{D_A} \left(\Phi_{(A,\bullet):p,\bullet}^{(A,\bullet):j,\bullet}\right)_1 [W]_{p,q}^{(A,s)} + \sum_{p=1}^{D}\sum_{q=1}^{D_A} \left(\Phi_{(A,\bullet):p,\bullet}^{(A,\bullet):j,\bullet}\right)_2 [W]_{p,q}^{(A,i)}
$$

$$
+ \sum_{s=1}^{n_e}\sum_{p=1}^{D} \left(\Phi_{(A,\bullet):p,\bullet}^{(A,\bullet):j,\bullet}\right)_3 [W]_{p,k}^{(A,s)} + \sum_{p=1}^{D} \left(\Phi_{(A,\bullet):p,\bullet}^{(A,\bullet):j,\bullet}\right)_4 [W]_{p,k}^{(A,i)}
$$

$$
+ \sum_{s=1}^{n_e}\sum_{p=1}^{D_A}\sum_{q=1}^{D} \left(\Phi_{(B,\bullet):\bullet,q}^{(A,\bullet):j,\bullet}\right)_1 [W]_{p,q}^{(B,s)} + \sum_{p=1}^{D_A}\sum_{q=1}^{D} \left(\Phi_{(B,\bullet):\bullet,q}^{(A,\bullet):j,\bullet}\right)_2 [W]_{p,q}^{(B,i)}
$$

$$
+ \sum_{s=1}^{n_e}\sum_{q=1}^{D} \left(\Phi_{(B,\bullet):\bullet,q}^{(A,\bullet):j,\bullet}\right)_3 [W]_{k,q}^{(B,s)} + \sum_{q=1}^{D} \left(\Phi_{(B,\bullet):\bullet,q}^{(A,\bullet):j,\bullet}\right)_4 [W]_{k,q}^{(B,i)}
$$

| Input | Input shape | Weight | Weight shape | Einsum |
|---|---|---|---|---|
| $[WW]_{p,q}^{(QK,s)}$ | $[b,d,h,D,D]$ | $\Phi_{(QK,\bullet):p,q}^{(G,\bullet):j}$ | $[e,d,D,D,D]$ | $(bdhpq, edjpq \to bej).usq(-2)$ |
| $[WW]_{p,q}^{(VO,s)}$ | $[b,d,h,D,D]$ | $\Phi_{(VO,\bullet):p,q}^{(G,\bullet):j}$ | $[e,d,D,D,D]$ | $(bdhpq, edjpq \to bej).usq(-2)$ |
| $[W]_p^{(G,s)}$ | $[b,d,n_e,D]$ | $\left(\Phi_{(G,\bullet):p}^{(G,\bullet):j}\right)_1$ | $[e,d,D,D]$ | $(bdnp, edpq \to beq).usq(-2)$ |
| $[W]_p^{(G,s)}$ | $[b,d,n_e,D]$ | $n_e\left(\Phi_{(G,\bullet):p}^{(G,\bullet):j}\right)_1$ | $[e,d,D,D]$ | $(bdnp, edpq \to benq)$ |
| $[W]_{p,q}^{(A,s)}$ | $[b,d,n_e,D,D_e]$ | $\left(\Phi_{(A,\bullet):\bullet,q}^{(G,\bullet):j}\right)_1$ | $[e,d,D,D_e]$ | $(bdnpq, edjq \to bej).usq(-2)$ |
| $[W]_{p,q}^{(A,i)}$ | $[b,d,n_e,D,D_e]$ | $\left(\Phi_{(A,\bullet):\bullet,q}^{(G,\bullet):j}\right)_2$ | $[e,d,D,D_e]$ | $(bdnpq, edjq \to benj)$ |
| $[W]_{p,q}^{(B,s)}$ | $[b,d,n_e,D_e,D]$ | $\left(\Phi_{(B,\bullet):\bullet,q}^{(G,\bullet):j}\right)_1$ | $[e,d,D,D]$ | $(bdnpq, edjq \to bej).usq(-2)$ |
| $[W]_{p,q}^{(B,i)}$ | $[b,d,n_e,D_e,D]$ | $\left(\Phi_{(B,\bullet):\bullet,q}^{(G,\bullet):j}\right)_2$ | $[e,d,D,D]$ | $(bdnpq, edjq \to benj)$ |
| $[b]^{(G,s)}$ | $[b,d,n_e]$ | $\left(\Phi_{(G,\bullet)}^{(G,\bullet):j}\right)_1$ | $[e,d,D]$ | $(bdn, edj \to bej).usq(-2)$ |
| $[b]^{(G,s)}$ | $[b,d,n_e]$ | $n_e\left(\Phi_{(G,\bullet)}^{(G,\bullet):j}\right)_1$ | $[e,d,D]$ | $(bdn, edj \to benj)$ |
| $[b]_q^{(A,s)}$ | $[b,d,n_e,D_e]$ | $\left(\Phi_{(A,\bullet):\bullet}^{(G,\bullet):j}\right)_1$ | $[e,d,D]$ | $(bdnq, edj \to bej).usq(-2)$ |
| $[b]_q^{(A,i)}$ | $[b,d,n_e,D_e]$ | $\left(\Phi_{(A,\bullet):\bullet}^{(G,\bullet):j}\right)_2$ | $[e,d,D]$ | $(bdnq, edj \to benj)$ |
| $[b]_q^{(B,s)}$ | $[b,d,n_e,D]$ | $\left(\Phi_{(B,\bullet):q}^{(G,\bullet):j}\right)_1$ | $[e,d,D,D]$ | $(bdnq, edjq \to bej).usq(-2)$ |
| $[b]_q^{(B,i)}$ | $[b,d,n_e,D]$ | $\left(\Phi_{(B,\bullet):q}^{(G,\bullet):j}\right)_1$ | $[e,d,D,D]$ | $(bdnq, edjq \to benj)$ |
| | | $\Phi_1^{(G,\bullet):j}$ | $[e,D]$ | $(ej \to ej).usq(0).usq(-2)$ |

Table 10: Pseudocode for $[E(W)]_j^{(G,i)}$.

| Input | Input shape | Weight | Weight shape | Einsum |
|---|---|---|---|---|
| $[WW]_{p,q}^{(QK,s)}$ | $[b,d,h,D,D]$ | $\Phi_{(QK,\bullet):p,q}^{(A,\bullet):j,\bullet}$ | $[e,d,D,D,D]$ | $(bdhpq, edjpq \rightarrow bej).usq(-2).usq(-1)$ |
| $[WW]_{p,q}^{(VO,s)}$ | $[b,d,h,D,D]$ | $\Phi_{(VO,\bullet):p,q}^{(A,\bullet):j,\bullet}$ | $[e,d,D,D,D]$ | $(bdhpq, edjpq \rightarrow bej).usq(-2).usq(-1)$ |
| $[W]_{p}^{(G,s)}$ | $[b,d,n_e,D]$ | $\left(\Phi_{(G,\bullet):p}^{(A,\bullet):j,\bullet}\right)_1$ | $[e,d,D,D]$ | $(bdnp, edjp \rightarrow bej).usq(-2).usq(-1)$ |
| $[W]_{p}^{(G,i)}$ | $[b,d,n_e,D]$ | $n_e\left(\Phi_{(G,\bullet):p}^{(A,\bullet):j,\bullet}\right)_1$ | $[e,d,D,D]$ | $(bdnp, edjp \rightarrow benj).usq(-1)$ |
| $[W]_{p,q}^{(A,s)}$ | $[b,d,n_e,D,D_e]$ | $\left(\Phi_{(A,\bullet):p,\bullet}^{(A,\bullet):j,\bullet}\right)_1$ | $[e,d,D,D]$ | $(bdnpq, edjp \rightarrow bej).usq(-2).usq(-1)$ |
| $[W]_{p,q}^{(A,i)}$ | $[b,d,n_e,D,D_e]$ | $\left(\Phi_{(A,\bullet):p,\bullet}^{(A,\bullet):j,\bullet}\right)_2$ | $[e,d,D,D]$ | $(bdnpq, edjp \rightarrow benj).usq(-1)$ |
| $[W]_{p,k}^{(A,s)}$ | $[b,d,n_e,D,D_e]$ | $\left(\Phi_{(A,\bullet):p,\bullet}^{(A,\bullet):j,\bullet}\right)_3$ | $[e,d,D,D]$ | $(bdnpk, edjp \rightarrow bejk).usq(-3)$ |
| $[W]_{p,k}^{(A,i)}$ | $[b,d,n_e,D,D_e]$ | $\left(\Phi_{(A,\bullet):p,\bullet}^{(A,\bullet):j,\bullet}\right)_4$ | $[e,d,D,D]$ | $(bdnpk, edjp \rightarrow benjk)$ |
| $[W]_{p,q}^{(B,s)}$ | $[b,d,n_e,D_e,D]$ | $\left(\Phi_{(B,\bullet):\bullet,q}^{(A,\bullet):j,\bullet}\right)_1$ | $[e,d,D,D]$ | $(bdnpq, edjq \rightarrow bej).usq(-2).usq(-1)$ |
| $[W]_{p,q}^{(B,i)}$ | $[b,d,n_e,D_e,D]$ | $\left(\Phi_{(B,\bullet):\bullet,q}^{(A,\bullet):j,\bullet}\right)_2$ | $[e,d,D,D]$ | $(bdnpq, edjq \rightarrow benj).usq(-1)$ |
| $[W]_{k,q}^{(B,s)}$ | $[b,d,n_e,D_e,D]$ | $\left(\Phi_{(B,\bullet):\bullet,q}^{(A,\bullet):j,\bullet}\right)_3$ | $[e,d,D,D]$ | $(bdnkq, edjq \rightarrow bejk).usq(-3)$ |
| $[W]_{k,q}^{(B,i)}$ | $[b,d,n_e,D_e,D]$ | $\left(\Phi_{(B,\bullet):\bullet,q}^{(A,\bullet):j,\bullet}\right)_4$ | $[e,d,D,D]$ | $(bdnkq, edjq \rightarrow benjk)$ |
| $[b]^{(G,s)}$ | $[b,d,n_e]$ | $\Phi_{(G,\bullet)}^{(A,\bullet):j,\bullet}$ | $[e,d,D]$ | $(bdn, edj \rightarrow bej).usq(-2).usq(-1)$ |
| $[b]^{(G,s)}$ | $[b,d,n_e]$ | $n_e\Phi_{(G,\bullet)}^{(A,\bullet):j,\bullet}$ | $[e,d,D]$ | $(bdn, edj \rightarrow benj).usq(-1)$ |
| $[b]_{q}^{(A,s)}$ | $[b,d,n_e,D_e]$ | $\left(\Phi_{(A,\bullet):\bullet}^{(A,\bullet):j,\bullet}\right)_1$ | $[e,d,D]$ | $(bdnq, edj \rightarrow bej).usq(-2).usq(-1)$ |
| $[b]_{q}^{(A,i)}$ | $[b,d,n_e,D_e]$ | $\left(\Phi_{(A,\bullet):\bullet}^{(A,\bullet):j,\bullet}\right)_2$ | $[e,d,D]$ | $(bdnq, edj \rightarrow benj).usq(-1)$ |
| $[b]_{k}^{(A,s)}$ | $[b,d,n_e,D_e]$ | $\left(\Phi_{(A,\bullet):\bullet}^{(A,\bullet):j,\bullet}\right)_3$ | $[e,d,D]$ | $(bdnk, edj \rightarrow bejk).usq(-3)$ |
| $[b]_{k}^{(A,i)}$ | $[b,d,n_e,D_e]$ | $\left(\Phi_{(A,\bullet):\bullet}^{(A,\bullet):j,\bullet}\right)_4$ | $[e,d,D]$ | $(bdnk, edj \rightarrow benjk)$ |
| $[b]_{q}^{(B,s)}$ | $[b,d,n_e,D]$ | $\left(\Phi_{(B,\bullet):q}^{(A,\bullet):j,\bullet}\right)_1$ | $[e,d,D,D]$ | $(bdnq, edjq \rightarrow bej).usq(-2).usq(-1)$ |
| $[b]_{q}^{(B,i)}$ | $[b,d,n_e,D]$ | $\left(\Phi_{(B,\bullet):q}^{(A,\bullet):j,\bullet}\right)_2$ | $[e,d,D,D]$ | $(bdnq, edjq \rightarrow benj).usq(-1)$ |
| | | $\Phi_1^{(A,\bullet):j,\bullet}$ | $[e,D]$ | $(ej \rightarrow ej).usq(0).usq(-2).usq(-1)$ |

Table 11: Pseudocode for $[E(W)]_{j,k}^{(A,i)}$.

$$+ \sum_{s=1}^{n_e} \left( \Phi_{(G,\bullet)}^{(A,\bullet):j,\bullet} \right)_1 [b]^{(G,s)} - n_e \left( \Phi_{(G,\bullet)}^{(A,\bullet):j,\bullet} \right)_1 [b]^{(G,i)}$$

$$+ \sum_{s=1}^{n_e} \sum_{q=1}^{D_A} \left( \Phi_{(A,\bullet):\bullet}^{(A,\bullet):j,\bullet} \right)_1 [b]_q^{(A,s)} + \sum_{q=1}^{D_A} \left( \Phi_{(A,\bullet):\bullet}^{(A,\bullet):j,\bullet} \right)_2 [b]_q^{(A,i)}$$

$$+ \sum_{s=1}^{n_e} \left( \Phi_{(A,\bullet):\bullet}^{(A,\bullet):j,\bullet} \right)_3 [b]_k^{(A,s)} + \left( \Phi_{(A,\bullet):\bullet}^{(A,\bullet):j,\bullet} \right)_4 [b]_k^{(A,i)}$$

$$+ \sum_{s=1}^{n_e} \sum_{q=1}^{D} \left( \Phi_{(B,\bullet):q}^{(A,\bullet):j,\bullet} \right)_1 [b]_q^{(B,s)} + \sum_{q=1}^{D} \left( \Phi_{(B,\bullet):q}^{(A,\bullet):j,\bullet} \right)_2 [b]_q^{(B,i)}$$

$$+ \Phi_1^{(A,\bullet):j,\bullet}.$$

**Shapes and pseudocode:** See Table 11.

## H.4.7 $[E(W)]_{j,k}^{(B,i)}$ PSEUDOCODE

$$[E(W)]_{j,k}^{(B,i)} = \sum_{s=1}^{h} \sum_{p=1}^{D} \sum_{q=1}^{D} \Phi_{(QK,\bullet):p,q}^{(B,\bullet):\bullet,k} [WW]_{p,q}^{(QK,s)}$$

$$+ \sum_{s=1}^{h} \sum_{p=1}^{D} \sum_{q=1}^{D} \Phi_{(VO,\bullet):p,q}^{(B,\bullet):\bullet,k} [WW]_{p,q}^{(VO,s)}$$

$$+ \sum_{s=1}^{n_e} \sum_{p=1}^{D} \left( \Phi_{(G,\bullet):p}^{(B,\bullet):\bullet,k} \right)_1 [W]_p^{(G,s)} - \sum_{p=1}^{D} n_e \left( \Phi_{(G,\bullet):p}^{(B,\bullet):\bullet,k} \right)_1 [W]_p^{(G,i)}$$

$$+ \sum_{s=1}^{n_e} \sum_{p=1}^{D} \sum_{q=1}^{D_A} \left( \Phi_{(A,\bullet):p,\bullet}^{(B,\bullet):\bullet,k} \right)_1 [W]_{p,q}^{(A,s)} + \sum_{p=1}^{D} \sum_{q=1}^{D_A} \left( \Phi_{(A,\bullet):p,\bullet}^{(B,\bullet):\bullet,k} \right)_2 [W]_{p,q}^{(A,i)}$$

$$+ \sum_{s=1}^{n_e} \sum_{p=1}^{D} \left( \Phi_{(A,\bullet):p,\bullet}^{(B,\bullet):\bullet,k} \right)_3 [W]_{p,j}^{(A,s)} + \sum_{p=1}^{D} \left( \Phi_{(A,\bullet):p,\bullet}^{(B,\bullet):\bullet,k} \right)_4 [W]_{p,j}^{(A,i)}$$

$$+ \sum_{s=1}^{n_e} \sum_{p=1}^{D_A} \sum_{q=1}^{D} \left( \Phi_{(B,\bullet):\bullet,q}^{(B,\bullet):\bullet,k} \right)_1 [W]_{p,q}^{(B,s)} + \sum_{p=1}^{D_A} \sum_{q=1}^{D} \left( \Phi_{(B,\bullet):\bullet,q}^{(B,\bullet):\bullet,k} \right)_2 [W]_{p,q}^{(B,i)}$$

$$+ \sum_{s=1}^{n_e} \sum_{q=1}^{D} \left( \Phi_{(B,\bullet):\bullet,q}^{(B,\bullet):\bullet,k} \right)_3 [W]_{j,q}^{(B,s)} + \sum_{q=1}^{D} \left( \Phi_{(B,\bullet):\bullet,q}^{(B,\bullet):\bullet,k} \right)_4 [W]_{j,q}^{(B,i)}$$

$$+ \sum_{s=1}^{n_e} \left( \Phi_{(G,\bullet)}^{(B,\bullet):\bullet,k} \right)_1 [b]^{(G,s)} - n_e \left( \Phi_{(G,\bullet)}^{(B,\bullet):\bullet,k} \right)_1 [b]^{(G,i)}$$

$$+ \sum_{s=1}^{n_e} \sum_{q=1}^{D_A} \left( \Phi_{(A,\bullet):\bullet}^{(B,\bullet):\bullet,k} \right)_1 [b]_q^{(A,s)} + \sum_{q=1}^{D_A} \left( \Phi_{(A,\bullet):\bullet}^{(B,\bullet):\bullet,k} \right)_2 [b]_q^{(A,i)}$$

$$+ \sum_{s=1}^{n_e} \left( \Phi_{(A,\bullet):\bullet}^{(B,\bullet):\bullet,k} \right)_3 [b]_j^{(A,s)} + \left( \Phi_{(A,\bullet):\bullet}^{(B,\bullet):\bullet,k} \right)_4 [b]_j^{(A,j)}$$

$$+ \sum_{s=1}^{n_e} \sum_{q=1}^{D} (\Phi_{(B,\bullet):q}^{(B,\bullet):\bullet,k})_1 [b]_q^{(B,s)} + \sum_{q=1}^{D} (\Phi_{(B,\bullet):q}^{(B,\bullet):\bullet,k})_2 [b]_q^{(B,i)}$$

$$+ \Phi_1^{(B,\bullet):\bullet,k}.$$

**Shapes and pseudocode:** See Table 12.

| Input | Input shape | Weight | Weight shape | Einsum |
|---|---|---|---|---|
| $[WW]_{p,q}^{(QK,s)}$ | $[b,d,h,D,D]$ | $\Phi_{(QK,\bullet):p,q}^{(B,\bullet):\bullet,k}$ | $[e,d,D,D,D]$ | $(bdhpq, edkpq \rightarrow bek).usq(-2).usq(-2)$ |
| $[WW]_{p,q}^{(VO,s)}$ | $[b,d,h,D,D]$ | $\Phi_{(VO,\bullet):p,q}^{(B,\bullet):\bullet,k}$ | $[e,d,D,D,D]$ | $(bdhpq, edkpq \rightarrow bek).usq(-2).usq(-2)$ |
| $[W]_{p}^{(G,s)}$ | $[b,d,n_e,D]$ | $\left(\Phi_{(G,\bullet):p}^{(B,\bullet):\bullet,k}\right)_1$ | $[e,d,D,D]$ | $(bdnp, edkp \rightarrow bek).usq(-2).usq(-2)$ |
| $[W]_{p}^{(G,i)}$ | $[b,d,n_e,D]$ | $n_e\left(\Phi_{(G,\bullet):p}^{(B,\bullet):\bullet,k}\right)_1$ | $[e,d,D,D]$ | $(bdnp, edkp \rightarrow benk).usq(-2)$ |
| $[W]_{p,q}^{(A,s)}$ | $[b,d,n_e,D,D_A]$ | $\left(\Phi_{(A,\bullet):p,\bullet}^{(B,\bullet):\bullet,k}\right)_1$ | $[e,d,D,D]$ | $(bdnpq, edkp \rightarrow bek).usq(-2).usq(-2)$ |
| $[W]_{p,q}^{(A,i)}$ | $[b,d,n_e,D,D_A]$ | $\left(\Phi_{(A,\bullet):p,\bullet}^{(B,\bullet):\bullet,k}\right)_2$ | $[e,d,D,D]$ | $(bdnpq, edkp \rightarrow benk).usq(-2)$ |
| $[W]_{p,j}^{(A,s)}$ | $[b,d,n_e,D,D_A]$ | $\left(\Phi_{(A,\bullet):p,\bullet}^{(B,\bullet):\bullet,k}\right)_3$ | $[e,d,D,D]$ | $(bdnpj, edkp \rightarrow bejk).usq(-3)$ |
| $[W]_{p,j}^{(A,i)}$ | $[b,d,n_e,D,D_A]$ | $\left(\Phi_{(A,\bullet):p,\bullet}^{(B,\bullet):\bullet,k}\right)_4$ | $[e,d,D,D]$ | $(bdnpj, edkp \rightarrow benjk)$ |
| $[W]_{p,q}^{(B,s)}$ | $[b,d,n_e,D_A,D]$ | $\left(\Phi_{(B,\bullet):\bullet,q}^{(B,\bullet):\bullet,k}\right)$ | $[e,d,D,D]$ | $(bdnpq, edkq \rightarrow bek).usq(-2).usq(-2)$ |
| $[W]_{p,q}^{(B,i)}$ | $[b,d,n_e,D_A,D]$ | $\left(\Phi_{(B,\bullet):\bullet,q}^{(B,\bullet):\bullet,k}\right)$ | $[e,d,D,D]$ | $(bdnpq, edkq \rightarrow benk).usq(-2)$ |
| $[W]_{j,q}^{(B,s)}$ | $[b,d,n_e,D_A,D]$ | $\left(\Phi_{(B,\bullet):\bullet,q}^{(B,\bullet):\bullet,k}\right)$ | $[e,d,D,D]$ | $(bdnjq, edkq \rightarrow bejk).usq(-3)$ |
| $[W]_{j,q}^{(B,i)}$ | $[b,d,n_e,D_A,D]$ | $\left(\Phi_{(B,\bullet):\bullet,q}^{(B,\bullet):\bullet,k}\right)$ | $[e,d,D,D]$ | $(bdnjq, edkq \rightarrow benjk)$ |
| $[b]^{(G,s)}$ | $[b,d,n_e]$ | $\left(\Phi_{(G,\bullet)}^{(B,\bullet):\bullet,k}\right)_1$ | $[e,d,D]$ | $(bdn, edk \rightarrow bek).usq(-2).usq(-2)$ |
| $[b]^{(G,s)}$ | $[b,d,n_e]$ | $n_e\left(\Phi_{(G,\bullet)}^{(B,\bullet):\bullet,k}\right)_1$ | $[e,d,D]$ | $(bdn, edk \rightarrow benk).usq(-2)$ |
| $[b]_{q}^{(A,s)}$ | $[b,d,n_e,D_A]$ | $\left(\Phi_{(A,\bullet):\bullet}^{(B,\bullet):\bullet,k}\right)_1$ | $[e,d,D]$ | $(bdnq, edk \rightarrow bek).usq(-2).usq(-2)$ |
| $[b]_{q}^{(A,i)}$ | $[b,d,n_e,D_A]$ | $\left(\Phi_{(A,\bullet):\bullet}^{(B,\bullet):\bullet,k}\right)_2$ | $[e,d,D]$ | $(bdnq, edk \rightarrow benk).usq(-2)$ |
| $[b]_{j}^{(A,s)}$ | $[b,d,n_e,D_A]$ | $\left(\Phi_{(A,\bullet):\bullet}^{(B,\bullet):\bullet,k}\right)_3$ | $[e,d,D]$ | $(bdnj, edk \rightarrow bejk).usq(-3)$ |
| $[b]_{j}^{(A,j)}$ | $[b,d,n_e,D_A]$ | $\left(\Phi_{(A,\bullet):\bullet}^{(B,\bullet):\bullet,k}\right)_4$ | $[e,d,D]$ | $(bdnj, edk \rightarrow benjk)$ |
| $[b]_{q}^{(B,s)}$ | $[b,d,n_e,D]$ | $\left(\Phi_{(B,\bullet):q}^{(B,\bullet):\bullet,k}\right)_1$ | $[e,d,D,D]$ | $(bdnq, edkq \rightarrow bek).usq(-2).usq(-2)$ |
| $[b]_{q}^{(B,i)}$ | $[b,d,n_e,D]$ | $\left(\Phi_{(B,\bullet):q}^{(B,\bullet):\bullet,k}\right)_2$ | $[e,d,D,D]$ | $(bdnq, edkq \rightarrow benk).usq(-2)$ |
|  |  | $\Phi_{1}^{(B,\bullet):\bullet,k}$ | $[e,D]$ | $(ek \rightarrow ek).usq(0).usq(-2).usq(-2)$ |

Table 12: Pseudocode for $[E(W)]_{j,k}^{(B,i)}$.

| Input | Input shape | Weight | Weight shape | Einsum |
|---|---|---|---|---|
| $[WW]_{p,q}^{(QK,s)}$ | $[b,d,h,D,D]$ | $\Phi_{(QK,\bullet):p,q}^{(G,\bullet)}$ | $[e,d,D,D]$ | $(bdhpq, edpq \to be).usq(-1)$ |
| $[WW]_{p,q}^{(VO,s)}$ | $[b,d,h,D,D]$ | $\Phi_{(VO,\bullet):p,q}^{(G,\bullet)}$ | $[e,d,D,D]$ | $(bdhpq, edpq \to be).usq(-1)$ |
| $[W]_{p}^{(G,s)}$ | $[b,d,n_e,D]$ | $\left(\Phi_{(G,\bullet):p}^{(G,\bullet)}\right)_1$ | $[e,d,D]$ | $(bdnp, edp \to be).usq(-1)$ |
| $[W]_{p}^{(G,i)}$ | $[b,d,n_e,D]$ | $n_e\left(\Phi_{(G,\bullet):p}^{(G,\bullet)}\right)_1$ | $[e,d,D]$ | $(bdnp, edp \to ben)$ |
| $[W]_{p,q}^{(A,s)}$ | $[b,d,n_e,D,D_A]$ | $\Phi_{(A,\bullet):p,\bullet}^{(G,\bullet)}$ | $[e,d,D]$ | $(bdnpq, edp \to be).usq(-1)$ |
| $[W]_{p,q}^{(A,i)}$ | $[b,d,n_e,D,D_A]$ | $\Phi_{(A,\bullet):p,\bullet}^{(G,\bullet)}$ | $[e,d,D]$ | $(bdnpq, edp \to ben)$ |
| $[W]_{p,q}^{(B,s)}$ | $[b,d,n_e,D_A,D]$ | $\left(\Phi_{(B,\bullet):\bullet,q}^{(G,\bullet)}\right)$ | $[e,d,D]$ | $(bdnpq, edq \to be).usq(-1)$ |
| $[W]_{p,q}^{(B,i)}$ | $[b,d,n_e,D_A,D]$ | $\left(\Phi_{(B,\bullet):\bullet,q}^{(G,\bullet)}\right)$ | $[e,d,D]$ | $(bdnpq, edq \to ben)$ |
| $[b]^{(G,s)}$ | $[b,d,n_e]$ | $\left(\Phi_{(G,\bullet)}^{(G,\bullet)}\right)_1$ | $[e,d]$ | $(bdn, ed \to be).usq(-1)$ |
| $[b]^{(G,i)}$ | $[b,d,n_e]$ | $n_e\left(\Phi_{(G,\bullet)}^{(G,\bullet)}\right)_1$ | $[e,d]$ | $(bdn, ed \to ben)$ |
| $[b]_{q}^{(A,s)}$ | $[b,d,n_e,D_A]$ | $\left(\Phi_{(A,\bullet):\bullet}^{(G,\bullet)}\right)_1$ | $[e,d]$ | $(bdnq, ed \to be).usq(-1)$ |
| $[b]_{q}^{(A,i)}$ | $[b,d,n_e,D_A]$ | $\left(\Phi_{(A,\bullet):\bullet}^{(G,\bullet)}\right)_2$ | $[e,d]$ | $(bdnq, ed \to ben)$ |
| $[b]_{q}^{(B,s)}$ | $[b,d,n_e,D]$ | $\left(\Phi_{(B,\bullet):q}^{(G,\bullet)}\right)_1$ | $[e,d,D]$ | $(bdnq, edq \to be).usq(-1)$ |
| $[b]_{q}^{(B,i)}$ | $[b,d,n_e,D]$ | $\left(\Phi_{(B,\bullet):q}^{(G,\bullet)}\right)_2$ | $[e,d,D]$ | $(bdnq, edq \to ben)$ |
| | | $\Phi_1^{(G,\bullet)}$ | $[e]$ | $(e \to e).usq(0).usq(-1)$ |

Table 13: Pseudocode for $[E(b)]^{(G,i)}$.

## H.4.8 $[E(b)]^{(G,i)}$ PSEUDOCODE

$$
\begin{aligned}
[E(b)]^{(G,i)} &= \sum_{s=1}^{h}\sum_{p=1}^{D}\sum_{q=1}^{D} \Phi_{(QK,\bullet):p,q}^{(G,\bullet)}[WW]_{p,q}^{(QK,s)} \\
&+ \sum_{s=1}^{h}\sum_{p=1}^{D}\sum_{q=1}^{D} \Phi_{(VO,\bullet):p,q}^{(G,\bullet)}[WW]_{p,q}^{(VO,s)} \\
&+ \sum_{s=1}^{n_e}\sum_{p=1}^{D} \left(\Phi_{(G,\bullet):p}^{(G,\bullet)}\right)_1 [W]_p^{(G,s)} - \sum_{p=1}^{D} n_e\left(\Phi_{(G,\bullet):p}^{(G,\bullet)}\right)_1 [W]_p^{(G,i)} \\
&+ \sum_{s=1}^{n_e}\sum_{p=1}^{D}\sum_{q=1}^{D_A} \Phi_{(A,\bullet):p,\bullet}^{(G,\bullet)}[W]_{p,q}^{(A,s)} + \sum_{p=1}^{D}\sum_{q=1}^{D_A} \Phi_{(A,\bullet):p,\bullet}^{(G,\bullet)}[W]_{p,q}^{(A,i)} \\
&+ \sum_{s=1}^{n_e}\sum_{p=1}^{D_A}\sum_{q=1}^{D} \left(\Phi_{(B,\bullet):\bullet,q}^{(G,\bullet)}\right)[W]_{p,q}^{(B,s)} + \sum_{p=1}^{D_A}\sum_{q=1}^{D} \left(\Phi_{(B,\bullet):\bullet,q}^{(G,\bullet)}\right)[W]_{p,q}^{(B,i)} \\
&+ [b]^{(G,i)} + \sum_{s=1}^{n_e} \left(\Phi_{(G,\bullet)}^{(G,\bullet)}\right)_1 [b]^{(G,s)} - n_e\left(\Phi_{(G,\bullet)}^{(G,\bullet)}\right)_1 [b]^{(G,i)}
\end{aligned}
$$

$$+ \sum_{s=1}^{n_e} \sum_{q=1}^{D_A} \left( \Phi_{(A,\bullet):\bullet}^{(G,\bullet)} \right)_1 [b]_q^{(A,s)} + \sum_{q=1}^{D_A} \left( \Phi_{(A,\bullet):\bullet}^{(G,\bullet)} \right)_2 [b]_q^{(A,i)}$$

$$+ \sum_{s=1}^{n_e} \sum_{q=1}^{D} \left( \Phi_{(B,\bullet):q}^{(G,\bullet)} \right)_1 [b]_q^{(B,s)} + \sum_{q=1}^{D} \left( \Phi_{(B,\bullet):q}^{(G,\bullet)} \right)_2 [b]_q^{(B,i)}$$

$$+ \Phi_1^{(G,\bullet)}.$$

**Shapes and pseudocode:** See Table 13.

## H.4.9 $[E(b)]_j^{(A,i)}$ PSEUDOCODE

$$[E(b)]_j^{(A,i)} = \sum_{s=1}^{h} \sum_{p=1}^{D} \sum_{q=1}^{D} \Phi_{(QK,\bullet):p,q}^{(A,\bullet):\bullet} [WW]_{p,q}^{(QK,s)}$$

$$+ \sum_{s=1}^{h} \sum_{p=1}^{D} \sum_{q=1}^{D} \Phi_{(VO,\bullet):p,q}^{(A,\bullet):\bullet} [WW]_{p,q}^{(VO,s)}$$

$$+ \sum_{s=1}^{n_e} \sum_{p=1}^{D} \left( \Phi_{(G,\bullet):p}^{(A,\bullet):\bullet} \right)_1 [W]_p^{(G,s)} - \sum_{p=1}^{D} n_e \left( \Phi_{(G,\bullet):p}^{(A,\bullet):\bullet} \right)_1 [W]_p^{(G,i)}$$

$$+ \sum_{s=1}^{n_e} \sum_{p=1}^{D} \sum_{q=1}^{D_A} \left( \Phi_{(A,\bullet):p,\bullet}^{(A,\bullet):\bullet} \right)_1 [W]_{p,q}^{(A,s)} + \sum_{p=1}^{D} \sum_{q=1}^{D_A} \left( \Phi_{(A,\bullet):p,\bullet}^{(A,\bullet):\bullet} \right)_2 [W]_{p,q}^{(A,i)}$$

$$+ \sum_{s=1}^{n_e} \sum_{p=1}^{D} \left( \Phi_{(A,\bullet):p,\bullet}^{(A,\bullet):\bullet} \right)_3 [W]_{p,j}^{(A,s)} + \sum_{p=1}^{D} \left( \Phi_{(A,\bullet):p,\bullet}^{(A,\bullet):\bullet} \right)_4 [W]_{p,j}^{(A,i)}$$

$$+ \sum_{s=1}^{n_e} \sum_{p=1}^{D_A} \sum_{q=1}^{D} \left( \Phi_{(B,\bullet):\bullet,q}^{(A,\bullet):\bullet} \right)_1 [W]_{p,q}^{(B,s)} + \sum_{p=1}^{D_A} \sum_{q=1}^{D} \left( \Phi_{(B,\bullet):\bullet,q}^{(A,\bullet):\bullet} \right)_2 [W]_{p,q}^{(B,i)}$$

$$+ \sum_{s=1}^{n_e} \sum_{q=1}^{D} \left( \Phi_{(B,\bullet):\bullet,q}^{(A,\bullet):\bullet} \right)_3 [W]_{j,q}^{(B,s)} + \sum_{q=1}^{D} \left( \Phi_{(B,\bullet):\bullet,q}^{(A,\bullet):\bullet} \right)_4 [W]_{j,q}^{(B,i)}$$

$$+ \sum_{s=1}^{n_e} \left( \Phi_{(G,\bullet)}^{(A\bullet):\bullet} \right)_1 [b]^{(G,s)} - n_e \left( \Phi_{(G,\bullet)}^{(A,\bullet):\bullet} \right)_1 [b]^{(G,i)}$$

$$+ \sum_{s=1}^{n_e} \sum_{q=1}^{D_A} \left( \Phi_{(A,\bullet):\bullet}^{(A,\bullet):\bullet} \right)_1 [b]_q^{(A,s)} + \sum_{q=1}^{D_A} \left( \Phi_{(A,\bullet):\bullet}^{(A,\bullet):\bullet} \right)_2 [b]_q^{(A,i)}$$

$$+ \sum_{s=1}^{n_e} \left( \Phi_{(A,\bullet):\bullet}^{(A,\bullet):\bullet} \right)_3 [b]_j^{(A,s)} + \left( \Phi_{(A,\bullet):\bullet}^{(A,\bullet):\bullet} \right)_4 [b]_j^{(A,i)}$$

$$+ \sum_{s=1}^{n_e} \sum_{q=1}^{D} \left( \Phi_{(B,\bullet):q}^{(A,\bullet):\bullet} \right)_1 [b]_q^{(B,s)} + \sum_{q=1}^{D} \left( \Phi_{(B,\bullet):q}^{(A,\bullet):\bullet} \right)_2 [b]_q^{(B,i)}$$

$$+ \Phi_1^{(A,\bullet):\bullet}.$$

**Shapes and pseudocode:** See Table 14.

## H.4.10 $[E(b)]_j^{(B,i)}$ PSEUDOCODE

$$[E(b)]_j^{(B,i)} = \sum_{s=1}^{h} \sum_{p=1}^{D} \sum_{q=1}^{D} \Phi_{(QK,\bullet):p,q}^{(B,\bullet):j} [WW]_{p,q}^{(QK,s)}$$

| Input | Input shape | Weight | Weight shape | Einsum |
|---|---|---|---|---|
| $[WW]_{p,q}^{(QK,s)}$ | $[b,d,h,D,D]$ | $\Phi_{(QK,\bullet):p,q}^{(A,\bullet):\bullet}$ | $[e,d,D,D]$ | $(bdhpq, edpq \to be).usq(-1).usq(-1)$ |
| $[WW]_{p,q}^{(VO,s)}$ | $[b,d,h,D,D]$ | $\Phi_{(VO,\bullet):p,q}^{(A,\bullet):\bullet}$ | $[e,d,D,D]$ | $(bdhpq, edpq \to be).usq(-1).usq(-1)$ |
| $[W]_p^{(G,s)}$ | $[b,d,n_e,D]$ | $\left(\Phi_{(G,\bullet):p}^{(A,\bullet):\bullet}\right)_1$ | $[e,d,D]$ | $(bdnp, edp \to be).usq(-1).usq(-1)$ |
| $[W]_p^{(G,i)}$ | $[b,d,n_e,D]$ | $n_e\left(\Phi_{(G,\bullet):p}^{(A,\bullet):\bullet}\right)_1$ | $[e,d,D]$ | $(bdnp, edp \to ben).usq(-1)$ |
| $[W]_{p,q}^{(A,s)}$ | $[b,d,n_e,D,D_A]$ | $\left(\Phi_{(A,\bullet):p,\bullet}^{(A,\bullet):\bullet}\right)_1$ | $[e,d,D]$ | $(bdnpq, edp \to be).usq(-1).usq(-1)$ |
| $[W]_{p,q}^{(A,i)}$ | $[b,d,n_e,D,D_A]$ | $\left(\Phi_{(A,\bullet):p,\bullet}^{(A,\bullet):\bullet}\right)_2$ | $[e,d,D]$ | $(bdnpq, edp \to ben).usq(-1)$ |
| $[W]_{p,j}^{(A,s)}$ | $[b,d,n_e,D,D_A]$ | $\left(\Phi_{(A,\bullet):p,\bullet}^{(A,\bullet):\bullet}\right)_3$ | $[e,d,D]$ | $(bdnpj, edp \to bej).usq(-2)$ |
| $[W]_{p,j}^{(A,i)}$ | $[b,d,n_e,D,D_A]$ | $\left(\Phi_{(A,\bullet):p,\bullet}^{(A,\bullet):\bullet}\right)_4$ | $[e,d,D]$ | $(bdnpj, edp \to benj)$ |
| $[W]_{p,q}^{(B,s)}$ | $[b,d,n_e,D_A,D]$ | $\left(\Phi_{(B,\bullet):\bullet,q}^{(A,\bullet):\bullet}\right)_1$ | $[e,d,D]$ | $(bdnpq, edq \to be).usq(-1).usq(-1)$ |
| $[W]_{p,q}^{(B,i)}$ | $[b,d,n_e,D_A,D]$ | $\left(\Phi_{(B,\bullet):\bullet,q}^{(A,\bullet):\bullet}\right)_2$ | $[e,d,D]$ | $(bdnpq, edq \to ben).usq(-1)$ |
| $[W]_{j,q}^{(B,s)}$ | $[b,d,n_e,D_A,D]$ | $\left(\Phi_{(B,\bullet):\bullet,q}^{(A,\bullet):\bullet}\right)_3$ | $[e,d,D]$ | $(bdnjq, edq \to bej).usq(-2)$ |
| $[W]_{j,q}^{(B,i)}$ | $[b,d,n_e,D_A,D]$ | $\left(\Phi_{(B,\bullet):\bullet,q}^{(A,\bullet):\bullet}\right)_4$ | $[e,d,D]$ | $(bdnjq, edq \to benj)$ |
| $[b]^{(G,s)}$ | $[b,d,n_e]$ | $\left(\Phi_{(G,\bullet)}^{(A,\bullet):\bullet}\right)_1$ | $[e,d]$ | $(bdn, ed \to be).usq(-1).usq(-1)$ |
| $[b]^{(G,i)}$ | $[b,d,n_e]$ | $n_e\left(\Phi_{(G,\bullet)}^{(A,\bullet):\bullet}\right)_1$ | $[e,d]$ | $(bdn, ed \to ben).usq(-1)$ |
| $[b]_q^{(A,s)}$ | $[b,d,n_e,D_A]$ | $\left(\Phi_{(A,\bullet):\bullet}^{(A,\bullet):\bullet}\right)_1$ | $[e,d]$ | $(bdnq, ed \to be).usq(-1).usq(-1)$ |
| $[b]_q^{(A,i)}$ | $[b,d,n_e,D_A]$ | $\left(\Phi_{(A,\bullet):\bullet}^{(A,\bullet):\bullet}\right)_2$ | $[e,d]$ | $(bdnq, ed \to ben).usq(-1)$ |
| $[b]_j^{(A,s)}$ | $[b,d,n_e,D_A]$ | $\left(\Phi_{(A,\bullet):\bullet}^{(A,\bullet):\bullet}\right)_3$ | $[e,d]$ | $(bdnj, ed \to bej).usq(-2)$ |
| $[b]_j^{(A,i)}$ | $[b,d,n_e,D_A]$ | $\left(\Phi_{(A,\bullet):\bullet}^{(A,\bullet):\bullet}\right)_4$ | $[e,d]$ | $(bdnj, ed \to benj)$ |
| $[b]_q^{(B,s)}$ | $[b,d,n_e,D]$ | $\left(\Phi_{(B,\bullet):q}^{(A,\bullet):\bullet}\right)_1$ | $[e,d,D]$ | $(bdnq, edq \to be).usq(-1).usq(-1)$ |
| $[b]_q^{(B,i)}$ | $[b,d,n_e,D]$ | $\left(\Phi_{(B,\bullet):q}^{(A,\bullet):\bullet}\right)_2$ | $[e,d,D]$ | $(bdnq, edq \to ben).unsq(-1)$ |
| | | $\Phi_1^{(A,\bullet):\bullet}$ | $[e]$ | $(e \to e).usq(0).usq(-1).usq(-1)$ |

Table 14: Pseudocode for $[E(b)]_j^{(A,i)}$.

| Input | Input shape | Weight | Weight shape | Einsum |
|---|---|---|---|---|
| $[WW]_{p,q}^{(QK,s)}$ | $[b,d,h,D,D]$ | $\Phi_{(QK,\bullet):p,q}^{(B,\bullet):j}$ | $[e,d,D,D,D]$ | $(bdhpq,edjpq \to bej).usq(-2)$ |
| $[WW]_{p,q}^{(VO,s)}$ | $[b,d,h,D,D]$ | $\Phi_{(VO,\bullet):p,q}^{(B,\bullet):j}$ | $[e,d,D,D,D]$ | $(bdhpq,edjpq \to bej).usq(-2)$ |
| $[W]_{p}^{(G,s)}$ | $[b,d,n_e,D]$ | $\left(\Phi_{(G,\bullet):p}^{(B,\bullet):j}\right)_1$ | $[e,d,D,D]$ | $(bdnp,edjp \to bej).usq(-2)$ |
| $[W]_{p}^{(G,i)}$ | $[b,d,n_e,D]$ | $n_e\left(\Phi_{(G,\bullet):p}^{(B,\bullet):j}\right)_1$ | $[e,d,D,D]$ | $(bdnp,edjp \to benj)$ |
| $[W]_{p,q}^{(A,s)}$ | $[b,d,n_e,D,D_e]$ | $\left(\Phi_{(A,\bullet):p,\bullet}^{(B,\bullet):j}\right)_1$ | $[e,d,D,D]$ | $(bdnpq,edjp \to bej).usq(-2)$ |
| $[W]_{p,q}^{(A,i)}$ | $[b,d,n_e,D,D_e]$ | $\left(\Phi_{(A,\bullet):p,\bullet}^{(B,\bullet):j}\right)_2$ | $[e,d,D,D]$ | $(bdnpq,edjp \to benj)$ |
| $[W]_{p,q}^{(B,s)}$ | $[b,d,n_e,D_e,D]$ | $\left(\Phi_{(B,\bullet):\bullet,q}^{(B,\bullet):j}\right)_1$ | $[e,d,D,D]$ | $(bdnpq,edjq \to bej).usq(-2)$ |
| $[W]_{p,q}^{(B,i)}$ | $[b,d,n_e,D_A,D]$ | $\left(\Phi_{(B,\bullet):\bullet,q}^{(B,\bullet):j}\right)_2$ | $[e,d,D,D]$ | $(bdnpq,edjq \to benj)$ |
| $[b]^{(G,s)}$ | $[b,d,n_e]$ | $\left(\Phi_{(G,\bullet)}^{(B,\bullet):j}\right)_1$ | $[e,d,D]$ | $(bdn,edj \to bej).usq(-2)$ |
| $[b]^{(G,i)}$ | $[b,d,n_e]$ | $n_e\left(\Phi_{(G,\bullet)}^{(B,\bullet):j}\right)_1$ | $[e,d,D]$ | $(bdn,edj \to benj)$ |
| $[b]_{q}^{(A,s)}$ | $[b,d,n_e,D_A]$ | $\left(\Phi_{(A,\bullet):\bullet}^{(B,\bullet):j}\right)_1$ | $[e,d,D]$ | $(bdnq,edj \to bej).usq(-2)$ |
| $[b]_{q}^{(A,i)}$ | $[b,d,n_e,D_A]$ | $\left(\Phi_{(A,\bullet):\bullet}^{(B,\bullet):j}\right)_2$ | $[e,d,D]$ | $(bdnq,edj \to benj)$ |
| $[b]_{q}^{(B,s)}$ | $[b,d,n_e,D]$ | $\left(\Phi_{(B,\bullet):q}^{(B,\bullet):j}\right)_1$ | $[e,d,D,D]$ | $(bdnq,edjq \to bej).usq(-2)$ |
| $[b]_{q}^{(B,i)}$ | $[b,d,n_e,D]$ | $\left(\Phi_{(B,\bullet):q}^{(B,\bullet):j}\right)_2$ | $[e,d,D,D]$ | $(bdnq,edjq \to benj)$ |
| | | $\Phi_1^{(B,\bullet):j}$ | $[e,D]$ | $(ej \to ej).usq(0).usq(-2)$ |

Table 15: Pseudocode for $[E(b)]_j^{(B,i)}$.

| Input | Input shape | Weight | Weight shape | Einsum |
|---|---|---|---|---|
| $[WW]_{p,q}^{(QK,s)}$ | $[b, d, h, D, D]$ | $\Phi_{(QK,\bullet):p,q}^{i}$ | $[e, d, D', D, D]$ | $(bdhpq, edipq \rightarrow bei)$ |
| $[WW]_{p,q}^{(VO,s)}$ | $[b, d, h, D, D]$ | $\Phi_{(VO,\bullet):p,q}^{i}$ | $[e, d, D', D, D]$ | $(bdhpq, edipq \rightarrow bei)$ |
| $[W]_{p}^{(G,s)}$ | $[b, d, n_e, D]$ | $\bar{\Phi}_{(G,\bullet):p}^{i}$ | $[e, d, D', D]$ | $(bdnp, edip \rightarrow bei)$ |
| $[W]_{p,q}^{(A,s)}$ | $[b, d, n_e, D, D_e]$ | $\Phi_{(A,\bullet):p,\bullet}^{i}$ | $[e, d, D', D]$ | $(bdnpq, edip \rightarrow bei)$ |
| $[W]_{p,q}^{(B,s)}$ | $[b, d, n_e, D_e, D]$ | $\Phi_{(B,\bullet):\bullet,q}^{i}$ | $[e, d, D', D]$ | $(bdnpq, ediq \rightarrow bei)$ |
| $[b]^{(G,s)}$ | $[b, d, n_e]$ | $\bar{\Phi}_{(G,\bullet)}^{i}$ | $[e, d, D']$ | $(bdn, edi \rightarrow bei)$ |
| $[b]_{q}^{(A,s)}$ | $[b, d, n_e, D_e]$ | $\Phi_{(A,\bullet):\bullet}^{i}$ | $[e, d, D']$ | $(bdnq, edi \rightarrow bei)$ |
| $[b]_{q}^{(B,s)}$ | $[b, d, n_e, D]$ | $\Phi_{(B,\bullet):q}^{i}$ | $[e, d, D', D]$ | $(bdnq, ediq \rightarrow bei)$ |
| | | $\Phi_{1}^{i}$ | $[e, D']$ | $(ei \rightarrow ei).usq(0)$ |

Table 16: Pseudocode for Invariant Layer.

$$
+ \sum_{s=1}^{h} \sum_{p=1}^{D} \sum_{q=1}^{D} \Phi_{(VO,\bullet):p,q}^{(B,\bullet):j} [WW]_{p,q}^{(VO,s)}
$$

$$
+ \sum_{s=1}^{n_e} \sum_{p=1}^{D} \left( \Phi_{(G,\bullet):p}^{(B,\bullet):j} \right)_{1} [W]_{p}^{(G,s)} - \sum_{p=1}^{D} n_e \left( \Phi_{(G,\bullet):p}^{(B,\bullet):j} \right)_{1} [W]_{p}^{(G,i)}
$$

$$
+ \sum_{s=1}^{n_e} \sum_{p=1}^{D} \sum_{q=1}^{D_A} \left( \Phi_{(A,\bullet):p,\bullet}^{(B,\bullet):j} \right)_{1} [W]_{p,q}^{(A,s)} + \sum_{p=1}^{D} \sum_{q=1}^{D_A} \left( \Phi_{(A,\bullet):p,\bullet}^{(B,\bullet):j} \right)_{2} [W]_{p,q}^{(A,i)}
$$

$$
+ \sum_{s=1}^{n_e} \sum_{p=1}^{D_A} \sum_{q=1}^{D} \left( \Phi_{(B,\bullet):\bullet,q}^{(B,\bullet):j} \right)_{1} [W]_{p,q}^{(B,s)} + \sum_{p=1}^{D_A} \sum_{q=1}^{D} \left( \Phi_{(B,\bullet):\bullet,q}^{(B,\bullet):j} \right)_{2} [W]_{p,q}^{(B,i)}
$$

$$
+ \left( \Phi_{(G,\bullet)}^{(B,\bullet):j} \right)_{1} [b]^{(G,s)} - n_e \left( \Phi_{(G,\bullet)}^{(B,\bullet):j} \right)_{1} [b]^{(G,i)}
$$

$$
+ \sum_{s=1}^{n_e} \sum_{q=1}^{D_A} \left( \Phi_{(A,\bullet):\bullet}^{(B,\bullet):j} \right)_{1} [b]_{q}^{(A,s)} + \sum_{q=1}^{D_A} \left( \Phi_{(A,\bullet):\bullet}^{(B,\bullet):j} \right)_{2} [b]_{q}^{(A,i)}
$$

$$
+ \sum_{s=1}^{n_e} \sum_{q=1}^{D} \left( \Phi_{(B,\bullet):q}^{(B,\bullet):j} \right)_{1} [b]_{q}^{(B,s)} + \sum_{q=1}^{D} \left( \Phi_{(B,\bullet):q}^{(B,\bullet):j} \right)_{2} [b]_{q}^{(B,i)}
$$

$$
+ \Phi_{1}^{(B,\bullet):j}.
$$

**Shapes and pseudocode:** See Table 15.

## H.5 Invariant Layers Pseudocode

$$
I(U)_i = \sum_{s=1}^{h} \sum_{p=1}^{D} \sum_{q=1}^{D} \Phi_{(QK,\bullet):p,q}^{i} [WW]_{p,q}^{(QK,s)}
$$

$$
+ \sum_{s=1}^{h} \sum_{p=1}^{D} \sum_{q=1}^{D} \Phi_{(VO,\bullet):p,q}^{i} [WW]_{p,q}^{(VO,s)}
$$

$$
+ \sum_{s=1}^{n_e} \sum_{p=1}^{D} \left( \Phi_{(G,\bullet):p}^{i} - \frac{1}{n_e} \sum_{s=1}^{n_e} \Phi_{(G,\bullet):p}^{i} \right) [W]_{p}^{(G,s)}
$$

Table 17: Ablation study on network components for generalization prediction. Kendall's $\tau$ is reported for models using only the MoE Transformer blocks, only the classifier hear, and both.

| Component Used | MoE Transformer blocks | Classifier head | MoE Transformer blocks + Classifier head |
|---|---|---|---|
| Kendall's $\tau$ | 0.775 | 0.597 | **0.788** |

$$+ \sum_{s=1}^{n_e} \sum_{p=1}^{D} \sum_{q=1}^{D_A} \Phi^i_{(A,\bullet):p,\bullet}[W]^{(A,s)}_{p,q}$$

$$+ \sum_{s=1}^{n_e} \sum_{p=1}^{D_A} \sum_{q=1}^{D} \Phi^i_{(B,\bullet):\bullet,q}[W]^{(B,s)}_{p,q}$$

$$+ \sum_{s=1}^{n_e} \left( \Phi^i_{(G,\bullet)} - \frac{1}{n_e} \sum_{s=1}^{n_e} \Phi^i_{(G,\bullet)} \right) [b]^{(G,s)}$$

$$+ \sum_{s=1}^{n_e} \sum_{q=1}^{D_A} \Phi^i_{(A,\bullet):\bullet}[b]^{(A,s)}_q$$

$$+ \sum_{s=1}^{n_e} \sum_{q=1}^{D} \Phi^i_{(B,\bullet):q}[b]^{(B,s)}_q$$

$$+ \Phi^i_1.$$

**Shapes and pseudocode:** See Table 16.

## I   ABLATION STUDY ON IMPORTANCE OF MOE TRANSFORMER BLOCKS IN PREDICTING MODEL PERFORMANCE

**Experiment Setup.** It is natural to ask whether the MoE Transformer blocks or the classification head contribute more to predicting a model's generalization performance. To investigate this, we conduct an ablation study on the AGNews-MoE dataset by restricting the input to the neural functional model. Specifically, we evaluate its performance when given access to: (1) both the MoE Transformer blocks and classification head weights, (2) only the MoE Transformer block weights, and (3) only the classification head weights. This allows us to assess which component is most predictive of model generalization.

**Results.** Table 17 from demonstrates that using only the MoE Transformer blocks results in a Kendall's $\tau$ of $0.775$, while using only the classifier head yields $0.597$. When both components are included, performance improves to $0.788$. This suggests that the MoE blocks contribute most to generalization prediction, while the classifier head provides complementary information.

## J   ABLATION STUDY ON THE EFFECT OF LAYER SIZE AND DEPTH

**Experiment Setup.** In this section, we examine how the number of layers and the hidden dimension of each MoE-NFN layer affect the model's ability to predict generalization on the AGNews-MoEs. We do so by varying the hidden dimensions in $\{2, 4, 6, 10\}$ and the number of layers in $\{1, 2\}$.

**Results.** Table 19 from Appendix L shows that MoE-NFN achieves consistently strong performance across a range of model sizes. Notably, even the smallest configuration, with a single layer and hidden size of 2, reaches a Kendall's $\tau$ of $0.784$. In contrast, the best performance is obtained with two layers of hidden size 10, achieving a Kendall's $\tau$ of $0.806$. This demonstrates that while increased capacity can improve performance, MoE-NFN remains highly effective even under constrained model sizes.

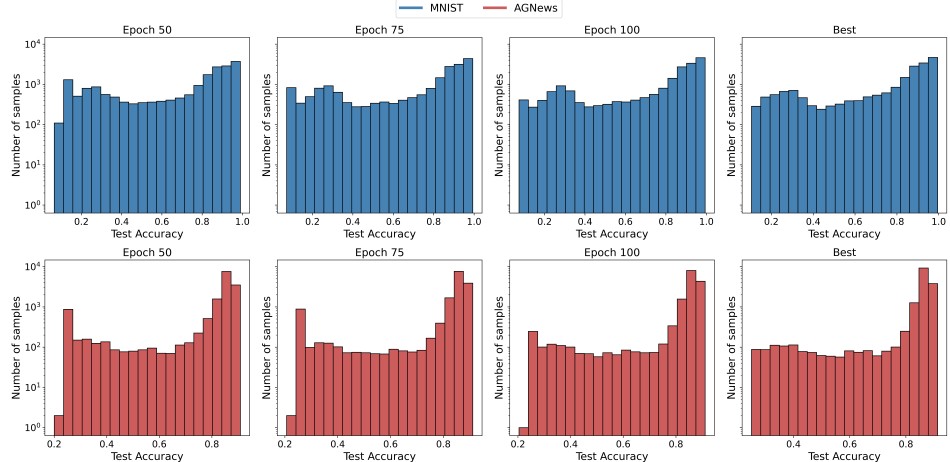

Figure 1: Histogram of test accuracy distribution in the MNIST-MoEs and AGNews-MoEs datasets.

# K    ADDITIONAL DATASET DETAILS

To explore a rich landscape of Transformer-MoE architectures, we systematically vary eight core hyperparameters in our study: top-K, activation function, training data fraction, optimizer (selected from SGD, SGDm, Adam, or RMSprop), learning rate, L2 regularization coefficient, initialization standard deviation, and dropout probability. Each plays a distinct role - train fraction dictates how much of the dataset is fed into training, while the optimizer governs the learning dynamics. Learning rate, regularization, and initialization standard deviation modulate convergence behavior, and dropout serves as a defense against overfitting. For top-K, we test values 1, 2, and 4 - representing how many expert modules process a given token. As for activation, models flip between ReLU and GeLU.

We treat each hyperparameter dimension independently, selecting representative values before exhaustively combining them into a sweeping configuration grid. Early experiments highlighted that the ideal hyperparameter landscape diverges drastically depending on optimizer type. Thus, we separate our configurations into two distinct families: one for Adam and RMSprop, and another for SGD and SGDm. Table 18 lays out the full matrix. These setups remain consistent across tasks to ensure apples-to-apples comparisons. All models undergo 100 training epochs, with performance snapshots at epochs 50, 75, 100, and the epoch of peak accuracy. Crashed runs are promptly discarded.

Table 18: Hyperparameter configurations of the MoE Transformer Model Zoos dataset

| Hyperparameter | SGD-SGDm | Adam-RMSprop |
|---|---|---|
| Top-K | [1,2,4] | [1,2,4] |
| Activation | [ReLU, GeLU] | [ReLU, GeLU] |
| Train Fraction | [1.0, 0.9, 0.8] | [1.0, 0.9, 0.8] |
| Dropout | [0.2, 0.15, 0.1, 0.05, 0] | [0.2, 0.15, 0.1, 0.05, 0] |
| Learning Rate - MNIST | [1e-3, 3e-3, 5e-3, 1e-2, 3e-2] | [3e-4, 5e-4, 1e-3, 5e-3, 3e-2] |
| Learning Rate - AGNews | [1e-3, 3e-3, 1e-2, 5e-2, 7e-2] | [3e-4, 1e-3, 5e-3, 3e-2, 5e-2] |
| Weight Init Standard Deviation | [0.1, 0.15, 0.2, 0.25] | [0.1, 0.2, 0.3, 0.4] |
| L2 Regularization - MNIST | [1e-6, 1e-4, 1e-2] | [1e-6, 1e-4, 1e-2] |
| L2 Regularization - AGNews | [1e-8, 1e-6, 1e-4] | [1e-8, 1e-6, 1e-4] |

**MNIST-MoE.**    The MNIST dataset (LeCun & Cortes, 2005), a staple in the vision benchmark canon, presents $28 \times 28$ grayscale images of handwritten digits ranging from 0 to 9. The goal: classify the digit shown. Our model begins with a 2D convolutional embedding that carves the image into patches, overlayed with fixed positional encodings to anchor spatial information. These em-

Table 19: Effect of MoE-NFN architecture (width and depth) on generalization prediction.

| Encoder Term | [2] | [2,2] | [4] | [4,4] | [6] | [6,6] | [10] | [10,10] |
|---|---|---|---|---|---|---|---|---|
| Kendall's $\tau$ | 0.784 | 0.794 | 0.788 | 0.797 | 0.775 | 0.799 | 0.781 | 0.806 |
| Params | 1.5M | 3.9M | 2.9M | 12.7M | 4.6M | 26.5M | 8.4M | 69.2M |

beddings pass through two Transformer-MoE blocks, which weave global dependencies across the image. In the MoE block, there are 4 experts, each is a two-layer feedforward network. The resulting representations are globally averaged and routed through a two-layer feedforward classifier, separated by ReLU, culminating in a ten-class probability distribution. Using our hyperparameter schema, we generate a massive 100,024 model samples for MNIST - 25,006 of which are checkpoints from selected epochs. Figure 1 shows the accuracy histogram, with the accuracy distributed across [0,1].

**AGNews-MoE.** The AG's News dataset (Zhang et al., 2015) offers a text classification challenge across four broad domains: World, Sports, Business, and Sci/Tech. For each article, the model predicts its corresponding topic based on its description. Our Transformer-MoE variant kicks off with token embeddings sourced from a pre-trained Word2Vec model, fused with fixed positional encodings to maintain sequence order. These flow into a dual-layer Transformer-MoE encoder that captures semantic interrelations across the input. In the MoE block, there are 4 experts, each is a two-layer feedforward network. The encoder output undergoes global average pooling, then feeds into a two-layer MLP with a ReLU bridge, concluding with a four-class softmax. Across this task, we generate 79,220 checkpoints derived from 19,805 unique configurations, capturing performance at epochs 50, 75, 100, and each model's best epoch. The accuracy distribution (Figure 1) reveals a pronounced peak between 50% and 90%, with a sharp mode around 80%, and a modest secondary cluster hovering near 25%.

**Computing Resources** The whole dataset is trained on a cluster of 4x NVIDIA A100 SXM4 80GB GPUs. We run 5 settings at a time on one GPU. The running time for a MNIST-MoE setting is 20 to 25 minutes, depending on the fraction of training data being set. The running time for an AGNews-MoE setting is 30 to 35 minutes.

# L ADDITIONAL EXPERIMENT DETAILS

## L.1 GENERAL DETAILS

**Training details** All models underwent training over 100 epochs with a batch size set to 16. Optimization was carried out using Adam, capped at a peak learning rate of $10^{-3}$ (In the case of MLP the learning rate is $10^{-4}$). To ease the model into learning, we implemented a linear learning rate warmup during the first 10 epochs. The loss was computed using the Binary Cross Entropy criterion.

**Computing resource** All experiments were conducted on a workstation equipped with an AMD Ryzen Threadripper PRO 5945WX processor (24 cores) and four NVIDIA GeForce RTX 3090 GPUs (24GB VRAM each). GPU driver version 570.86.15 and CUDA 12.8 were used. Each experiment was completed in under 12 hours using this hardware configuration.

**Number of parameters** An overview of parameter counts for each model is presented in Table 20. Complete architectural specifications and hyperparameter settings can be found in Appendices L.2 and L.3. For the baseline models, hyperparameters were carefully tuned to their optimal configurations; any further increase in parameter size likely leads to overfitting rather than improved performance.

## L.2 ARCHIECTURE AND HYPERPARAMETERS OF MOE-NFN

The MoE-NFN architecture is structured around three core modules, each tailored to manage the weight processing in a Transformer MoE system. The embedding and classification components

are both implemented using standard multi-layer perceptrons (MLPs) with ReLU activation, each independently handling a distinct part of the input.

At the heart of the model lies the Transformer MoE block, which is governed by an invariant architecture featuring multiple MoE-NFN equivariant polynomial layers. These layers specifically target the two MLP segments within the Transformer block and are activated using ReLU. Once processed, their output is funneled into an invariant polynomial layer of MoE-NFN, which further distills the representation. All intermediate outputs - vectorized by design - are concatenated and fed into a terminal MLP head equipped with a Sigmoid activation function to generate the prediction.

For our experiments, the embedding component consists of a single-layer MLP with 100 hidden units. The classification module is slightly deeper, comprising two MLP layers, each also with 100 hidden units. Within the invariant MoE-NFN core, a single equivariant polynomial layer with 4 hidden channels is used to process the Transformer weights, followed by an invariant polynomial layer that yields a 5-dimensional vector per input layer. These outputs are then combined and passed through another MLP, which expands them into a 100-dimensional vector space. Ultimately, the concatenated outputs from all three branches are directed through a final classification layer to produce the model's prediction.

### L.3 ARCHITECTURE AND HYPERPARAMETERS FOR OTHER BASELINES

Here we describe the architecture of all baselines:

- **Transformer-NFN** (Tran et al., 2025) This model comprises three primary modules responsible for processing the input weights. The embedding is processed by a single layer MLP, while classifier component utilizes two-layer MLPs, each with 100 hidden units. The Transformer core is modeled using an invariant architecture that integrates 2 Transformer-NFN equivariant polynomial layers, with 12 hidden channels. These are followed by an invariant polynomial layer to finalize the transformation. Outputs from each module are encoded as vectors, concatenated, and passed through a concluding MLP head (100 hidden units, Sigmoid activation) for prediction. To make this architecture compatible with Transformer-MoE inputs, we omit gating weights and average the expert-specific weights to form a unified feed-forward layer, suitable for Transformer-NFN. However, this adaptation breaks the model's original equivariance under the new group action introduced by Transformer-MoE.

- **MLP** In this baseline, all model component weights are flattened and processed individually through dedicated MLPs. The embedding and Transformer-MoE components are each fed into a single-layer MLP with 64 hidden neurons. The classifier component, by contrast, is modeled with a two-layer MLP containing 256 neurons per layer. Outputs from all three branches are concatenated and passed through a final prediction head: a two-layer MLP with 100 hidden neurons in each layer.

- **XGBoost** (Chen & Guestrin, 2016), **LightGBM** (Ke et al., 2017), **Random Forest** (Breiman, 2001): For these tree-based models, we flatten the weights from all components and input them directly into the respective regressors. We used consistent hyperparameter settings across the three models: maximum tree depth of 10, minimum child weight of 50, and a cap of 256 leaves per tree.

- **SVR** (Vapnik et al., 1996): All input weights are first flattened and then reduced to 1000 dimensions via Principal Component Analysis (PCA)(Pearson, 1901; Hotelling, 1933). The resulting feature set is processed by a linear Support Vector Regression (SVR) model using a linear kernel. We adopt the default configuration provided by the scikit-learn library.

### L.4 $\mathcal{G}_{\mathcal{U}}$ TRANSFORMATION EXPERIMENT

In this experiment, we keep all training settings the same as the AGNews-MoE performance prediction experiment. We retrain each of the baseline metanetwork and evaluate the trained model on both the original test set and an augmented version of the original test set. Then we record the Kendall's $\tau$ metric for both test sets and compute the gap between them.

Table 20: Number of parameters for all models

| Model | MNIST | AGNews |
|---|---|---|
| MoE-NFN | 3.088M | 2.984M |
| Transformer-NFN | 2.511M | 2.406M |
| MLP | 11.359M | 11.255M |

The augmented version of the test set of AGNews-MoE dataset is produced by applying randomly selected transformations from the group $\mathcal{G}_{\mathcal{U}}$ to the original model weights. These transformations yield new models that are functionally identical but differ in parameterization. We uniformly sample the permutations $\tau_h$, $\tau_e$, and $\pi_e^{(i)}$, sample the scalars $\gamma_W$ and $\gamma_b$ from the interval $[0, 1]$, and sampling each entry of the transformation matrices $M_k^{(i)}$ and $M_v^{(i)}$ from a uniform distribution over $[-100, 100]$.

## M  BROADER IMPACTS

This work contributes to the foundational understanding of functional equivalence in neural network architectures, particularly Mixture-of-Experts (MoE), with implications that extend to the design and interpretation of modern AI systems. By rigorously characterizing the symmetry-induced redundancies in MoE models, our analysis enables the development of more parameter - efficient, interpretable, and robust architectures. These insights are especially relevant for metanetworks - neural systems that reason over other networks - where ensuring functional identity is critical for tasks like model editing, transfer learning, and interpretability.

The societal benefits of this research stem from its potential to reduce computational waste by elliviate the computational need to evaluate language model. In domains such as healthcare and environmental science, where large-scale models are increasingly deployed for predictive diagnostics or climate modeling, such efficiency gains can reduce energy consumption, and make cutting-edge AI more accessible to under-resourced settings. Moreover, by deepening the theoretical understanding of neural network symmetries, this work contributes to safer and more transparent AI development, helping mitigate risks associated with model redundancy, overparameterization, and brittleness.

Overall, the theoretical advancements presented in this paper support the broader movement toward efficient, reliable, and responsible AI - enhancing both the scalability of current models and the interpretability of their inner workings, which are crucial for high-stakes and mission-critical applications.

