# OpenReview forum: "Equivariant Metanetworks for Mixture-of-Experts Weights"
_ICLR.cc/2026/Conference — ICLR 2026 Conference Withdrawn Submission_

### Official Review · Reviewer_y77G · 2025-11-01

**Soundness:** 3
**Presentation:** 3
**Contribution:** 2
**Rating:** 4
**Confidence:** 3

**Summary:**

This paper explores the foundational concept of functional equivalence in Mixture-of-Experts (MoE) architectures and demonstrates its application to the design of equivariant metanetworks. Functional equivalence refers to the condition where distinct parameterizations of a neural network result in identical input–output mappings. The authors formalize the group actions that preserve functional behavior in MoE architectures, addressing both dense and sparse gating settings. Key theoretical contributions include characterizations of functional equivalence via permutation and affine transformations of expert and gating modules. The framework is rigorously justified through proofs based on ReLU network properties and exponential function linear independence. These theoretical insights are then applied to build equivariant metanetworks for MoE Transformers, ensuring consistent output for functionally equivalent input weights. Two new datasets—MNIST-MoEs and AGNews-MoEs—are introduced to empirically evaluate metanetworks. Experiments show that their proposed model, MoE-NFN, outperforms baselines, including Transformer-NFN, in generalization prediction. The metanetwork is shown to be robust under group action transformations, affirming its equivariance. Overall, the paper offers a principled mathematical and empirical investigation into weight-function symmetry in MoEs and how to exploit it in metanetwork design.

**Strengths:**

This is the first work to study theoretical characterization of functional equivalence in both dense and sparse MoE models, and develops an equivariant metanetwork architecture (MoE-NFN) grounded in theoretical insights. Rich theoretic justifications are provided.
The experiments show that MoE-NFN largely outperforms reasonable baselines. And the release of large modelzoo datasets of MoE transformers adds significant value to the paper and will help stimulate more study of metanetworks.

**Weaknesses:**

The gating mechanism is quite intuitive and maybe a bit simple (not necessarily a bad thing) for applying to the MoE. The equivariant layer part very much based on existing work of NFN / Transformer NFN. Also a large portion of theoretical contributions are very much based on existing work.

Experiments limited to one task of generalization prediction of test accuracy (on two datasets). Evaluation measurement also only limited to Kendall correlation.

**Questions:**

- Could you tell more detail why top-1 is not covered in the proof of Theorem 3.4?

- I can probably understand using Kendall correlation for evaluation between predicted and ground-truth accuracy rankings for MoEs and that is reasonable. What about more absolute error measurements?

- Could you clarify which from the theoretical contributions are more original and specific to MoE setups?

I put initial score on the negative side of borderline but I would be willing to raise it if my concerns are solved.

---

### Official Review · Reviewer_pWAZ · 2025-11-01

**Soundness:** 3
**Presentation:** 3
**Contribution:** 3
**Rating:** 6
**Confidence:** 4

**Summary:**

This work introduces equivariant metanetworks for Mixtures of Experts architectures. The authors formalize the symmetries in MoEs, combined with multi-head attention modules. They introduce datasets of (semi-)trained MoEs and perform experiments on predicting MoE generalization. The proposed method outperforms all competing methods in all experiments.

**Strengths:**

The manuscript reads very well, despite being very math heavy.
The mathematical formalism is very clear and concise as well.

The research question is very topical; metanetworks are becoming increasingly relevant in the community, and MoEs are an important neural network module. Hence, studying the symmetries in MoEs and proposing equivariant architectures is very significant.

The introduced dataset is an important addition in the model zoo datasets.

**Weaknesses:**

The performance gains compared to TransformerNFN are, in most cases, marginal. The benefit of incorporating MoE symmetries in the metanetwork is thus, not demonstrated convincingly.

The breadth of the experiments is also quite small, including only a single task and two datasets for evaluation.

**Questions:**

1. Are there any experiments (possibly on synthetic datasets) that the authors can perform to convincingly demonstrate the benefit of incorporating the MoE symmetries in the architecture? In other words, is there a setting in which TransformerNFN would perform poorly where the proposed method performs well?

2. Can the MoE symmetries be addressed with data augmentation instead of being explicitly accounted for? What is the performance of TransformerNFN trained with data augmentations on MoE symmetries?

---

### Official Review · Reviewer_MHtZ · 2025-11-01

**Soundness:** 4
**Presentation:** 2
**Contribution:** 1
**Rating:** 0
**Confidence:** 3

**Summary:**

The paper studies functional equivalence for Mixture-of-Experts (MoE) and Sparse MoEs. On the theory side, the authors give a constructive definition of a group action on these models parameter spaces. They further show that these actions are exactly the symmetries preserving functional equivalence in the gating components of MoE models. Building on this, the authors construct an equivariant metanetwork for MoE-Transformer architectures and validate it on two datasets of MoE-Transformer checkpoints.

**Strengths:**

- The paper has strong theoretical impact: it tackles the identifiability of MoE gating which is a fundamental yet poorly understood mechanism in modern state-of-the-art LLMs.
- The paper has strong practical impact: it introduces an equivariant metanetwork for MoE-Transformers, constructed in a principled way and grounded in the presented theory.

**Weaknesses:**

There are strong overlaps without proper citation in the submitted paper.
See the bibliography at the bottom.

- Prop A.1, A.2, A.3, A.4 in [S] <-> Prop A.1 in [3].
- Lemma B.2 in [S] <-> Lemma B.3 in [3].
- Block of Def and Rmk B.4–B.6 in [S] <-> App. B.1 in [3].
- Thm B.7 in [S] <-> Thm B.5 in [3].
- Thm C.5 in [S] <-> Thm C.4 in [3].

Partial overlap and partially referenced:

- Sec. 4 and App. D [S] <-> Thm 4.3 and App. C in [1]

In particular, certain dependencies within [S] are concerning. Given the overlaps documented above, the attribution of novelty is undermined and concerns arise regarding the main contributions, which are as follows:

- Thm 3.1 is Thm B.7 in appendix of [S] which is equivalent to Thm B.5 in [3].
- Thm 3.4 is Thm C.5 in appendix of [S] which is equivalent to Thm C.4 in [3].

For context only: [1-3], all published, also exhibit non-trivial mutual overlaps, which further complicates novelty attribution in [S]. I have not investigated these in depth, as all four papers [S, 1-3] are unusually long for conference submissions, with substantial appendices. A full audit of overlapping proofs is infeasible under normal review time constraints.

I will not speculate about authorship or intent.

**Bibliography:**
[S] submitted paper
[1] Tran et al., _Equivariant Neural Functional Networks for Transformers_
[2] Tran et al., _Equivariant Polynomial Functional Networks_
[3] Tran et al., _On Linear Mode Connectivity of Mixture-of-Experts Architectures_

**Questions:**

Could you address the identified weaknesses with appropriate rigor and detail?

---

> ### Author Response · Authors · 2025-11-12
>
> We would like to sincerely clarify the relationship between our submission [S] and the cited works [1], [2], and [3] as follows:
>
> - All these works are centered around the symmetry properties of neural architectures.
>
> - Works [1], [2], and [3] are independent. The length of [1] and [2] arises from the detailed derivations of equivariant metanetworks based on parameter-sharing principles ([1] focuses on Transformers, while [2] focuses on MLP/CNNs). In contrast, [3] is lengthy mainly due to its theoretical proof on the symmetry of Mixture-of-Experts architectures and its visualization of Linear Mode Connectivity.
>
> - [S] and [1,2] belong to the same line of research on equivariant metanetwork design, each addressing different neural architectures. Although the metanetwork computations may appear similar at first glance, their formulations are distinct. These derivations are included primarily for readers who wish to verify correctness; readers not interested in the technical details can simply skip them and refer to the provided implementations.
>
> - [S] and [3] indeed overlap in the theoretical results regarding the symmetry of Mixture-of-Experts architectures. We sincerely acknowledge that this overlap was our mistake.
>
> We sincerely thank the Reviewer for carefully identifying these issues. We fully agree that they are substantial and cannot be adequately addressed within the rebuttal phase. Consequently, we have made the necessary decision to withdraw our submission.
>
> As the last author and advisor to the student authors, I take full responsibility for the mistakes in this work. The students made their best effort to extend our previous research and continue this line of inquiry within our group. This experience serves as an important lesson for all of us on our academic journey.
>
> We remain sincerely grateful for the Reviewer’s thoughtful feedback and understanding, which will greatly help us in revising the paper for future submissions.

---

### Official Review · Reviewer_UtwA · 2025-11-06

**Soundness:** 3
**Presentation:** 3
**Contribution:** 3
**Rating:** 6
**Confidence:** 3

**Summary:**

The paper proposes a new neural network architecture that is equivariant w.r.t. the symmetries in a Transformer MoE. The specific symmetries considered in this work are the permutation symmetry for the experts and the translations for the gating logits, which complements previous works that characterized the symmetries in standard Transformers. The paper shows that the proposed architecture captures all universal symmetries. The contributions include a new MoE Transformer Zoos dataset. Empirically, the proposed architecture is demonstrated to perform favorably against previous methods that lack the specific symmetries that MoE introduce.

**Strengths:**

- comprehensive characterization of the MoE symmetries and formal invariance proofs for dense and sparse cases
- largish scale evaluation with around 179k checkpoints
- MoE Transformers are amongst the most relevant architectures due to current frontier LLMs relying on them, so the work has high relevance

**Weaknesses:**

Experiments should be more extensive to better understand the benefits of explicitly accounting for the MoE symmetries. Currently, they are limited to predicting the test accuracy of MoEs. Additional experiments could for example include those considered in the weight space literature. The results on MNIST-MoEs show that the proposed network underperforms LightGBM in certain settings, which is surprising and deserves further investigation and discussion

**Questions:**

How often do the non-degeneracy conditions hold in the trained MoEs? What's the frequency of near-duplicate experts or linearly dependent gating differences?

---

### Author Response · Authors · 2025-11-12
**Withdrawal Notice and Acknowledgment to Reviewers**

We thank the Reviewers for their valuable feedback on our submission. In light of the issues raised, we have decided to withdraw the paper. We remain sincerely grateful for the Reviewers’ thoughtful comments and understanding, which will greatly assist us in revising the work for future submissions.

---

### Note · Authors · 2026-01-08

I have read and agree with the venue's withdrawal policy on behalf of myself and my co-authors.